# Exploring the Flavor Symmetry Landscape

Alfredo Glioti [ID],[*a]  Riccardo Rattazzi [ID],[†b,c]  Lorenzo Ricci [ID],[‡d] and Luca Vecchi [ID],[§e]

[a] *Université Paris-Saclay, CNRS, CEA, Institut de Physique Théorique, 91191, Gif-sur-Yvette, France*
[b] *Theoretical Particle Physics Laboratory, Institute of Physics, EPFL, Lausanne, Switzerland*
[c] *Center for Cosmology and Particle Physics, New York University, 726 Broadway, New York, NY 10003, U.S.A.*
[d] *Maryland Center for Fundamental Physics, Department of Physics, University of Maryland, College Park, MD 20742, USA*
[e] *Istituto Nazionale di Fisica Nucleare, Sezione di Padova, I-35131 Padova, Italy*

## Abstract

We explore flavor dynamics in the broad scenario of a strongly interacting light Higgs (SILH). Our study focuses on the mechanism of partial fermion compositeness, but is otherwise as systematic as possible. Concretely, we classify the options for the underlying flavor (and CP) symmetries, which are necessary in order to bring this scenario safely within the range of present or future explorations. Our main goal in this context is to provide a practical map between the space of hypotheses (the models) and the experimental ground that will be explored in the medium and long term, in both indirect and direct searches, in practice at HL-LHC and Belle II, in EDM searches and eventually at FCC-hh. Our study encompasses scenarios with the maximal possible flavor symmetry, corresponding to minimal flavor violation (MFV), scenarios with no symmetry, corresponding to the so-called flavor anarchy, and various intermediate cases that complete the picture. One main result is that the scenarios that allow for the lowest new physics scale have intermediate flavor symmetry rather than the maximal symmetry of MFV models. Such optimal models are rather resilient to indirect exploration via flavor and CP violating observables, and can only be satisfactorily explored at a future high-energy collider. On the other hand, the next two decades of indirect exploration will significantly stress the parameter space of a large swath of less optimal but more generic models up to mass scales competing with those of the FCC-hh.

---

[*]alfredo.glioti@ipht.fr
[†]riccardo.rattazzi@epfl.ch
[‡]lricci@umd.edu
[§]luca.vecchi@pd.infn.it

# 1    Introduction

Finding an explanation for the observed pattern of fermion masses and mixing remains one of the grand goals of particle physics. While the Standard Model (SM) does not address that question, the simplicity of its flavor structure and the remarkable experimental adequacy it implies clearly stand out. The core of that success resides in the Glashow-Iliopoulos-Maiani (GIM) mechanism [1] and consists of a powerful set of selection rules determined by the structurally robust *minimality* of the flavor-violating couplings. SM extensions aiming at a natural explanation of the Higgs sector dynamics are instead, and unfortunately, not endowed with the same minimality. Flavor has thus always been the crux of such otherwise well-motivated models. In order to adequately describe experimental flavor data either some clever mechanism has to be invoked or the scale of new physics has to be pushed up. While the first option may be viewed as ad hoc, the second, besides looking less plausible on the grounds of naturalness, is also less amenable to direct experimental study. When exploring the energy frontier it is thus fair to ask which model structures, and how clever, are needed in order to ensure meaningful new physics can exist at the energy of the machine without conflicting with flavor data. In general, the cleverness of the mechanisms that suppress unwanted flavor violation corresponds to flavor symmetry assumptions. One would grossly expect a bigger flavor symmetry group and a smaller set of symmetry breaking couplings ($\equiv$ spurions) to imply a stronger suppression of flavor and CP violation. Correspondingly that should allow for a lower scale of new physics. Given the quantum numbers of the SM fermions, the biggest possible group choice is $U(3)^5$ while the minimal set of spurions just coincides with the three SM Yukawa matrices $Y_{u,d,e}$. The choice of these limiting options determines the hypothesis of so-called Minimal Flavor Violation (MFV) [2]. The *minimal* in the label refers to the set of spurions, but considering the group choice is *maximal*, the hypothesis could equally well be labeled Maximal Flavor Conservation. MFV is the most straightforward hypothesis to implement at the level of SM effective field theory. Examples of the implementation of MFV in concrete models of electroweak symmetry breaking date back to composite technicolor models [3] and to supersymmetric models with gauge [4] or gaugino mediation [5, 6].

The above facts explain the widespread use of MFV when studying the generic implications of new physics, in particular in recent studies of Higgs couplings (see e.g. [7]). However, a broader perspective on flavor is greatly desirable. That is mainly because, given the uncertainty of our vision on new physics, the characterization of the space of flavor hypotheses and its broad exploration can only help organize our expectations. That is particularly true for scenarios of composite Higgs where the underlying dynamics cannot be fully specified, and where the most concrete constructions we possess [8] are based on warped compactifications [9] through holography. For instance, when taking the dynamics of new physics into account, it may well be that scenarios other than MFV offer the best option to lower the scale.

A broad perspective on flavor inheres, in our mind, in any strategic vision of the future of particle physics. The next decade, besides advances in the measurement of Higgs properties at the HL-LHC, will witness great progress in the experimental study of flavor and CP violation. That will especially be through more precise b-physics studies at LHCb [10] and Belle II [11], but also through kaon physics studies, like at NA62 and KOTO [12], searches for lepton flavor violation, like at Mu2e [13], Mu3e [14] and MEG II [15], and finally searches for electric dipole moments [16, 17]. But the next decade will also be a crucial time to plan the next big step in the exploration of the energy frontier. A map of the options in the space of flavor hypotheses

seems to us a necessary tool to have. To be explicit and concrete: already with the flavor data presently available, composite Higgs models with the least elaborate flavor sector, e.g. the structurally attractive "flavor anarchy scenario", are pushed up to scales in the $20 \div 30$ TeV range, where even FCC would struggle. The next decade of flavor may change the landscape either by finding a deviation from the SM or by making the constraints stronger. In either case, it will be important to be able to correlate the forthcoming precision flavor measurements with direct searches at the future big machine, which can only be achieved by classifying and evaluating the available flavor physics hypotheses. In this paper, we shall partially address this task by classifying the flavor symmetry options in the scenario of partial compositeness.

This paper is organized as follows. We start in Section 2 by presenting our hypotheses, reviewing the framework of partial compositeness, and illustrating how they determine the structure of the effective field theory. The scenario of "flavor anarchy" is reviewed and the most constraining experimental bounds on that model are critically updated. After a few preliminary considerations, in Sec. 3 we present a number of representative flavor symmetry options for the quark sector, analyzing the corresponding phenomenology in Sections 4 and 5. The reader's guide in Section 3.2 will support the reader in this journey. Section 6 is devoted to a discussion of a few flavor options for leptons. Section 7 provides a detailed and extended summary of our results along with a perspective on the next two decades of experimental exploration. Our results are summarized in Figs. 5, 6, 7, 8, 9 where for each scenario we show the present constraints as well as the expected reach of the forthcoming explorations. Several appendices contain the basic technical results.

## 2  Working assumptions and main concepts

The main goal of our study is to design a plausible framework in which to correlate low-energy observations in flavor physics with searches at the energy frontier. As we already stated, we shall be working under the assumption that flavor violation is controlled by some symmetry. We must stress, however, that, in order to at least partially achieve our goal, the mere hypotheses on flavor symmetry are not sufficient. Additional hypotheses, and parameters, characterizing the underlying dynamics are necessary. For instance, as the coefficients of effective operators from new physics depend on both couplings and masses, extra assumptions on the couplings are needed in order to translate low-energy constraints into constraints on the mass scale. Another class of important assumptions pertains to the embeddability of the low-energy EFT in a plausible and well-motivated scenario, in particular in connection with electroweak symmetry breaking. While at this stage these statements seem somewhat abstract, this section should make them more concrete to the reader. The rest of our study will also illustrate the synergy among the different structural requests. Indeed, we cannot avoid pointing out that the approach we are going to follow is regretfully not the dominant one in the now vast and ever increasing literature on SMEFT as well as in the older literature on MFV. The vast majority of those studies content themselves with a parametrization of the low-energy Lagrangian, without any serious attempt to depict what it means, or if it means anything at all, from a microscopic perspective. That approach is usually justified under the umbrella of generality. We are not objecting to the notion of generality per se but to the absence of the minimal set of self-consistent assumptions that are worth testing and that are necessary to produce a *story*. Our story will consist in the correlation between low-energy flavor data, high-energy data, and possibly electroweak data. It will follow from

a set of assumptions, which is specific but encompasses a vast landscape of models. Indeed the same methodology can be followed on the basis of other assumptions. For instance, one could assume the alternative flavor dynamics depicted in Refs. [18–21].

As mentioned in the introduction, we shall focus on the scenario where flavor is controlled by the so-called *partial compositeness* mechanism. The notion of partial compositeness was first developed in the early 90s [22] in the context of technicolor models and has become the flavor dynamics of choice of "modern" composite Higgs models (see [23] for a comprehensive review), where the Higgs boson arises as a pseudo-Nambu-Goldstone boson. However, the structural hypotheses underlying partial compositeness apply more broadly. In our study we shall apply them to the slightly broader scenario of a Strongly Interacting Light Higgs (SILH) [24], upon which further specific hypotheses can be added, at will.

In Subsection 2.1 we shall introduce the basic concepts and assumptions that underlie the class of models we want to study. These will, in particular, give rise to power counting rules controlling the low-energy effective Lagrangian and the resulting phenomenology (see Subsection 2.2. In Subsection 2.3 we shall instead illustrate the scenario of flavor anarchy, which will serve as the starting point of our exploration.

## 2.1 Partial compositeness

We will here schematically present the hypotheses and the concepts that underlie our work.

We will study flavor in a class of models of physics beyond the Standard Model that satisfy the following hypotheses.

- New physics relevant for electroweak symmetry breaking, flavor, and future collider searches is broadly characterized by a mass scale $m_*$. That is to say that new states are more or less clustered around the scale $m_*$ and that inverse powers of $m_*$ control the operator expansion in the effective Lagrangian below threshold. Indeed, as clarified below, the structure of the effective Lagrangian is also controlled by further hypotheses concerning the interaction strengths and approximate symmetries.

- There exists a hierarchically large window of energies above $m_*$ and below some UV scale $\Lambda$, with $\Lambda \gg m_*$, where physics is described by two QFT sectors weakly coupled to one another. One sector, which we indicate by SM′, consists of the gauge fields and fermions of the SM, i.e. of the SM fields minus the Higgs doublet. The other is an approximately scale (or more plausibly conformal) invariant sector which we simply indicate as the *strong sector*. At energies below $m_*$ the strong sector reduces to just the SM Higgs doublet. One could for instance picture the scale $m_*$ as the scale of confinement and the massive states and Higgs as the resulting bound states. But the strong sector could also be modeled holographically, in which case $m_*$ simply corresponds to the mass of Kaluza-Klein states [8].

- The only relevant or marginally irrelevant couplings[1] between the two sectors are (see Fig. 1 for a pictorial representation):

    1. the weak gauging of a $SU(3) \times SU(2) \times U(1)$ subgroup of the global symmetry of the strong sector;

---

[1] By this we mean couplings associated with operators whose dimension is not significantly larger than 4.

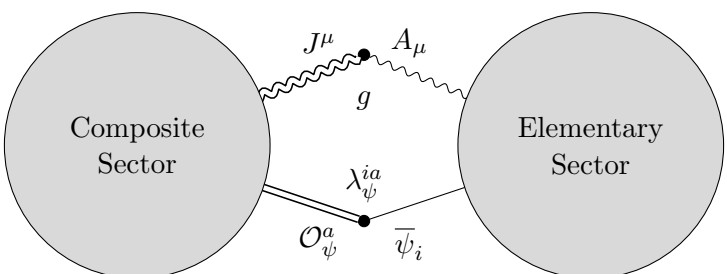

Figure 1: Schematic representation of the dominant interactions between the strongly-coupled sector and SM′ in partial compositeness.

2. a linear mixing between the elementary fermion fields $\psi^i$ of SM′ and composite fermionic operators $\mathcal{O}_\psi^a$ of the strong sector. The $\mathcal{O}_\psi^a$ have then, by hypothesis, scaling dimension $\lesssim 5/2$. The mixings connect the SM′ fermions to the Higgs dynamics, giving rise, below the scale $m_*$, to the ordinary SM Yukawa interactions.

Based on the above hypotheses, at energies above $m_*$ the system is described by a Lagrangian of the form

$$\mathcal{L} = \mathcal{L}_{\mathrm{SM}'} + \mathcal{L}_{\mathrm{strong}} + g A_\mu J^\mu + \lambda_\psi^{ia} \overline{\psi}^i \mathcal{O}_\psi^a + \dots \tag{2.1}$$

where $\mathcal{L}_{\mathrm{SM}'}$ and $\mathcal{L}_{\mathrm{strong}}$ purely depend on fields of the corresponding sectors. By $g$, $A_\mu$ and $J^\mu$ we collectively indicate respectively the SM gauge couplings, the SM gauge fields, and the corresponding currents from the strong sector.[2] The $\psi = q_L, u_R, d_R, \ell_L, e_R$ collectively indicate the SM fermion fields, and the couplings $\lambda_\psi^{ia}$ represent the seeds of the Yukawa interactions. The ellipses indicate operators with dimension strictly $> 4$ whose relevance is suppressed by the existence of a large window of approximate scaling above $m_*$. More precisely, given the upper edge of the energy window sits at $E \sim \Lambda \gg m_*$, the effect of these operators at low energy is expected to be suppressed by positive powers of $m_*/\Lambda$. Some effects may nonetheless still be dominated by these terms. In particular, neutrino masses could still arise from bilinear interactions of the form (see [25] for more general options)

$$\frac{\lambda^{ij}}{\Lambda^{\Delta_T - 1}} \overline{\ell}_L^i \ell_L^{cj} \mathcal{O}_T \tag{2.2}$$

where $\ell_L^i$ are the SM lepton doublets, while $\mathcal{O}_T$ is some composite scalar of dimension $\Delta_T$ transforming as a triplet under the electroweak $SU(2)_L$. At the scale $m_*$, $\mathcal{O}_T$ is simply matched to a Higgs bilinear triplet $\mathcal{O}_T \to m_*^{\Delta_T - 2} HH$, giving eventually a Majorana mass matrix $m_\nu^{ij} \propto (\lambda^{ij} v^2 / \Lambda)(m_*/\Lambda)^{\Delta_T - 2}$.

When trying to depict the phenomenology of the above scenario, further hypotheses can in general be considered, to either help manage the analysis or to help make the model more plausible, either structurally or phenomenologically. Here is a list of these extra hypotheses.

- The hierarchical separation $\Lambda \gg m_*$ is natural, either because the strong sector CFT does not possess strongly relevant deformations or because its only relevant deformations

---

[2]Of course the coupling to $A_\mu$ is expressed as linear only indicatively. In general, like when scalars are present, there can appear nonlinear contact terms, which however do not correspond to independent couplings.

are controlled by symmetries. While the addition of this extra hypothesis would make the scenario more plausible in light of the hierarchy problem, the phenomenological implications would be unaffected.

- In order to even qualitatively describe the low-energy phenomenology, the sole parameter $m_*$ is insufficient and some hypothesis on the mutual couplings of the strong sector states is necessary. The minimal hypothesis one can make is that these interactions are characterized by a single coupling $g_*$. As we shall deduce in the next section, consistency demands $g_*$ to be larger than the largest SM couplings (the gauge couplings and the top Yukawa), hence the appellative *strong sector*. The "one-coupling situation" is actually realized in the most minimal holographic incarnations of partial compositeness, with $g_* \sim 4\pi/\sqrt{N}$ and $N$ denoting the number of weakly coupled Kaluza-Klein modes below the 5D cut-off.[3] A variant of this situation occurs in large $N$ gauge theories, where the emergent coupling scales like $g_* \sim 4\pi/\sqrt{N}$ for mesons and $g_* \sim 4\pi/N$ for glueballs. In the next section, we will illustrate the concrete implementation and implication of the one-coupling one-scale ($g_*$ and $m_*$) hypothesis, which we will follow throughout this paper. Relaxing it, a much richer structure of scales and couplings becomes available. In Appendix A.2 we discuss some of the qualitatively new features encountered when the one-coupling hypothesis is relaxed. As it turns out, these do not correspond to more phenomenologically favorable scenarios.

- In the limit in which the couplings to the SM′ are turned off, the strong sector may be endowed with specific global symmetries. Their presence can help make the phenomenology more realistic without excessive tunings of parameters. Following the indication of the electroweak precision data, we shall always assume Custodial Symmetry $SU(2)_L \times SU(2)_R \sim SO(4)$. We shall also optionally consider the extension of $SO(4)$ to $O(4)$ via an internal $P_{\mathrm{LR}}$ parity [27]. This option helps ease constraints both in the electroweak and flavor sectors. A review of custodial symmetry and of its implementation in partial compositeness is given in Appendix A.1, to which we shall often refer. A further option is to have the Higgs as a pseudo-Nambu-Goldstone boson. The simplest realization of that, and compatible with custodial symmetry, is to have the Higgs doublet identified with the coset space $SO(5)/SO(4)$ [8]. That hypothesis explains, at least in part, the separation of mass between the Higgs particle, at $\sim 125$ GeV, and the other resonances of the strong sector, at $m_* \gtrsim 1$ TeV. The pseudo-Nambu-Goldstone hypothesis can also give rise to selection rules that suppress classes of FCNC, to the Higgs boson in particular [28]. We shall treat this hypothesis as an important option, but most of our results in the domain of flavor physics do not strongly rely on it.

- Along with the $U(3)^5$ symmetry of $\mathcal{L}_{\mathrm{SM}'}$ the strong sector can itself be endowed with a flavor symmetry under which the fermion operators $\mathcal{O}_\psi^a$ transform. In fact at least baryon and lepton number are always assumed respected by the strong sector in order to avoid fast proton decay and heavy neutrinos. CP invariance, as we shall see, is also another phenomenologically well-motivated symmetry of the strong sector. The combination of all these symmetries, the full symmetry group of the model, is partially broken by the mixing couplings $\lambda_\psi^{ia}$. The choice of symmetry and the pattern of breaking through $\lambda_\psi^{ia}$ give rise to a landscape of models whose exploration is the primary goal of

---

[3]This is for instance explained in Appendix A of Ref. [26].

this paper.

## 2.2 Effective Lagrangian

The centerpiece of our study is the generic low-energy effective Lagrangian that is derived from the hypotheses of the previous section, as we now explain. Our approach is standard in the Composite Higgs literature (see for instance [23, 24]).

Throughout the paper we will make the minimal "one-coupling one-scale" hypothesis, and limit ourselves to comments on the more general cases. In App. A.2 we illustrate the less minimal case of two-couplings and one-scale and show how it is phenomenologically less favorable, hence our limitation to the minimal option.

Concretely, the one-scale hypothesis implies that the masses of the resonances are all roughly of the same order, i.e. $m_*$. The parameter $m_*$ is then the analog of the hadron mass scale in confining gauge theories, at both small and large $N$.[4] The one-coupling hypothesis dictates instead that amplitudes with $n$ legs scale like $g_*^{n-2}$. Again an example of that is offered by $SU(N)$ pure Yang-Mills at large $N$ where the rule is satisfied with $g_* \sim 4\pi/N$. Notice however that, while large $N$ gauge theories serve as partial inspiration for the structure of the theories we are considering, we are in no way committing to a specific microscopic description of our models. In fact the strong sector may not even correspond to an ordinary Lagrangian gauge theory. In other words, it could consist of a CFT that cannot be obtained through the RG flow generated by a relevant deformation (e.g. a gauge coupling) in an ordinary weakly coupled Lagrangian field theory.

The "one coupling one-scale" hypothesis is readily implemented when writing the Lagrangian that describes the interactions of resonances at the scale $m_*$. Indicating collectively by $\Phi$ the fields that interpolate for the resonances (and taking them as dimensionless) the original microscopic Lagrangian of Eq. (2.1) is matched at $E \sim m_*$ to[5]

$$\mathcal{L} = \mathcal{L}_{\mathrm{SM}'} + \frac{m_*^4}{g_*^2} \mathcal{L}_{\mathrm{res}} \left( \Phi, \frac{\partial_\mu}{m_*}, \frac{gA_\mu}{m_*}, \frac{\lambda_\psi^{ia}\overline{\psi}^i}{m_*^{3/2}} \right) + \dots , \tag{2.3}$$

where the dots now indicate, besides all effects suppressed by the high cut-off $\Lambda$, the more important corrections to the matching coming from loops of both the SM$'$ and strong sector fields. These effects are thus controlled by $g^2/16\pi^2$, $\lambda_\psi^2/16\pi^2$, $g_*^2/16\pi^2$, etc. Notice also that we have absorbed the linear couplings of the strong sector to $A_\mu$ and to $\psi^i$ into $\mathcal{L}_{\mathrm{res}}$. When comparing to Eq. (2.1), this then simply implies the matching

$$\frac{\delta}{\delta(gA_\mu)} \frac{m_*^4}{g_*^2} \mathcal{L}_{\mathrm{res}} \to J^\mu , \qquad \frac{\delta}{\delta(\lambda_\psi^{ia}\overline{\psi}^i)} \frac{m_*^4}{g_*^2} \mathcal{L}_{\mathrm{res}} \to \mathcal{O}_\psi^a . \tag{2.4}$$

By expanding Eq. (2.3) in powers of $\Phi$ and by normalizing the fields canonically one can easily check that the $n$-resonance amplitude is $\propto g_*^{n-2}$. Implicit in this derivation is that $\mathcal{L}_{\mathrm{res}}$ is a generic function of its arguments. A corollary is that any expectation values of $\Phi$ must

---

[4]In large$-N$ QCD there exist heavier states but, with the exclusion of baryons, their width grows with their mass and one does not expect narrow resonances way above the hadron mass scale.

[5]Although we use the same notation for the high scale parameters in Eq. (2.1) and the low scale ones in Eq. (2.3) it is understood that RG evolution must be taken into account.

be $O(1)$, which, when expressed in terms of the canonical fields $\Phi_c \equiv (m_*/g_*)\Phi$, gives the expected scale $f$ of symmetry breaking vacuum expectation values (VEVs)

$$f = \frac{m_*}{g_*} \, . \tag{2.5}$$

When considering the Higgs field, the smallness of the ratio $(\langle H\rangle)^2/f^2 = v^2/2f^2$ (with $v \approx 246$ GeV) gives then a measure of the tuning in the vacuum dynamics.

Integrating out all resonances except the Higgs doublet, Eq. (2.3) leads to an effective low-energy Lagrangian of the form

$$\mathcal{L}_{\mathrm{EFT}} = \mathcal{L}_{\mathrm{SM}'} + \frac{m_*^4}{g_*^2}\widehat{\mathcal{L}}_{\mathrm{EFT}}\left(\frac{g_* H}{m_*}, \frac{D_\mu}{m_*}, \frac{\lambda_\psi^{ia}\overline{\psi}^i}{m_*^{3/2}}, \frac{g_*^2}{16\pi^2}, \frac{g^2}{16\pi^2}, \frac{[\lambda_\psi^*]^{ia}\lambda_\psi^{ib}}{16\pi^2}\right), \tag{2.6}$$

where again $\widehat{\mathcal{L}}_{\mathrm{EFT}}$ is a polynomial with unknown coefficients of order unity and where we have included the formal dependence on the effects of loops (i.e. the dependence on $g^2/16\pi^2$, $[\lambda_\psi^*]^{ia}\lambda_\psi^{ib}/16\pi^2$ and $g_*^2/16\pi^2$). We can now quantify what is meant by strongly-coupled sector: such dynamics is "strong" in the very concrete sense that $g_* \gtrsim g$. Indeed, among the interactions described by Eq. (2.6) we find corrections to the kinetic terms of the SM gauge fields with relative size $\sim g^2/g_*^2$. If we had $g^2/g_*^2 \gtrsim O(1)$, the propagation of the SM gauge fields would be dominated by strong sector effects, somewhat in contradiction with the intuitive notion that the two sectors can be treated as separate in first approximation. We will thus assume $g^2/g_*^2 \lesssim O(1)$, and, by a similar reasoning, $\lambda_\psi/g_* \lesssim 1$.

The Lagrangian in Eq. (2.6) provides a rule to compute up to $O(1)$ coefficients the interactions of the low-energy effective theory. For example, the SM up-type Yukawa coupling $Y_u^{ij}$ is given by

$$\mathcal{L}_{\mathrm{EFT}} \supset c_{ab}\frac{m_*^4}{g_*^2}\frac{g_* H}{m_*}\frac{\lambda_q^{ia}\overline{q}^i}{m_*^{3/2}}\frac{[\lambda_u^*]^{jb}u^j}{m_*^{3/2}} \tag{2.7}$$

$$\implies \quad Y_u^{ij} = c_{ab}\frac{\lambda_q^{ia}[\lambda_u^*]^{jb}}{g_*} \equiv g_*\varepsilon_q^{ia}c_{ab}[\varepsilon_u^*]^{jb}.$$

with $c_{ab}$ a set of $O(1)$ coefficients. The matrices $\varepsilon_q \equiv \lambda_q/g_*$ and $\varepsilon_u \equiv \lambda_u/g_*$ parametrize the mixing between the SM fermions $q$ and $u$ with the corresponding strong sector resonances interpolated by respectively $\mathcal{O}_q$ and $\mathcal{O}_u$. Their eigenvalues then offer a measure of the degree of compositeness of the corresponding eigenstates in $u$ and $q$ field space. The coupling in Eq. (2.7) can also be pictured as being generated by the diagram in Fig. 2 (left): the strong sector resonances have a Yukawa coupling $\propto g_*c_{ab}$, which they *share* with the elementary fermions through the mixing matrices ($\varepsilon_q$, $\varepsilon_u$).

In a similar manner, Eq. (2.6) also dictates the presence of 4-$q$ interactions of the form

$$\mathcal{L}_{\mathrm{EFT}} \supset \frac{g_*^2}{m_*^2}c_{abcd}\varepsilon_q^{ia}[\varepsilon_q^*]^{jb}\varepsilon_q^{kc}[\varepsilon_q^*]^{ld}\overline{q}^i\gamma_\mu q^j\overline{q}^k\gamma^\mu q^l \, . \tag{2.8}$$

Again, as shown in Fig. 2 (right), this can be diagrammatically depicted as an effective 4-fermion interaction $\propto g_*^2c_{abcd}/m_*^2$ among the strong sector resonances which is *shared* with the elementary fermions through the $\varepsilon_q$'s mixing matrices.

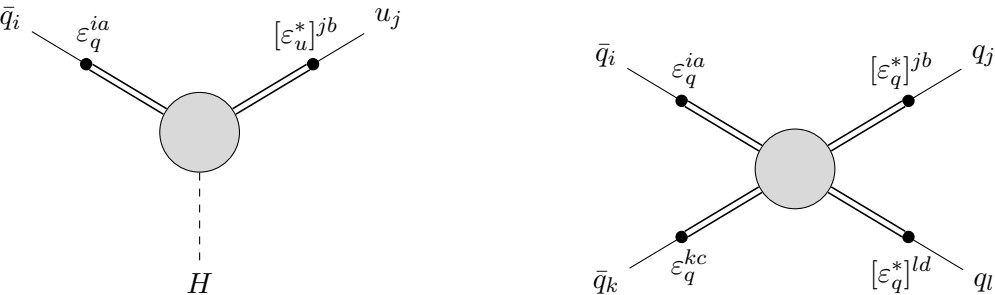

Figure 2: Pictorial representation of the up-type Yukawa and 4-$q$ vertices.

Eq. (2.6) will be used throughout this paper to estimate the size of the Wilson coefficients of the effective field theory below $m_*$. Besides flavor, these interactions will affect in particular Higgs couplings and precision electroweak quantities, as already studied in numerous papers (see for instance [7,23,24]). Our hypotheses, implemented through the Lagrangian in Eq. (2.3), allow us to express any physical observable in terms of a few basic objects: the couplings $g_*$, $g$, the fermion mixing matrices $\varepsilon_\psi$, and the fundamental mass scale $m_*$. In particular that allows us to correlate the indirect effects (flavor, Higgs, and electroweak) with the direct searches for resonances. Searches for top partners and heavy vector states have been amply studied along those lines in the scenario of composite Higgs (see e.g. [29,30]). We believe that is a crucial conceptual advantage with respect to analyses purely based on the notion of SMEFT (see for instance [31–33] and [34]), where the absence of clear dynamical hypotheses does not allow to correlate different measurements, thus limiting the perspective.

## 2.3 Lessons from anarchy

As we already mentioned, the strong sector can in general be endowed with global symmetries. That will result in selection rules satisfied by the Wilson coefficients like $c_{ab\cdots}$ and $c_{abcd}$ just discussed. As concerns flavor, the interplay among the strong sector symmetries and the structure of the mixings $\varepsilon_\psi$ will give rise to a landscape of options which is the main target of our study. However, in order to better appreciate the relevance and the need for family symmetries, it is important to first depict the situation where no such symmetry is assumed. That is the framework of "anarchic partial compositeness", where no family quantum numbers are assumed besides the overall baryon and lepton numbers, thus corresponding to the minimal symmetry hypothesis. Such minimality remarkably sets the stage for a dynamical explanation of the observed hierarchical pattern of fermion masses and mixings. As no flavor symmetry is associated with the strong dynamics indices $a, b, \cdots$ of the fermionic operators $\mathcal{O}_\psi^a$, the mixing couplings at the UV cut-off scale $\lambda_\psi^{ia}(\Lambda)$ are assumed to be generic unstructured matrices. Also, generically the scaling dimensions can be ordered as $\Delta_\psi^1 < \Delta_\psi^2 < \Delta_\psi^3 < \dots$. Parametrizing the dimensions as $\Delta_\psi^a = 5/2 + \gamma_\psi^a$, the relation between the mixing couplings at the IR scale and UV scale is then

$$\lambda_\psi^{ia}(m_*) \simeq \lambda_\psi^{ia}(\Lambda) \left(\frac{m_*}{\Lambda}\right)^{\gamma_\psi^a} \equiv \lambda_\psi^{ia}(\Lambda)e^{-\gamma_\psi^a L}\,, \qquad L \equiv \ln \Lambda/m_* \qquad (2.9)$$

A large hierarchy between $\Lambda$ and $m_*$, and consequently a sizeable $L$, will then induce a hierarchical structure in the Yukawa couplings at the scale $m_*$, even for a generic $\lambda_\psi^{ia}(\Lambda)$.

In particular, for the case in Eq. (2.7), mass ratios and mixings between generation $a$ and generation $b$ are respectively controlled by

$$e^{(\gamma_u^a - \gamma_u^b + \gamma_q^a - \gamma_q^b)L}, \qquad \text{and} \qquad e^{(\gamma_q^a - \gamma_q^b)L}. \tag{2.10}$$

For instance, for $\Lambda/m_* \sim 10^5$, differences in scaling dimension in the range $\Delta\gamma \sim 0.1 - 0.2$ are immediately seen to give a flavor structure in the right ballpark. A detailed analysis confirms that the observed spectrum of fermion masses and mixings is indeed reproduced even for generic $\lambda_\psi^{ia}(\Lambda)$. Flavor is thus explained dynamically.

As it inevitably happens in any predictive solution of the flavor problem, though, "anarchic partial compositeness" possesses sizeable sources of flavor violation at the scale $m_*$ below which the flavor structure becomes locked into the SM Yukawa couplings. The absence of signatures beyond the Cabibbo-Kobayashi-Maskawa (CKM) paradigm forces these sources of flavor violation to be small, which in turn implies that $m_*$ must sit well above the TeV scale. Strong constraints arise from flavor and CP-violation in both the quark and lepton sectors. In what follows we will quantify the most relevant bounds, beginning with the quark sector.

According to Eq. (2.6), and in the absence of any symmetry controlling CP violating phases, the electric dipole moment (EDM) of the neutron is expected to receive a contribution of order $d_n \sim d_{u/d} \sim e m_{u/d}/m_*^2$ from the quark dipole interactions $d_{u/d}$. To be compatible with the strong experimental constraint $|d_n| \leq 1.8 \times 10^{-26} e$ cm [35], the scale $m_*$ must satisfy

$$m_* \gtrsim 40 \div 60 \text{ TeV}. \tag{2.11}$$

In models where the dipole interactions are suppressed by a loop factor[6] $g_*^2/16\pi^2$, the bound on $m_*$ is reduced by a factor of $g_*/4\pi$

$$\frac{m_*}{g_*} \gtrsim 3 \div 5 \text{ TeV}. \tag{2.12}$$

Even for $g_* \sim 1$ this is still not enough to lower $m_*$ to the TeV scale: the resonances are forced above the reach of the LHC. Besides quark EDMs, also flavor-conserving, CP-odd, non-renormalizable interactions affect the neutron EDM, a fact which is often not well-appreciated.[7] Consider indeed the purely gluonic dimension-6 CP-violating operator [36]. Eq. (2.6) dictates the presence of the term

$$\mathcal{L}_{\text{EFT}} \supset c_* \frac{g_s^3(m_*)}{g_*^2 m_*^2} \frac{1}{3!} f^{abc} G_{\mu\rho}^a G_\nu^{b\rho} G_{\alpha\beta}^c \epsilon^{\mu\nu\alpha\beta}, \tag{2.13}$$

with $c_*$ an $O(1)$ coefficient.[8] Up to a dimensionless $O(1)$ factor associated with the non-perturbative QCD dynamics, we can then estimate

$$\frac{d_n}{e} \approx c(1 \text{ GeV}) \frac{g_s^3(m_*)}{g_*^2 m_*^2} \frac{\Lambda_{\text{QCD}}}{4\pi}, \tag{2.14}$$

---

[6] An example of this situation is offered by holographic realizations where the 5D bulk dynamics is characterized by a single scale where the 5D couplings all become strong. See for instance the discussion in Appendix A of Ref. [26].

[7] We assume here that the effects of the QCD topological angle are tamed by an axion.

[8] Notice that this result is, for instance, matched to the result of integrating out $N$ massive quarks by identifying $1/g_*^2 \sim N/16\pi^2$. That is compatible with the interpretation of $16\pi^2/g_*^2$ as a parameter that somehow counts the number of degrees of freedom in the strong sector.

where $c(1\,\text{GeV})$ is the QCD scale value of the 3-gluon operator running Wilson Coefficient $c(\mu)$ defined by the UV boundary condition $c(m_*) = c_*$. Computing first the RG running from the scale $m_*$ to 1 TeV and then down to $\sim 1$ GeV using the results of [37] we get $c(1\,\text{GeV}) \approx 0.3\,c(1\,\text{TeV}) \approx 0.3\,c(m_*)\,(\alpha_s(m_*)/\alpha_s(1\,\text{TeV}))^{15/14}$. Imposing the current experimental constraint, we find

$$m_* g_* \gtrsim 110\,\text{TeV}\,. \tag{2.15}$$

Even though for relatively large values of $g_*$ (or equivalently for a small number of colored constituents) this bound is weaker than that in Eq. (2.11), this result unequivocally demonstrates that sizable sources of CP-violations are not compatible with new physics at the TeV scale, irrespective of the flavor structure of the underlying model. Notice that even combining Eq. (2.15) with the favorable case of Eq. (2.12), one cannot lower $m_*$ below $\sim 20 \div 25$ TeV.

Assuming a similar anarchic structure in the leptonic sector leads to even more severe constraints. For example, the impressive recent bound on the EDM of the electron, $|d_e| \leq 4.1 \times 10^{-30} e$ cm [38] translates into the lower bound

$$m_* \gtrsim 2200\ \text{TeV}(g_*/4\pi)\,, \tag{2.16}$$

where we took $d_e \sim 2m_e/m_*^2(g_*^2/16\pi^2)$, conservatively including a loop factor in the dipole operator.

Constraints from flavor-violating processes only consolidate the picture (see e.g. [39]). Starting from the quark sector, a particularly severe bound comes from the CP asymmetry in D-meson decays. This reads $m_* \gtrsim 120\,\text{TeV}(g_*/4\pi)$,[9] where the $g_*/4\pi$ factor is conservatively introduced assuming a 1-loop suppression of the dipoles; the latter may or may not be present depending on the underlying dynamics, as discussed before. Meson-antimeson oscillations are also very significant. In particular, updating the analysis of [39] the constraint from $\epsilon_K$ reads $m_* \gtrsim 20 \div 25$ TeV. Furthermore, an anarchic flavor structure in the leptonic sector leads to even stronger bounds. Specifically, the non-observation of the transition $\mu \to e\gamma$ gives $m_* \gtrsim 250\ \text{TeV}(g_*/4\pi)$ in the presence of a $g_*^2/16\pi^2$ suppression of the dipole interaction [25].

In conclusion: in the quark sector anarchic partial compositeness is incompatible with new physics below $\lesssim 20$ TeV; in the lepton sector new physics must be above a few hundred TeV. Not only are these values outside the direct reach of the LHC but also practically outside the reach of any foreseeable future machine. How can we modify the scenario so as to bring it within the reach of present or future explorations at the energy frontier? We see two main alternatives. The first is to drastically modify our assumptions while sticking to models that dynamically explain the flavor structure. One example of that is offered by models that rely on multiple mass scales, with typically the highest scale associated with the smallest fermion masses and the lowest scale associated with the top quark mass (see for instance [18, 21, 40]). Another example consists in forgetting the hypothesis of partial compositeness and to allow bilinear couplings to play a role, at least for the light families [19, 20].

The second alternative, perhaps more humble, is to maintain the hypothesis of partial compositeness and that of a single mass scale but to assume the existence of flavor symmetries [25, 41–47]. Consistently with the options spelled out in Sections 2.1, we shall then assume the strongly-coupled sector is invariant under

$$\text{CP} \times \mathcal{G}_{\text{strong}}\,, \tag{2.17}$$

---

[9]In quoting this bound we assumed that the unknown factor $\text{Im}[\Delta R^{\text{NP}}]$ defined in App. C.4 is $\text{Im}[\Delta R^{\text{NP}}] = 1$ and we required that the CP asymmetry $\Delta a_{CP}$ is less than twice the current experimental uncertainty. See the discussion around (C.39) for details.

for some flavor group $\mathcal{G}_{\text{strong}}$. The strong bounds reviewed in this section arise because anarchic partial compositeness has completely unstructured coefficients $c_{ab\cdots}$, and hence many more flavor- and CP-violating parameters than the SM. By assuming flavor and CP symmetry, the $c_{ab\cdots}$'s are forced to comply with severe selection rules. That reduces the number of independent flavor- and CP-violating couplings and allows to relax the constraints, at least partly. In particular, the assumption of CP-invariance is motivated by the severity of the constraints in Eqs. (2.11, 2.15, 2.16) which all arise from CP-odd, flavor-conserving observables. CP-invariance of the strong sector corresponds to the existence of a basis where the coefficients $c_{ab\cdots}$ in Eq. (2.6) are all real, while CP violating phases purely reside in the mixing couplings $\lambda_\psi^{ia}$. As $\mathcal{G}_{\text{strong}}$ selection rules control the insertions of $\lambda_\psi^{ia}$ in physical quantities, the appearance of CP-violating phases will also be controlled.[10]

In the rest of the paper, we will provide a systematic exploration of the landscape of symmetry-based scenarios compatible with $m_* \sim$ few TeV. This approach does not offer a neat dynamical explanation for the observed flavor structure: the flavor symmetry breaking spurions now just parametrize the flavor structure precisely like the Yukawas do in the SM. Yet, in so doing we are able to significantly lower the scale of new physics and hence to identify and explore the patterns and correlations that current and future experiments are more likely to probe.

# 3    General considerations and plan of the paper

Among all symmetry-based scenarios for physics beyond the SM, those based on MFV generally cope better with the stringent bounds from flavor violation. MFV will therefore be the natural starting point of our analysis. As it turns out, however, it won't be our safest option.

As it will shortly become evident, in partial compositeness any implementation of MFV forces a strict *family universality* in some of the mixing matrices $\lambda_\psi$ of Eq. (2.1). That leads in turn to severe constraints from flavor-conserving observables [43, 45, 47]. A logical possibility to try and improve this state of things is then to consider a less-symmetric flavor hypothesis relaxing as much as possible the most severe flavor and CP bounds while avoiding the strictures of full flavor-universality. A good part of our paper will be devoted to the exploration of such structural compromise. But first, we must study the possible implementations of MFV.

We will analyze two representative implementations of MFV and propose a few simple and motivated less-symmetric, and thus less-universal, variations. We think the models we select for study well represent the different classes of scenarios and that a more thorough classification would not add significantly more information. Specifically, we will analyze scenarios with (partial) universality in the right-handed quark sector in Section 4 and scenarios with (partial) universality in the left-handed quark sector in Section 5. Leptons are discussed in Sections 6.

The remainder of this section is devoted to a number of general considerations. Sections 3.1 will introduce the basic setup all our models are based on, identify the realizations of MFV that it admits, and single out those that will be studied in detail in the rest of the paper. Subsequently, because the amount of material discussed in the paper is quite significant, Section 3.2 will provide a user's guide supporting the navigation of our study.

---

[10]A similar mechanism of control was presented in Ref. [48].

## 3.1  Setup and maximally symmetric scenarios (MFV)

In order to be able to present quantitative considerations we have to specify a concrete setup. We do it here by focusing for definiteness on the quark sector. Our analysis will be based on the realizations of partial compositeness that are minimal in terms of field content and interactions. More involved incarnations are not expected to possess qualitatively new features.

The absolutely minimal realization of partial compositeness contains three families of fermionic operators $\mathcal{O}_q$, $\mathcal{O}_u$, $\mathcal{O}_d$ with the correct quantum numbers to mix with $q_L$, $u_R$, $d_R$, respectively. Yet, in order to encompass a wider range of flavor hypotheses as well as phenomenologically well-motivated implementations of custodial symmetry (see Appendix A.1), we shall also consider a next-to-minimal realization. That is characterized by three families of $\mathcal{O}_{q_u}$, $\mathcal{O}_{q_d}$, $\mathcal{O}_u$, $\mathcal{O}_d$, involving *two* distinct types of composite operators mixing with $q_L$, one ($\mathcal{O}_{q_u}$) associated with up-quarks and the other ($\mathcal{O}_{q_d}$) with down-quarks.[11] Sections 4 analyzes the second, richer, option. We will come back to the minimal case of a single $\mathcal{O}_q$ in Sections 5. In both scenarios, CP is assumed to be a good symmetry of the strong sector so as to avoid the constraints from EDMs reported in Sections 2.3. In order to reproduce the CKM phase, CP is allowed to be maximally broken by the mixings $\lambda_\psi$.

Consider our next-to-minimal scenario, with two doublets $\mathcal{O}_{q_u}$ and $\mathcal{O}_{q_d}$. According to Eq. (2.1), the flavor-violating interactions between the quarks of the SM$'$ and the strongly-coupled sector are controlled by

$$\mathcal{L}_{\mathrm{mix}} = \lambda_{q_u}^{ia}\bar{q}_L^i\mathcal{O}_{q_u}^a + \lambda_{q_d}^{ia}\bar{q}_L^i\mathcal{O}_{q_d}^a + \lambda_u^{ia}\bar{u}_R^i\mathcal{O}_u^a + \lambda_d^{ia}\bar{d}_R^i\mathcal{O}_d^a, \tag{3.1}$$

where $i, j, \cdots = 1, 2, 3$ and $a = 1, 2, 3$ are flavor indices, the contraction of the gauge and Lorentz indices is suppressed, and the parameters $\lambda_{q_u,q_d,u,d}$ are complex matrices in flavor space. The composites, as we shall now explain, are characterized by suitable quantum numbers under the strong sector flavor symmetry while the Higgs scalar is obviously assumed flavor neutral.

In this setup, MFV can be realized in five distinct ways, summarized in Table 1. In the first four options, the strong sector is assumed to be endowed with a flavor symmetry $\mathcal{G}_{\mathrm{strong}} = U(3)_U \times U(3)_D$ (first column) under which the fermionic operators transform as $\mathcal{O}_{q_u,u} \in (\mathbf{3}, \mathbf{1})$ and $\mathcal{O}_{q_d,d} \in (\mathbf{1}, \mathbf{3})$. The fifth option corresponds to $\mathcal{G}_{\mathrm{strong}} = U(3)_{U+D}$ with all the composites $\mathcal{O}_{q_u,q_d,u,d}$ transforming as a $\mathbf{3}$. Indicating the SM quark flavor symmetry

$$\mathcal{G}_{\mathrm{quarks}} = U(3)_q \times U(3)_u \times U(3)_d \,, \tag{3.2}$$

the different scenarios are determined by the pattern of explicit breaking of $\mathcal{G}_{\mathrm{strong}} \times \mathcal{G}_{\mathrm{quarks}}$ induced by the mixing matrices $\lambda_{q_u,q_d,u,d}$. Each construction is characterized by two classes of mixings, the *universal* and the *non-universal* class. The universal mixings, indicated in the third column of Table 1, are proportional to identity matrices in flavor space and break $\mathcal{G}_{\mathrm{strong}} \times \mathcal{G}_{\mathrm{quarks}}$ to a diagonal subgroup $\mathcal{G}_F$ indicated in the fourth column of the table. The non-universal mixings, indicated in the fifth column, are proportional to the SM Yukawa couplings and can be viewed as spurions breaking $\mathcal{G}_F$ down to baryon number and hypercharge.

---

[11]This option is not ideal in scenarios of flavor anarchy, as the generation of hierarchical couplings for $q_L$ would require a correlation between $\lambda_{q_u}$ and $\lambda_{q_d}$ as well as in the scaling dimensions of $\mathcal{O}_{q_u,q_d}$. There is no problem here since we do not aim at explaining the origin of flavor hierarchies. We also emphasize that the occurrence of multiple partners for $q_L$ is natural in minimal PNGB Higgs models with composite fermions in $SO(5)$ multiplets corresponding to tensor products of $\mathbf{5} \in SO(5)$. See for instance [23] for more details.

| Name | $\mathcal{G}_{\text{strong}}$ | Universal $\lambda_\psi$ | $\mathcal{G}_F$ | Non-universal $\lambda_\psi$ |
|---|---|---|---|---|
| Right Univ. | $U(3)_U \times U(3)_D$ | $\lambda_u \propto \mathbf{1}, \lambda_d \propto \mathbf{1}$ | $U(3)_q \times U(3)_{U+u} \times U(3)_{D+d}$ | $\lambda_{q_u} \propto Y_u, \lambda_{q_d} \propto Y_d$ |
| Left Univ. $(Q_u Q_d)$ | $U(3)_U \times U(3)_D$ | $\lambda_{q_u} \propto \mathbf{1}, \lambda_{q_d} \propto \mathbf{1}$ | $U(3)_{q+U+D} \times U(3)_u \times U(3)_d$ | $\lambda_u \propto Y_u^\dagger, \lambda_d \propto Y_d^\dagger$ |
| Mixed Univ. $(Q_u D)$ | $U(3)_U \times U(3)_D$ | $\lambda_{q_u} \propto \mathbf{1}, \lambda_d \propto \mathbf{1}$ | $U(3)_{q+U} \times U(3)_u \times U(3)_{D+d}$ | $\lambda_u \propto Y_u^\dagger, \lambda_{q_d} \propto Y_d$ |
| Mixed Univ. $(Q_d U)$ | $U(3)_U \times U(3)_D$ | $\lambda_u \propto \mathbf{1}, \lambda_{q_d} \propto \mathbf{1}$ | $U(3)_{q+D} \times U(3)_{U+u} \times U(3)_d$ | $\lambda_{q_u} \propto Y_u, \lambda_d \propto Y_d^\dagger$ |
| Left Univ. $(Q)$ | $U(3)_{U+D}$ | $\lambda_{q_u} \propto \mathbf{1}, \lambda_{q_d} \propto \mathbf{1}$ | $U(3)_{q+U+D} \times U(3)_u \times U(3)_d$ | $\lambda_u \propto Y_u^\dagger, \lambda_d \propto Y_d^\dagger$ |

Table 1: The five distinct realizations of MFV within the theory described by Eq. (3.1).

The fact that the only sources of flavor non-universality are proportional to $Y_u$ and $Y_d$ explicitly realizes minimal flavor violation. The naming of the five scenarios presented in the first column is determined by which SM fermions mix universally.

In the scenarios of the first four rows of Table 1, $\mathcal{O}_{q_u}$ and $\mathcal{O}_{q_d}$ are distinguished by their flavor quantum numbers. For these scenarios, Yukawas are generated, as depicted in Fig. 2, independently in the up and down sector. Instead, in the scenario of the fifth row, $\mathcal{O}_{q_u}$ and $\mathcal{O}_{q_d}$ are indistinguishable: in practice that is as if there was a single type of composite doublets, the mixing to which controls both up and down Yukawas.

Notice that each scenario in the table admits multiple realizations according to the custodial symmetry assignments of the various $\mathcal{O}_\psi$. We will not present a systematic discussion of all the possible combinations of quantum numbers and limit ourselves to making comments where needed. Even if less systematic, this approach helps to contain the length of the analysis while providing the relevant information.

In the next section, we will analyze in detail the scenario of Right Universality, corresponding to the first row in Table 1. Similar considerations apply to the other scenarios. The scenarios in the second and fifth lines have in practice the same mixing structure, even though the symmetry of the fifth is a subgroup of that of the second.

The universal couplings in the third column are consequential even if flavor symmetric. For instance, in the scenario of Right Universality (Sections 4), $\lambda_u^{ia} \propto \delta^{ia}$ implies that all three $u_R^i$ mix with the same strength to the strong sector. At the same time, the $\lambda_u^{ia}$ cannot be too small because they must reproduce the observed top quark mass, see Eq. (2.7). As a consequence, sizable flavor-conserving deviations from the SM are introduced and result in important constraints discussed in Sections 4.1. Those constraints motivate us to look for alternatives with only Partial Right Universality in subsections 4.2 and 4.3. We will play a similar game also with models based on Left Universality (Sections 5) investigating also Partial Left Universality. In those scenarios, for the sake of demonstration, we will focus on the class of models based on the fifth row of Table 1. Models based on the second line basically imply the same results. The approach illustrated in Sections 4.1 and Sections 5.1 can be straightforwardly extended to models with Mixed Universality as well (see Table 1). A superficial analysis indicates that those scenarios do not bring urgent novelties and we leave their detailed exploration for the future.

## 3.2   Reader's guide

Sections 4 and 5 are devoted to a systematic investigation of the hypothesis of universality and partial universality in the quark sector. For each model, we will present its "General structure", subsequently discuss the "Experimental constraints", and finally provide a "Summary" of our results. The analysis is complemented by three appendices.

To ease the reading of the paper and help the reader navigate through its intricacies, in the following we schematically anticipate the content of the main body and the appendices and present our main conventions.

**Main body**

- **General structure.** This includes a detailed discussion of the basic hypotheses defining the model, along the lines already anticipated in the previous subsection. For each model, we state the symmetry $\mathcal{G}_{\text{strong}}$, identify the universal couplings and the unbroken $\mathcal{G}_F$, and finally present the explicit expression of the mixing parameters $\lambda_{q_u,q_d,u,d}$ in a flavor basis.

- **Experimental constraints.** Here we analyze the bounds on the parameters of the model. The EFT operators most relevant to our discussion can be grouped into four classes: 4-fermions, EW vertices, dipoles, and bosonic (see Table 4). The constraints from bosonic operators are independent of the flavor structure and are the same for all our models. These are therefore discussed separately in an appendix. "Experimental constraints" analyzes the other three classes of operators one by one. A separate analysis of the neutron EDM is also presented for some models.

- **Summary.** Here we present our main conclusions on each model. A visual summary is presented in the left panels of Figs. 5, 6, 7, 8, 9. Experimental prospects are discussed in the conclusions of the paper (Section 7).

**Appendices**

- Appendix A presents material necessary to complement the theoretical picture underlying our work.

  App. A.1 contains a discussion of the custodial $SO(4)$ and $P_{\text{LR}}$ symmetries, and how to implement them in our models. As we will see, such symmetries will play a significant role in suppressing not only electroweak constraints but also $\Delta F = 1$ transitions.

  App. A.2 discusses the main qualitative consequences of abandoning the one-coupling hypothesis we adopt throughout the paper.

- Appendix B contains a collection of the effective operators used in our experimental analysis as well as a detailed discussion of how their Wilson coefficients are determined in our models.

  App. B.1 includes a list of all the SM-invariant operators relevant to our work. Their coefficients in the SM-gauge basis, denoted by $\mathcal{C}_{\mathcal{X}}$, are determined as follows. The rules presented in Sections 2.2 allow to establish the general structure in terms of $\lambda_\psi, m_*, g_*$, at the desired order in $\lambda_\psi/g_*$, up to numbers of order unity. The explicit form of the $\lambda_\psi$'s in the gauge basis is found in the "General structure" for each model. The Wilson coefficients in the mass basis, denoted by $\widetilde{\mathcal{C}}_{\mathcal{X}}$, are finally derived via appropriate rotations as discussed in this appendix.

  App. B.2, for convenience, analyzes in detail the transition to the mass basis for the coefficients $\lambda_\psi \lambda_{\psi'}^\dagger$ of the quark bilinears. Their expressions in the mass basis are collected for all the flavor models in Tables 5 and 6.

App. B.3 contains the matching of the coefficients $\widetilde{\mathcal{C}}_\mathcal{X}$ of the SM-invariant operators to those of the low-energy operators we use in deriving the experimental bounds.

App. B.4 contains an estimate of the neutron EDM as a function of the effective operators. Our discussion emphasizes the role of the strange quark electric dipole moment.

- Appendix C contains a summary of all the experimental constraints. Only the most relevant ones are discussed in the main text. For each observable, we derive two bounds on the new physics parameter affecting it. The first corresponds to the 95% CL extreme of the allowed region that gives the weakest constraint. The second corresponds to asking that the new physics contribution be smaller than half the size of the 95% CL interval unless specified otherwise. For observables whose allowed interval is well centered around the SM prediction, the two methods give roughly the same bound. In that case, we only present one bound. However, there are observables for which the data are centered a little away from the SM, in which case the second method gives a stronger bound. In those cases, we present two bounds. The above comment concerns mostly low-energy measurements where the observables are affected linearly by the new physics. Instead, for the high-energy observables associated with 4-fermion contact interactions, it can in principle happen that the new physics contribution to scattering amplitude is comparable to or even bigger than the SM one. In those cases, the bound depends on whether new physics interferes constructively or destructively with the SM. This again produces two different bounds but for a different reason.

App. C.1 contains a discussion of the constraints on bosonic operators, which apply to all models.

App. C.2 analyzes the electroweak $\widehat{T}$ parameter. This is also described by a purely bosonic operator, but its coefficient is more model-dependent and deserves a separate discussion.

App. C.3 analyzes the flavor-dependent bounds. A collection of the tree-level bounds on anomalous $Z$ couplings is provided by Tables 9 and 10, those on tree $\Delta F = 1$ transitions are shown in Tables 11 and 12, and finally the constraints on $\Delta F = 2$ observables are in Tables 13 and 14.

App. C.4 discusses flavor-violating Higgs couplings, CP-violation in $D$ meson decays, and $K \to \pi\nu\nu$.

App. C.5 contains rough estimates of how the most constraining bounds analyzed earlier will evolve in the near future.

App. C.6 presents an assessment of the status and prospects of direct searches of fermionic and bosonic resonances, to be compared to the indirect constraints analyzed in the rest of the paper.

## Conventions

— $\psi = q_u, q_d, u, d$ is a label that distinguishes the fermionic operators $\mathcal{O}_\psi$ of the composite sector, see (2.1). By an abuse of notation $\psi$ may also denote a generic SM fermion field $(q_L, u_R, d_R)$ coupled to them.

— The relation between the SM fields in the gauge basis, $q_L = (u_L, d_L), u_R, d_R$, and those in the mass basis (identified with a "tilde"), is

$$u_L = U_u \widetilde{u}_L, \qquad d_L = U_d \widetilde{d}_L \tag{3.3}$$

$$u_R = V_u \widetilde{u}_R, \qquad d_R = V_d \widetilde{d}_R. \tag{3.4}$$

An explicit (approximate) expression for the matrices $U_{u,d}, V_{u,d}$ in each model is given in App. B.1.

— The family index is denoted by $i, j, \cdots = 1, 2, 3$ both in the gauge and the mass basis. Explicitly:

$$(\widetilde{u}^1, \widetilde{u}^2, \widetilde{u}^3) = (u, c, t) \tag{3.5}$$

$$(\widetilde{d}^1, \widetilde{d}^2, \widetilde{d}^3) = (d, s, b).$$

— The SM fields in the mass basis are collectively denoted by $f = \widetilde{u}, \widetilde{d}$ (see for example Eqs. (B.43, B.44)).

— $Y_{u,d}$ are the Yukawa matrices in the gauge basis, $\mathcal{L}_{\text{SM}} \supset [Y_u]^{ij} \bar{q}_L^i \widetilde{H} u_R^j + [Y_d]^{ij} \bar{q}_L^i H d_R^j + \text{hc.}$ In the mass basis, they are given by

$$\widetilde{Y}_u \equiv U_u^\dagger Y_u V_u = \begin{pmatrix} y_u & 0 & 0 \\ 0 & y_c & 0 \\ 0 & 0 & y_t \end{pmatrix}, \qquad \widetilde{Y}_d \equiv U_d^\dagger Y_d V_d = \begin{pmatrix} y_d & 0 & 0 \\ 0 & y_s & 0 \\ 0 & 0 & y_b \end{pmatrix} \tag{3.6}$$

with $y_{u,c,t}, y_{d,s,b}$ real and positive eigenvalues. The numerical values renormalized at 3 TeV are shown in Table 7.

— The CKM is expressed in the Wolfenstein parametrization in terms of $\lambda_C, A, \rho, \eta$. We use the numerical values shown in Table 8.

— The electroweak scale is defined as

$$v^2 = \frac{1}{\sqrt{2} G_F} = (246 \text{ GeV})^2. \tag{3.7}$$

— A generic effective operator in the gauge basis (see Table 4) is denoted by $\mathcal{O}_{\mathcal{X}}$. Its coefficient, still in the gauge basis, is $\mathcal{C}_{\mathcal{X}}$. When expressed in the mass basis it is instead denoted by $\widetilde{\mathcal{C}}_{\mathcal{X}}$.

— The operators $\mathcal{O}_{\mathcal{Y}}$ of the low-energy Lagrangian below the weak scale are by definition always written in terms of the $f$'s. Their Wilson coefficients are denoted as $C_{\mathcal{Y}}$ (see App. B.3 for a matching between the $C_{\mathcal{Y}}$'s and the $\widetilde{\mathcal{C}}_{\mathcal{X}}$'s).

# 4    Universality in the right-handed sector

In this section we explore universality, total or partial, in the right-handed quark sector, starting with the scenario in the first row of Table 1. A summary of the currently allowed parameter space in the most compelling scenarios is shown in the left panels of Figs. 5, 7 and 8. In Appendix C we collect tables with all the most relevant indirect constraints. The lowest allowed values of $m_*$ are also probed by direct searches (see Appendix C.6).

## 4.1 Right Universality (RU)

### 4.1.1 General structure

In models with Right Universality (RU), the strong sector is endowed with a

$$\mathcal{G}_{\text{strong}} = U(3)_U \times U(3)_D \tag{4.1}$$

global symmetry under which $\mathcal{O}_{q_u,u} \in (\mathbf{3},\mathbf{1})$ and $\mathcal{O}_{q_d,d} \in (\mathbf{1},\mathbf{3})$. The matrices $\lambda_{q_u,q_d,u,d}$ of Eq. (3.1) can be viewed as spurions with $\mathcal{G}_{\text{strong}} \times \mathcal{G}_{\text{quarks}}$ quantum numbers

$$\lambda_{q_u} \in (\overline{\mathbf{3}},\mathbf{1},\mathbf{3},\mathbf{1},\mathbf{1}), \qquad\qquad \lambda_{q_d} \in (\mathbf{1},\overline{\mathbf{3}},\mathbf{3},\mathbf{1},\mathbf{1}),$$
$$\lambda_u \in (\overline{\mathbf{3}},\mathbf{1},\mathbf{1},\mathbf{3},\mathbf{1}), \qquad\qquad \lambda_d \in (\mathbf{1},\overline{\mathbf{3}},\mathbf{1},\mathbf{1},\mathbf{3}). \tag{4.2}$$

The above properties in fact apply to the first four rows of Tab. 1, not just to RU. What distinguishes RU is the basic assumption $\lambda_{u,d}^{ia} \propto \delta^{ia}$, which leaves intact

$$\mathcal{G}_F \equiv U(3)_q \times U(3)_{U+u} \times U(3)_{D+d} \subset \mathcal{G}_{\text{strong}} \times \mathcal{G}_{\text{quarks}}. \tag{4.3}$$

The SM Yukawas, which according to Eq. (2.7) are given by $Y_u \sim \lambda_{q_u}\lambda_u^\dagger/g_*$, $Y_d \sim \lambda_{q_d}\lambda_d^\dagger/g_*$, are then simply related to the matrices $\lambda_{q_u,q_d}$ by a proportionality constant. Notice that the Yukawas transform under $\mathcal{G}_F$ like in the SM: $Y_u \in (\mathbf{3},\overline{\mathbf{3}},\mathbf{1})$ and $Y_d \in (\mathbf{3},\mathbf{1},\overline{\mathbf{3}})$.

In principle the alignment $\lambda_{u,d}^{ia} \propto \delta^{ia}$ may be justified through additional dynamical hypotheses. For instance, one simple extreme option is to assume that, rather than elementary, the fermions $u_R, d_R$ are massless composites from the strong sector, with $\mathcal{G}_{\text{strong}}$ quantum numbers $u_R \in (\mathbf{3},\mathbf{1})$ and $d_R \in (\mathbf{1},\mathbf{3})$. In that case, the flavor symmetry of the quark sector reduces to $U(3)_q \times \mathcal{G}_{\text{strong}}$ and the mixing between the elementary and composite sector reduces to $\lambda_{q_u}^{ia}\bar{q}_L^i\mathcal{O}_{q_u}^a + \lambda_{q_d}^{ia}\bar{q}_L^i\mathcal{O}_{q_d}^a$ with $\lambda_{q_u,q_d} \propto Y_{u,d}$. That situation implies, according to our rules, compositeness parameters $\varepsilon_u, \varepsilon_d$ of order unity. Nevertheless, as we emphasized at the end of Section 2.3, our aim here is not to explain the origin of the structures in Eq. (4.4), but rather to explore their phenomenological consequences. Because of that, we will treat $\varepsilon_u, \varepsilon_d$ as free parameters throughout our analysis.

Denoting by $y_i = \sqrt{2}m_i/v$ the eigenvalues of the SM Yukawas, the full set of $\lambda_\psi$'s in models with RU reads

$$\text{RU}: \quad \begin{cases} \lambda_{q_u} \sim \dfrac{1}{\varepsilon_u}\begin{pmatrix} y_u & 0 & 0 \\ 0 & y_c & 0 \\ 0 & 0 & y_t \end{pmatrix}, & \lambda_{q_d} \sim \dfrac{1}{\varepsilon_d}V_{\text{CKM}}\begin{pmatrix} y_d & 0 & 0 \\ 0 & y_s & 0 \\ 0 & 0 & y_b \end{pmatrix}, \\[20pt] \lambda_u \sim g_*\begin{pmatrix} \varepsilon_u & 0 & 0 \\ 0 & \varepsilon_u & 0 \\ 0 & 0 & \varepsilon_u \end{pmatrix}, & \lambda_d \sim g_*\begin{pmatrix} \varepsilon_d & 0 & 0 \\ 0 & \varepsilon_d & 0 \\ 0 & 0 & \varepsilon_d \end{pmatrix}. \end{cases} \tag{4.4}$$

where $\sim$ indicates that the equalities are up to overall $O(1)$ numbers. According to the discussion below Eq. (2.6) we will also demand

$$\frac{y_t}{g_*} \lesssim \varepsilon_u \lesssim 1 \qquad \frac{y_b}{g_*} \lesssim \varepsilon_d \lesssim 1. \tag{4.5}$$

In the RU scenario, we just described the only CP-odd parameter in the $\lambda_\psi$'s coincide with the CKM phase (we are ignoring topological angles, of course). It is the freedom in

performing field re-definitions granted by the symmetry group $\mathcal{G}_{\text{strong}} \times \mathcal{G}_{\text{quarks}}$ that allows us to reach this conclusion. Yet, in order to better appreciate the robustness of our analysis we would like to briefly assess what would happen if we had chosen the smaller symmetry $\mathcal{G}'_{\text{strong}} = SU(3)_U \times SU(3)_D \times U(1)_{U+D}$ instead of $\mathcal{G}_{\text{strong}}$. The non-abelian part of the symmetry group is the same and guarantees we can still realize minimal flavor violation precisely as described above. Similarly, $U(1)_{U+D}$ appears in both groups as it is essentially the baryon number, which is always assumed in our models (see Section 2.1). The only difference between $\mathcal{G}_{\text{strong}}$ and $\mathcal{G}'_{\text{strong}}$ is that the latter does not include a $U(1)_{U-D}$ under which the composite up-type fermions carry a charge opposite to the down-type composites. As a result, a RU model based on the symmetry $\mathcal{G}'_{\text{strong}} \times \mathcal{G}_{\text{quarks}}$ would possess an additional observable CP-odd phase. It is intuitively clear that the new phase can appear only if one can build $SU(3)$ invariants that are not $U(3)$ singlets, namely via combinations that involve the Levi-Civita tensor of $SU(3)$. This requirement in turn results either in Wilson coefficients with more insertions of $\lambda_{q_u,q_d}$ than the ones identified in a $U(3)$-invariant theory, or operators of dimensionality higher than six. In either case, the effect of the new phase is subleading. We thus expect scenarios based on the symmetry group $\mathcal{G}'_{\text{strong}}$ would lead to signatures indistinguishable from the scenarios with $\mathcal{G}_{\text{strong}}$, which we will discuss in detail. We also anticipate the same qualitative pattern should continue to hold if one were to replace the $U(n)$ flavor symmetries of all the models discussed in this paper with their $SU(n)$ subgroups. For this reason, we will solely focus on scenarios with unitary groups, as opposed to special-unitary symmetries. A systematic analysis of the implications of relaxing this additional hypothesis is beyond the scope of the present paper.

In the next subsection, we present an analysis of the main experimental constraints on the $\mathcal{G}_{\text{strong}}$ scenario. Previous work can be found in [43,45] where, because of the reason explained in the paragraph above (4.4), it was called "right-handed compositeness".

### 4.1.2  Experimental constraints

As anticipated in Section 3.2, we will now discuss the phenomenological constraints on 4-fermion operators, EW vertices and dipoles. Those on the bosonic operators, independent of the specific flavor assumptions, are collected in App. C.1.

Because flavor non-universality is controlled, in this model, by insertions of $\lambda_{q_u,q_d}/g_*$, and since $\varepsilon_{u,d} \equiv \lambda_{u,d}/g_*$ can a priori be sizable (see Eq. (4.5)), it is natural to organize our discussion according to an expansion in power of $\lambda_{q_u,q_d}/g_*$. In any case, multiple insertions of (the possibly unsuppressed) $\varepsilon_{u,d}$ do not alter the flavor or the CP structure of the Wilson coefficients.

### 4-fermion operators

We will be working at tree level, meaning that we will not consider effects involving virtual elementary fermions. We have checked that those effects always give subdominant constraints. At zeroth order in $\lambda_{q_u,q_d}/g_*$ we have two types of *flavor-universal* operators

$$\mathcal{O}_{uu} = (\bar{u}_R \gamma^\mu u_R)(\bar{u}_R \gamma_\mu u_R), \qquad \mathcal{O}_{dd} = (\bar{d}_R \gamma^\mu d_R)(\bar{d}_R \gamma_\mu d_R), \qquad (4.6)$$

with expected coefficients $\sim g_*^2 \varepsilon_u^4/m_*^2$ and $\sim g_*^2 \varepsilon_d^4/m_*^2$ respectively. Measures of the dijet angular distributions at LHC can constrain such operators. At present, CMS [49] and ATLAS

[50] released only early run 2 bounds on these coefficients,[12] which read

$$m_* \gtrsim (4.8 \div 7.8) g_* \varepsilon_u^2 \, \text{TeV} \,, \qquad m_* \gtrsim (3.0 \div 3.6) g_* \varepsilon_d^2 \, \text{TeV} \,, \qquad \text{LHC@37 fb}^{-1} \,, \qquad (4.7)$$

at 95% CL. As it is usual, the two different values for the constraint from each operator correspond to respectively constructive and destructive interference with the SM. As a reference, we also present the projected bounds for the LHC at 300 fb$^{-1}$ integrated luminosity [51]

$$m_* \gtrsim (8.2 \div 13) g_* \varepsilon_u^2 \, \text{TeV} \,, \qquad m_* \gtrsim (5.1 \div 5.8) g_* \varepsilon_d^2 \, \text{TeV} \,, \qquad \text{LHC@300 fb}^{-1}(\text{proj.}) \,. \qquad (4.8)$$

We see that, in order to allow for $m_* \sim$ TeV, the size of $\varepsilon_{u,d}$ must be reduced as much as possible. However, in so doing, $\lambda_{q_u,q_d} = Y_{u,d}/\varepsilon_{u,d}$ increases and flavor violation is enhanced, as we shall now discuss in more detail.

Consider the next operators involving powers of $\lambda_{q_u,q_d}$. The chiral structure of 4-quark operators requires even powers of the $\lambda_{q_u,q_d}$ (similarly, mass terms and dipole operators must involve odd powers of $\lambda_{q_u,q_d}$). The up sector turns out to be numerically less relevant and we focus, therefore, on the down sector. At second order in $\lambda_{q_u,q_d}$, one finds the following $\bar{q}_L q_L \bar{d}_R d_R$ operators

$$\frac{1}{g_*^2 m_*^2} \left[ c_u (\lambda_{q_u} \lambda_{q_u}^\dagger)^{ij} (\lambda_d \lambda_d^\dagger)^{lk} + c_d (\lambda_{q_d} \lambda_d^\dagger)^{ik} (\lambda_d \lambda_{q_d}^\dagger)^{lj} + c_d' (\lambda_{q_d} \lambda_{q_d}^\dagger)^{ij} (\lambda_d \lambda_d^\dagger)^{lk} \right] \, \bar{q}_L^i d_R^k \, \bar{d}_R^l q_L^j . \quad (4.9)$$

The only flavor-violating effects are mediated by $c_u$ and have $\Delta F = 1$. The remaining terms are diagonal in the mass basis. Analogous results apply for $\bar{q}_L q_L \bar{u}_R u_R$ operators where the only active source of flavor violation is $\lambda_{q_d} \lambda_{q_d}^\dagger$. As it turns out, however, purely hadronic $\Delta F = 1$ transitions are far less significantly constrained than semileptonic $\Delta F = 1$ or $\Delta F = 2$. We will thus neglect the constraints from these operators.

Consider now operators involving 4 powers of $\lambda_{q_u,q_d}$. At tree level the induced operators are $\mathcal{O}_{qq}^{(1)}$ and $\mathcal{O}_{qq}^{(3)}$ of Table 4. When focusing on the relevant $\Delta F = 2$ subset it is however more convenient to use the $\mathcal{O}_1^{f_i f_j}$ basis in Eq. (B.43). Its coefficient $C_1^{f_i f_j}$ is proportional to a linear combination of the three structures $(\lambda_{q_u} \lambda_{q_u}^\dagger)^2$, $(\lambda_{q_d} \lambda_{q_d}^\dagger)^2$ and $(\lambda_{q_u} \lambda_{q_u}^\dagger)(\lambda_{q_d} \lambda_{q_d}^\dagger)$. As $y_t^2 \gg y_b^2$ the largest effects are generated in the down quark sector (oscillations of the $K, B_d, B_s$ mesons) by the first structure. In the up sector that structure is diagonal while the others give weaker bounds. From Eq. (2.6), rotating to the mass basis, for the down sector we estimate

$$\begin{aligned}
C_1^{f_i f_j} &= (V_{\text{CKM}}^\dagger \lambda_{q_u} \lambda_{q_u}^\dagger V_{\text{CKM}})^{ij} (V_{\text{CKM}}^\dagger \lambda_{q_u} \lambda_{q_u}^\dagger V_{\text{CKM}})^{ij} \frac{c}{g_*^2 m_*^2} \\
&= (V_{\text{CKM}}^\dagger \widetilde{Y}_u \widetilde{Y}_u^\dagger V_{\text{CKM}})^{ij} (V_{\text{CKM}}^\dagger \widetilde{Y}_u \widetilde{Y}_u^\dagger V_{\text{CKM}})^{ij} \frac{c}{g_*^2 \varepsilon_u^4 m_*^2} \\
&\simeq c \frac{y_t^4}{g_*^2 \varepsilon_u^4 m_*^2} ([V_{\text{CKM}}^*]^{3i} V_{\text{CKM}}^{3j})^2 \,.
\end{aligned} \qquad (4.10)$$

By the current data [52], the most stringent constraint comes from $B_d$ oscillation and reads

$$m_* \gtrsim \frac{6.6}{g_* \varepsilon_u^2} \, \text{TeV} \,. \qquad (4.11)$$

Slightly weaker constraints are provided by $B_s, K$ oscillations and are reported for completeness in Table 13. Notice that, unlike Eq. (4.7), these constraints favor larger $\varepsilon_u$.

Finally, semi-leptonic $\Delta F = 1$ transitions in our setup are dominantly produced by modifications of the EW vertices and will be discussed next.

---

[12]See App. C.3 for details.

**EW vertices**

The next class of operators are the so-called EW vertices in Table 4, which after taking into account the Higgs VEV give rise to modifications of the $W, Z$ couplings to quarks. Here we observe the same trend as in 4-fermion interactions: the observables involving right-handed quarks favor smaller $\varepsilon_u, \varepsilon_d$ whereas those involving left-handed quarks prefer larger $\varepsilon_u, \varepsilon_d$.

At zeroth order in $\lambda_{q_u,q_d}$, the relevant operators are $\mathcal{O}_{Hu,Hd}$, involving right handed quarks. $\mathcal{O}_{Hu,Hd}$, however, violate custodial $SO(4)$ and their occurrence depends on the $SO(4)$ assignments of $\mathcal{O}_u$ and $\mathcal{O}_d$ (see A.1). In particular, when $\mathcal{O}_u$ and $\mathcal{O}_d$ are either in the $(\mathbf{1}, \mathbf{1})$ or $(\mathbf{1}, \mathbf{3})$ the $\mathcal{O}_{Hu,Hd}$ are not generated at tree-level. For $\mathcal{O}_{u,d} \in (\mathbf{1}, \mathbf{2})$, $\mathcal{O}_{Hu,Hd}$ are generated at $O(\varepsilon_u^2)$ and $O(\varepsilon_d^2)$. According to the analysis in [53], the resulting corrections to the $Z$ couplings imply the constraints[13] (see Table 9)

$$m_* \gtrsim (1.7 \div 2.1)\, g_* \varepsilon_u \ \text{TeV}\,, \qquad m_* \gtrsim (1.4 \div 1.6)\, g_* \varepsilon_d \ \text{TeV}\,. \tag{4.12}$$

Notice that in the case $\mathcal{O}_u \in (\mathbf{1}, \mathbf{2})$, loop corrections from top exchanges are also expected to generate a sizeable contribution to $\hat{T}$, as reviewed in App. C.2. Yet, Eq. (4.12) leads to stronger bounds unless $g_* \varepsilon_u \gtrsim 2 \div 3$.

At second order in the $\lambda_{q_u,q_d}$ we find the operators $\mathcal{O}_{Hq}^{(1,3)}$ with coefficients given by a linear combination of $\propto \lambda_{q_u} \lambda_{q_u}^\dagger / m_*^2$ and $\propto \lambda_{q_d} \lambda_{q_d}^\dagger / m_*^2$ (see Eq. (2.6)). Bounds on these operators come from both $\Delta F = 0$ and $\Delta F = 1$ processes, as we now report. A detailed analysis is provided in App. C.3.

The main $\Delta F = 0$ effect is a correction to the $Z$ coupling of $b_L$, on which LEP/SLC data give the very significant bound

$$m_* \gtrsim \frac{(2.2 \div 2.8)}{\varepsilon_u}, \frac{0.05}{\varepsilon_d} \ \text{TeV}\,. \tag{4.13}$$

The two terms in the previous expression are related to the fact that in this model we have two different partners for the SM doublets. Exchanges of up-type partners $\mathcal{O}_{q_u}$ give the leading term $(\delta g_b/g_b \sim y_t^2 v^2/(2\varepsilon_u^2 m_*^2))$, while the subleading is coming from $\mathcal{O}_{q_d}$ $(\delta g_b/g_b \sim y_b^2 v^2/(2\varepsilon_d^2 m_*^2))$. Indeed LHC now also provides a constraint on the $Z\bar{t}t$ coupling which gives a weaker but still interesting bound:

$$m_* \gtrsim \frac{0.9}{\varepsilon_u} \text{TeV}\,. \tag{4.14}$$

The bound in Eq. (4.13) is significant in view of the fact that $\varepsilon_u \lesssim 1$. As discussed in App. A.1 this class of anomalous $Z$-couplings can be further protected by enlarging the custodial group to $O(4) = SO(4) \rtimes P_{\text{LR}}$ and by assigning suitable quantum numbers to $\mathcal{O}_{q_u}$. The mechanism protects either couplings to $d_L^i$ or to $u_L^i$. Considering Eqs. (4.13) and (4.14) it is clearly preferable to protect the down type, which is in fact achieved by the choice $\mathcal{O}_{q_u} \in (\mathbf{2}, \mathbf{2})_{2/3}$ and $\mathcal{O}_u \in (\mathbf{1}, \mathbf{1})_{2/3}$ under $SO(4) \times U(1)_X$.

Assuming that is the case the main bound in Eq. (4.13) is eliminated leaving only the one due to the exchange of the $\mathcal{O}_{q_d}$ partners.[14] In that case, the leading correction to the $Z$

---

[13]Given the highly asymmetric experimental interval, we take as reference the largest boundary and half of the interval. We always make this choice in the rest of the paper unless otherwise stated (see the discussion on App. C in Section 3.2).

[14]In principle, as discussed in App. A.1, also the subdominant bound in Eq. (4.13) can be eliminated assuming $\mathcal{O}_{q_d} \in (\mathbf{2}, \mathbf{2})_{2/3}$. However, given that the constraint is quite mild, we do not discuss this possibility further.

coupling to $b_L$ comes instead from the operators

$$[\mathcal{O}_{qD}^{(1)}]^{ij} \equiv \bar{q}_L^i \gamma^\mu q_L^j \; \partial^\nu B_{\nu\mu} \,, \qquad\qquad [\mathcal{O}_{qD}^{(3)}]^{ij} \equiv \bar{q}_L^i \sigma^a \gamma^\mu q_L^j \; (D^\nu W_{\nu\mu})^a \,, \qquad (4.15)$$

whose coefficients are $\sim g^{(')}\lambda_{q_u}\lambda_{q_u}^\dagger/(g_*^2 m_*^2)$. The correction to the $Z$ coupling is then $\delta g_b/g_b \sim \cos\theta_W(y_t^2 m_Z^2)/(\varepsilon_u^2 g_*^2 m_*^2)$ corresponding to a $\sim g^2/(2\cos\theta_W g_*^2)$ reduction. The bound is

$$m_* \gtrsim \frac{(1.1 \div 1.3)}{g_* \varepsilon_u} \; \text{TeV} \,. \qquad (4.16)$$

Notice that, since $\varepsilon_u \lesssim 1$, this bound is always weaker than that in Eq. (4.11). It is also weaker than that in Eq. (4.14) unless $1 \lesssim g_* \lesssim 2$.

Let us now consider $\Delta F = 1$ effects. Here $B$ decays into leptons play the main role with Kaon decays subdominant, as discussed in Sec. C.4.3. The transition $b \to s\ell^+\ell^-$ provides the strongest constraint. The relevant effective Hamiltonian is described in App. B.3 and consists of a linear combination of four operators $\mathcal{O}_{9/10}$, $\mathcal{O}_{9/10}'$ (see Eq. (B.33)). As the operators $\mathcal{O}_{Hq}^{(1,3)}$ involve only left quarks $C_{10}'$ is however not generated. Moreover the Wilson coefficients $C_9$ and $C_9'$ are suppressed by a factor $(1 - 4\sin^2\theta_w) \sim 0.08$ with respect to $C_{10}$ and can be ignored. Focusing on the only remaining $\mathcal{O}_{10}$, in the mass basis we have

$$\begin{aligned} C_{10} &= \frac{1}{2} c_u^{(1,3)} \frac{(V_{\text{CKM}}^\dagger \lambda_{q_u} \lambda_{q_u}^\dagger V_{\text{CKM}})^{23}}{m_*^2} \frac{\sqrt{2}}{4 G_F} \frac{16\pi^2}{V_{\text{CKM}}^{tb}(V_{\text{CKM}}^{ts})^* e^2} \\ &\simeq \frac{1}{2} c_u^{(1,3)} \frac{4\sqrt{2}\pi^2}{G_F m_*^2} \frac{y_t^2}{e^2 \varepsilon_u^2} \,. \end{aligned} \qquad (4.17)$$

The constraint on $C_{10}$ is dominated by the branching ratio for $B_s \to \mu^+\mu^-$. As shown in [54] the data for this process favor a non-zero, yet small $C_{10}$, at roughly one standard deviation. As explained at the beginning of this section, we then considered two different hypotheses, one more and one less conservative, which translate into a weaker and a stronger bound [54]

$$m_* \gtrsim \frac{6.5 \div 8.3}{\varepsilon_u} \text{TeV} \,. \qquad (4.18)$$

Notice that these bounds are stronger than those from the $Z\bar{b}b$ vertex in Eq. (4.13). Again, these bounds can be relaxed by implementing the $P_{\text{LR}}$ protection mechanism which can suppress all the $Z\bar{d}_L^i d_L^{\,j}$ vertex corrections, whether or not flavor diagonal. Thus with the charge assignment $\mathcal{O}_{q_u} \in (\mathbf{2}, \mathbf{2})_{2/3}$ of $O(4) \times U(1)_X$ the main semi-leptonic $\Delta F = 1$ effects are also controlled by the operators in Eq. (4.15) and mediated by $Z$ boson and photon exchange. As shown in App. B.3, the interactions mediated by photons dominate and generate a value for $C_9$ of the order of $C_{10}$ in Eq. (4.17) times a suppression factor $\sim e^2/g_*^2$. $C_9$ is constrained by the various $b \to s\ell^+\ell^-$ transitions, including the theoretically more uncertain angular distributions. To estimate the bound we rely again on the analysis of [54], which follows two approaches, a less restrictive "data-driven" one and a more restrictive "model dependent" one. The resulting bounds on $m_*$ then read

$$m_* \gtrsim \frac{1.2 \div 3.5}{g_* \varepsilon_u} \text{TeV} \,. \qquad (4.19)$$

Like for Eq. (4.16), these are always weaker than Eq. (4.11), but for $g_* \lesssim 4$ they can be stronger than Eq. (4.14).

### Dipoles

Dipole operators (see Table 4) first arise at linear order in $\lambda_{q_u,q_d}/g_*$ with their coefficients exactly aligned with the SM fermion mass matrix. These are then real-diagonal and do not contribute to either flavor violation or EDMs. Flavor violation only arises at the next order, the cubic, in the $\lambda_{q_u,q_d}/g_*$. The additional insertions of $\lambda_{q_u,q_d}/g_*$ can originate in two ways. The first way is through $q_L$ wave-function renormalization. These, according to the rules of our EFT construction, come as a linear combination of $\lambda_{q_u}\lambda_{q_u}^\dagger/g_*^2$ and $\lambda_{q_d}\lambda_{q_d}^\dagger/g_*^2$ while the combination $\lambda_{q_u}\lambda_{q_d}^\dagger$ is forbidden by $\mathcal{G}_{\text{strong}}$. Yet, such wavefunction contributions do not lead to flavor violation because precisely the same correction affects the Yukawa coupling and the dipole. The second way is through loops involving virtual elementary quarks $q_L^i$'s, in which case the result can depend on both $\lambda_{q_u,q_d}^\dagger\lambda_{q_u,q_d}/16\pi^2$ and $\lambda_{q_u}^\dagger\lambda_{q_d}/16\pi^2$. The best constrained dipole-mediated $\Delta F = 1$ processes involve the down-type quarks. Thus focusing on $d^i \to d^{j\neq i}$, and indicating the different SM vector bosons $V = B, W, G$ in an obvious notation, the Wilson coefficients of dipole operators have a generic structure

$$
\begin{aligned}
[\mathcal{C}_{dV}]^{ij} &= \frac{g_V}{g_* 16\pi^2}\frac{1}{m_*^2}\left[(c_{1V}\lambda_{q_u}\lambda_{q_u}^\dagger\lambda_{q_d} + c_{2V}\lambda_{q_d}\lambda_{q_d}^\dagger\lambda_{q_d})\lambda_d^\dagger\right]^{ij} &&(4.20)\\
&\to c_{1V}g_V[V_{\text{CKM}}^\dagger\widetilde{Y}_u^2 V_{\text{CKM}}]^{ij}\frac{\widetilde{Y}_d^{jj}}{16\pi^2\varepsilon_u^2 m_*^2}\,,
\end{aligned}
$$

with $c_{1V}, c_{2V} = O(1)$. Here in the second line, we performed a rotation to the mass basis. The term controlled by $c_{2V}$ is diagonal after rotation and thus drops for $j \neq i$. To constrain our parameters we took into account the RG running from 3 TeV to $m_b$, and then applied the bound on the Wilson coefficient $C_7 \simeq c_{1V}\sqrt{2}y_t^2/(4G_F\varepsilon_u^2 m_*^2)$ (see App. B.3 for the definition) derived in Ref. [55] from the data on the inclusive decay $B \to X_s\gamma$. A more detailed explanation is offered in App. C.3. The result is

$$
m_* \gtrsim \frac{0.45 \div 0.68}{\varepsilon_u}\text{TeV}\,, \tag{4.21}
$$

which is weaker than Eq. (4.14).

Dipole operators also affect the neutron EDM, but in a very suppressed way. Consider the quark EDMs, or similarly the quark chromo-EDMs. These require at least additional 8 powers of $\lambda_{q_u,q_d}$ beyond the leading order. By a superficial analysis, we found the leading contribution to the electric dipole matrix of down quarks in Eq. (B.41) arises at two loops and scales like

$$
\begin{aligned}
\text{Im}[\mathcal{C}_{d\gamma}] &\sim \frac{e}{m_*^2}\left(\frac{g_*^2}{16\pi^2}\right)^2\text{Tr}\left[x_u x_d x_u^2\frac{\lambda_{q_d}\lambda_d^\dagger}{g_*}\right] - \text{h.c.} &&(4.22)\\
&\sim \frac{e}{m_*^2}\left(\frac{g_*^2}{16\pi^2}\right)^2\text{Tr}\left[\frac{(Y_u Y_u^\dagger)}{g_*^2\varepsilon_u^2}\frac{(Y_d Y_d^\dagger)}{g_*^2\varepsilon_d^2}\frac{(Y_u Y_u^\dagger)^2}{g_*^4\varepsilon_u^4}Y_d\right] - \text{h.c.}\,,
\end{aligned}
$$

where $x_{u,d} = \lambda_{q_u,q_d}\lambda_{q_u,q_d}^\dagger/g_*^2$. The $d$ quark entry, expressed in terms of the Yukawa eigenvalues, reads $[\widetilde{\mathcal{C}}_{d\gamma}]^{11} \sim (e/m_*^2)\left(g_*^2/16\pi^2\right)^2 y_t^4 y_c^2 y_b^2/(g_*^8\varepsilon_u^6\varepsilon_d^2)J_{\text{CP}}\bar{Y}_d^{11}$, with $J_{\text{CP}} \sim \lambda_C^6$ the SM Jarlskog invariant.[15] The resulting bound on $m_*$ is negligible even for the smallest allowed values

---

[15]This result is consistent with the MFV analysis performed in [56].

$\varepsilon_u \sim y_t/g_*$, $\varepsilon_d \sim y_b/g_*$. Similar considerations apply to the dipoles of the up-quarks.[16] Furthermore, contributions to the coefficient of the 3-gluon operator of Eq. (2.13) are utterly small because the construction of a CP-odd flavor-invariant combination of $\lambda_{q_u,q_d}$'s requires a large number of spurions. Below the weak scale, also operators with four fermions contribute to the neutron EDM, but these require a loop of light fermions and their effect is thus effectively more suppressed than quark dipoles. We conclude that models with RU structurally evade the stringent constraints from the neutron EDM reviewed in Sections 2.3.

### 4.1.3 Summary

While all bounds are obviously better satisfied by large $m_*$, we found that processes involving right and left handed fermions favor respectively a smaller and a larger amount of composite-ness $\varepsilon_{u,d}$. More precisely, the compositeness bounds in Eq. (4.7) (Eq. (4.8)) push towards small $\varepsilon_u$ whereas $\Delta F = 2$ transitions Eq. (4.11) prefer maximal $\varepsilon_u$. Combining these constraints we find the following lower bound on $m_*$

$$m_* \gtrsim 5.6 \div 7.2 \,\text{TeV} \,(7.4 \div 9.2 \,\text{TeV}) \,, \tag{4.24}$$

obtained for an optimal choice of $\varepsilon_u$: $\varepsilon_u^{\text{opt}} \sim 1/\sqrt{g_*}$ for present constraints. The number in parenthesis corresponds to the projected bounds for the end of run-3 of the LHC. As mentioned above, because we expect the operators in Eq. (4.6) to receive sizable effects from the exchange of spin-1 resonances, we assume the stronger bound applies. We emphasize that Eq. (4.24) is independent of the choice for the $SO(4) \times U(1)_X$ representations for the composite fermions and so of all the discussion in App. A.1.

Further constraints arise from anomalous $Z$ couplings to the left-handed quarks, in particular from semileptonic B-decays in Eq. (4.18). Yet, this bound is more model dependent, meaning that it can be avoided by suited representations for the composite quarks, as by the choice in Eq. (A.2). For generic $SO(4) \times U(1)_X$ representations, however, Eq. (4.18) applies. Combining the latter with the compositiness bounds of Eq. (4.7) (Eq. (4.8)) we get

$$m_* > 5.9 \div 8.1 \,(7 \div 9.6) \, g_*^{\frac{1}{3}} \,\text{TeV}, \tag{4.25}$$

obtained for $\varepsilon_u \sim 1/g_*^{1/3}$, which is even stronger than Eq. (4.24) for large $g_*$.

A similar trend is seen in the parameter $\varepsilon_d$, though this is far less significant. The parameter $\varepsilon_d$ is required to be small by Eq. (4.7) but cannot be too small otherwise the subdominant $Zb_Lb_L$ bound on the right in Eq. (4.13) becomes relevant. Still, in the rather vast range

---

[16]To prove the claim above, note that the coefficient $\mathcal{C}_{d\gamma}^{ij}$ is formally a spurion with exactly the same flavor quantum numbers of the Yukawa matrix $Y_d^{ij}$. We are interested in asking under which conditions the three diagonal elements of $\widetilde{\mathcal{C}}_{d\gamma}$ have imaginary parts in the mass basis, i.e. in the field basis in which $Y_d$ is diagonal. The phases of the dipole moments, being physical observables, are associated with three independent flavor-invariant combinations of the spurions $\lambda_{q_u,q_d,u,d}$. The simplest choice for these flavor invariants is represented by

$$\text{Tr}[\mathcal{C}_{d\gamma}Y_d^\dagger], \qquad \text{Tr}[\mathcal{C}_{d\gamma}(Y_d^\dagger Y_d)Y_d^\dagger], \qquad \text{Tr}[\mathcal{C}_{d\gamma}(Y_d^\dagger Y_d)^2 Y_d^\dagger]. \tag{4.23}$$

Indeed, as long as the eigenvalues of $Y_d$ are all non-degenerate, the three diagonal elements of $\mathcal{C}_{d\gamma}$ in the mass basis can be expressed as combinations of the quantities shown in (4.23). If all those invariants are real, i.e. CP-even, there cannot be any dipole moment. Conversely, one can estimate the order at which an EDM can appear by exploring which structures $\mathcal{C}_{d\gamma}$ support CP-odd invariants.

$0.02 \lesssim \varepsilon_d^{\mathrm{opt}} \lesssim 0.5$ both bounds are subdominant compared to those from the bosonic opera-
tors in Eq. (C.4).

In conclusion, the strongest and most robust constraint on this model is Eq. (4.24). This
significantly exceeds the constraints from the bosonic operators, as shown in the summary
plot in the left panel of Fig. 5.

## 4.2   Partial Up-Right Universality (puRU)

### 4.2.1   General structure

The main problem of RU is that a unique parameter controls the partial compositeness of
$u_R$ and $c_R$ (which are subject to an upper bound from contact interactions) and that of $t_R$
(which is subject to a lower bound from flavor violation). This tension can be alleviated in
a scenario with a smaller symmetry in the strong sector which allows $t_R$ to have a mixing
strength different than that of $u_R$ and $c_R$. This defines the scenario of Partial Up-Right
Universality (puRU).

The model is obtained by replacing $\mathcal{G}_{\mathrm{strong}} = U(3)_U \times U(3)_D$ with[17]

$$\mathcal{G}_{\mathrm{strong}} = U(2)_U \times U(1)_U \times U(3)_D \,, \tag{4.26}$$

and by assigning the three families of composite fermions to the representations

$$\mathcal{O}_{q_u,u} \in (\mathbf{2} \oplus \mathbf{1}_U, \mathbf{1}) \,, \qquad \mathcal{O}_{q_d,d} \in (\mathbf{1}, \mathbf{3}) \,. \tag{4.27}$$

Here $\mathbf{1}$ denotes a complete singlet, while the subscript $_U$ indicates the presence of the $U(1)_U$
charge. The $U(1)_U$, besides distinguishing the third family of up-type composites (which
partners the top quark), is necessary to guarantee baryon number conservation. According
to Eq. (4.27) the spurions of Eq. (3.1) transform under $\mathcal{G}_{\mathrm{strong}} \times \mathcal{G}_{\mathrm{quarks}}$ as (with obvious
notation)

$$\lambda_{q_u} \equiv \lambda_{q_u}^{(2)} \oplus \lambda_{q_u}^{(1)} \in (\overline{\mathbf{2}} \oplus \overline{\mathbf{1}}_{\mathbf{U}}, \mathbf{1}, \mathbf{3}, \mathbf{1}, \mathbf{1}) \,, \qquad \lambda_{q_d} \in (\mathbf{1}, \overline{\mathbf{3}}, \mathbf{3}, \mathbf{1}, \mathbf{1}) \,,$$
$$\lambda_u \equiv \lambda_u^{(2)} \oplus \lambda_u^{(1)} \in (\overline{\mathbf{2}} \oplus \overline{\mathbf{1}}_U, \mathbf{1}, \mathbf{1}, \mathbf{3}, \mathbf{1}) \,, \qquad \lambda_d \in (\mathbf{1}, \overline{\mathbf{3}}, \mathbf{1}, \mathbf{1}, \mathbf{3}) \,. \tag{4.28}$$

The final defining ingredient of Partial Up-Right Universality is the assumption that the
couplings $\lambda_{u,d}$ in Eq. (3.1) respect

$$\mathcal{G}_F = U(3)_q \times U(2)_{U+u} \times U(1)_{U+u} \times U(3)_{D+d} \subset \mathcal{G}_{\mathrm{strong}} \times \mathcal{G}_{\mathrm{quarks}} \,. \tag{4.29}$$

This assumption and the reproduction of the SM Yukawa couplings imply that by suitable
field redefinitions, the mixings can be put in the form

$$\mathrm{puRU}: \begin{cases} \lambda_{q_u} \sim \frac{1}{\varepsilon_u} \begin{pmatrix} y_u & 0 \\ 0 & y_c \\ ay_c & by_c \end{pmatrix} \oplus \frac{1}{\varepsilon_{u_3}} \begin{pmatrix} 0 \\ 0 \\ y_t \end{pmatrix} \,, \quad \lambda_{q_d} \sim U_d \frac{1}{\varepsilon_d} \begin{pmatrix} y_d & 0 & 0 \\ 0 & y_s & 0 \\ 0 & 0 & y_b \end{pmatrix} \,, \\[3ex] \lambda_u \sim g_* \begin{pmatrix} \varepsilon_u & 0 \\ 0 & \varepsilon_u \\ 0 & 0 \end{pmatrix} \oplus g_* \begin{pmatrix} 0 \\ 0 \\ \varepsilon_{u_3} \end{pmatrix} \,, \qquad \lambda_d \sim g_* \begin{pmatrix} \varepsilon_d & 0 & 0 \\ 0 & \varepsilon_d & 0 \\ 0 & 0 & \varepsilon_d \end{pmatrix} \,. \end{cases} \tag{4.30}$$

---

[17]As emphasized at the end of Sections 4.1, we will not discuss the alternative formulation with the strong
flavor symmetry replaced by $\mathcal{G}'_{\mathrm{strong}} = SU(2)_U \times SU(3)_D \times U(1)_{U+D}$.

The decomposition of $\lambda_{q_u}$ and $\lambda_u$ into the direct sum of a $2 \times 3$ and a $1 \times 3$ matrix corresponds to the quantum number assignments in Eq. (4.28). The structure of $\lambda_u$ realizes the breaking pattern $U(2)_U \times U(1)_U \times U(3)_u \to U(2)_{U+u} \times U(1)_{U+u}$. In particular, the $1 \times 3$ spurion $\lambda_u^{(1)}$ provides the exclusive mixing between the right handed top $u_R^3$ and the third family composite $\mathcal{O}_u^3$. The corresponding mixing parameter $\varepsilon_{u_3}$ crucially differs from that of the first two families, $\varepsilon_u$. The structure in the down sector is the same as in RU, apart from the fact that the residual $U_d$ rotation does not precisely coincide with the CKM matrix. That is because, as shown in Eq. (4.30), $U(3)_q \times U(2)_U \times U(1)_U$ rotations are now not enough to put the up sector mixing $\lambda_{q_u}$ in diagonal form. Indeed by an $U(3)_q$ rotation, we can always orient $\lambda_{q_u}^{(1)}$ along the third family, while the residual $U(2)_q \times U(2)_U$ only allows to diagonalize the upper $2 \times 2$ block of $\lambda_{q_u}^{(2)}$, as shown in Eq. (4.30). The diagonal entries of this block control the up and charm mass, while the residual third row, parametrized by $a, b$, mostly affects mixing angles. Our baseline assumption is that $a, b$ are (complex) numbers not larger than order unity. This corresponds to assuming the entries of $\lambda_{q_u}^{(2)}$ are all $\lesssim y_c/\varepsilon_u$, which does not seem implausible. In any case, the conclusions do not drastically depend on this assumption. In the end $\lambda_{q_u}^{(1)}$ has larger entries than $\lambda_{q_u}^{(2)}$, which helps account the large size of the top Yukawa and the small CKM mixing between the light families and the third. The resulting SM up-type Yukawa coupling is finally the sum of two independent pieces[18]

$$Y_u = \frac{1}{g_*}\lambda_{q_u}^{(2)}[\lambda_u^{(2)}]^\dagger + \frac{1}{g_*}\lambda_{q_u}^{(1)}[\lambda_u^{(1)}]^\dagger = \begin{pmatrix} y_u & 0 & 0 \\ 0 & y_c & 0 \\ ay_c & by_c & y_t \end{pmatrix}. \tag{4.31}$$

As already mentioned, the down sector is the same as in RU, implying $Y_d \sim \lambda_{q_d}\lambda_d^\dagger/g_*$. As usual, the Yukawa matrices can be diagonalized via bi-unitary transformations, i.e. $Y_u = U_u\widetilde{Y}_uV_u^\dagger$ and $Y_d = U_d\widetilde{Y}_dV_d^\dagger$ with $\widetilde{Y}_u \sim \mathrm{diag}(y_u, y_c, y_t)$ and $\widetilde{Y}_d \sim \mathrm{diag}(y_d, y_s, y_b)$. With the parametrization adopted in Eq. (4.30), we have $V_d = \mathbf{1}$ and

$$V_{\mathrm{CKM}} = U_u^\dagger U_d. \tag{4.32}$$

The two matrices $U_u$ and $V_u$ can be computed analytically in the limit $y_u \ll y_c \ll y_t$ (see Appendix B.2). Since $U_u \approx \mathbf{1}$ up to corrections of order $y_c^2/y_t^2 = O(10^{-5})$, we find that $V_{\mathrm{CKM}} \approx U_d$ to a very good approximation.

Besides the *real* coefficients $c_{ab\cdots} = O(1)$ arising from Eq. (2.6), the scenario of Partial Up-RU is described by five additional real parameters

$$\frac{y_c}{g_*} \lesssim \varepsilon_u \lesssim 1, \quad \frac{y_t}{g_*} \lesssim \varepsilon_{u_3} \lesssim 1, \quad \frac{y_b}{g_*} \lesssim \varepsilon_d \lesssim 1, \quad |a| \sim 1, \quad |b| \sim 1. \tag{4.33}$$

As concerns phases, besides $\arg[a]$ and $\arg[b]$, we can remove all but one phase in $U_d$, so that the model features a total of three physical phases. In the mass basis, one combination defines the CKM phase via Eq. (4.32) whereas the other two are hidden in $U_u$ and $V_u$ and appear only in higher-dimensional operators.

Like in the RU scenario, the key symmetry hypothesis (4.29) could be given the dynamical interpretation that $u_R^i$ and $d_R^i$ are chiral composite states of the strong dynamics transforming

---

[18]We implicitly rescaled the $y_i$'s so that the order one numbers that generically appear in the following expression become exactly 1. This way the $y_i$'s in (4.30) are indeed the eigenvalues of the Yukawas up to small $y_i/y_{j>i}$ corrections.

respectively as $\mathbf{2} \oplus \mathbf{1}_U$ of $U(2)_U \times U(1)_U$ and $\mathbf{3}$ of $U(3)_D$. Of course that would be plausible as long as $\varepsilon_u, \varepsilon_{u_3}, \varepsilon_d$ are all $O(1)$.

In the next subsection, we will present an analysis of the experimental constraints on models with Partial Up-Right Universality. A first more qualitative study already appeared a decade ago in [57]. Our analysis, besides being based on the latest data, is more in depth. In particular, we shall demonstrate that the presence of the two extra phases $\arg[a]$ and $\arg[b]$ does not introduce sizable EDMs. Subsequently, we will discuss the other constraints, using the detailed study of Sec. 4.1.2 as a reference.

### 4.2.2 Suppression of the EDMs

As also reviewed in Sec. 2.3, the non-observation of an EDM for the neutron implies very strong constraints on generic CP violating new physics. While MFV structurally evades the constraints (see Sec. 4.1.2), it is interesting to see what happens with a weaker flavor assumption like Partial Up-Right Universality. This subsection is devoted to that.

In puRU, and actually, in all the models we consider in this paper, the dominant contributions to the neutron EDM are induced by the quark electric dipole moments and, at comparable order, by the chromo-electric quark dipole operators. The effect of four-fermion operators is suppressed by insertions of the light quarks' masses whereas the pure gluon operator in (2.12) has a coefficient that contains a large number of mixing parameters, and is hence negligible. We therefore focus on the effect of the quark electric dipole moment, $d_f^{ii} = \sqrt{2}\, v \, \mathrm{Im}[\widetilde{\mathcal{C}}_{f\gamma}]^{ii}$, which we recall is controlled by the imaginary part of the coefficient $\widetilde{\mathcal{C}}_{f\gamma}$ defined in Eq. (B.41) (see also Tab. 4 and Eq. (B.42)). Completely analogous considerations apply to the chromo-electric dipole interactions, controlled by the imaginary part of $\widetilde{\mathcal{C}}_{fG}$ again displayed in Tab. 4.

Like for the RU scenario, we organize our analysis of $\mathcal{C}_{f\gamma}$ as an expansion in $\lambda_{q_u,q_d}$, given these control the sources of flavor and CP violation. In the down sector of models with Partial Up-RU, as in scenarios of MFV, the coefficients $\mathcal{C}_{f\gamma}$ are obviously aligned with the SM Yukawas at leading order in an expansion in $\lambda_{q_d}$ and are, therefore, real and diagonal in the mass basis. The up sector deserves a separate discussion, though. At leading order, the spurion structure of the Wilson coefficients is the same as in Eq. (4.31) except for the relative size of the two contributions. Generically, we have

$$\mathcal{C}_{u\gamma} \propto \lambda_{q_u}^{(2)}[\lambda_u^{(2)}]^\dagger + r_\gamma \lambda_{q_u}^{(1)}[\lambda_u^{(1)}]^\dagger = \begin{pmatrix} y_u & 0 & 0 \\ 0 & y_c & 0 \\ a y_c & b y_c & r_\gamma y_t \end{pmatrix}, \tag{4.34}$$

where $r_\gamma$ is an $O(1)$ real coefficient parametrizing the relative size of the contribution of the two spurion structures as they enter in the dipole operators, having fixed to 1 the ratio of their contribution to the Yukawa coupling in Eq. (4.31). In full analogy, in the chromo-electric dipole a similar coefficient $r_G$ controls the relative size of the two spurion structures. Now, $r_\gamma - 1$ (and $r_G - 1$) measure the misalignment with the up-type Yukawa matrix in Eq. (4.31). Yet, independently from the values of $r_\gamma$, no quark electric dipole moments are generated at this order because the coefficients in Eq. (4.34), once rotated in the mass basis, have real entries. That is because one can remove all the phases from the parameters $a, b$ of (4.30) by performing a rotation of the first two generations of fundamental quarks

$$u_R^1 \to u_R^1\, e^{-i\arg[a]}\,, \quad u_R^2 \to u_R^2\, e^{-i\arg[b]}\,, \quad q_L^1 \to q_L^1\, e^{-i\arg[a]}\,, \qquad q_L^2 \to q_L^2\, e^{-i\arg[b]}\,, \tag{4.35}$$

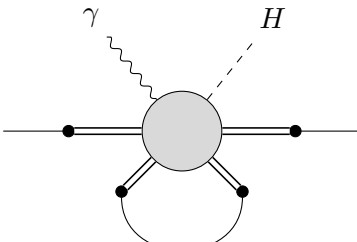

Figure 3: Representative 1-loop contribution to the neutron EDM in puRU. The double lines represent fermionic states of the composite dynamics, the solid single lines denote elementary fermions, the dashed line the Higgs, and the wavy line a photon.

and composite fermions

$$\mathcal{O}_u^1 \to \mathcal{O}_u^1 \, e^{-i\arg[a]}, \quad \mathcal{O}_u^2 \to \mathcal{O}_u^2 \, e^{-i\arg[b]}, \quad \mathcal{O}_{q_u}^1 \to \mathcal{O}_{q_u}^1 \, e^{-i\arg[a]}, \quad \mathcal{O}_{q_u}^2 \to \mathcal{O}_{q_u}^2 \, e^{-i\arg[b]}. \quad (4.36)$$

Once this re-definition is performed both $Y_u$ and $\mathcal{C}_{u\gamma;uG}$ are real. The mass basis is then reached via a real orthogonal transformation that does not introduce imaginary parts in Eq. (4.34). Thus up-type EDM is induced at this order. The above field re-definitions have obviously moved the two extra phases of the model to the down sector. But we have already argued that no dipole can be generated in the down sector because of MFV, so those phases are unphysical at this order.[19]

Therefore, the only way to generate a quark EDM, or more generally to induce a physical CP-violating phase, is to include the effects of *both* $\lambda_{q_d}$ and $\lambda_{q_u}$. The first effects occur at 1-loop and are controlled by $\lambda_{q_u,q_d}^\dagger \lambda_{q_u,q_d}/16\pi^2$ (see Eq. (2.6)). A representative Feynman diagram is shown in Fig. 3.

In the up-sector, the leading (1-loop) contribution to the imaginary part of the Wilson coefficient $\widetilde{\mathcal{C}}_{u\gamma}$ in the mass basis takes the form

$$
\begin{aligned}
\mathrm{Im}[\widetilde{\mathcal{C}}_{u\gamma}]^{ij} &= c_{u\gamma} \frac{1}{m_*^2} \frac{e}{16\pi^2 g_*} \mathrm{Im}\left[ U_u^\dagger \lambda_{q_d} \lambda_{q_d}^\dagger \left( \lambda_{q_u}^{(2)} [\lambda_u^{(2)}]^\dagger + r_\gamma \lambda_{q_u}^{(1)} [\lambda_u^{(1)}]^\dagger \right) V_u \right]^{ij} \quad (4.37) \\
&\sim c_{u\gamma} \frac{1}{m_*^2} \frac{e}{16\pi^2} \mathrm{Im}\left[ V_{\mathrm{CKM}} \frac{\widetilde{Y}_d^2}{\varepsilon_d^2} V_{\mathrm{CKM}}^\dagger \left( \widetilde{Y}_u + y_t(r_\gamma - 1) U_u^\dagger \widetilde{P} V_u \right) \right]^{ij}.
\end{aligned}
$$

where $c_{u\gamma}$ is an $O(1)$ real coefficient from Eq. (2.6). The matrices $U_u, V_u$ were defined above Eq. (4.32), and we introduced the projector onto the third family

$$\widetilde{P} = \frac{1}{g_* y_t} \lambda_{q_u}^{(1)} [\lambda_u^{(1)}]^\dagger = \begin{pmatrix} 0 & 0 & 0 \\ 0 & 0 & 0 \\ 0 & 0 & 1 \end{pmatrix}. \quad (4.38)$$

To better follow the steps that lead to the second line of (4.37), the reader will find in Appendix B a discussion and a compendium of the relevant flavor structures in the mass basis.

---

[19]The possibility to eliminate the two phases in the up sector corresponds to the fact, which we checked, that all flavor invariants built out of $\lambda_{q_u}$ and $\lambda_u$ are real.

The first term in brackets in Eq. (4.37) does not contribute to the EDMs because it has real diagonal elements. The $u^i$ EDM purely comes from the second term and reads ($v$ is the Higgs VEV)

$$d_u^{ii} = \sqrt{2}v\,\text{Im}[\widetilde{\mathcal{C}}_{u\gamma}]^{ii} \sim 2c_{u\gamma}(r_\gamma - 1)\frac{m_t}{m_*^2}\frac{e}{16\pi^2}\frac{y_b^2}{\varepsilon_d^2}\text{Im}\left[[V_{\text{CKM}}]^{i3}[V_{\text{CKM}}^*]^{33}[U_u^*]^{33}[V_u]^{3i}\right]. \quad (4.39)$$

The $i = 1, 2$ components are suppressed both by the CKM entries and by $[V_u]^{31,32} \sim y_c/y_t$ (see Appendix B.2). Applying the standard estimate $d_n \sim d_u^{11}$ for the neutron EDM, see also Eq. (B.50), and making the generic assumption $c_{u\gamma}(r_\gamma - 1) \sim 1$ one then has

$$d_n \sim e\frac{y_b^2}{8\pi^2}\frac{\text{Re}[a]A\eta\lambda_C^3}{m_*^2\varepsilon_d^2}m_c \implies m_* > \frac{0.06}{\varepsilon_d}\,\text{TeV}. \quad (4.40)$$

This constraint is comparable to precision electroweak physics (shown in parentheses in the first row of Tab. 9) but still worth noting. The bound becomes very significant only for $\varepsilon_d$ approaching its smallest allowed value in Eq. (4.33), when it reads $m_* \gtrsim 4.6\,g_*$ TeV. In order to have $m_*$ as low as the bound from the $\hat{S}$ parameter in Eq. (C.4), it is therefore mandatory to remain within the range $0.03 \lesssim \varepsilon_d \lesssim 1$. Similar considerations and bounds arise when considering the chromo-electric dipole moments. The contribution of the EDMs of the charm and top quarks are instead negligible, as claimed in Appendix B.4.

Consider now the down sector EDMs. Based on the previous discussion, the only contributions with a chance to provide an imaginary coefficient must involve mixings from both the up and down sectors. At the 1-loop level the only such terms are

$$\begin{aligned}[\mathcal{C}_{d\gamma}]^{ij} &= c_{d\gamma}\frac{1}{m_*^2}\frac{1}{16\pi^2 g_*} \qquad\qquad\qquad\qquad\qquad (4.41)\\ &\times \left[\left(\lambda_{q_u}^{(2)}[\lambda_{q_u}^{(2)}]^\dagger + r_\gamma'\lambda_{q_u}^{(1)}[\lambda_{q_u}^{(1)}]^\dagger\right)\lambda_{q_d}\lambda_d^\dagger\right]^{ij}.\end{aligned}$$

which in the mass basis become (see Eq. (4.32) and Table 5)

$$\begin{aligned}&U_d^\dagger\left(\lambda_{q_u}^{(2)}[\lambda_{q_u}^{(2)}]^\dagger + r_\gamma'\lambda_{q_u}^{(1)}[\lambda_{q_u}^{(1)}]^\dagger\right)\frac{\lambda_{q_d}\lambda_d^\dagger}{g_*}V_d \qquad\qquad (4.42)\\ &= V_{\text{CKM}}^\dagger\left[\frac{\widetilde{Y}_u^2}{\varepsilon_u^2} + y_t^2\left(\frac{r_\gamma'}{\varepsilon_{u_3}^2} - \frac{1}{\varepsilon_u^2}\right)U_u^\dagger\widetilde{P}U_u\right]V_{\text{CKM}}\widetilde{Y}_d,\end{aligned}$$

where we used completion relations $[\lambda_u^{(2)}]^\dagger\lambda_u^{(2)} = \varepsilon_u^2 g_*^2\mathbf{1}_{2\times 2}$ and $[\lambda_u^{(1)}]^\dagger\lambda_u^{(1)} = \varepsilon_{u_3}^2 g_*^2$, and the orthogonality $[\lambda_u^{(2)}]^\dagger\lambda_u^{(1)} = 0$. Since the above term is proportional to the product of a Hermitean matrix and a real diagonal mass matrix, the diagonal entries of $\widetilde{\mathcal{C}}_{d\gamma}$ are real as well. Thus, in the puRU scenario, the 1-loop contribution to the EDMs of down-type quarks exactly vanish.

We conclude that in the puRU scenario, the neutron EDM vanishes at leading order, while at 1-loop it produces the rather weak bound of Eq. (4.40). The neutron EDM bound hence does not require to go as far as MFV in symmetry space.

### 4.2.3 Experimental constraints

The most relevant constraints on the model come from the same observables considered in RU. In this subsection, we will therefore use Sec. 4.1.2 as a useful reference for our discussion, but only quote the most relevant bounds and emphasize the qualitative new features characterizing Partial Up-Right Universality. The full list of constraints can be found in App. C.

#### 4-fermion operators

In models of puRU, we expect the 4-fermion operators with the largest coefficients to be again those of Eq. (4.6). The bounds from fermion compositeness are therefore formally the same as in the previous model, see Eqs. (4.7, 4.8),

$$m_* \gtrsim (4.8 \div 7.8)\, g_* \varepsilon_u^2 \,\mathrm{TeV} \qquad m_* \gtrsim (3.0 \div 3.6)\, g_* \varepsilon_d^2 \,\mathrm{TeV} \quad \mathrm{LHC@37\ fb^{-1}}\,, \tag{4.43}$$

$$m_* \gtrsim (8.2 \div 13)\, g_* \varepsilon_u^2 \,\mathrm{TeV} \qquad m_* \gtrsim (5.1 \div 5.8)\, g_* \varepsilon_d^2 \,\mathrm{TeV} \quad \mathrm{LHC@300\ fb^{-1}\,(proj.)}\,, \tag{4.44}$$

though here $\varepsilon_u \neq \varepsilon_{u_3}$. As for the RU scenario of Sec. 4.1.2 $\Delta F = 1$ operators do not provide the leading constraints. As concerns instead $\Delta F = 2$ operators, we again find that the dominant constraint comes from $B_d$ oscillations. The bound in Eq. (4.11) becomes

$$m_* \gtrsim \frac{6.6}{g_* \varepsilon_{u_3}^2} \;\mathrm{TeV}\,. \tag{4.45}$$

Crucially, as opposed to the case of RU, the flavor bound, (4.45), and those from compositeness, (4.43) and (4.44), depend on different $\varepsilon$'s.

#### EW vertices

The bounds from the modified $Z$-couplings to the light families reported in Eq. (4.12) continue to apply also in scenarios with puRU and constrain $\varepsilon_u, \varepsilon_d$.

On the other hand, the bounds of Eqs. (4.13, 4.14, 4.16) involve the third generation and are the same as in those equations up to the replacement $\varepsilon_u \to \varepsilon_{u_3}$. In particular, the $Z\bar{t}t$ coupling of (4.14) becomes

$$m_* \gtrsim \frac{0.9}{\varepsilon_{u_3}} \,\mathrm{TeV}\,. \tag{4.46}$$

The same is true for the constraint on $C_{10}$ coming from $B_s \to \mu^+\mu^-$, which we report here because one of the most relevant for this model:

$$m_* \gtrsim \frac{6.5 \div 8.3}{\varepsilon_{u_3}} \,\mathrm{TeV}\,. \tag{4.47}$$

As discussed in Sec. 4.1.2 and also in App. A.1, the latter may be relaxed by an appropriate choice of $SO(4) \rtimes P_{\mathrm{LR}}$ representation for the composite fermions. In that case, the constraint on $C_9$ implies similarly to Eq. (4.19)

$$m_* \gtrsim \frac{1.2 \div 3.5}{g_* \varepsilon_{u_3}} \,\mathrm{TeV}\,. \tag{4.48}$$

Recalling that $y_t/g_* \lesssim \varepsilon_{u_3} \lesssim 1$, we see that the bound (4.48) is always weaker than (4.45).

**Dipoles**

In Sec. 4.2.2 we argued that corrections to the electric dipole moments are very weak. Here we hence focus on flavor-violating effects.

As in RU, the most relevant observable is $B \to X_s \gamma$. The coefficient of the associated dipole operator has a structure similar to the first line of Eq. (4.20), but now with $\lambda_{q_u}[\lambda_{q_u}]^\dagger$ replaced by the sum of two terms, $\lambda_{q_u}^{(2)}[\lambda_{q_u}^{(2)}]^\dagger$ and $\lambda_{q_u}^{(1)}[\lambda_{q_u}^{(1)}]^\dagger$. Using Table 5 and imposing the experimental constraint on $C_7$ we obtain numerically the same bound of Eq. (4.21) but with $\varepsilon_u$ replaced by $\varepsilon_{u_3}$:

$$m_* \gtrsim \frac{0.45 \div 0.68}{\varepsilon_{u_3}} \, \text{TeV} \, . \tag{4.49}$$

This is always weaker than (4.46), and becomes stronger than Eq. (4.45) only for $g_* \gtrsim 10$. Additional flavor-violating transitions in the up-sector involving the $D$ meson are negligible.

### 4.2.4 Summary

The main advantage of Partial Up-RU over plain RU lies in the difference between the parameters $\varepsilon_{u_3}$ and $\varepsilon_u$, controlling the compositeness of respectively $t_R$ and $u_R$, $c_R$. That allows to relax the tension between flavor and collider constraints. At the same time, puRU still features minimal down-type flavor violation and thus evades the most important flavor constraints. The novel flavor-violating parameters appear only in the up-quark sector, where the bounds are much weaker.

Let us first consider the favorite ranges for the compositeness parameters. Eqs. (4.47) and (4.45) are both controlled by $\varepsilon_{u_3}$ and favor a maximally composite $t_R$: the optimal value for $\varepsilon_{u_3}$ is $\varepsilon_{u_3}^{\text{opt}} \sim 1$. Happily, as $\varepsilon_{u_3} \neq \varepsilon_u$, this choice does not conflict with Eq. (4.43). The constraint on $m_*$ from this equation is made negligible by choosing sufficiently small $\varepsilon_u$. Indeed for $\varepsilon_u \lesssim 0.3$ Eq. (4.43) becomes even weaker than the universal constraint on $C_H$ in Eq. (C.4). Notice however, that significantly smaller values, roughly $\varepsilon_u \lesssim 0.1$, are disfavored by flavor violation in the down sector (this is shown in parentheses in the first row of Tab. 13). We therefore conclude that in the regime $\varepsilon_{u_3} \sim 1$ and $0.1 \lesssim \varepsilon_u \lesssim 0.3$ the bounds from flavor violation and flavor non-universality are minimized. As concerns instead the parameter $\varepsilon_d$, it appears in Eq. (4.40), Eq. (4.43), and in Tab. 9. Like in scenarios with RU, it is only mildly constrained. Taking $0.03 \lesssim \varepsilon_d \lesssim 0.5$, all bounds in which this parameter appears are weaker than the universal constraints.

In view of the above, the leading constraints on $m_*$ are from flavor violation via top compositeness and from the universal observables of Sec. C.1. On the flavor side, for composite fermions carrying generic representations under the custodial group, the strongest constraint on $m_*$ comes from semi-leptonic $b$ decays, Eq. (4.47), and implies

$$m_* \gtrsim \frac{6.5 \div 8.3}{\varepsilon_{u_3}} \, \text{TeV} > 6.5 \div 8.3 \, \text{TeV} \, . \tag{4.50}$$

This is a very significant bound compared to the universal constraints of Sec. C.1 and clearly pushes the new physics beyond the reach of the (HL-)LHC. Invoking custodial protection for the $Z d_L^i d_L^j$ couplings (assuming, for instance, the representations in Eq. (A.2)), this bound is removed and the strongest constraint becomes that from $B_d$ oscillations shown in Eq. (4.45).

The two bounds of Eq. (4.45) and Eq. (C.4) determine an absolute lower bound

$$m_* \gtrsim 2.4 \text{ TeV}, \tag{4.51}$$

obtained for the optimal value $g_* \sim 3$. This can be seen in the interplay between the flavor (blue) and the universal (dashed black area) constraints in Fig. 7. Importantly, for scales as low as the ones shown in Eq. (4.51) our indirect bounds become complementary to direct searches of top partners and spin-1 resonances. A qualitative discussion of the latter can be found in App. C.6. A more comprehensive study of direct effects is well beyond the purpose of this paper.

In conclusion, assuming custodial protection for the $Z$ couplings, the choice $0.1 \lesssim \varepsilon_u^{\text{opt}} \lesssim 0.3$, $0.03 \lesssim \varepsilon_d^{\text{opt}} \lesssim 0.5$, $\varepsilon_{u3}^{\text{opt}} \sim 1$ and $g_* \sim 3$ allows to depict a scenario where $m_*$ can be as low as $\sim 2.4$ TeV. Once one accepts the symmetry assumptions, this scenario is not implausible in the light of naturalness. Moreover, $g_* \sim 3$ is also the optimal value to reproduce the proper Higgs quartic in the simplest models of pseudo-NG Higgs (see for instance [29]). In addition, the sizeable compositeness of right handed fermions allows us to defend their interpretation as composites chiral states of the strong dynamics.

## 4.3 Partial Right Universality (pRU)

### 4.3.1 General structure

In the previous subsection, a separation between the top quark and the $u, c$ quarks was introduced in order to reduce the tension between flavor-violating and flavor-conserving observables, which characterized RU. An analogous structure can be obviously considered for the down sector, even though there is no strong phenomenological motivation for that. Indeed a good part of the success of Partial Up RU is precisely rooted in the minimality of flavor-violation in the down sector, which we would thus be spoiling. Nevertheless, the study of the less minimal scenario is structurally important, as it provides an appreciation of the genericity of new physics constraints (on $m_*$ and $g_*$ principally) in the landscape of flavor models. We carry out such a study in the present subsection by analyzing the scenario we call Partial Right Universality. That can be viewed intuitively as an up-down symmetric version of the framework of Sec. 4.2.

In this scenario flavor violation again originates from Eq. (3.1). But now the strong sector has a smaller global symmetry

$$\mathcal{G}_{\text{strong}} = U(2)_U \times U(1)_U \times U(2)_D \times U(1)_D, \tag{4.52}$$

and the composite fermions transform as $\mathcal{O}_{q_u,u} \in (\mathbf{2} \oplus \mathbf{1}, \mathbf{1}_U)$, $\mathcal{O}_{q_d,d} \in (\mathbf{1}, \mathbf{2} \oplus \mathbf{1}_D)$, where the notation $\mathbf{1}_{U/D}$ indicates a singlet under $SU(2)_{U/D}$ carrying $U(1)_{U/D}$ charges.

Similarly to puRU, the spurions $\lambda_{q_u,q_d,u,d}$ transform under $\mathcal{G}_{\text{strong}} \times \mathcal{G}_{\text{quarks}}$ as

$$\lambda_{q_u} \equiv \lambda_{q_u}^{(2)} \oplus \lambda_{q_u}^{(1)} \in (\overline{\mathbf{2}} \oplus \overline{\mathbf{1}}_U, \mathbf{1}, \mathbf{3}, \mathbf{1}, \mathbf{1}), \qquad \lambda_{q_d} \equiv \lambda_{q_d}^{(2)} \oplus \lambda_{q_d}^{(1)} \in (\mathbf{1}, \overline{\mathbf{2}} \oplus \overline{\mathbf{1}}_D, \mathbf{3}, \mathbf{1}, \mathbf{1}),$$
$$\lambda_u \equiv \lambda_u^{(2)} \oplus \lambda_u^{(1)} \in (\overline{\mathbf{2}} \oplus \overline{\mathbf{1}}_U, \mathbf{1}, \mathbf{1}, \mathbf{3}, \mathbf{1}), \qquad \lambda_d \equiv \lambda_d^{(2)} \oplus \lambda_d^{(1)} \in (\mathbf{1}, \overline{\mathbf{2}} \oplus \overline{\mathbf{1}}_D, \mathbf{1}, \mathbf{1}, \mathbf{3}). \tag{4.53}$$

The key difference is of course that here also $\lambda_{q_d,d}$ decompose in a $3 \times 2$ plus a $3 \times 1$ matrices.

Finally, the couplings $\lambda_{u,d}$ are assumed to respect the following subgroup of $\mathcal{G}_{\text{strong}} \times \mathcal{G}_{\text{quarks}}$:

$$\mathcal{G}_F = U(3)_q \times U(2)_{U+u} \times U(1)_{U+u} \times U(2)_{D+d} \times U(1)_{D+d}. \tag{4.54}$$

The form of $\lambda_{q_u,q_d}$ is a priori arbitrary, and only constrained by the requirement of reproducing the SM masses and mixings. Explicitly, one can prove that there exists a field basis in which the mixing parameters are:

$$
\text{pRU}: \begin{cases}
\lambda_{q_u} \sim \frac{1}{\varepsilon_u}\begin{pmatrix} y_u & 0 \\ 0 & y_c \\ ay_c & by_c \end{pmatrix} \oplus \frac{1}{\varepsilon_{u_3}}\begin{pmatrix} 0 \\ 0 \\ y_t \end{pmatrix}, & \lambda_{q_d} \sim \widetilde{U}_d \frac{1}{\varepsilon_d}\begin{pmatrix} y_d & 0 \\ 0 & y_s \\ a'y_s & b'y_s \end{pmatrix} \oplus \widetilde{U}_d \frac{1}{\varepsilon_{d_3}}\begin{pmatrix} 0 \\ 0 \\ y_b \end{pmatrix}, \\[3em]
\lambda_u \sim g_* \begin{pmatrix} \varepsilon_u & 0 \\ 0 & \varepsilon_u \\ 0 & 0 \end{pmatrix} \oplus g_* \begin{pmatrix} 0 \\ 0 \\ \varepsilon_{u_3} \end{pmatrix}, & \lambda_d \sim g_* \begin{pmatrix} \varepsilon_d & 0 \\ 0 & \varepsilon_d \\ 0 & 0 \end{pmatrix} \oplus g_* \begin{pmatrix} 0 \\ 0 \\ \varepsilon_{d_3} \end{pmatrix}.
\end{cases}
\tag{4.55}
$$

As in the previous model, we write $\lambda_{q_{u/d}}$ is such a way that the diagonal entries control the quark masses and the third row controls the mixing angles through the complex numbers $a$, $b$, $a'$ and $b'$. The parameter $\varepsilon_d$, $\varepsilon_{d_3}$ are constrained to lie in the range

$$
\frac{y_s}{g_*} \lesssim \varepsilon_d \lesssim 1, \qquad \frac{y_b}{g_*} \lesssim \varepsilon_{d_3} \lesssim 1,
\tag{4.56}
$$

while $\varepsilon_u, \varepsilon_{u_3}$ satisfy the same relations of Eq. (4.33). For definiteness, from now on we will assume $b_R$ is more composite than $d_R$ and $s_R$, i.e. $\varepsilon_d \lesssim \varepsilon_{d_3}$. Note that the same $3 \times 3$ special unitary matrix $\widetilde{U}_d$ appears in front of both the doublet and singlet components of $\lambda_{q_d}$.[20] Via field redefinitions we can remove from $\widetilde{U}_d$ all the phases but one, leaving a total of 5 physical phases including those in $a, b, a', b'$. Similarly to puRU, for this model we assume $a, b, a'$ and $b'$ are $O(1)$ with arbitrary phases.

The Yukawa couplings $Y_{u,d}$ are both written as a sum of two independent pieces, analogously to Eq. (4.31). The matrices $U_{u/d}$ and $V_{u/d}$ that diagonalize them can again be expressed analytically (see Appendix B.2) in the approximation of hierarchical quark masses. The matrix $\widetilde{U}_d$ is instead related to the CKM matrix, given $V_{\mathrm{CKM}} = U_u^\dagger U_d$, with $U_u = \mathbf{1} + O(y_c^2/y_t^2)$ and $U_d = \widetilde{U}_d + O(y_s^2/y_b^2)$.

Analogously to RU and puRU (see for example the discussion above (4.4)), the structure (4.55) would naturally emerge if $u_R^i$ and $d_R^i$ were chiral composite states of the strong dynamics transforming respectively as $\mathbf{2} \oplus \mathbf{1}_U$ of $U(2)_U \times U(1)_U$ and $\mathbf{2} \oplus \mathbf{1}_D$ of $U(2)_D \times U(1)_D$, and provided $\varepsilon_u, \varepsilon_{u_3}, \varepsilon_{d_3}, \varepsilon_d$ are all $O(1)$.

Models with pRU are somewhat reminiscent of the $U(2)_{\mathrm{RC}}^3$ scenarios studied in [44–46]. But in principle, the two approaches differ significantly, both in the underlying symmetry hypothesis as well as in the way that the symmetry is broken. Indeed, in [44–46] a $U(2)^3$ symmetry is approximately shared by both the composite sector and the fundamental fermions

---

[20]To see this one first introduces a 3 by 3 unitary matrix $\widetilde{U}_d'$ such that $\lambda_{q_d}^{(1)} = \widetilde{U}_d'(0,0,y_b)^t/\varepsilon_{d_3}$, and then defines $\lambda_{q_d}^{(2)} = \widetilde{U}_d'\lambda_{q_d}^{(2)'}$, where $\lambda_{q_d}^{(2)'}$ is an arbitrary 3 by 2 matrix. The latter can always be put in the standard form via rotations $\widetilde{U}_2$ involving the first two generations of $q_L$ (and obviously also a rotation of the doublet component of $\mathcal{O}_d$, which to avoid cluttering we leave it as understood):

$$
\lambda_{q_d}^{(2)'} = \widetilde{U}_2 \frac{1}{\varepsilon_d}\begin{pmatrix} y_d & 0 \\ 0 & y_s \\ a'y_s & b'y_s \end{pmatrix}.
\tag{4.57}
$$

Finally, because $\widetilde{U}_2$ has no effect on $(0,0,y_b)^t$, the matrix that appears in (4.55) can be defined to be $\widetilde{U}_d \equiv \widetilde{U}_d'\widetilde{U}_2$.

$q_L, u_R, d_R$, and a priori may be broken by any of the bilinear couplings $\overline{\psi}\mathcal{O}_\psi$ allowed by gauge invariance. The authors decide to focus on the minimal set of symmetry-breaking spurions that can reproduce the SM masses and mixing angles, but other choices may be made. In pRU the hypotheses are radically different. First, the flavor symmetry is larger, namely $U(2)_{U+u} \times U(2)_{D+d} \times U(3)_q \subset \mathcal{G}_F$. Furthermore, our $\mathcal{G}_F$ is broken in the most general way by the couplings to $q_L$, and no model-dependence is left over other than the one encoded in the parameters $a, b, a', b'$ and $\widetilde{U}_d$. Nevertheless, the concrete implementation of the $U(2)_{\mathrm{RC}}^3$ scenarios considered in [45] (see their Eq. (76)) is a particular case of our setup.[21] The phenomenology is therefore similar.

### 4.3.2 Experimental constraints

This model inherits many of the constraints of the previous model. However, the lack of MFV in the down sector also gives rise to new flavor transitions that, as we explore in this section, tend to push $m_*$ to higher scales.

**4-fermion operators**

The main constraints from 4-fermion operators derived in Sec. 4.2 for puRU apply to this scenario as well, though with important differences. In particular pRU is still subject to the strong bounds from dijet searches, see Eqs. (4.43, 4.44), though here the $\varepsilon_d$ parameter in Eqs. (4.43, 4.44) controls only the compositeness of $d_R$, $s_R$ and is independent from $\varepsilon_{d_3}$, which instead controls $b_R$ compositeness.

In complete analogy to puRU, $\Delta F = 2$ transitions lead to significant constraints. In addition to Eq. (4.45), new $\Delta F = 2$ effects appear in pRU. The most relevant one involves the following operator with four right-handed quarks

$$\mathcal{O}_1'^{d_i d_j} = (\bar{d}_{jR}\gamma^\mu d_{iR})(\bar{d}_{jR}\gamma_\mu d_{iR}), \tag{4.59}$$

defined in App. B.3.

The coefficient is estimated to be

$$C_1'^{d_i d_j} \sim \frac{1}{m_*^2 g_*^2}\left(\left[V_d^\dagger(\lambda_d^{(1)}[\lambda_d^{(1)}]^\dagger + r\lambda_d^{(2)}[\lambda_d^{(2)}]^\dagger)V_d\right]^{ij}\right)^2 \sim \frac{\varepsilon_{d_3}^4 g_*^2}{m_*^2}([V_d]^{3i}[V_d^*]^{3j})^2. \tag{4.60}$$

The dominant bound comes from the $B_d$ system and leads to

$$m_* \gtrsim 22\, g_*\, \varepsilon_{d_3}^2 \text{ TeV}, \tag{4.61}$$

which clearly pushes $\varepsilon_{d_3}$ towards small values. Other contributions to $\Delta F = 2$ processes, reported in Tab. 13 for completeness, are weaker. In particular, the "Higgs-mediated" transitions, discussed in detail in App. C.4, always turn out to be negligible within our flavor hypothesis.

---

[21]Specifically, in the u-quark sector they assume zeros in the $q_L^3$ couplings to $\mathcal{O}_{qu}^{(2)}$:

$$\lambda_{qu} \sim \begin{pmatrix} \boldsymbol{\Delta}_{2\times 2} \\ \mathbf{0}_{1\times 2} \end{pmatrix} \oplus \begin{pmatrix} \mathbf{V}_{2\times 1} \\ \lambda_{Lu} \end{pmatrix} \tag{4.58}$$

whereas we allow the most general structure of that coupling. In either case, after appropriate flavor rotations one recovers our $\lambda_{qu}$ in (4.55).

**EW vertices**

The bounds from the modified $Z$-couplings to the light families reported in Eq. (4.12) also apply to the pRU scenario and set important constraints on $\varepsilon_u, \varepsilon_d$.

Regarding the $Z$ coupling to heavy families, on the other hand, we obtain the same bounds of Eqs. (4.13, 4.14, 4.16) up to the key replacements $\varepsilon_u \to \varepsilon_{u_3}$ and $\varepsilon_d \to \varepsilon_{d_3}$. For example, in pRU Eq. (4.13) becomes

$$m_* \gtrsim \frac{2.2 \div 2.8}{\varepsilon_{u_3}}, \ \frac{0.05}{\varepsilon_{d_3}} \, \text{TeV} \,. \tag{4.62}$$

Like in all models studied so far, the $\Delta F = 1$ transitions $b \to s$ provide powerful constraints, see for instance Eqs. (4.47, 4.48). The misalignment in the right-handed down sector implies further contributions to $b \to s$ transitions. The most important is a tree-level contribution to the $C'_{10}$ Wilson coefficient (defined in Eq. (B.33)) proportional to the structure

$$\frac{1}{m_*^2} \left[ V_d^\dagger (\lambda_d^{(1)} [\lambda_d^{(1)}]^\dagger + r \lambda_d^{(2)} [\lambda_d^{(2)}]^\dagger) V_d \right]^{32} = \frac{g_*^2}{m_*^2} (\varepsilon_{d_3}^2 - r \varepsilon_d^2)[V_d^*]^{33}[V_d]^{32} \,, \tag{4.63}$$

where $r$ is an unknown real coefficient of order unity. Observing that $[V_d]^{32} \sim y_s/y_b$, we get

$$C'_{10} \sim \frac{y_s/y_b}{V_{\text{CKM}}^{tb}(V_{\text{CKM}}^{ts})^*} \frac{2\sqrt{2}\pi^2 g_*^2}{G_F m_*^2 e^2} \varepsilon_{d_3}^2 \,. \tag{4.64}$$

In view of the above, the observed branching fraction $B_s \to \mu\mu$ [54] then implies

$$m_* \gtrsim 7.1 \, g_* \varepsilon_{d_3} \, \text{TeV} \,. \tag{4.65}$$

Custodial protection can as usual be invoked to relax some of these bounds. For instance, within the assumptions of Eq. (A.2), the constraint (4.65) is removed. Yet, the milder

$$m_* \gtrsim (1.0 \div 2.4)\varepsilon_{d_3} \, \text{TeV} \tag{4.66}$$

remains as a result of the operator $\mathcal{O}'_9$ defined in (B.33), whose Wilson coefficient is given by Eq. (4.64) with a further $\sim e^2/g_*^2$ suppression.

**Dipoles**

As in the previous setup (see Sec. 4.2.2) we find that tree-level corrections to the electric dipole moments of the quarks vanish. The first non-trivial contributions arise at 1-loop in both up and down sectors and appear via a generalization of the structure in Eq. (4.40). In particular, we have a total of four independent contributions to the up-type and the down-type quark dipoles:

$$\text{Im}[\widetilde{\mathcal{C}}_{u\gamma}] \sim \frac{1}{g_* m_*^2} \frac{e}{16\pi^2} \text{Im}\Big[ U_u^\dagger \Big( \lambda_{q_d}^{(2)}[\lambda_{q_d}^{(2)}]^\dagger \lambda_{q_u}^{(2)}[\lambda_u^{(2)}]^\dagger + r_u \lambda_{q_d}^{(1)}[\lambda_{q_d}^{(1)}]^\dagger \lambda_{q_u}^{(2)}[\lambda_u^{(2)}]^\dagger$$
$$+ r'_u \lambda_{q_d}^{(2)}[\lambda_{q_d}^{(2)}]^\dagger \lambda_{q_u}^{(1)}[\lambda_u^{(1)}]^\dagger + r''_u \lambda_{q_d}^{(1)}[\lambda_{q_d}^{(1)}]^\dagger \lambda_{q_u}^{(1)}[\lambda_u^{(1)}]^\dagger \Big) V_u \Big] \,, \tag{4.67}$$

$$\text{Im}[\widetilde{\mathcal{C}}_{d\gamma}] \sim \frac{1}{g_* m_*^2} \frac{e}{16\pi^2} \text{Im}\Big[ U_d^\dagger \Big( \lambda_{q_u}^{(2)}[\lambda_{q_u}^{(2)}]^\dagger \lambda_{q_d}^{(2)}[\lambda_d^{(2)}]^\dagger + r_d \lambda_{q_u}^{(1)}[\lambda_{q_u}^{(1)}]^\dagger \lambda_{q_d}^{(2)}[\lambda_d^{(2)}]^\dagger$$
$$+ r'_d \lambda_{q_u}^{(2)}[\lambda_{q_u}^{(2)}]^\dagger \lambda_{q_d}^{(1)}[\lambda_d^{(1)}]^\dagger + r''_d \lambda_{q_u}^{(1)}[\lambda_{q_u}^{(1)}]^\dagger \lambda_{q_d}^{(1)}[\lambda_d^{(1)}]^\dagger \Big) V_d \Big] \,, \tag{4.68}$$

The two main contributions to $d_n$ arising from Eq. (4.67) are due to the up-quark (see also Appendix B.4) and read

$$\frac{1}{g_* m_*^2} \frac{e}{16\pi^2} \text{Im}\left[ U_u^\dagger \left( \lambda_{q_d}^{(1)} [\lambda_{q_d}^{(1)}]^\dagger \lambda_{q_u}^{(1)} [\lambda_u^{(1)}]^\dagger \right) V_u \right]^{11} \Rightarrow d_n|_u \sim \frac{e m_c}{m_*^2 \varepsilon_{d_3}^2} \frac{y_b^2}{8\pi^2} \lambda_C^3 \, \text{Im}[a A(\rho - i\eta)] \,, \tag{4.69}$$

$$\frac{1}{g_* m_*^2} \frac{e}{16\pi^2} \text{Im}\left[ U_u^\dagger \left( \lambda_{q_d}^{(2)} [\lambda_{q_d}^{(2)}]^\dagger \lambda_{q_u}^{(1)} [\lambda_u^{(1)}]^\dagger \right) V_u \right]^{11} \Rightarrow d_n|_u \sim \frac{e m_c}{m_*^2 \varepsilon_d^2} \frac{y_s^2}{8\pi^2} \lambda_C \, \text{Im}[ab'^*] \,. \tag{4.70}$$

These are enhanced by small $\varepsilon_{d_3}$ and by small $\varepsilon_d$, respectively, but they only lead to mild bounds

$$m_* \gtrsim \frac{0.06}{\varepsilon_{d_3}} \text{ TeV} \,, \qquad\qquad m_* \gtrsim \frac{0.01}{\varepsilon_d} \text{ TeV} \,. \tag{4.71}$$

Indeed, the first constraint is comparable to the one from anomalous Z-couplings in Eq. (4.62), while the second is weaker than that from $\widehat{S}$ in Eq. (C.4), as long as $\varepsilon_d \gtrsim 0.004$.

In the down-sector, as discussed in Appendix B.4, both the $d$- and the $s$-quark EDMs may contribute significantly to $d_n$. The dominant terms in Eq. (4.68) come from the structure $\lambda_{q_u}^{(1)} [\lambda_{q_u}^{(1)}]^\dagger \lambda_{q_d}^{(1)} [\lambda_d^{(1)}]^\dagger$; referring to Eq. (B.50) we find

$$d_n|_d \sim c_d \sqrt{2} v \text{Im}[\widetilde{\mathcal{C}}_{d\gamma}]^{11} \sim c_d \frac{e m_s}{m_*^2 \varepsilon_{u_3}^2} \frac{y_t^2}{8\pi^2} \lambda_C^3 \, \text{Im}[a' A(1 - \rho + i\eta)] \,, \tag{4.72}$$

$$d_n|_s \sim \frac{c_s}{N_c} \sqrt{2} v \text{Im}[\widetilde{\mathcal{C}}_{d\gamma}]^{22} \sim \frac{c_s}{N_c} \frac{e m_s}{m_*^2 \varepsilon_{u_3}^2} \frac{y_t^2}{8\pi^2} A \lambda_C^2 \, \text{Im}[b] \,. \tag{4.73}$$

Notice that the $s$-quark contribution to the $d_n$ is parametrically larger by a factor $1/\lambda_C$ but simultaneously suppressed by $c_s/(c_d N_c)$. As a result, taking $c_d \sim 1$ we obtain:

$$m_* \gtrsim \begin{cases} 1.7/\varepsilon_{u_3} \text{ TeV} & \text{(d)} \\ \sqrt{c_s} \, 2.0/\varepsilon_{u_3} \text{ TeV} & \text{(s)} \,. \end{cases} \tag{4.74}$$

With our conservative estimate $c_s \sim 1$ the two bounds are comparable, whereas the down-quark contribution clearly dominates when $|c_s| \ll 1$. While significant, these bounds are still compatible with new physics at the edge of the LHC reach. Yet, the relevance of the bound from the $s$-quark EDM motivates further investigations of this contribution in lattice QCD. The $\varepsilon_u$-dependent bounds are much weaker because suppressed by $y_c^2/(y_t^2 \lambda_C)$ and can be ignored.

Let us next turn to the flavor-violating dipoles. From $B \to X_s \gamma$ we still obtain the same bound found in puRU (see Eq. (4.49)).

$$m_* \gtrsim \frac{0.45 \div 0.68}{\varepsilon_{u_3}} \text{ TeV} \,. \tag{4.75}$$

Yet, the less minimal flavor structure in the down sector produces new effects, non-vanishing $C_7$ and $C_7'$, at the first non-trivial order in $\lambda_{q_u, q_d}$ (we refer again to Appendix B.3 and in particular to Eq. (B.39) for the definition). That originates from the misalignment between the

down-Yukawa $Y_d \propto \lambda_{qd}^{(2)}[\lambda_d^{(2)}]^\dagger + \lambda_{qd}^{(1)}[\lambda_d^{(1)}]^\dagger$ and the corresponding dipole operator coefficients $\mathcal{C}_{d\gamma} \propto \lambda_{qd}^{(2)}[\lambda_d^{(2)}]^\dagger + r\lambda_{qd}^{(1)}[\lambda_d^{(1)}]^\dagger$ with $r \neq 1$. That leads to

$$\frac{4G_F}{\sqrt{2}}V_{\text{CKM}}^{tb}(V_{\text{CKM}}^{ts})^*\frac{m_b}{16\pi^2}C_7 = c_7\frac{1}{g_*m_*^2}\frac{v}{\sqrt{2}}\left[U_d^\dagger\lambda_{qd}^{(1)}[\lambda_d^{(1)}]^\dagger V_d\right]_{23}, \tag{4.76}$$

$$\frac{4G_F}{\sqrt{2}}V_{\text{CKM}}^{tb}(V_{\text{CKM}}^{ts})^*\frac{m_b}{16\pi^2}C_7' = c_7'\frac{1}{g_*m_*^2}\frac{v}{\sqrt{2}}\left[V_d^\dagger\lambda_d^{(1)}[\lambda_{qd}^{(1)}]^\dagger U_d\right]_{23}.$$

Because $[V_d]_{23,32} \sim y_s/y_b \gg [U_d]_{23,32}$ (see Appendix B.2 for the explicit expression), we estimate

$$C_7' \sim c_7'\frac{m_s}{m_b}\frac{b'4\sqrt{2}\pi^2}{G_F V_{\text{CKM}}^{tb}(V_{\text{CKM}}^{ts})^*m_*^2}, \tag{4.77}$$

while $C_7$ is suppressed by an additional power of $m_s/m_b$. In previous models, on the contrary, $C_7'$ was generated only through loops of the elementary fermions, as in Eq. (4.21). Making the conservative assumption $c_7' \sim 1$, corresponding to the statement that dipole interactions are parametrically of "tree-level" size, and using [58], we derive a significant bound

$$m_* \gtrsim (4.5 \div 5.2)\,\text{TeV} \qquad (|c_7'| = 1). \tag{4.78}$$

This result is particularly relevant because it is completely independent of the mixing parameters $\varepsilon_{d_3}, \varepsilon_d$. A weaker constraint applies however to theories in which the dipole interactions first arise at 1-loop order in the strong interactions, i.e. $c_7' \sim g_*^2/16\pi^2$. In that case, the lower bound on the mass of the new physics becomes

$$m_* \gtrsim (0.36 \div 0.41)\,g_*\,\text{TeV} \qquad (|c_7'| = g_*^2/16\pi^2), \tag{4.79}$$

and is weaker than the universal constraint on $C_H$ shown in (C.4).

### 4.3.3 Summary

A scenario with flavor symmetry in the down sector reduced from $U(3)$ down to $U(2)$, corresponding to Partial Right Universality (pRU), does not fare better than the Partial Up-Right Universality (puRU) scenario of Sections 4.2. All the constraints of that scenario continue to apply and new important effects appear.

In the most general case, with no specific assumption on the custodial quantum numbers of the composite fermions, the strongest constraint comes from $B_s \to \mu\mu$. That is the same as in puRU and gives the lower bound (see Eq. (4.47))

$$m_* > \frac{6.5 \div 8.3}{\varepsilon_{u_3}}\,\text{TeV}. \tag{4.80}$$

A scenario within collider reach must then necessarily involve custodial protection of the effect leading to the above bound. Assuming for instance the representations of Eq. (A.2) the bound is eliminated and other constraints become more relevant. Let us discuss them in turn.

Effects controlled by the compositeness of the up-type quarks are precisely the same as in puRU. Specifically, the largest effects controlled by $\varepsilon_{u_3}$ are seen in $\Delta F = 2$ oscillations, see Eq. (4.45), and electric dipoles, see Eq. (4.74). They are both minimized by taking the optimal value $\varepsilon_{u_3}^{\text{opt}} \sim 1$. Concerning $u_R, c_R$ compositeness, as in puRU we find that in the

optimal range $0.1 \lesssim \varepsilon_u^{\mathrm{opt}} \lesssim 0.3$ the dijet bounds of Eq. (4.43) are under control and no large $\Delta F = 2$ effects are induced (see the first row of Tab. 13).

In the down-sector, the situation departs significantly from puRU, where new sizable flavor-violating effects are controlled by the two parameters $\varepsilon_d$ and $\varepsilon_{d_3}$. In particular, Eq. (4.61), Eqs. (4.65, 4.66) and Eq. (4.71) add to the flavor bounds found in puRU. Eq. (4.65) is removed by invoking the custodial protection also in the bottom-right sector, for example assuming the representations in Eq. (A.2). The other effects may be suppressed by taking $\varepsilon_d, \varepsilon_{d_3}$ in the appropriate range. For example, in the range $0.004 \lesssim \varepsilon_d^{\mathrm{opt}} \lesssim \varepsilon_{d_3}^{\mathrm{opt}}$ and $0.03 \lesssim \varepsilon_{d_3}^{\mathrm{opt}} \lesssim 0.2$ they become less important than the universal constraints of Appendix (C.4). A far more significant novelty is instead the new contribution to $B \to X_s \gamma$. In scenarios in which the relevant $\Delta F = 1$ dipole is generated at tree-level a very strong bound independent of all the other parameters, arises (see Eq. (4.78))

$$m_* \gtrsim 4.4 \div 6.3\,\mathrm{TeV}. \tag{4.81}$$

To render new physics accessible at the (HL-)LHC it is therefore mandatory that dipoles are generated at 1-loop level. In this more optimistic scenario the new correction to $B \to X_s \gamma$ results in Eq. (4.79), which is less significant than the universal constraint on $\mathcal{C}_H$ in Eq. (C.4). This is illustrated in the left panel of Fig. 8.

In conclusion, in pRU the absolute lower bound on the new physics scale is the same as in puRU (see (4.51)):

$$m_* \gtrsim 2.4\ \mathrm{TeV}. \tag{4.82}$$

Yet, the parameter region where such lowest $m_*$ becomes possible is more constrained. Not only do we need $g_* \sim 3$, $\varepsilon_{u_3}^{\mathrm{opt}} \sim 1$, and $0.1 \lesssim \varepsilon_u^{\mathrm{opt}} \lesssim 0.3$, as in puRU, but also the exclusive generation of dipoles at 1-loop (check the dashed green line of Fig. 8) and down sector mixing parameters in the smaller range $0.004 \lesssim \varepsilon_d^{\mathrm{opt}} \lesssim \varepsilon_{d_3}^{\mathrm{opt}}$ and $0.03 \lesssim \varepsilon_{d_3}^{\mathrm{opt}} \lesssim 0.2$.

## 5    Universality in the left-handed sector

In this section, we shall explore scenarios with universality, full or partial, in the couplings of the left-handed quarks $q_L$. As discussed in Sec. 3.1, this can be realized in either implementation of the mixing with $q_L$, that is for the case of two triplets $\mathcal{O}_{q_u}$ and $\mathcal{O}_{q_d}$ and for the case of a single triplet $\mathcal{O}_q$. In the former case, the mixing is described by Eq. (3.1) and MFV can be realized in two distinct ways, shown in the second and last lines of Table 1. In the latter case, the mixing takes instead the form

$$\mathcal{L}_{\mathrm{mix}} = \lambda_q^{ia} \overline{q}_L^i \mathcal{O}_q^a + \lambda_u^{ia} \overline{u}_R^i \mathcal{O}_u^a + \lambda_d^{ia} \overline{d}_R^i \mathcal{O}_d^a, \tag{5.1}$$

with $a = 1, 2, 3$. In the following, we shall focus on scenarios with a single triplet $\mathcal{O}_q^a$, as they have no analog among the cases discussed in Sections 4. As before, we first analyze the most symmetric realization and then consider departures from it. The phenomenology of models with left universality with two composite doublets has a similar phenomenology and will not be studied in detail.

## 5.1 Left Universality (LU)

### 5.1.1 General structure

As opposed to the scenarios discussed in Sections 4, within the framework in Eq. (5.1) there is a unique realization of MFV which we call Left Universality (LU), as was anticipated in the last row of Tab. 1. It is implemented by postulating the strong sector is endowed with a symmetry

$$\mathcal{G}_{\text{strong}} = U(3)_Q \,, \tag{5.2}$$

under which $\mathcal{O}_{q,u,d} \in \mathbf{3}$. The matrices $\lambda_{q,u,d}$ of Eq. (5.1) can be viewed as spurions with $\mathcal{G}_{\text{strong}} \times \mathcal{G}_{\text{quarks}}$ quantum numbers

$$\lambda_q \in (\overline{\mathbf{3}}, \mathbf{3}, \mathbf{1}, \mathbf{1}) \,, \qquad \lambda_u \in (\overline{\mathbf{3}}, \mathbf{1}, \mathbf{3}, \mathbf{1}) \,, \qquad \lambda_d \in (\overline{\mathbf{3}}, \mathbf{1}, \mathbf{1}, \mathbf{3}) \,. \tag{5.3}$$

The mixing with the left-handed fermions is then postulated to be proportional to the identity, i.e. $\lambda_q^{ia} \propto \delta^{ia}$, so as to respect

$$\mathcal{G}_F = U(3)_{Q+q} \times U(3)_u \times U(3)_d \subset \mathcal{G}_{\text{strong}} \times \mathcal{G}_{\text{quarks}} \,. \tag{5.4}$$

$\mathcal{G}_F$ is finally explicitly broken by $\lambda_{u,d}$. As $Y_u \sim \lambda_q \lambda_u^\dagger / g_* \propto \lambda_u^\dagger$ and $Y_d \sim \lambda_q \lambda_d^\dagger / g_* \propto \lambda_d^\dagger$, the only two sources of flavor violation are proportional to the SM Yukawa couplings thus realizing MFV. Without loss of generality the matrices $\lambda_{q,u,d}$ can be written as

$$\text{LU}: \begin{cases} \lambda_q \sim \begin{pmatrix} \varepsilon_q & 0 & 0 \\ 0 & \varepsilon_q & 0 \\ 0 & 0 & \varepsilon_q \end{pmatrix} g_*, \\[2em] \lambda_u \sim \dfrac{1}{\varepsilon_q} \begin{pmatrix} y_u & 0 & 0 \\ 0 & y_c & 0 \\ 0 & 0 & y_t \end{pmatrix}, \\[2em] \lambda_d \sim \dfrac{1}{\varepsilon_q} \begin{pmatrix} y_d & 0 & 0 \\ 0 & y_s & 0 \\ 0 & 0 & y_b \end{pmatrix} V_{\text{CKM}}^\dagger \end{cases} \tag{5.5}$$

where

$$\frac{y_t}{g_*} \lesssim \varepsilon_q \lesssim 1. \tag{5.6}$$

A straightforward generalization of the discussion presented above (4.4) for RU reveals that a dynamical interpretation of the flavor structure (5.5) could be given if we interpreted $q_L^i$ as chiral composite states of the strong dynamics transforming as $\mathbf{3}$ of $U(3)_Q$, provided $\varepsilon_q = O(1)$.

This scenario has been previously investigated in [43, 45] where, because of the reason we just explained, it was called "left-handed compositeness".

### 5.1.2 Experimental constraints

Following the same scheme adopted in Sections 4, we will now present the main constraints on this scenario, starting from four-fermion interactions, then modified vector couplings, and finally dipole operators.

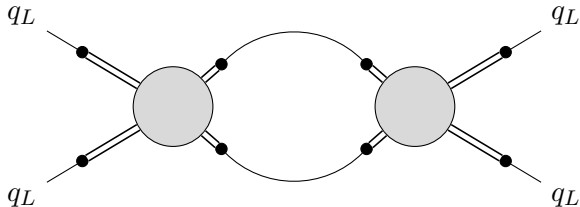

Figure 4: Dominant $\Delta F = 2$ processes in LU. The double lines represent fermionic states of the composite dynamics, the solid single lines denote elementary fermions.

**4-fermion operators**

As the $\lambda_{u,d}$ encapsulate the SM Yukawa structure it is reasonable to treat them as small and to expand in powers of them. At the zeroth order in the expansion, we encounter the operators

$$\mathcal{O}_{qq}^{(1)} = (\bar{q}_L \gamma^\mu q_L)(\bar{q}_L \gamma_\mu q_L)\,, \qquad \mathcal{O}_{qq}^{(3)} = (\bar{q}_L \gamma^\mu \sigma^a q_L)(\bar{q}_L \gamma_\mu \sigma^a q_L)\,, \tag{5.7}$$

with flavor-universal coefficients $\sim \varepsilon_q^4 g_*^2/m_*^2$. This operator affects elastic quark scattering and is constrained by the study of dijet events at the LHC. Imposing again the bounds of [49–51] we find

$$m_* \gtrsim (5.2 \div 8.7)\, g_* \varepsilon_q^2 \,\text{TeV}\,, \qquad\qquad \text{LHC@37 fb}^{-1}\,, \tag{5.8}$$

$$m_* \gtrsim (9.0 \div 14)\, g_* \varepsilon_q^2 \,\text{TeV}\,, \qquad\qquad \text{LHC@300 fb}^{-1}\,(\text{proj.})\,. \tag{5.9}$$

Remaining at tree level, but going to the next order of the expansion in $\lambda_{u,d} \propto Y_{u,d}$ we find operators $(\bar{q}_L d_R)(\bar{d}_R q_L)$ and $(\bar{q}_L u_R)(\bar{u}_R q_L)$ proportional to respectively $(Y_d)_{ij}(Y_d)^\dagger_{kl}$ and $(Y_u)_{ij}(Y_u)^\dagger_{kl}$. As such, these operators only induce flavor violation in charged currents and no FCNC. The other structures such as $(\bar{q}_L \gamma_\mu q_L)(\bar{d}_R \gamma^\mu d_R)$ at order $\lambda_d^2$ or $(\bar{d}_R \gamma_\mu d_R)(\bar{d}_R \gamma^\mu d_R)$ at order $\lambda_d^4$, along with the analogues involving $u_R$, are easily seen not to produce any flavor violation. Genuine FCNC can therefore only appear through loops. The leading effect involves external $q_L$ lines and is induced at 1-loop via the diagrams with the structure shown in Fig. 4. The corresponding effective operator $\mathcal{O}_1$ (see Eq. (B.43)) is generated dominantly by loops with virtual $u_R$'s, and with a coefficients $\sim (\lambda_u^\dagger \lambda_u)^2 \varepsilon_q^4/(16\pi^2 m_*^2) \sim (Y_u Y_u^\dagger)^2/(16\pi^2 m_*^2)$. That is the same as Eq. (4.10), but with a further suppression, $g_*^2 \varepsilon_u^4/16\pi^2$. The most severe bound is from $B_d$ mixing and reads

$$m_* \gtrsim 0.53 \text{ TeV}\,. \tag{5.10}$$

We conclude that, remarkably, in this scenario, FCNC mediated by 4-fermion operators are not a concern. In fact, Eq. (5.10) is always weaker than the universal constraint from the electroweak $\widehat{S}$ parameter reported in Eq. (C.4).

**EW vertices**

By far the most significant constraints on this model is due to the modifications of the quark couplings to vectors. In Tab. 10 we collect the bounds from anomalous $Z$-couplings. The most relevant ones are

$$m_* \gtrsim (4.5 \div 5.0) g_* \varepsilon_q \text{ TeV}\,, \tag{5.11}$$

from the couplings of $u_L, d_L$ [53] and

$$m_* \gtrsim \frac{0.5}{\varepsilon_q} \text{ TeV} \,, \tag{5.12}$$

from the anomalous $t_R$ coupling [59].

Yet, a stronger constraint comes from the $W$ couplings. The operator $[\mathcal{O}_{Hq}^{(3)}]^{ij}$ (see Tab. 4) comes with a flavor-conserving coefficient $c\delta^{ij}\varepsilon_q^2 g_*^2/m_*^2$ and generates a universal correction to the $W$ coupling to left-handed quarks

$$\frac{g}{\sqrt{2}}(1 + \delta g_W)\bar{u}V_{\text{CKM}}\gamma^\mu P_L d W_\mu^+ \,, \tag{5.13}$$

with $\delta g_W = c\varepsilon_q^2 g_*^2 v^2/m_*^2$ and $c = O(1)$. In practice, this is equivalent to the replacement $V_{\text{CKM}} \to (1 + \delta g_W)V_{\text{CKM}}$ in the SM vertex. This deformation is in tension with the experimental CKM-unitarity test: $(1 + \delta g_W)^2 \sum_i |V_{\text{CKM}}^{ui}|^2 - 1 = (1.5 \pm 0.7) \times 10^{-3}$ [60]. As in our model, $V_{\text{CKM}}$ is unitary, this reduces to a constraint on $(1 + \delta g_W)^2 - 1 \approx 2\delta g_W$. The conservative request that $\delta g_W$ not exceed twice the 1-$\sigma$ error translates into

$$m_* \gtrsim 9.3\, g_*\varepsilon_q \,\text{TeV} \,. \tag{5.14}$$

This bound is obviously stronger than (5.11), but also than (5.8) since $\varepsilon_q \lesssim 1$.

Like for $\Delta F = 2$ transitions, the bounds from $\Delta F = 1$ transitions are also rather innocuous. At leading order in the $\lambda_{u,d}$ expansion, the only flavor-violating structure is due to the operator

$$[O_{Hud}]^{ij} \equiv \left(\widetilde{H}^\dagger i D_\mu H\right) \bar{u}_R^i \gamma^\mu d_R^j \,, \tag{5.15}$$

which appears in the effective Lagrangian with a coefficient

$$[\mathcal{C}_{Hud}]^{ij} = c\,\frac{(\lambda_u\lambda_d^\dagger)^{ij}}{m_*^2} = c\,\frac{(Y_uY_d^\dagger)^{ij}}{\varepsilon_q^2 m_*^2} \,, \qquad\qquad c = O(1). \tag{5.16}$$

$[O_{Hud}]$ is the only vertex correction operator not included in Table 4. We made this choice because of its limited relevance to our study. Its main effect is the creation of a coupling of the $W$ to the right-handed charged current. The leading constraint comes from LHC data on the $Wtb$ vertex [59] and reads $m_* \gtrsim 0.06/\varepsilon_q \,\text{TeV}$. Even for the smallest possible choice, $\varepsilon_q \sim y_t/g_*$, the resulting bound on $m_*/g_*$ is weaker than those from universal effects reported in App. C.4.

### Dipoles

At tree level, the Yukawas and the dipole coefficients are perfectly aligned and proportional to $\lambda_q\lambda_u^\dagger \propto Y_u$ and $\lambda_q\lambda_d^\dagger \propto Y_d$, in respectively the up and down sector. Flavor violation is thus absent at leading order, and only appears at 1-loop, in the form of a middle insertion of $\lambda_u\lambda_u^\dagger$ or $\lambda_d\lambda_d^\dagger$. The strongest constraint is again from $B \to X_s\gamma$, through a misaligned correction $\mathcal{C}_{d\gamma} \propto \lambda_q\lambda_u^\dagger\lambda_u\lambda_d^\dagger$ to the down sector dipole matrix coefficient. In the mass basis, this leads to the same coefficient displayed in the second line of Eq. (4.20), but with $\varepsilon_u$ replaced by $\varepsilon_q$. The bound on $C_7$ then translates into (see Eq. (4.21))

$$m_* \gtrsim \frac{0.45 \div 0.68}{\varepsilon_q} \text{ TeV} \,, \tag{5.17}$$

which is comparable to, if not slightly stronger than, the LHC bound in Eq. (5.12). Notice, however, that this bound is always slightly weaker than that from $\mathcal{C}_H$, shown in Eq. (C.4).

Furthermore, as in Sections 4.1, MFV plus CP guarantees that in the LU scenario, there is no significant contribution to the neutron EDM.

### 5.1.3 Summary

To conclude, the most relevant constraints on LU are the CKM unitarity bound in Eq. (5.14) and the universal bounds in Eq. (C.4). Recalling that $g_* \varepsilon_q \gtrsim y_t$, the former implies

$$m_* \gtrsim 7.5 \, \text{TeV} \,, \tag{5.18}$$

for the optimal, namely the minimal, $\varepsilon_q^{\text{opt}} \sim y_t/g_*$. We see that in Left Universality, like for Right Universality, $m_*$ is strongly constrained from below, precisely because of universal effects. The tension is so significant that the scenario is pushed out of reach of the (HL-)LHC. The universal constraint from $\mathcal{C}_H$ starts to compete with Eq. (5.18) only at large couplings $g_* \gtrsim 9$. Furthermore, in this region also $\Delta F = 1$ transitions acquire some relevance, still remaining weaker than universal constraints. In particular, loop induced $b \to s\gamma$ transition in Eq. (5.17) are enhanced by small $\varepsilon_q \gtrsim y_t/g_*$. For $g_* \gtrsim 9$ the optimal value for $\varepsilon_q$ is then obtained by combining Eq. (5.17) and Eq. (5.18) and reads $\varepsilon_q^{\text{opt}} \sim 0.3/\sqrt{g_*}$. This gives $m_* \gtrsim 2.5\sqrt{g_*}$ that is still milder than the bound on $\mathcal{C}_H$.

The allowed regime is shown in the left plot of Fig. 6. The lesson from these results is rather clear. There is no such thing as a pure "flavor hypothesis" disconnected from its concrete dynamical incarnation: the MFV scenario (as introduced in Ref. [2]) is not sufficiently specified to infer its full implications and constraints.

For completeness, we mention that the very same conclusions, and in particular the same lower bound in Eq. (5.18), can be derived for LU scenarios with two composite $\mathcal{O}_{q_u,q_d}$. The main difference is that now the structures $\lambda_u \lambda_d^\dagger$ and $\lambda_d \lambda_u^\dagger$ are forbidden by the larger $U(3)_{Q_u} \times U(3)_{Q_d}$ symmetry. The modest constraints associated with the operator of (5.15) and with dipoles (e.g. Eq. (5.17)) are thus eliminated.

## 5.2 Partial Left Universality (pLU)

### 5.2.1 General structure

It is natural to ask whether, also in the LU scenario, a reduction of universality allows to lower $m_*$ closer to the TeV scale. Notice however that, for the models with a single $\mathcal{O}_q$ we are considering here, a reduced universality implies the loss of the MFV structure in *both* the up and the down sector. This, as we will show below, implies more severe constraints compared with the scenario of Partial Right Up Universality, discussed in Sec. 4.2. Yet, the main constraints can be relaxed through additional dynamical assumptions (e.g. the hypothesis that the dipoles are generated at 1-loop), making the scenario as viable as Partial Right Universality, discussed in Sec. 4.3.

Let us then consider the scenario of Partial Left Universality (pLU), defined by the reduced strong sector symmetry

$$\mathcal{G}_{\text{strong}} = U(2)_Q \times U(1)_Q \,, \tag{5.19}$$

with the three families of composite fermions transforming as $\mathcal{O}_{q,u,d} \in (\mathbf{2} \oplus \mathbf{1_Q})$. The matrices $\lambda_\psi$ can then be viewed as spurions of $\mathcal{G}_{\text{strong}} \times \mathcal{G}_{\text{quarks}}$ transforming as

$$\lambda_q \equiv \lambda_q^{(2)} \oplus \lambda_q^{(1)} \in (\overline{\mathbf{2}} \oplus \overline{\mathbf{1}}_Q, \mathbf{3}, \mathbf{1}, \mathbf{1}),$$ (5.20)
$$\lambda_u \equiv \lambda_u^{(2)} \oplus \lambda_u^{(1)} \in (\overline{\mathbf{2}} \oplus \overline{\mathbf{1}}_Q, \mathbf{1}, \mathbf{3}, \mathbf{1}),$$
$$\lambda_d \equiv \lambda_d^{(2)} \oplus \lambda_d^{(1)} \in (\overline{\mathbf{2}} \oplus \overline{\mathbf{1}}_Q, \mathbf{1}, \mathbf{1}, \mathbf{3}),$$

with $\mathbf{1}_Q$ indicating singlets of $U(2)_Q$ carrying the same $U(1)_Q$ charge. Furthermore the left-handed mixings $\lambda_q$ are postulated to respect the flavor subgroup $\mathcal{G}_F \subset \mathcal{G}_{\text{strong}} \times \mathcal{G}_{\text{quarks}}$

$$\mathcal{G}_F = U(2)_{Q+q} \times U(1)_{Q+q} \times U(3)_u \times U(3)_d.$$ (5.21)

These assumptions, combined with the requirement of reproducing the SM Yukawas, lead to the following explicit expression for the $\lambda_\psi$'s:

$$\text{pLU}: \quad \begin{cases} \lambda_q \sim \begin{pmatrix} \varepsilon_q & 0 \\ 0 & \varepsilon_q \\ 0 & 0 \end{pmatrix} g_* \oplus \begin{pmatrix} 0 \\ 0 \\ \varepsilon_{q3} \end{pmatrix} g_* \\[2em] \lambda_u \sim \frac{1}{\varepsilon_q}\begin{pmatrix} y_u & 0 \\ 0 & y_c \\ a^* y_c & b^* y_c \end{pmatrix} \oplus \frac{1}{\varepsilon_{q3}}\begin{pmatrix} 0 \\ 0 \\ y_t \end{pmatrix} \\[2em] \lambda_d \sim \frac{1}{\varepsilon_q}\begin{pmatrix} y_d & 0 \\ 0 & y_s \\ a'^* y_s & b'^* y_s \end{pmatrix} \widetilde{O}_d \oplus \frac{1}{\varepsilon_{q3}}\begin{pmatrix} 0 \\ 0 \\ y_b \end{pmatrix} \end{cases}$$ (5.22)

where

$$\frac{y_c}{g_*} \lesssim \varepsilon_q \lesssim 1, \qquad \frac{y_t}{g_*} \lesssim \varepsilon_{q3} \lesssim 1,$$ (5.23)

are real parameters. In the following, we will make the phenomenologically preferred choice $\varepsilon_q \lesssim \varepsilon_{q3}$. The explicit expression of $\lambda_q$ follows directly from our symmetry hypothesis, while the $\lambda_{u,d}$ can be put in the above form by appropriate field redefinitions. Specifically, we used $U(3)_{u,d}$ rotations to orient the $3 \times 1$ matrices $\lambda_{u,d}^{(1)}$ along the third family and the remaining $U(2)_u \times U(2)_{Q+q}$ to diagonalize the upper $2 \times 2$ block of $\lambda_u^{(2)}$. After that, we are only left with $U(2)_d$ freedom, and thus the upper $2 \times 2$ block of $\lambda_d^{(2)}$ can only be diagonalized on its left with a unitary $2 \times 2$ matrix $\widetilde{O}_d$ remaining on its right. Re-defining the phases of the fields we can make $\widetilde{O}_d$ completely real (and hence orthogonal) whereas the parameters $a, b, a', b'$ are in general complex. Their values, as well as that of the angle $\theta$ defining $\widetilde{O}_d$, are partially constrained by the requirement of reproducing the correct quark masses and CKM mixing angles. More precisely, out of the 5 real parameters $|a|, |b|, |a'|, |b'|, \theta$, three combinations are mapped into the CKM angles, and the two extra ones represent two physical angles beyond the SM. Instead, only one combination out of the four phases in $a, b, a', b'$ is needed to match the CKM phase, resulting in 3 additional physical phases on top of the SM one.

Notice that we could alternatively choose the phases of the fields so as to make either $\lambda_u$ or $\lambda_d$ in Eq. (5.22) fully real. For instance, we could make $a$ and $b$ real, in which case the

matrix $\widetilde{O}_d$ would be complex. In view of that, also in this scenario, quarks EDMs are purely induced by loop effects so as to involve both the $u$ and $d$ elementary fermions.

The SM Yukawas are of the form $Y_{u,d} \sim \lambda_q^{(2)}[\lambda_{u,d}^{(2)}]^\dagger + \lambda_q^{(1)}[\lambda_{u,d}^{(1)}]^\dagger$ and explicitly read

$$Y_u \sim \begin{pmatrix} y_u & 0 & ay_c \\ 0 & y_c & by_c \\ 0 & 0 & y_t \end{pmatrix}, \qquad Y_d \sim \widetilde{U}_d \begin{pmatrix} y_d & 0 & a'y_s \\ 0 & y_s & b'y_s \\ 0 & 0 & y_b \end{pmatrix}, \qquad \widetilde{U}_d \equiv \begin{pmatrix} \widetilde{O}_d^T & 0 \\ 0 & 1 \end{pmatrix}. \quad (5.24)$$

Following our conventions, they are diagonalized according to $Y_{u,d} = U_{u,d}\bar{Y}_{u,d}V_{u,d}^\dagger$, where $U_u = \mathbf{1} + O(y_c/y_t)$, $U_d = \widetilde{U}_d + O(y_s/y_b)$ (see Appendix B.2 for more details). The CKM matrix $V_{\mathrm{CKM}} = U_u^\dagger U_d$ is thus (we use $s_\theta = \sin\theta$ and $c_\theta = \cos\theta$ for brevity)

$$V_{\mathrm{CKM}} \sim \begin{pmatrix} c_\theta & s_\theta & \frac{y_s}{y_b}(a'c_\theta + b's_\theta) - \frac{y_c}{y_t}a \\ -s_\theta & c_\theta & \frac{y_s}{y_b}(b'c_\theta - a's_\theta) - \frac{y_c}{y_t}b \\ -\frac{y_s}{y_b}a'^* + \frac{y_c}{y_t}(-b^*s_\theta + a^*c_\theta) & -\frac{y_s}{y_b}b'^* + \frac{y_c}{y_t}(a^*s_\theta + b^*c_\theta) & 1 \end{pmatrix}$$

$$(5.25)$$

up to corrections of order $y_s^2/y_b^2$, $y_c^2/y_t^2$, and $y_s y_c/(y_b y_t)$. In order to reproduce the observed CKM structure quantitatively, we need

$$\theta \sim \lambda_C. \quad (5.26)$$

The parameters $a, b, a', b'$ are less constrained, though, but the choice with the least amount of fine-tuning is

$$|a'| \sim \lambda_C \qquad\qquad |a|, |b|, |b'| \sim 1 \quad (5.27)$$

as it guarantees that $V_{\mathrm{CKM}}^{13,23}$ are roughly of the correct order of magnitude without the need for any accidental cancellation. For concreteness, we will assume such structure in the following but we emphasize that taking $|a|, |a'|, |b|, |b'| \sim 1$ will not affect our conclusions qualitatively.

Like pRU, also Partial Left Universality shares some similarities with the $U(2)_{\mathrm{LC}}^3$ models of [44–46]. There are however important differences. essentially the same considerations made for pRU in Section 4.3 apply here as well.

### 5.2.2 Experimental constraints

**4-fermion operators**

Like for Left Universality, the four-fermion operators with the largest coefficients are expected to be $\mathcal{O}_{qq}^{(1)/(3)}$ in Eq. (5.7). The LHC constraint from dijets in Eq. (5.8) continues to hold, though now $\varepsilon_q$ refers to the compositeness of the two light families, which can be taken to be smaller than the one of the third family, $\varepsilon_{q3}$.

The lack of complete Left Universality, however, gives rise to flavor-violating transitions at relatively low order in the $\lambda_\psi$'s. In particular, $\Delta F = 2$ effects arise at tree-level. The most relevant ones occur in the $b, d$ sector and are mediated by the $\mathcal{O}_1$ operators of Eq. (B.43). The leading contribution to the Wilson coefficient is mediated by mixing to the third family and reads

$$C_1^{bd} \sim (U_d^\dagger \lambda_q^{(1)}[\lambda_q^{(1)}]^\dagger U_d)_{bd}^2/(g_*^2 m_*^2) \sim A^2 \lambda_C^6 g_*^2 \varepsilon_{q3}^4/m_*^2, \quad (5.28)$$

where we used $U_d = U_u V_{\text{CKM}}$ and the approximate expression for $U_u$ given in App. B.2. The experimental data then imply

$$m_* \gtrsim 10\, g_* \varepsilon_{q3}^2 \text{ TeV} .\tag{5.29}$$

The full list of $\Delta F = 2$ constraints is summarized in Tab. 14.

### EW vertices

The choice $\varepsilon_{q3} \gtrsim \varepsilon_q$, corresponding to a smaller degree of compositeness of the first two families, allows for a relaxation of some of the constraints previously found in LU. In particular the constraint in Eq. (5.14), $m_* \gtrsim 9.3\, g_* \varepsilon_q$ TeV, still holds and is always more constraining than that from dijets, shown in Eq. (5.8).

However, $\varepsilon_q$ now refers to the two light families and is allowed to be smaller than $\sim y_t/g_*$. The other bounds on anomalous $Z$ couplings are weaker except, possibly, for that on the anomalous $t_R$ coupling

$$m_* \gtrsim \frac{0.5}{\varepsilon_{q3}} \text{ TeV} ,\tag{5.30}$$

which is the same as Eq. (5.12) with $\varepsilon_q \to \varepsilon_{q3}$.

As concerns flavor violation, compared with LU, the misalignment of the third family induces new effects. Among them the most significant ones are $b \to s$ transitions, in particular the branching fraction $B_s \to \mu\mu$, mediated by the $\mathcal{O}_{10}$ operator. Its Wilson coefficient schematically reads

$$
\begin{aligned}
C_{10} &\sim \frac{2\sqrt{2}\pi^2}{G_F m_*^2 e^2} \frac{1}{V_{\text{CKM}}^{tb}(V_{\text{CKM}}^{ts})^*} \left[ U_d^\dagger (c_1 \lambda_q^{(1)} [\lambda_q^{(1)}]^\dagger + c_2 \lambda_q^{(2)} [\lambda_q^{(2)}]^\dagger) U_d \right]^{23} \\
&\sim \frac{2\sqrt{2}\pi^2}{G_F m_*^2} \frac{\varepsilon_{q3}^2 g_*^2}{e^2} .
\end{aligned}
\tag{5.31}
$$

and implies the bound

$$m_* \gtrsim (8.0 \div 10)\, g_* \varepsilon_{q3} \text{ TeV} ,\tag{5.32}$$

which rather strong, given $\varepsilon_{q3} \gtrsim y_t/g_*$, and never weaker than the bound in Eq. (5.29). Yet, like in previous cases, custodial protection can be invoked. That is, assuming the additional $P_{\text{LR}}$ and choosing appropriate representations for the composite fermions, e.g. those in Eq. (A.3), the bound in Eq. (5.32) is relaxed. The same transition is then dominantly controlled by the $\mathcal{O}_9$ operator (see Eq. (B.33)) and entails, like in previous cases, the additional suppression factor $\sim e^2/g_*^2$. The bound then reduces to

$$m_* \gtrsim (1.5 \div 4.3)\, \varepsilon_{q3} \text{ TeV} ,\tag{5.33}$$

which is always weaker than Eq. (5.29) in the allowed range $\varepsilon_{q3} \gtrsim y_t/g_*$.

### Dipoles

As in the previously studied scenarios, Partial Left Universality is free from tree-level electric dipole moments. That is because, as we discussed before, CP-violating phases can all be shifted to either the up or the down sector (as in Sec. 4.2.2), so that the physical CP-violating effects must involve both sectors. For EDMs, that can only happen at loop level, via the exchange of virtual elementary fermions. At 1-loop order, we get four independent contributions to up-type and down-type dipoles, namely

$$
\text{Im}[\widetilde{\mathcal{C}}_{u\gamma}] \sim \frac{1}{g_* m_*^2} \frac{e}{16\pi^2} \text{Im}\Big[ U_u^\dagger \Big( \lambda_q^{(2)} [\lambda_d^{(2)}]^\dagger \lambda_d^{(2)} [\lambda_u^{(2)}]^\dagger + r_u \lambda_q^{(1)} [\lambda_d^{(1)}]^\dagger \lambda_d^{(2)} [\lambda_u^{(2)}]^\dagger
$$
$$
+ r_u' \lambda_q^{(2)} [\lambda_d^{(2)}]^\dagger \lambda_d^{(1)} [\lambda_u^{(1)}]^\dagger + r_u'' \lambda_q^{(1)} [\lambda_d^{(1)}]^\dagger \lambda_d^{(1)} [\lambda_u^{(1)}]^\dagger \Big) V_u \Big] , \tag{5.34}
$$

$$
\text{Im}[\widetilde{\mathcal{C}}_{d\gamma}] \sim \frac{1}{g_* m_*^2} \frac{e}{16\pi^2} \text{Im}\Big[ U_d^\dagger \Big( \lambda_q^{(2)} [\lambda_u^{(2)}]^\dagger \lambda_u^{(2)} [\lambda_d^{(2)}]^\dagger + r_d \lambda_q^{(1)} [\lambda_u^{(1)}]^\dagger \lambda_u^{(2)} [\lambda_d^{(2)}]^\dagger
$$
$$
+ r_d' \lambda_q^{(2)} [\lambda_u^{(2)}]^\dagger \lambda_u^{(1)} [\lambda_d^{(1)}]^\dagger + r_d'' \lambda_q^{(1)} [\lambda_u^{(1)}]^\dagger \lambda_u^{(1)} [\lambda_d^{(1)}]^\dagger \Big) V_d \Big] . \tag{5.35}
$$

Up-type dipoles ($\propto \text{Im}[\widetilde{\mathcal{C}}_{u\gamma}^{11}]$) turn out to be much smaller than down-type dipoles ($\propto \text{Im}[\widetilde{\mathcal{C}}_{d\gamma}^{11}]$), so the most relevant contributions to the neutron EDM come, by far, from the latter.

Taking $c_d \sim 1$ in Eq. (B.50), the two dominant contributions from the $d$-quark dipole read

$$
\frac{1}{g_* m_*^2} \frac{e}{16\pi^2} \text{Im}\left[ U_d^\dagger \left( \lambda_q^{(1)} [\lambda_u^{(1)}]^\dagger \lambda_u^{(1)} [\lambda_d^{(1)}]^\dagger \right) V_d \right]^{11} \Rightarrow d_n|_d \sim \frac{e m_d}{m_*^2 \varepsilon_{q_3}^2} \frac{y_t^2}{8\pi^2} \lambda_C^3 \frac{y_s}{y_b} \text{Im}[a'^* A(1-\rho+i\eta)] ,
$$
$$
\tag{5.36}
$$

$$
\frac{1}{g_* m_*^2} \frac{e}{16\pi^2} \text{Im}\left[ U_d^\dagger \left( \lambda_q^{(1)} [\lambda_u^{(1)}]^\dagger \lambda_u^{(2)} [\lambda_d^{(2)}]^\dagger \right) V_d \right]^{11} \Rightarrow d_n|_d \sim \frac{e m_d}{m_*^2 \varepsilon_q^2} \frac{y_t y_c}{8\pi^2} \lambda_C^3 \text{Im}[a A(1-\rho+i\eta)] ,
$$
$$
\tag{5.37}
$$

Assuming for concreteness that $\text{Im}[a'] \sim |a'| \sim \lambda_C$,[22] as discussed around Eq. (5.27), these lead to

$$
m_* \gtrsim \frac{0.03}{\varepsilon_{q_3}} \text{ TeV} , \qquad m_* \gtrsim \frac{0.02}{\varepsilon_q} \text{ TeV} . \tag{5.38}
$$

Similar considerations can be made about the strange quark. The leading contributions arising from Eq. (5.35) are (see the estimate in Eq. (B.50))

$$
d_n|_s \sim \frac{c_s}{N_c} \sqrt{2} v \text{Im}[\widetilde{\mathcal{C}}_{d\gamma}]^{22} \sim \frac{c_s e m_s}{m_*^2 \varepsilon_{q_3}^2 N_c} \frac{y_t^2}{8\pi^2} A \lambda_C^2 \frac{y_s}{y_b} \text{Im}[b'^*] , \tag{5.39}
$$

$$
d_n|_s \sim \frac{c_s}{N_c} \sqrt{2} v \text{Im}[\widetilde{\mathcal{C}}_{d\gamma}]^{22} \sim \frac{c_s e m_s}{m_*^2 \varepsilon_q^2 N_c} \frac{y_t y_c}{8\pi^2} A \lambda_C^2 \text{Im}[b] . \tag{5.40}
$$

Numerically, we obtain

$$
m_* \gtrsim \sqrt{c_s} \frac{0.29}{\varepsilon_{q_3}} \text{ TeV} , \qquad m_* \gtrsim \sqrt{c_s} \frac{0.12}{\varepsilon_q} \text{ TeV} . \tag{5.41}
$$

---

[22] Yet, notice that taking $a' \sim 1$ only strengthens the first bound in Eq. (5.38) of a factor $\sim 1/\sqrt{\lambda_c} \sim 2$.

For $c_s \sim 1$ the strange-quark contributions to $d_n$ happen to be respectively a factor $\sim 130$ and $\sim 30$ larger than those of the $d$-quark. On the other hand, for the value $c_s \sim 0.01$ suggested by [61] the strange and down quark leads to comparable constraints.

Splitting the third family implies new effects also in flavor-violating dipoles. Specifically, on the basis of symmetry considerations alone, the operator $\mathcal{O}_7$ can be generated at tree-level (see Eq. (B.38)). Focusing on the transition $B \to X_s \gamma$, we find

$$C_7 \sim c_7 \frac{m_s}{m_b} \frac{4\sqrt{2}\pi^2 b'}{G_F V_{\text{CKM}}^{tb}(V_{\text{CKM}}^{ts})^* m_*^2} \,, \tag{5.42}$$

$C_7' \sim C_7 m_s/m_b$ is instead more suppressed. The experimental constraints [55] translate into a very significant absolute lower bound

$$m_* \gtrsim (4.9 \div 7.5)\,\text{TeV} \qquad (|c_7| = 1)\,. \tag{5.43}$$

This is however reduced if the UV physics is such that dipole interactions appear with at least a 1-loop factor $g_*^2/16\pi^2$ in Eq. (2.6). In that case, Eq. (5.43) is relaxed to a milder

$$m_* \gtrsim (0.39 \div 0.60)g_*\,\text{TeV} \qquad (|c_7| = g_*^2/16\pi^2)\,, \tag{5.44}$$

which is competitive with the constraint on the universal parameter $\mathcal{C}_H$ shown in Eq. (C.4).

Before concluding, notice that Eq. (5.35) also induces flavor-violating EW dipoles and in particular a non-zero $C_7$ that, similarly to Eq. (4.20), leads to

$$m_* \gtrsim \frac{0.45 \div 0.68}{\varepsilon_{q_3}}\,\text{TeV}\,, \tag{5.45}$$

which is comparable or at most mildly stronger than (5.30). Again, in the limit $\varepsilon_{q_3} \sim y_t/g_*$ this becomes competitive with the universal parameter $\mathcal{C}_H$ shown in Eq. (C.4).

### 5.2.3 Summary

Partial Left Universality improves over the LU scenario of Sections 5.1 by breaking the universality in the left-handed sector and allowing a separation between the compositeness of the light and the third generations of $q_L$. Taking $\varepsilon_q < \varepsilon_{q_3}$ we can relax the constraint from the unitarity of the CKM matrix in Eq. (5.14), which is by far the most stringent bound found in LU. However, the loss of universality also leads to new important constraints controlled by $\varepsilon_q$, $\varepsilon_{q_3}$, as well as new effects that are $\varepsilon_q, \varepsilon_{q_3}$-independent. Let us analyze them in turn.

First, while $\varepsilon_q \ll 1$ is favored by the constraints we just mentioned, a too small $\varepsilon_q$ is disfavored by the resulting large corrections to the neutron EDM, see (5.38) and (5.41). The optimal range of $\varepsilon_q$ actually depends on whether the strange quark contribution to $d_n$ in Eq. (B.50) is sizable ($c_s \sim 1$) or not ($c_s \sim 0.01$). In the end, for $0.05\sqrt{c_s} \lesssim \varepsilon_q^{\text{opt}} \lesssim 0.1$, all the constraints controlled by $\varepsilon_q$ are weaker than the universal ones, shown in Eq. (C.4).

Moving next to the effects controlled by $\varepsilon_{q_3}$, the strongest bound arises from $B_s \to \mu\mu$ and is shown in Eq. (5.32). Using $\varepsilon_{q_3} \gtrsim y_t/g_*$ that translates into

$$m_* \gtrsim (8.0 \div 10)\,g_* \varepsilon_{q_3}\,\text{TeV} \gtrsim (6.5 \div 8.1)\,\text{TeV}\,. \tag{5.46}$$

Unless further hypotheses are made the new physics is thus out of the reach of direct searches at the LHC. To improve the situation we can invoke the $P_{\text{LR}}$ symmetry. With appropriate

representations for the composite quarks under the custodial symmetry, as those in Eq. (A.3), the strong bound in Eq. (5.32) is removed and replaced with the milder Eq. (5.33). In that case, the main constraints controlled by $\varepsilon_{q3}$ are Eq. (5.29), from $\Delta F = 2$, and the comparable Eqs. (5.45) and (5.30), from respectively $B \to X_s \gamma$ and the $Z\bar{t}_R t_R$ coupling. The first of these three constraints grows with $\varepsilon_{q3}$ and is minimized by picking its smallest allowed value $\sim y_t/g_*$

$$m_* \gtrsim 10, g_* \varepsilon_{q3}^2 \text{ TeV} \gtrsim 6.6/g_* \text{ TeV} . \tag{5.47}$$

This bound is just slightly stronger than the flavor universal bound on $C_{2W}$ discussed around Eq. (C.4). The two other constraints, Eqs. (5.45) and (5.30), are instead enhanced at small $\varepsilon_{q3}$ (these effects are indeed enhanced by the compositeness of the right-handed quarks). There is thus an optimal choice of $\varepsilon_{q3}$ allowing for the weakest constraint on $m_*$. Using conservatively the largest of the two values in Eq. (5.45), we find $\varepsilon_{q3}^{\text{opt}} \sim 0.4/g_*^{1/3}$ for $g_* \gtrsim 3$ and simply $\varepsilon_{q3}^{\text{opt}} = y_t/g_*$ for $g_* \lesssim 3$. Then for $g_* \gtrsim 3$ the optimal bound reads

$$m_* \gtrsim 2.5 \times \left(\frac{g_*}{3}\right)^{1/3} \text{ TeV} , \tag{5.48}$$

while for $g_* \lesssim 3$ it coincides with Eq. (5.47). Thus with $P_{\text{LR}}$ and an optimal choice for the couplings, the bound in Eq. (5.46) can be relaxed to $m_* \gtrsim 2.5$ TeV, at the rough edge of the LHC reach in direct searches.

Finally, pLU suffers from novel flavor-violating effects that are independent of $\varepsilon_q, \varepsilon_{q3}$. The most relevant ones are mediated by dipoles. We found that, when these operators are generated at tree-level, new physics is pushed beyond the direct LHC reach: $m_* \gtrsim (4.9 \div 7.5)$ TeV as shown in Eq. (5.43)) and in Fig. 9. Again, also this constraint can be relaxed, by assuming dipole operators arise at $O(g_*^2/16\pi^2)$. In that case (see Eq. (5.44)) the constraint is weaker than that from $\mathcal{C}_H$ in Eq. (C.4).

In conclusion, in the generic pLU scenario, $m_*$ may still lie within the direct reach of FCC-hh, but somewhat beyond that of the LHC (see Eqs. (5.46) and (5.43)). However in the special cases where dipoles are generated at 1-loop and $P_{\text{LR}}$ protects the $Z$ couplings, $m_*$ can be lowered around the edge of the LHC direct reach, see Eq. (5.48). More precisely the lowest $m_*$ is attained for $\varepsilon_{q3}^{\text{opt}} \sim y_t/g_*$, $0.05\sqrt{c_s} \lesssim \varepsilon_q^{\text{opt}} \lesssim 0.1$, and $g_* \sim 3$ (see the green region in Fig. 9). The strength of the assumptions of the optimal scenario is not unlike in the case of pRU. However, there is perhaps some reason to consider the present scenario structurally less plausible. Indeed, the preferred choice $\varepsilon_{q3}^{\text{opt}} \sim y_t/g_*$ corresponds to a maximally composite right-handed top, i.e. $\lambda_u^{(1)}/g_* \sim 1$. That is contrary to the naive expectation that the left handed quarks should mix more strongly, given their quasi-universal mixing to the strong sector.

Alternative scenarios of Partial LU may be realized within the framework of Eq. (3.1) with two partners of $q_L$, instead of Eq. (5.1). Nevertheless, we expect the associated phenomenology to be qualitatively similar in all such constructions because, as opposed to models with universality in the right-handed sector, here there appears to be no structural way to depart from universality while simultaneously remaining sufficiently close to MFV in the down sector.

# 6  Lepton Sector

Before presenting our conclusions we would like to investigate the extension of our program to leptons. This section will be very brief and qualitative. A more dedicated study is beyond

the scope of the present paper.

The crucial difference between leptons and quarks is that the generation of the neutrino masses requires the presence of additional interactions violating the overall lepton number. The simplest option is that these arise from the bilinear mixing of Eq. (2.2), which below the scale $m_*$ maps to the usual dimension-5 $\ell\ell HH$ SM operator. A more sophisticated option is that the strong sector itself is endowed with some tiny intrinsic lepton number violation, which again gives rise to $\ell\ell HH$ through partial compositeness [25, 62, 63]. In the first scenario, the resulting lepton flavor-violating effects enjoy the same suppression as the neutrino masses and can therefore be neglected (except of course when considering neutrino oscillations). Flavor violation at a level relevant for laboratory experiments is then purely due to the couplings underlying the charged lepton yukawas. In the second scenario, instead, more options are available. For instance, the masses of neutrinos and of charged leptons may or may not originate from the same $\ell\mathcal{O}$ mixing. In the former case, again all relevant sources of lepton flavor violation observable in the laboratory are associated with the dynamics underlying the charged lepton masses, while in the latter more sources are active, corresponding to additional model dependence. In any case, it seems fair to conclude that, while the mechanism underlying neutrino masses implies in principle extra model dependence, in the simplest options all relevant flavor violation (beyond neutrino oscillations) is associated with charged lepton masses. In what follows we work under that assumption.

Purely focusing on charged leptons, we can follow the same logic as in the quark sector. For concreteness, we consider only the minimal scenario where three families of operators $\mathcal{O}_\ell$ mix with the $\ell_L$ and three families of $\mathcal{O}_e$ mix with the $e_R$

$$\mathcal{L}_{\text{mix}} = \lambda_\ell^{ia}\bar{\ell}_L^i\mathcal{O}_\ell^a + \lambda_e^{ia}\bar{e}_R^i\mathcal{O}_e^a. \tag{6.1}$$

As neutrino masses are being neglected, the notion of minimal flavor violation is unambiguously defined just within the charged sector. We will proceed with a discussion of MFV and we then move to less symmetric scenarios.

### Lepton universality

Postulating that the strong sector has a $U(3)_L$ flavor symmetry, two realizations of MFV can be identified

- **Right-lepton universality**, implemented by taking $\lambda_e^{ia} = g_*\varepsilon_e\delta^{ia}$ and $\lambda_\ell^{ia} \sim Y_e^{ia}/\varepsilon_e$.

- **Left-lepton universality**, implemented by taking $\lambda_\ell^{ia} = g_*\varepsilon_\ell\delta^{ia}$ and $\lambda_e^{ia} \sim (Y_e^\dagger)^{ia}/\varepsilon_\ell$.

Either choice preserves individual lepton numbers, in such a way that no lepton flavor violation is generated other than the negligible one associated with neutrino masses.

As concerns flavor-universal effects, important constraints currently arise from four-fermion operators. The relevant ones are

$$[\mathcal{O}_{\ell\ell}]^{ijkl} = (\bar{\ell}_L^i\gamma^\mu\ell_L^j)(\bar{\ell}_L^k\gamma_\mu\ell_L^l), \qquad [\mathcal{O}_{ee}]^{ijkl} = (\bar{e}_R^i\gamma^\mu e_R^j)(\bar{e}_R^k\gamma_\mu e_R^l), \tag{6.2}$$

respectively for right- and left-universality, with expected coefficients $g_*^2\varepsilon_e^4/m_*^2$ and $g_*^2\varepsilon_\ell^4/m_*^2$. The strongest constraints come from electron processes. Taking the current experimental bounds from [64], i.e. $\widetilde{\mathcal{C}}_{ee}^{1111}, \widetilde{\mathcal{C}}_{\ell\ell}^{1111} < 0.066\,\text{TeV}^{-2}$, we get

$$m_* \gtrsim 3.9\,g_*\varepsilon^2\,\text{TeV}, \tag{6.3}$$

with $\varepsilon$ standing for either $\varepsilon_e$ or $\varepsilon_\ell$ in respectively Right- or Left-lepton Universality.

Other very significant flavor-universal effects are the corrections to the couplings of vector bosons to leptons. These are generated by operators $\mathcal{O}_{H\ell}^{(1)}$, $\mathcal{O}_{H\ell}^{(3)}$ and $\mathcal{O}_{He}$, which, in an obvious notation, are the analogs of the operators defined in Table 4 for the quark sector. These operators generically appear with coefficients of order $\lambda^2/m_*^2$, even though, with proper choices of the quantum numbers of $\mathcal{O}_\ell$ and $\mathcal{O}_e$, the appearance of the corresponding custodial breaking couplings can be avoided. Disregarding momentarily this option, the LEP/SLC data then give roughly the generic constraint $\lambda^2 v^2/m_*^2 \lesssim 10^{-3}$. For both the Left- and the Right-lepton universal case this constraint then implies

$$m_* \gtrsim 10^{\frac{3}{2}} (g_* v \varepsilon) \text{ (universal mixing)}, \qquad\qquad m_* \gtrsim 10^{\frac{3}{2}} (y_\tau v/\varepsilon) \text{ (tau mixing)}, \qquad (6.4)$$

where, again, $\varepsilon$ stands either for $\varepsilon_e$ or $\varepsilon_\ell$ in respectively Right- or Left-lepton Universality. This last bound is always stronger than Eq. (6.3). The "optimal" scenario allowing the lowest $m_*$ is obtained for $\varepsilon_{\ell,e} \sim \sqrt{y_\tau/g_*} \sim 0.1/\sqrt{g_*}$, where the two bounds coincide with

$$m_* \gtrsim 10^{3/2}\sqrt{y_\tau g_*}\, v \simeq 0.8 \sqrt{g_*} \text{ TeV}. \qquad (6.5)$$

This choice of parameters allows $m_*$ as low as $1 \div 2$ TeV in the fiducial range of $g_*$.

Alternatively, notice that one of the two bounds in Eq. (6.4) can be relaxed by a proper choice of the custodial quantum numbers for $\mathcal{O}_{\ell,e}$, in complete analogy to what discussed in Appendix A.1 for quarks. For instance, in the case of Right-lepton Universality one can decide to relax the "universal mixing" in Eq. (6.4) assuming either the representation $\mathcal{O}_e \in (\mathbf{1},\mathbf{1})_{-1}$ or $\mathcal{O}_e \in (\mathbf{1},\mathbf{3})_{-1}$ under the custodial $SO(4)$. With this choice the resulting bounds are controlled dominantly by the "tau mixing" in Eq. (6.4), which prefers large $\varepsilon_e$, and Eq. (6.3) which pushes towards small $\varepsilon_e$. It is then worth asking what is the maximum allowed value for $\varepsilon_e$ in this setup. Taking $0.03 \lesssim \varepsilon_e \lesssim 0.5$ those bounds are nevertheless below those from the purely bosonic operators. Completely analogous considerations apply to the case of Left Universality, where now it is the choice $\mathcal{O}_\ell \in (\mathbf{2},\mathbf{2})_0$ that allows to relax the "Unversal mixing" bound in Eq. (6.4).

As concerns the bounds from other observables, one finds them to be negligible. For instance, contributions to the electron EDMs are proportional either to powers of the neutrino masses or to flavor-invariant CP-odd combinations of the quark mixings. In either case, the result is utterly small. We thus conclude that MFV in the lepton sector is perfectly compatible with new physics at the TeV scale and $\varepsilon = O(1)$.

## Partial lepton universality

The safeness of the MFV scenario invites the exploration of alternative scenarios with a smaller symmetry, i.e. with only partial universality (see also [47]). Indeed, as we will now show, these other scenarios become rather quickly problematic. For concreteness, we shall mostly focus on the leptonic version of pRU, but analogous considerations apply to models with Partial LU. Leptonic Partial RU assumes the strong sector has a $U(2)_L \times U(1)_L$ symmetry, which, combined with the accidental symmetry of the elementary lepton sector, results in an overall $U(3)_\ell \times U(3)_e \times U(2)_L \times U(1)_L$. This symmetry is in turn explicitly broken first down to $U(3)_\ell \times U(2)_{L+e} \times U(1)_{L+e}$ by $\lambda_e$ and secondly down to $U(1)_Y \times U(1)_{L+\ell+e}$ (hypercharge and

total lepton number) by $\lambda_\ell$:

$$
\text{(lept.) pRU :}
\begin{cases}
\lambda_e \sim \begin{pmatrix} \varepsilon_e & 0 \\ 0 & \varepsilon_e \\ 0 & 0 \end{pmatrix} g_* \oplus \begin{pmatrix} 0 \\ 0 \\ \varepsilon_{e_3} \end{pmatrix} g_* \\[20pt]
\lambda_\ell \sim \dfrac{1}{\varepsilon_e} \begin{pmatrix} y_e & 0 \\ 0 & y_\mu \\ a_\ell y_\mu & b_\ell y_\mu \end{pmatrix} \oplus \dfrac{1}{\varepsilon_{e_3}} \begin{pmatrix} 0 \\ 0 \\ y_\tau \end{pmatrix}.
\end{cases}
\tag{6.6}
$$

By a transformation analogous to Eq. (4.35) and Eq. (4.36) one can take $a_\ell, b_\ell$ real and move all the phases to the neutrino sector, effectively secluding CP violation. This robustly avoids any constraint from the electron EDM. The sources of CP-violation available in the quark sector can only affect such observable via flavor-invariant combinations of the couplings $\lambda_{q,u,d}$, which can only arise via multi-loop diagrams involving both up and down sectors and are thus negligibly small.

Flavor violation provides instead important constraints, with the most significant ones from $\mu \leftrightarrow e$ transitions. Consider indeed the coefficient of dipole interactions. In this model, they are proportional to

$$
\left\{ U_e^\dagger \lambda_\ell^{(1)} [\lambda_e^{(1)}]^\dagger V_e \right\}^{ij} = g_* y_\tau [U_e^*]^{3i} V_e^{3j} ,
\tag{6.7}
$$

with $U_e$ and $V_e$ the matrices that diagonalize the charged lepton Yukawa: $U_e^\dagger Y_e V_e = \widetilde{Y}_e$. Using approximate expressions analogous to those of Appendix B.2 we find that the dominant $\mu \leftrightarrow e$ transition is given by the $i = 2, j = 1$ entry, corresponding to the helicity amplitude $\mu_L \to e_R \gamma$. Observing that $U_e^{32} \sim b_\ell (y_\mu/y_\tau)^2$ and $V_e^{31} \sim a_\ell y_\mu/y_\tau$, the Wilson coefficient (see Eq. (B.41)) of the dipole operator is estimated to be

$$
[\widetilde{\mathcal{C}}_{e\gamma}]^{21} \sim ce(a_\ell b_\ell^*) \frac{y_\tau}{m_*^2} (y_\mu/y_\tau)^3,
\tag{6.8}
$$

with either $c = O(1)$ or, possibly, $c \sim g_*^2/16\pi^2$ in UV completions where dipoles arise at 1-loop (see discussion in Sections 2). For generic $a_\ell, b_\ell = O(1)$ this result is smaller than what is expected in anarchic scenarios, where one has $\widetilde{\mathcal{C}}_{e\gamma}^{21} \sim ce\sqrt{y_e y_\mu}/m_*^2$, again with $c$ either $O(1)$ or, possibly, $O(g_*^2/16\pi^2)$.[23] In any case, for $a_\ell b_\ell \sim 1$, Eq. (6.8) does *not* feature enough suppression to allow $m_* \sim$ TeV. More precisely, the current best bound $\text{BR}(\mu \to e\gamma) < 3.1 \times 10^{-13}$ [15] translates roughly into

$$
m_* \gtrsim 60 \,\text{TeV} \ (c \sim 1), \qquad\qquad m_* \gtrsim 5 \, g_* \,\text{TeV} \ (c \sim \frac{g_*^2}{16\pi^2}).
\tag{6.9}
$$

Notice that even with a loop suppressed dipole the constraint is still much stronger than those from bosonic operators. A similar result holds for the scenario of partial Left-lepton universality, with the only difference that now the leading constraint is due to the other helicity amplitude, i.e. $\mu_R \to e_L \gamma$. We conclude that once one deviates from the maximally symmetric MFV scenario in the lepton sector, a less generic structure for the flavor breaking mixings

---

[23]It is however accidentally comparable to the result in the models with $U(1)^3$ flavor symmetry of Ref. [25], where $[\widetilde{\mathcal{C}}_{e\gamma}]^{21} \sim ce y_e/m_*^2$.

is necessary in order to let $m_*$ in the range of LHC searches. In the case of partial Right-lepton Universality which we have considered more explicitly, this concretely corresponds to assuming $a_\ell b_\ell^* \ll 1$. This choice can indeed, and expectedly, be motivated by symmetry as $a_\ell \to 0$ and $b_\ell \to 0$ control the conservation of respectively the first and second generation individual lepton numbers. In particular $a_\ell \sim m_e/m_\mu$ and $b_\ell \sim 1$, in which case all $U(1)_e$ breaking effects are consistently controlled by $\lambda_e,$[24] one can indeed lower $m_*$ down to the TeV. The same considerations apply to the case of partial Left-lepton Universality.

To compare, a similar analysis of $\tau \to \mu\gamma$ in leptonic pRU gives a dipole coefficient

$$[\widetilde{\mathcal{C}}_{e\gamma}]^{32} \sim c\,e b_\ell \frac{y_\tau}{m_*^2}(y_\mu/y_\tau), \tag{6.10}$$

The bound $|[\widetilde{\mathcal{C}}_{e\gamma}]^{32}| \lesssim 2.7\times10^{-6}/\,\mathrm{TeV}^2$ [25] reads $m_* \gtrsim \sqrt{cb_\ell}\,8$ TeV. Again the same qualitative conclusions also apply to pLU.

Constraints from $\mu \to eee$ and $\mu \to e$ conversions in nuclei are currently less relevant unless the tau is significantly composite, though that might change in the future. To illustrate this point we analyze $\mu \to e$ conversions in gold, as with $\mathrm{BR}(\mu\,\mathrm{Au} \to e\,\mathrm{Au}) < 7 \times 10^{-13}$ [65] it is at present slightly more stringent. In general, those processes are dominantly induced by the off-diagonal entries of the same $\mathcal{O}_{H\ell}^{(1)}$, $\mathcal{O}_{H\ell}^{(3)}$, $\mathcal{O}_{He}$ mentioned earlier. In leptonic pRU the largest rate is due to $\mathcal{O}_{He}$, whose coefficient scales as

$$[\widetilde{\mathcal{C}}_{He}]^{21} \sim \frac{1}{m_*^2}\left\{ V_e^\dagger \lambda_e^{(1)}[\lambda_e^{(1)}]^\dagger V_e \right\}^{21} = \frac{g_*^2}{m_*^2}\varepsilon_{e_3}^2 [V_e^*]^{32}V_e^{31} \sim \frac{g_*^2}{m_*^2}\varepsilon_{e_3}^2 a_\ell b_\ell^* \frac{y_\mu^2}{y_\tau^2}. \tag{6.11}$$

In models with leptonic Partial Left Universality, $\mu \to e$ conversions in nuclei are instead mainly mediated by $\mathcal{O}_{H\ell}^{(1,3)}$, but completely analogous considerations can be made. Imposing the current best bound $|\widetilde{\mathcal{C}}_{He}^{21}| \lesssim 4.9 \times 10^{-6}/\mathrm{TeV}^2$ (for pRU) and $|\widetilde{\mathcal{C}}_{H\ell}^{21}| \lesssim 4.9 \times 10^{-6}/\mathrm{TeV}^2$ (for pLU) [25] we have

$$m_* \gtrsim 27\, g_*\varepsilon\sqrt{|a_\ell b_\ell^*|}\ \mathrm{TeV}\,, \tag{6.12}$$

where $\varepsilon$ either stands for $\varepsilon_{e_3}$ in leptonic pRU or $\varepsilon_{\ell_3}$ in pLU, and $a_\ell, b_\ell$ denote the dimensionless numbers that appear in Eq. (6.6) or the corresponding expressions in pLU. Eq. (6.12) represents a significant constraint on the mass scale of models with Partial Universality in the lepton sector and maximally composite tau's, i.e. $\varepsilon \sim 1$. Yet, it is evident that taking the value $\varepsilon \sim \sqrt{y_\tau/g_*} \ll 1$ used to minimize Eq. (6.4), or even smaller, those observables become less constraining than Eq. (6.9) even when the latter features a loop suppression. The $\mu \to e$ rate can be further relaxed invoking the same $P_{\mathrm{LR}}$ protection anticipated when discussing the $Z$-couplings. In fact, with the same choices of custodial representation for the composite partners mentioned below (6.5), we can relax (6.12) by a factor $\sim e/g_*$ (see below Eq. (4.17)). This would relax (6.12) down to $m_* \gtrsim 11\,\varepsilon\sqrt{|a_\ell b_\ell^*|}$ TeV, which nevertheless is still significant with $\varepsilon \sim 1$.

Importantly, $\mu \to e$ conversions might soon become competitive even at relatively small $\varepsilon$, thanks to the planned experimental updates. Indeed, the sensitivity on the branching ratio for both $\mu \to e$ conversions in nuclei [13] and $\mu \to 3e$ [14] is expected to improve by about a factor $\gtrsim 10^4$, whereas the sensitivity to $\mathrm{BR}(\mu \to e\gamma)$ will improve by $\sim 5$ [66]. If this is indeed the case, and no discovery is made, Eq. (6.12) will be replaced by

$$m_* \gtrsim (200 \div 300)\, g_*\varepsilon\sqrt{|a_\ell b_\ell^*|}\ \mathrm{TeV} \qquad \text{(projected)}. \tag{6.13}$$

---

[24]Correspondingly $[\lambda_\ell^{(2)}]^\dagger \lambda_\ell^{(2)}$ has one eigenvalue of order $y_e^2/\varepsilon_e^2$ and another of order $y_\mu^2/\varepsilon_e^2$.

Taking the value $\varepsilon \sim \sqrt{y_\tau/g_*}$ that minimizes the constraints from the anomalous $Z$ couplings, the bound in Eq. (6.13) reads $m_* \gtrsim (20 \div 30) \sqrt{g_*|a_l b_l^*|}$ TeV. Interestingly, for any $g_* \lesssim 4\pi$ that is stronger than the one coming from the projected sensitivity to $\mu \to e\gamma$ in models with dipoles generated at 1-loop, which reads $m_* \gtrsim 8\,g_*$ TeV (namely the projected version of Eq. (6.9)). We thus see that in the near future, $\mu \to e$ conversions will be able to set the strongest constraint on the new physics scale when $\varepsilon \gtrsim \sqrt{y_\tau/g_*}$. However, in regions of the parameter space with smaller $\varepsilon$ its relevance relative to $\mu \to e\gamma$ drastically reduces. In scenarios with $P_{\mathrm{LR}}$ protection for the $Z$-couplings, the projected bound (6.13) would become $m_* \gtrsim (50 \div 150)\,\varepsilon\sqrt{|a_\ell b_\ell^*|}$ TeV and thus remain a non-negligible constraint only for sizable mixings.

In conclusion, in models with partial universality, $\mu \to e\gamma$ is and will remain an important and structurally robust probe of non-MFV models. Yet, $\mu \to e$ conversions will provide a complementary probe of scenarios with $\varepsilon \gtrsim \sqrt{y_\tau/g_*}$.

# 7   Summary and Prospects

The goal of this study has been to classify and explore the flavor physics options in the Strongly Interacting Light Higgs (SILH) scenario [24]. While aiming at being as systematic as possible, we focused on models where the Yukawas are generated by partial fermion compositeness. We based our analysis on a generic low-energy effective Lagrangian descending from basic and plausible assumptions of symmetry and dynamics. Concretely, these assumptions (spelled out in Section 2) determine power counting rules for the Wilson coefficients that fix their dependence on the fundamental couplings and mass scales. This dependence is fixed up to expectedly $O(1)$ unknown coefficients given we are parametrizing a general scenario rather than a full-fledged and specific QFT.[25] With a corresponding $O(1)$ *squiggliness*, our approach thus allows to correlate the reach of direct searches at the energy frontier and that of indirect searches in flavor, electroweak, and Higgs physics. In particular, each different flavor physics hypothesis is associated with a lower bound (and indirect reach) on the scale of new physics $m_*$. Thus, besides its inner structural plausibility, each different flavor hypothesis can be evaluated conceptually, as concerns naturalness, and practically, as concerns direct discoverability at the LHC or FCC. The possibility to make correlations, and produce a *story*, is an obvious advantage of a perspective based on concrete hypotheses, like ours, compared to an approach based instead on a structureless SMEFT.

The least structured, hence most generic, realization of partial compositeness is given by the so-called flavor anarchy scenario. That is also arguably the most attractive scenario as it allows for a dynamical explanation of the flavor structure through RG evolution. As it is well known, however, in flavor anarchy the new physics scale $m_*$ is strongly constrained and especially so in the lepton sector. Nonetheless, we believe its structural simplicity, still makes flavor anarchy a scenario worth of consideration, at least when limited to the quark sector. Our analysis of the possible scenarios indeed starts, in Section 2.3, with a reassessment of flavor anarchy in the quark sector. There we also fill some holes present in the literature.

The state of affairs with flavor anarchy compels us to consider other scenarios where suit-

---

[25]Like for all models of Composite Higgs, all the scenarios we study admit explicit realizations through warped compactifications. We do not expect these realizations to provide significantly stronger parameter correlations, or stricter predictions, than those implied by our generic EFT. See nonetheless the remarks in footnote 27.

able symmetries control flavor and CP violating observables thus permitting lower values of $m_*$. In these scenarios, we trade the ability to dynamically explain the flavor structure with the possibility to lower $m_*$ in a range that is both more compatible with naturalness and more amenable to direct exploration at present and future colliders. The bulk of our paper consists of a systematic classification and study of flavor symmetry models based on partial compositeness. In order to control the constraints from EDMs, throughout our study we have also assumed CP is an exact symmetry of the composite sector, which is explicitly, and arbitrarily, broken by the elementary-composite mixings that define partial compositeness. We started our exploration from models with the maximal possible flavor symmetry, corresponding to the so-called Minimal Flavor Violation (MFV), of which we identified the five possible realizations shown in Table 1. These models offer the strongest protection on flavor and CP violating observables, but their rigid structure implies stronger constraints from flavor-preserving observables that push $m_*$ at the margin of the direct reach of the LHC. Departing from MFV we have then explored scenarios with increasingly smaller flavor symmetry groups. Our findings are reported in detail below. One main result is that the scenarios presently allowing for the lowest new physics scale are those that strike a balance between the extreme rigidity of MFV while still providing significant protection for flavor and CP violating processes.

Before proceeding to a discussion of our results, two comments are in order as concerns the methodology of our analysis and its relation to previous work. Our exploration of the space of flavor symmetry hypotheses, starting from the highest symmetry and incrementally reducing it, is systematic in practice but not in the strict sense. Indeed, analyzing each and every symmetry group option and scenario would not have just been virtually impossible, but also not more informative. The several scenarios we have analyzed practically cover the full range of possible phenomenologies. In particular, they include the scenarios that optimize the present constraints on $m_*$ as well as those with less symmetry and hence stronger constraints. Our study thus allows a perspective on how special the optimal scenarios are and or how quickly one falls into the class of less structured but more constrained models akin to flavor anarchy. A number of the models we have studied, the optimal ones in particular, are not new and have previously appeared in the literature.[26] Ref. [67] in fact already offered an analysis based on the SILH hypotheses and somewhat in the same spirit as ours, albeit more limited in scope. In view of all that we cannot and do not want to claim great originality, in particular from the model building perspective. What instead distinguishes our study is its high degree of systematicity on the front of flavor scenarios, its phenomenological comprehensiveness and the broad perspective it therefore offers. In particular, each scenario has been analyzed under all the possible flavor independent structural hypotheses. Those in particular concern custodial symmetries ($SO(4)$ and $P_{\rm LR}$) and the possible loop suppression of dipole operators (which does arise in holographic realization, but may not in general), which importantly affect sensitivity and constraints on $m_*$. Moreover, we have correlated the reach from flavor and CP observables with that from all other LHC searches and, at a more qualitative level, with that of the FCC-hh.[27] Our study then provides a map of the Strongly Interacting Light Higgs scenario with

---

[26]More precisely the puRU model was discussed in [57] while the $U(2)^2$ models of [44–46] practically coincide with our pRU and pLU models with a further restriction of their parameters. Those studies however did not investigate in detail EDMs nor loop-induced $b \to s$ transitions. Furthermore, they did not realize the crucial role played by $P_{\rm LR}$ in suppressing the otherwise very constraining $\Delta F = 1$ observables.

[27]A more quantitative study on the sensitivity projections of future colliders is outside the scope of the present paper and left for future work. In particular, it would be interesting to frame our analysis in a more defined framework (e.g. holographic realizations or simplified models as in [29]), aiming at a more concrete

| Label | Observable | RU | puRU | pRU | LU | pLU |
|---|---|---|---|---|---|---|
| A | $pp \to jj$ | ✓ | | | | |
| B | $\Delta F = 2\,(B_d)$ | ✓ | ✓ | ✓ | | ✓ |
| C | $B_s \to \mu^+\mu^-$ | ✓ | ✓ | ✓ | | ✓ |
| D | nEDM | | | (✓) | | |
| E | $B^0 \to K^{*0}e^+e^-\ (C_7')$ | | | ✓ | | |
| F | $B \to X_s\gamma\ (C_7)$ | | | | | ✓ |
| G | $W$-coupling | | | | ✓ | |

Table 2: Most constraining observables for the models shown in Figs. 5-9. A checkmark indicates that the observable appears in the corresponding figure. When the checkmark is in parenthesis, the corresponding line appears only in the projection. Less relevant effects are discussed in the main text.

partial flavor compositeness and a guide to its present and future experimental exploration. The same approach can obviously be applied under other model building hypotheses. For instance, remaining within the SILH scenario, one may consider alternative flavor dynamics or relax the hypothesis of a single overall scale and coupling (e.g. [18, 19, 21, 25]). The recent study in Ref. [68] offers instead an example based on a concrete but different class of UV flavor models. Among recent related studies we should also mention Ref. [32], similar to ours as concerns the breadth of its phenomenological analysis, but based on hypotheses that only partially overlap with ours, which are derived from a well defined UV picture. Ref. [32] interestingly points out the impact FCC-ee would have on flavor physics, an aspect which we did not investigate and which would be clearly worth considering in future work.

## 7.1 Main results

Besides reassessing the constraints on flavor anarchy, we studied in detail five different flavor symmetry scenarios, which we consider to be the most representative. The studied scenarios offer indeed a rather complete overview of the relation between flavor hypotheses and overall phenomenology. Two of them, Right Universality (RU) and Left Universality (LU), are possible realizations of MFV. The other three, Partial Right Up Universality (puRU), Partial Right Universality (pRU), and Partial Left Universality (pLU), are scenarios with non-maximal flavor symmetry. On the one hand, the maximally symmetric cases RU and LU cope quite well (or even remarkably well in the case of LU) with flavor observables, but are strongly constrained by flavor-preserving observables. On the other hand, the cases with partial universality (puRU, pRU, and pLU) are principally constrained by flavor-violating processes.

The most important experimental constraints on the five scenarios are summarized in the left panels of Figs. 5, 6, 7, 8, 9. The most relevant observables for the various constraints in the Figures are pinpointed in Tab. 2. Details of the current bounds are presented in Appendix C and are discussed in dedicated sections in the main text. We find it convenient to outline the main lessons implied by our study in a number of bullet points. Future prospects for the next ten or twenty years are discussed subsequently and summarized in the right panels of the figures.

---

correlation between our results and direct searches.

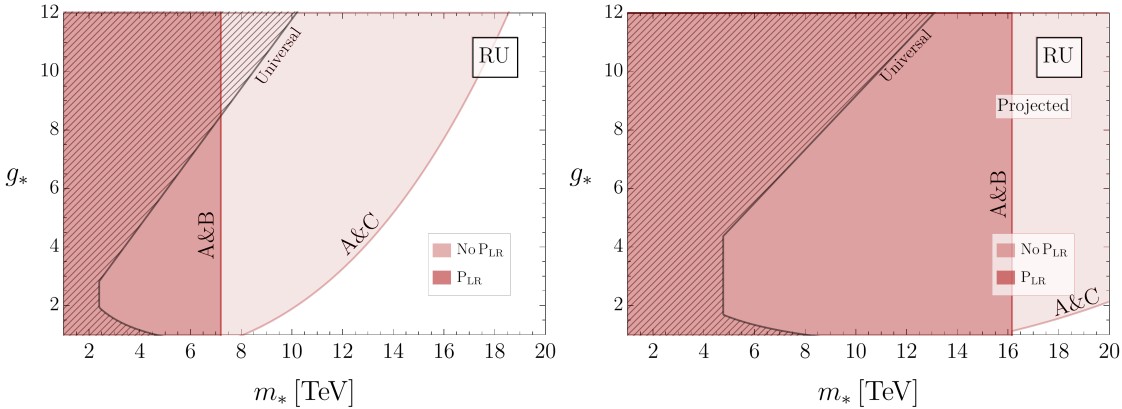

Figure 5: Leading current constraints (left) and projected sensitivity (right) in models with Right Universality (RU). The compositeness parameters $\varepsilon_\psi^{\mathrm{opt}}$ are chosen in the optimal range discussed in the summary of each model to minimize the constraint (or sensitivity) on $m_*$ for each $g_*$. See Sec. 4.1.3 for more details. The colored region is constrained by the observable indicated by the associated letters. The legend can be found in Tab. 2. The "Universal" shaded region is obtained from the bosonic flavor-universal observables discussed in App. C.1. In all our plots MFV is assumed in the leptonic sector.

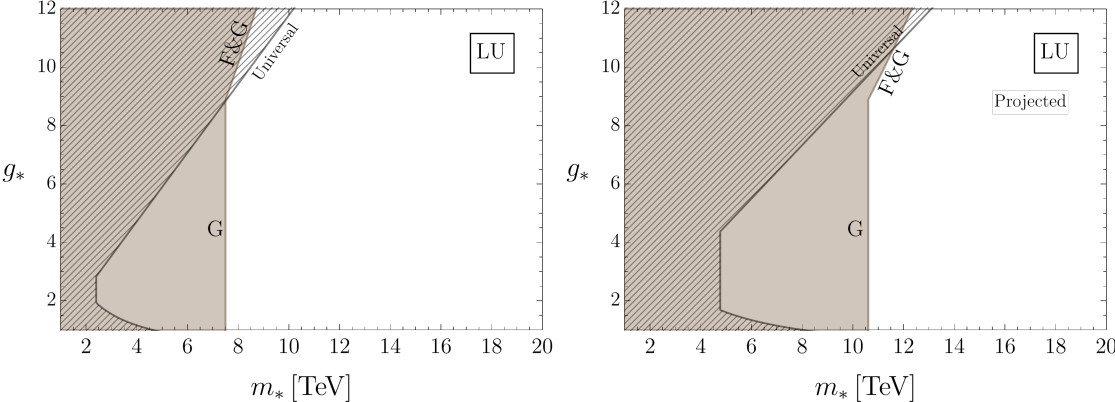

Figure 6: Same as Fig. 5 for models with Left Universality (LU). See Sec. 5.1.3 for more details.

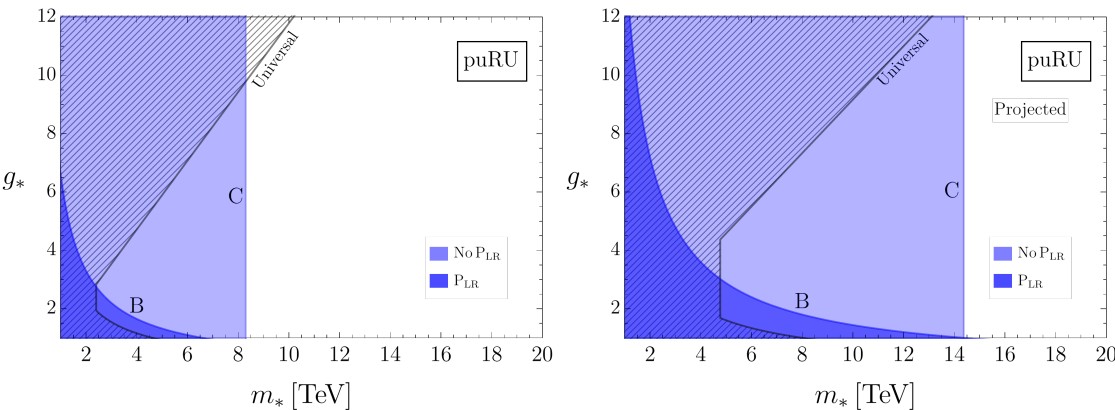

Figure 7: Same as Fig. 5 for models with Partial Up-Right Universality (puRU). See Sec. 4.2.4 for more details.

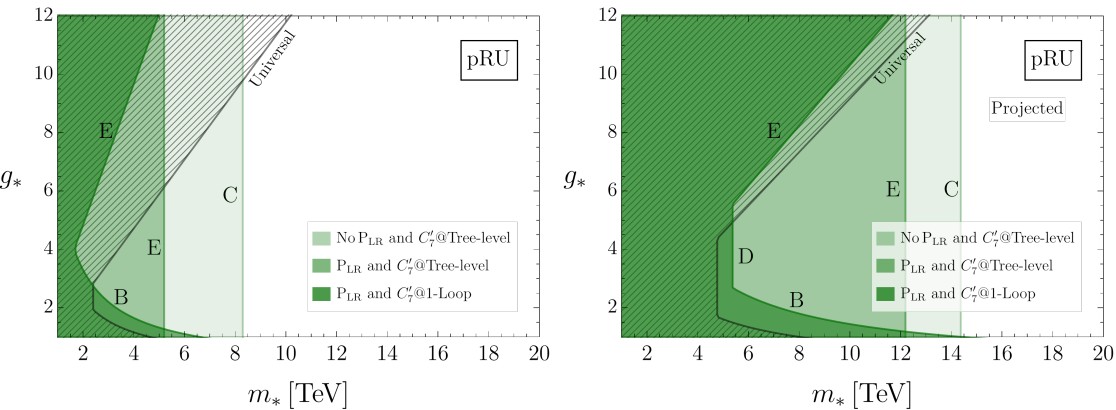

Figure 8: Same as Fig. 5 for models with Partial Right Universality (pRU). See Sec. 4.3.3 for more details.

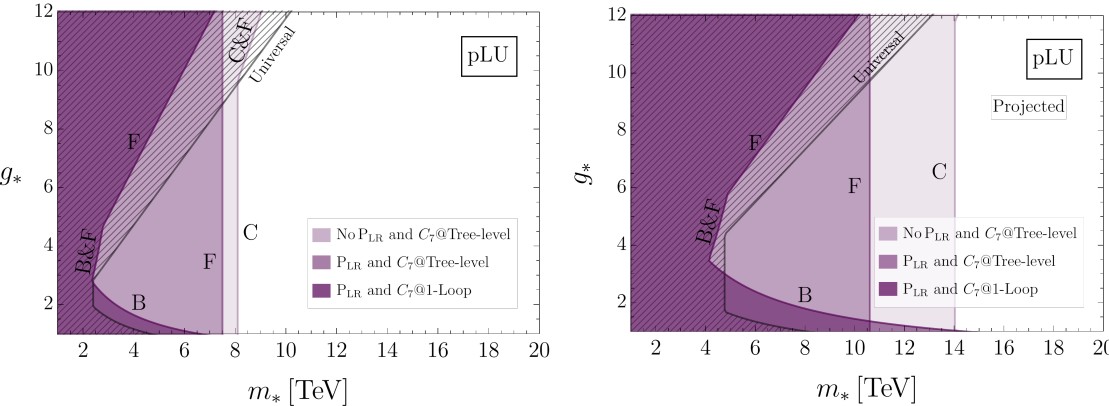

Figure 9: Same as Fig. 5 for models with Partial Left Universality (pLU). See Sec. 5.2.3 for more details.

- **No symmetry: flavor anarchy.** In the anarchic scenario the strongest constraints arise in the lepton sector, from the electron EDM and from $\mu \to e\gamma$. If taken at face value, those constraints already place this scenario well outside the direct reach of FCC-hh (see Section 2.3) and in a region of extreme fine tuning of the electroweak scale. As indicated by several studies, the constraints from the quark sector are instead weaker and possibly at the edge of the direct reach of FCC-hh. One may then optimistically entertain the possibility that anarchy strictly applies only to the quark sector, while in the lepton sector some extra symmetry allows for a lower mass scale (see e.g. [18, 21, 25]). With that perspective in mind, we reanalyzed the constraints in the quark sector, presenting, in particular, a more thorough analysis of the corrections to the EDM of the neutron, also accounting for the contribution of the gluonic Weinberg operator, which was wrongly neglected in previous studies. We confirm that the strongest constraints are from the usual suspects ($\epsilon_K$, $b \to s\gamma$, neutron EDM), but along with a perhaps surprising new entry: the CP asymmetry in $D$-meson decays [39]. Under the most favorable dynamical circumstances, these observables lead to $m_* \gtrsim 20 \div 40$ TeV. Besides implying a fine-tuning of order $10^{-3} \div 10^{-4}$ in electroweak symmetry breaking, this range is definitely above the direct reach of the LHC and indeed at the edge of FCC-hh.

- **Symmetric scenarios: general properties.** When going from the anarchic to the flavor symmetric quark scenarios the relevance of the various observables is reshuffled, with EDMs, $\epsilon_K$, and D meson decays somewhat losing their preeminence. With the exclusion of LU, in all generic incarnations of these scenarios, the dominant constraints arise from $\Delta F = 1$ FCNC transitions in the $B$ sector, in particular from semileptonic decays and $b \to s\gamma$. Furthermore, in more structured incarnations, possessing a combination of custodial $P_{\mathrm{LR}}$ protection and dipole operator loop suppression, $\Delta F = 1$ transitions lose their dominance in favor of a combination of $\Delta F = 2$ transitions, neutron EDM and universal observables. This general property can be clearly seen in Figs. 5, 6, 7, 8, 9.

  It must also be noticed that the dominant flavor-violating constraints arise in the down sector. That is for two reasons. The first is that, precisely like in the SM, the CKM-induced transitions, which are also the only ones in MFV scenarios, are controlled in the down sector by the top quark yukawa, and by the smaller bottom yukawa in the up sector. The second is that novel, non-CKM, effects are controlled, in the up and in the down sector, by the corresponding intergenerational mass ratios (see the matrices $U_{u,d}, V_{u,d}$ in App. B.1). As the ratios are larger for the down-type quarks, effects in the down-sector turn out to be more important than in the up sector.

- **Maximal symmetry: MFV.** On the extreme opposite of anarchy one has the scenarios with MFV. We studied two of the five possible incarnations of MFV within our setup, corresponding respectively to universality in the right-handed (RU, see Section 4.1) and in the left-handed (LU, see Section 5.1) quark sector. The former (RU) is significantly constrained by the combination of flavor breaking *and* flavor-conserving observables (dijet) as shown in the lines labeled "A&B", "A&C" in Fig. 5. The latter (LU) is instead constrained by the couplings of the electroweak bosons, the $W$-couplings in particular (see the line "G" in Fig. 6). In both cases, the scale of new physics is comfortably outside the LHC reach. These cases with extreme symmetry perfectly exemplify the importance of the interplay between flavor symmetry structure (i.e. MFV) and dynamics (partial compositeness and SILH): considering only one of the two aspects one cannot draw

reliable conclusions about a new physics scenario, in particular on its testability or on its viability.

- **Intermediate symmetry: puRU, pRU, pLU.** As already mentioned, in all these scenarios $\Delta B = 1$ transitions give the strongest constraints. In particular, $B$-decays are more constraining than $K$-decays, as shown in Tabs. 11 and 12 and in Appendix C.4.3. Indeed they alone push $m_*$ above the $7 \div 9$ TeV range in all our models with partial universality (see the lines labeled "C"). However, the very same $P_{\mathrm{LR}}$ symmetry often invoked to suppress corrections to the $Z$ coupling to $b$ in composite Higgs models can also significantly relax them. In Figs. 5, 7, 8, 9 the relaxation of the constraints in models with custodial (labeled by "$P_{\mathrm{LR}}$") over the generic ones (labeled "no $P_{\mathrm{LR}}$") is manifest.

  Furthermore, while the more symmetric scenarios (RU, LU and puRU) are safe from flavor-violating dipoles, in the less symmetric ones (pRU and pLU) $b \to s\gamma$ re-emerges as important. Yet, this transition, may or may not originate at tree level in the strong sector. The relevance of $b \to s\gamma$ then depends on addition dynamical assumptions. For instance, in holographic realizations, the dipoles arise at 1-loop, with a further $\sim g_*^2/16\pi^2$ suppression. We then find that, for tree-level generated dipole operators, $b \to s\gamma$ implies a lower bound $m_* \gtrsim 4 \div 6$ TeV (see the "E" line of Fig. 8, coming from $C_7'$ in Tab. 11, and the "F" of Fig. 9, coming from $C_7$ in Tab. 12). Thus, only with loop suppressed dipoles are pRU and pLU allowed in the LHC range. In conclusion, in pRU and pLU, both 1-loop generated dipoles and $P_{\mathrm{LR}}$ protection are necessary in order to let $m_*$ as low as $2 \div 3$ TeV, as shown in Fig. 8, 9. In partial Up-Right Universality (puRU), instead, only $P_{\mathrm{LR}}$ protection is sufficient to lower $m_*$ to the same range.

  As concerns $\Delta F = 2$ effects, $B$ and $K$ physics lead to comparable constraints (see Tabs. 13 and 14). These are subdominant in the generic models, but provide some of the strongest constraints in the nongeneric ones (those endowed with $P_{\mathrm{LR}}$ and loop suppressed dipoles). More precisely, they dominate along with the universal ones for moderate to small $g_*$ ($g_* \lesssim 3$). That is shown by the line labeled "B" in Figs. 5, 7, 8 and 9. Notice that these effects, unlike $\Delta F = 1$ ones, are independent of structural microphysics assumptions (like for instance the presence of $P_{\mathrm{LR}}$), but they do depend on the compositeness parameters $\varepsilon_\psi$. The bounds shown in the figure correspond to the optimal choice, which typically consists of a maximally composite $t_R$.

- **The neutron EDM and its strange quark contribution.** An essential aspect of our models is the simultaneous breakdown of CP and flavor symmetries by the mixing parameters $\lambda_\psi$. That appears to be the only option to control the size of flavor diagonal CP violating observables like the neutron EDM. Also, as it turns out, an automatic consequence of that hypothesis is that in all our models CP-violation purely arises from the combined effects of mixings in the up and down sector (see for example the discussion in Sections 4.2.2). That implies quark EDMs can only appear at higher order in the $\lambda_\psi$ through loop effects, thus providing a further suppression. The predicted $d_n$ then ends up below the current experimental sensitivity, unless extreme choices for the mixing parameters are made.

  Importantly, it is not necessarily the EDMs of the valence quarks that dominate: the contribution of the strange quark EDM may be comparable to or even larger than the

down quark EDM. Specifically, with a coefficient $c_s \sim 1$ in Eq. (B.50), as suggested by naive dimensional analysis, the impact of the strange on the new physics scale $m_*$ is comparable to the down in pRU (see Eq. (4.74)) and a factor of 10 larger in pLU (see Eqs. (5.38) and (5.41)). In particular in the pLU scenario, even by taking the surprisingly small value $c_s \sim 0.01$, suggested by lattice simulation of [61], the strange-quark contribution remains comparable to that of the down quark. Our results thus motivate further studies of the parameter $c_s$.

- **Higgs-couplings.** Our flavor hypotheses allow for a full characterization of Higgs couplings in the SILH scenario, going beyond Refs. [7, 24], where only universal effects were fully characterized. Overall the anomalous couplings correspond to dimension-6 operators obtained by multiplying the structures already existing in the SM by the gauge invariant scalar factor $H^\dagger H$. As such, the resulting effects are suppressed by at least a factor $g_*^2 v^2/m_*^2 \equiv v^2/f^2$ and possibly by additional factors controlled by symmetries.

  Let us first recall the result for the bosonic universal operators which give flavor independent effects. In the parametrization of Ref. [24], the most relevant such operators are $\mathcal{O}_H$, $\mathcal{O}_g$ and $\mathcal{O}_\gamma$ in Table 4.[28] The first leads to a universal $O(v^2/f^2)$ correction of all Higgs couplings. Its strength is independent of further assumptions and we thus took the constraint it implies as a reference throughout our study. The other two operators correct the $hgg$ and $h\gamma\gamma$ vertices, with a strength depending on whether the Higgs is or isn't a PNGB. As explained in App. C.1, in the former case, the present LHC constraints already place $m_* \gtrsim 15 \div 20$ TeV, well outside the direct reach of the LHC. In the latter case, the resulting constraints are weaker than that from $\mathcal{O}_H$, and compatible with $m_*$ around the TeV scale.

  Consider then the Higgs coupling to fermions, which is the real novelty of our study in this context. The leading deviations from the SM are controlled by $\mathcal{O}_{uH,dH}$. While the implications depend on the specific flavor scenario, in all cases flavor-violating vertices $h\bar{f}_i f_j$ ($i \neq j$) are phenomenologically irrelevant in the fiducial range of parameters we focused on (see Appendix C.4.1). In particular, in the case of RU and LU, the effect reduces to universal rescalings controlled by two independent parameters $c_u$ and $c_d$ (see Eq. C.14) in respectively the up and down sector. Moving to the partially universal cases only adds the possibility to differentiate the third family from the two lighter ones. So in the case of puRU, the couplings are essentially given by the analog of Eq. C.14, but with an independent further parameter $c_t$ controlling dominantly the $h\bar{t}t$ vertex (see Eq. (C.18)). When going to pRU and pLU that parameter freedom is extended to the down sector allowing an independent parameter $c_b$ controlling dominantly the $h\bar{b}b$ vertex (see Eq. (C.19)). These qualitative considerations apply to all flavor symmetric SILH scenarios we have considered, irrespective of whether the Higgs is a pseudo-Nambu-Goldstone boson or not.

  Nevertheless, it is worth stressing that in extreme regions of the parameter space, it might at least in principle be possible (see Eq. (C.18)) to produce $h\bar{t}c$ and $h\bar{t}u$ couplings of order $y_t v^2/f^2$, thus producing sizeable $t \to ch$, $t \to uh$ transitions. However, a more dedicated investigation of that region of parameter space is required in order to assess its full viability. We leave that for future work.

---

[28]Operators that modify the Higgs trilinear or the $hZ\gamma$ vertex, under all the circumstances of our study, represent weaker probes of new physics and correspondingly produce subdominant constraints [7, 24].

- **Top compositeness and PNGB Higgs.**

  In all of our scenarios, with the exclusion of RU, the lowest allowed value of $m_*$ corresponds to $O(1)$ compositeness of $t_R$. This fact is nicely compatible with the structure of models with partial universality which possess an $SU(2) \times U(1)$ flavor symmetry in the strong sector: the $t_R$ would just emerge as a composite singlet of the flavor symmetry. Indeed it also fits nicely in the scenario where the Higgs boson arises as a PNGB associated with $G \to H$ symmetry breaking in the strong sector, for instance with $SO(5) \to SO(4)$ [8]. In that case, the Higgs potential and other $G$-breaking couplings such as the $hgg$ and $h\gamma\gamma$ vertices only arise through small effects induced by the SM gauge coupling and the fermion mixings $\lambda_\psi = \varepsilon_\psi g_*$. With a composite $t_R$ the corresponding effect proportional to $\varepsilon_{u_3}$ cannot be present as it would represent an unsuppressed source of $G$-breaking, contradicting the very hypothesis that the strong sector is, *per se*, exactly $G$-symmetric. In this situation, the leading $G$-breaking effects are controlled by $\lambda_{q_3} \sim y_t/\varepsilon_{u_3} \sim y_t$, which both minimizes fine-tuning in electroweak symmetry breaking and eliminates the strong constraints on $m_*$ from $h \to gg$. Moreover, this situation nicely espouses the parameter range $g_* \sim 3$, where the constraints on $m_*$ are the weakest. Indeed, for $\lambda_{q_3} \sim y_t$ and $g_* \sim 3$ the Higgs quartic is generically in the correct ballpark (see, e.g., [29]). Hence, no additional fine-tuning is here required to explain the Higgs potential besides the usual one in $g_*^2 v^2/m_*^2$. Indeed, a measure of fine-tuning in our best case scenarios for puRU, pRU, and pLU is presently given by $g_*^2 v^2/m_*^2 \sim (5 \div 10)\%$, where we took $g_* \sim 3$ and $m_* \sim 2.5 \div 3.0$ TeV. That is the minimum one can achieve in SILH models.

- **Leptons and flavor symmetries.** MFV is also very efficient at controlling flavor-violation and CP-violation in the leptonic sector. But, contrary to quarks, in such a case there exists no urgent reason to depart from it, since bounds from flavor-conserving observables are easily tamed because of the tiny Yukawa couplings. Nevertheless, it is worth to point out that as soon as one departs from complete universality, $\mu \to e\gamma$ becomes so constraining that $m_*$ is pushed outside the direct reach of the LHC. Indeed even under the most optimistic assumption of loop suppressed dipoles (see Eq. (6.9)), one generically has $m_* \gtrsim 5 \, g_*$ TeV. Currently, $\mu \to e$ conversions lead to weaker bounds, that may be further relaxed by invoking the $P_{\text{LR}}$ symmetry. Furthermore, in generic scenarios for neutrino masses, CP violation is secluded from the charged sector and so the electron EDM remains robustly under control. In conclusion, by far the less constrained scenario for leptons is realized with MFV. That is the hypothesis underlying all the plots in Figs. 5, 6, 7, 8, 9.

## 7.2   Prospects

The next two decades will witness a substantial influx of new data from HL-LHC, Belle II, as well as a wealth of upcoming experiments. It is therefore relevant to ask how their results will be able to affect the landscape of models we have explored. To address this question here, we rely on the future projections for the experimental sensitivity collected in Appendix C.5. While the bases of these projections are somewhat rough, we think our analysis still provides a sufficiently informative picture of the expected future exploration of the parameter space. In particular, it offers an idea of what portion will be explored in the next two decades and what will be left for a distant future machine like FCC-hh.

- **Doomsday for Flavor Anarchy?** As emphasized already, anarchic partial compositeness in the leptonic sector is firmly excluded by existing constraints from the electron EDM and $\mu \to e\gamma$. We therefore assume some non-generic solution of the leptonic flavor problem and focus on the quark sector. In the near future, scenarios with anarchy only in the quark sector will be probed up to $m_* \sim 40 \div 50$ TeV at HL-LHC via $\Delta F = 2$ transitions alone, provided $\epsilon_K$ improves the sensitivity on $m_*$ by a factor of $\sim 2$, as suggested by [69]. Another important flavor-violating observable is the CP asymmetry in $D$ meson decays. However, the large theoretical uncertainty on this quantity prevents us from quantifying the improvement in the sensitivity. Instead, we confidently attribute great importance to the prospect [17] to increase in the medium term the experimental sensitivity on $d_n$ by another factor of $10 \div 100$. That will explore the tens to hundreds of TeV range in $m_*$ and either make a discovery or push also quark flavor anarchy "comfortably" outside the reach of FCC-hh.

- **Mid-term indirect vs long-term direct.** The next two decades will further test the parameter space of our flavor symmetric models[29]. The main implications are visually summarized in the right panels of Figs. 5, 6, 7, 8, 9, under the assumption that no discovery is made. As in the previous discussion, we focus on the quark sector, assuming a proper protection mechanism for the dangerous effects in the lepton sector (this can be achieved for instance by MFV, as discussed in detail in Sec. 6).

  The relative importance of the main constraining observables is not expected to change in the near future. That is seen by comparing the left and right panels of the Figures 5, 6, 7, 8, 9.

  All in all, the projected indirect searches will probe the most generic versions of all our flavor symmetric models up to $m_* \sim 10 \div 20$ TeV. The scenario that will be more under pressure is the generic version of RU, which will be probed well above 20 TeV by the combination of high-$p_T$ observables (dijet), semileptonic B-decays ($B_s \to \mu\mu$) and meson oscillations, as shown in Fig. 5. In the more structured versions of puRU, pRU and pLU, those featuring the $P_{LR}$ protection and a 1-loop suppression for dipoles, the projected indirect reach is lowered down to roughly $m_* \gtrsim 5$ TeV, as shown in Figs. 7, 8 and 9, but it remains $\sim 15$ TeV for pRU.

  In summary, in the absence of a discovery in the next two decades the generic versions of our flavor models will be potentially, but marginally, within the reach of the thinkable future machines. Indeed, FCC-hh will be able to directly search for spin-1 resonances up to roughly $m_* \sim 20 \div 30$ TeV [70], whereas a $\sim 10$ TeV muon-collider will be able to probe $m_* \gtrsim 50$ TeV [71], albeit indirectly. More structured scenarios, with custodial protection and, in some cases, 1-loop generated dipoles, will certainly remain more amenable for direct exploration. In those cases, the projected sensitivity $m_* \sim 5$ TeV, will leave space for direct searches of top-partner at FCC-hh [72] and at a 10 TeV muon-collider [71].

- **A not so small nEDM, after all...** In our projection plots we conservatively assumed the measurement of the neutron EDM will improve by a factor 10 [17] (along with the lattice result of $|c_s| \sim 0.01$ for the strange quark contribution). Under those

---

[29]That is true in general, as illustrated for instance in Ref. [68], which is based on a different class of models.

assumptions, in most scenarios, the nEDM will remain a weaker probe of $m_*$ than flavor-violating transitions or universal bosonic operators. The only notable exception is pRU, in which the projected version of Eq. (4.74) becomes the solid vertical line (labeled "D") around $m_* \sim 6$ TeV in the right panel of Fig. 8. Indeed, in pRU, a more aggressive $O(100)$ improvement in the measurements of the nEDM will probe the new physics scale up to 20 TeV, making it the most relevant bound for that scenario. On the other hand, even such a remarkable improvement would affect only mildly puRU and pLU.

- **The future of lepton flavor violation.** In the near future, the already remarkably strong constraints on $m_*$ arising from the electron EDM and $\mu \to e\gamma$ are expected to further improve by roughly a factor of 10 [73] and 5 [15] respectively. In addition, both $\mu \to e$ conversions in nuclei and $\mu \to 3e$ will experience a drastic improvement by about four orders of magnitudes [13, 14]. Obviously, the safest scenario for leptonic physics is and will remain MFV. Yet, the various experimental constraints will further restrain the space of alternative options. In particular, in Sec. 6, we discussed partial universal models as minimal departures from MFV. These models avoid electron EDM bounds secluding CP-violation in the neutrino sector but the absence of universality leads to a sizeable rate for $\mu \to e\gamma$. Conservatively assuming a 1-loop suppression of dipole operators, the corresponding sensitivities will roughly be up to $m_* \lesssim 8\, g_*$ TeV. Even though $\mu \to e$ conversions are now negligible, thanks to the aforementioned experimental updates, they will soon test a significant portion of the parameter space of models with partial lepton universality. In scenarios with sizable lepton compositeness ($\varepsilon \sim 1$) the projected bound $m_* \gtrsim (200 \div 300)\, g_* \varepsilon$ TeV (see Eq. (6.13)) will dominate over $\mu \to e\gamma$, even in the most generic case in which the latter is mediated at tree-level. Yet, $\mu \to e$ conversions and $\mu \to eee$ transitions are much rarer in models with smaller lepton compositeness ($\varepsilon \ll 1$) and may be further suppressed invoking the $P_{\mathrm{LR}}$ protection. In that respect, $\mu \to e\gamma$ will remain the most robust and model-independent probe of scenarios with reduced lepton flavor symmetry.

## 7.3 Concluding Remarks

There is no doubt that only by pushing particle collisions to ever higher energies will we tear through the curtain of mystery that lies behind the SM. In that respect the experimental study of low-energy flavor and CP violation could appear as just poking into that curtain, trying to figure out what may and what may not be. However, as we know from the study of plausible scenarios of new physics, indirect searches often provide remarkably strong constraints. That is particularly true in the broad scenario we studied in this paper, where the Higgs field is a composite of some strong dynamics. Under a variety of flavor and dynamical hypotheses, which we discussed at length, we assessed the indirect reach for the forthcoming two decades, mostly driven by LHCb, Belle II, and EDM searches. That should be compared to the expectations at a machine like FCC-hh, which can be roughly quantified as $10 - 15$ TeV for colored resonances like the top-partners and $20 - 30$ TeV for Drell-Yan produced vector resonances. Our results, partly synthesized in Figures 5, 6, 7, 8, 9, show that one can basically divide the scenarios into three classes. The first consists of the "cleverest scenarios", endowed with optimal flavor and generalized custodial symmetry allowing the overall new physics scale $m_*$ as low as $\sim 2$ TeV. As we show, the forthcoming two decades will significantly push the reach of $m_*$ by roughly a factor of 2. In the absence of a discovery, that would still leave ample

space for exploration at a machine like FCC-hh. The second class is instead made of more generic (and perhaps more plausible) models that are already more constrained and which in the forthcoming two decades will be probed at a level that competes with FCC-hh. Finally, there are the scenarios with minimal symmetry, like flavor anarchy, which already now are marginally within FCC-hh reach and for which the last chance for even an indirect signal will be during the next two decades.

Our study, which is based on a structured set of dynamical and symmetry assumptions, allows to correlate the full palette of new physics searches, including flavor and CP violation, Higgs coupling, electroweak, and particle production. It seems to us it offers a more useful input to strategic studies than, for instance, many SMEFT approaches where no structured assumption is often made. On the sunny side, it shows that there are concrete and plausible scenarios for which we won't have to wait until 2070 to learn something interesting.

## Acknowledgments

We would like to thank K. Agashe, R. Barbieri, T. Bhattacharya, M. Ekhterachian, D. E. Kaplan, G. Isidori, R. Sundrum, A. Wulzer for discussions, and especially R. Barbieri and G. Isidori for valuable comments on a very preliminary version of our paper.

The work of R. R. is partially supported by the Swiss National Science Foundation under contract 200020-213104. The work of L. R. is supported by NSF Grant No. PHY-2210361 and by the Maryland Center for Fundamental Physics. The work of L. V. was partly supported by the Italian MIUR under contract 202289JEW4 (Flavors: dark and intense), the Iniziativa Specifica "Physics at the Energy, Intensity, and Astroparticle Frontiers" (APINE) of Istituto Nazionale di Fisica Nucleare (INFN), and the European Union's Horizon 2020 research and innovation programme under the Marie Sklodowska-Curie grant agreement No 860881-HIDDeN.

## A    Completing the theoretical picture

### A.1    Custodial symmetries

An approximate custodial symmetry is an accidental property of the SM and also an experimental fact. Custodial symmetry is thus an essential ingredient in models of composite Higgs. In its absence, the operator $\mathcal{O}_T$ of Table 4 would arise unsuppressed in the effective Lagrangian and contribute sizeably to the electroweak parameter $\widehat{T}$ [74].[30] Considering the experimental constraint [53] ($\Delta \widehat{T} \lesssim 2 \times 10^{-3}$) we then have

$$\Delta \widehat{T} \sim \frac{g_*^2 v^2}{m_*^2} \qquad \longrightarrow \qquad m_* > 5.5\, g_* \text{ TeV}. \tag{A.1}$$

To avoid this bound, in our study we *always* assume the strong sector possesses an $SO(4) \sim SU(2)_L \times SU(2)_R$ custodial symmetry. More precisely, partial compositeness forces the strong sector to possess, on top of possible flavor symmetries, a symmetry $SU(3) \times SO(4) \times U(1)_X$,

---

[30]We do not take into account the recent CDF $W$-boson mass measurement, which may push the preferred value for $\Delta \hat{T}$ away from 0 [75]. Although this result is intriguing and may represent a hint of New Physics we still find it too premature and do not include it in our analysis.

of which a subgroup $SU(3)_c \times SU(2)_L \times U(1)_Y$ with $Y = T_L^3 + X$ is gauged. The factor $U(1)_X$ is necessary to embed hypercharge compatibly with Eq. (3.1).

Custodial $SO(4)$ can also protect $Z$ couplings to right-handed quarks [27] associated with the operators $\mathcal{O}_{Hu}$ and $\mathcal{O}_{Hd}$ of Tab. 4. The realization of that possibility depends on the $SO(4) \times U(1)_X$ representation of the composite partners $\mathcal{O}_{u,d}$. The main group theoretic reason is that $H^\dagger D_\mu H$ corresponds to the $T^{3R} = 0$ component of a $(\mathbf{1}, \mathbf{3})$ of $SO(4)$. As shown in the table, corrections to $C_{Hu}$, or equivalently to the $Z\bar{u}_R^i u_R^j$ vertex, are avoided with the choice $\mathcal{O}_u \in (\mathbf{1}, \mathbf{1})_{2/3}$ or $\mathcal{O}_u \in (\mathbf{1}, \mathbf{3})_{2/3}$. Similarly $C_{Hd}$, or equivalently $Z\bar{d}_R^i d_R^j$ vertex, is protected by the choice $\mathcal{O}_d \in (\mathbf{1}, \mathbf{1})_{-1/3}$ or $\mathcal{O}_d \in (\mathbf{1}, \mathbf{3})_{-1/3}$.

Corrections to the vertices $Z\bar{q}_L q_L$ are not protected by $SO(4)$ per se, but they can be protected when $SO(4)$ is enlarged to $O(4) = SO(4) \rtimes P_{\rm LR}$ with the composite partners eigenstates of $P_{\rm LR}$ [27].[31] More precisely, the corrections to $Z\bar{d}_L^i d_L^j$ vanish when $\mathcal{O}_q \in (\mathbf{2}, \mathbf{2})_{2/3}$. Similarly corrections to $Z\bar{u}_L^i u_L^j$ vanish when $\mathcal{O}_q \in (\mathbf{2}, \mathbf{2})_{-1/3}$. For some of the models discussed in the main text, two different composite operators $\mathcal{O}_{q_u}$ and $\mathcal{O}_{q_d}$ mix with the SM doubltes $q$ and are responsible, respectively, for the up and the down Yukawas (see Eq. (3.1)). In that case, reading from Tab. 3, the choice that offers the strongest protection of both $\Delta\widehat{T}$ and vertex corrections is given by

$$\mathcal{O}_{q_u} \in (\mathbf{2}, \mathbf{2})_{2/3}, \qquad \mathcal{O}_{q_d} \in (\mathbf{2}, \mathbf{2})_{-1/3}, \qquad \mathcal{O}_u \in (\mathbf{1}, \mathbf{1})_{2/3}, \qquad \mathcal{O}_d \in (\mathbf{1}, \mathbf{1})_{-1/3}. \qquad \text{(A.2)}$$

This choice avoids large corrections to the experimentally well measured $Z\bar{d}_L^i d_L^j$ from the largest mixings associated with the top quark sector. At the same time it protects $Zu_R^i u_R^j$ and $Zd_R^i d_R^j$ and eliminates contributions to $\Delta\widehat{T}$ from $\lambda_{u,d}$, which offers more model building freedom in the top sector.

In models where a single $q_L$ partner $\mathcal{O}_q$ is responsible for both the up and the down Yukawas the optimal representation is

$$\mathcal{O}_q \in (\mathbf{2}, \mathbf{2})_{2/3}, \qquad\qquad \mathcal{O}_u \in (\mathbf{1}, \mathbf{1})_{2/3}, \qquad\qquad \mathcal{O}_d \in (\mathbf{1}, \mathbf{3})_{2/3}. \qquad \text{(A.3)}$$

and suppresses the dominant corrections to $Z\bar{d}_L^i d_L^j$ and $Z\bar{u}_R^i u_R^j$ but cannot protect the $Z\bar{d}_R^i d_R^j$ coupling.

## A.2 Beyond the one-coupling hypothesis

Within the one-coupling one-scale assumptions, the description of Eq. (2.1) flows at scales of order $m_*$ into a set of interactions among the SM degrees of freedom (except the Higgs) and the lightest resonances (including the Higgs) of the strongly-coupled sector of the form in Eq. (2.3). After having integrated out all the resonances except the Higgs one is then left with the effective field theory in Eq. (2.6).

Generic quantum field theories possess couplings of different sizes, though. For example, asymptotically-free large $N$ gauge theories with a number $N_f \ll N$ of fermions in the

---

[31]This condition is often met accidentally in composite Higgs models [76].

| Operator | Representation | $\Delta\widehat{T}$ | Protected coupling |
|----------|----------------|---------------------|--------------------|
| $\mathcal{O}_q$ | $(\mathbf{2},\mathbf{1})_{1/6}$ | — | $\times$ |
| | $(\mathbf{2},\mathbf{2})_{2/3}$ | $\lambda_q^4$ | $Z d_L^i d_L^j$ |
| | $(\mathbf{2},\mathbf{2})_{-1/3}$ | $\lambda_q^4$ | $Z u_L^i u_L^j$ |
| $\mathcal{O}_u$ | $(\mathbf{1},\mathbf{1})_{2/3}$ | — | $Z u_R^i u_R^j$ |
| | $(\mathbf{1},\mathbf{2})_{1/6}$ | $\lambda_u^4$ | $\times$ |
| | $(\mathbf{1},\mathbf{3})_{2/3}$ | $\lambda_u^2$ | $Z u_R^i u_R^j$ |
| | $(\mathbf{1},\mathbf{3})_{-1/3}$ | $\lambda_u^2$ | $\times$ |
| $\mathcal{O}_d$ | $(\mathbf{1},\mathbf{1})_{-1/3}$ | — | $Z d_R^i d_R^j$ |
| | $(\mathbf{1},\mathbf{2})_{1/6}$ | $\lambda_d^4$ | $\times$ |
| | $(\mathbf{1},\mathbf{3})_{2/3}$ | $\lambda_d^2$ | $\times$ |
| | $(\mathbf{1},\mathbf{3})_{-1/3}$ | $\lambda_d^2$ | $Z d_R^i d_R^j$ |

Table 3: Minimal $SO(4)\times U(1)_X$ embeddings for the quark-partners. In the third column we report the parametric dependence of $\Delta\widehat{T}$ from the elementary-composite mixing. The symbol — denotes that the mixing does not break custodial symmetry. The last column shows which of the $Z$ couplings is protected by tree-level zero-momentum corrections. The $\times$ denotes that none of the couplings is protected. Notice that, in order to protect $Z$ couplings to left-handed quarks, the strong sector must respect the $P_{\mathrm{LR}}$ symmetry.

fundamental representation generate two types of resonances of mass $\sim m_*$ (baryons are parametrically heavier): "mesons" $\Phi_M$, with self-couplings of order $g_M \sim 4\pi/\sqrt{N}$, and "glueballs" $\Phi_G$, with self-coupling $g_G \sim 4\pi/N \ll g_M$. In order to assess the robustness of the results derived in the bulk of the paper, it is important to assess how the EFT in Eq. (2.6) would be affected by relaxing the one-coupling hypothesis. In this appendix we address this question by exploring the simplest possible case of two different couplings.

When including two types of resonances, and after having canonically normalized the resonance fields, Eq. (2.3) is replaced by

$$
\begin{aligned}
\mathcal{L} \;=\; & \mathcal{L}_{\mathrm{SM}'} \\
+ \; & \frac{m_*^4}{g_G^2}\widehat{\mathcal{L}}_G\left(\frac{g_G\Phi_G}{m_*^{d_G}}, \frac{D_\mu}{m_*}, \frac{\lambda_\psi\overline{\psi}}{m_*^{3/2}}, \frac{g^2}{16\pi^2}, \frac{\lambda_\psi^\dagger\lambda_\psi}{16\pi^2}, \frac{g_*^2}{16\pi^2}\right) \\
+ \; & \frac{m_*^4}{g_M^2}\widehat{\mathcal{L}}_M\left(\frac{g_M\Phi_M}{m_*^{d_M}}, \frac{g_G\Phi_G}{m_*^{d_G}}, \frac{D_\mu}{m_*}, \frac{\lambda_\psi\overline{\psi}}{m_*^{3/2}}, \frac{g^2}{16\pi^2}, \frac{\lambda_\psi^\dagger\lambda_\psi}{16\pi^2}, \frac{g_*^2}{16\pi^2}\right) ,
\end{aligned}
\tag{A.4}
$$

with $\widehat{\mathcal{L}}_{M,G}$ interpreted as polynomials with coefficients of order unity.[32] Crucially, to retain a consistent counting the factor of $1/g_G^2 \gg 1/g_M^2$ must appear only in front of purely-glueball interactions. This way, and in this case only, the hypothesis that the glueballs remain more weakly-coupled than mesons is ensured. Analogously, scenarios with more couplings can be described by isolating the most weakly-coupled states, then adding interactions among them and the next to weakly-coupled ones, and so on as in Eq. (A.4).

---

[32]By including factors of $D_\mu/m_*$ from the second and third lines of Eq. (A.4), we are tacitly assuming that at least some of the $\Phi_G$'s and some of the $\Phi_M$'s are charged under the full SM gauge group. If all glueballs were neutral, on the other hand, all covariant derivatives in the second line should be replaced by $\partial_\mu/m_*$.

The key players in the EFT of partial compositeness are the Higgs and the fermionic resonances that mix with the SM fermions, aka the top-partners. The implications of Eq. (A.4) are qualitatively the same as those of Eq. (2.3) whenever Higgs and top-partners are resonances of the same type. Scenarios in which those states are both mesons have couplings $g_* = g_M$ in Eq. (2.3), whereas $g_* = g_G$ when the Higgs doublet and the top-partners are all glueball-like resonances. Qualitative differences arise if top-partners and Higgs belong to two different classes of resonances. As we will demonstrate below, these scenarios are less attractive than the scenarios discussed in the rest of the paper because certain effective operators are enhanced and, in some cases, the Higgs couplings are less SM-like. Intermediate possibilities with some top-partners of a given type and others of another type result in a phenomenology that is qualitatively a mixture of the two cases analyzed below.

### Higgs = glueball, Top-partners = mesons

We start by comparing scenarios in which the Higgs is a glueball-like resonance and the top-partners are mesons to the one-coupling scenarios analyzed in Eq. (2.6), which we take as a reference. In the former framework, from Eq. (A.4) follows that the SM fermion compositeness is given by $\varepsilon^{(M)} \sim \lambda_\psi / g_M$, with the superscript recalling that the mixing is with a mesonic state. Because the trilinear interaction between the top-partners and the Higgs is $\sim g_G$, see the third line of Eq. (A.4), the SM Yukawa coupling is now of order

$$Y \sim g_G \, \varepsilon_L^{(M)} [\varepsilon_R^{(M)}]^\dagger \,. \tag{A.5}$$

In the framework of reference we instead have $\varepsilon \sim \lambda_\psi / g_*$ and $Y \sim g_* \varepsilon_L \varepsilon_R^\dagger$. The mixings in the two scenarios thus differ by a factor $\varepsilon^{(M)}/\varepsilon \sim \sqrt{g_*/g_G}$. As a result, in the class of models discussed in this appendix, the operators involving multiple fermions are generically enhanced by powers of $g_M/g_G$. For example, the Wilson coefficient of 4-fermion operators obtained from Eq. (A.4) goes as (for ease of notation we will not distinguish $\lambda_\psi$ and $\lambda_\psi^\dagger$ in the reminder of this appendix)

$$\varepsilon_i^{(M)} \varepsilon_j^{(M)} \varepsilon_k^{(M)} \varepsilon_l^{(M)} \frac{g_M^2}{m_*^2} \sim \varepsilon_i \varepsilon_j \varepsilon_k \varepsilon_l \frac{g_*^2}{m_*^2} \frac{g_M^2}{g_G^2} \,, \tag{A.6}$$

and is larger by a factor $g_M^2/g_G^2 \gg 1$ compared to those derived from Eq. (2.3). Dipole interactions of course scale the same way whereas the coupling of effective operators with a fermion current coupled to a Higgs current go as

$$\varepsilon_i^{(M)} \varepsilon_j^{(M)} \frac{g_G^2}{m_*^2} \sim \varepsilon_i \varepsilon_j \frac{g_*^2}{m_*^2} \frac{g_G}{g_*} \,, \tag{A.7}$$

and are generically suppressed compared to scenarios with $g_* = g_M$, though comparable to scenarios where $g_* = g_G$. The coefficients of operators involving gauge bosons, in particular some of those contributing to the electroweak parameters, depend on whether the gauge bosons mix with glueball or meson states and can be estimated analogously. The deviations in the Higgs couplings are instead always controlled by $g_G^2 v^2/m_*^2$ in the exotic scenario and they are, so, typically smaller by a factor $g_G^2/g_*^2 \leq 1$ compared to one-coupling models with a comparable new physics scale $m_*$.

For completeness, let us also compare the radiative corrections to the Higgs potential induced by the fermion mixing. Conservatively considering a fully composite $t_R$, where $\varepsilon_L^{(M)} \sim$

$y_t/g_G$ and the dominant corrections to the potential come from loops of the left-handed quarks, at leading order we have

$$\delta V_H = \frac{m_*^4 N_c}{16\pi^2}(\varepsilon_L^{(M)})^\dagger \varepsilon_L^{(M)} \widehat{V}\left(\frac{g_G H}{m_*}\right) \quad \Longrightarrow \quad \delta m_H^2 \sim \frac{y_t^2 N_c}{16\pi^2}m_*^2, \quad \delta\lambda_H \sim \frac{y_t^2 N_c}{16\pi^2}g_G^2. \quad \text{(A.8)}$$

We thus find that the correction to the Higgs mass is qualitatively unaffected by the presence of multiple couplings. On the other hand, the quartic tends to be smaller (or at most comparable) with respect to one-coupling scenarios with $g_* \sim g_M$.

In summary, scenarios with a glueball-like Higgs and mesonic top-partners have more SM-like Higgs couplings compared to generic one-coupling scenarios with mesonic resonances and the same level of fine-tuning, but have to cope with larger coefficients for the 4-fermion interactions.

## Higgs = meson, Top-partners = glueballs

Models with a mesonic Higgs and glueball-like top-partners manifest the same enhancement of the multi-fermion interactions we just saw, as well as an additional undesirable boost in the Higgs self-couplings.

According to Eq. (A.4), the SM Yukawa now scales as

$$Y \sim \frac{g_G^2}{g_M}\varepsilon_L^{(G)}\varepsilon_R^{(G)}, \quad\quad\quad\quad\quad \text{(A.9)}$$

where the fermion compositeness is measured by $\varepsilon^{(G)} \sim \lambda_\psi/g_G \sim \varepsilon\sqrt{g_* g_M}/g_G > \varepsilon$. The mixing parameters are so always larger compared to the reference one-coupling models, and one can verify that the coefficient of the 4-fermion interactions are enhanced accordingly

$$\varepsilon_i^{(G)}\varepsilon_j^{(G)}\varepsilon_k^{(G)}\varepsilon_l^{(G)}\frac{g_G^2}{m_*^2} \sim \varepsilon_i\varepsilon_j\varepsilon_k\varepsilon_l\frac{g_*^2}{m_*^2}\frac{g_M^2}{g_G^2}. \quad\quad \text{(A.10)}$$

The coefficient of operators with two fermions and two Higgses is

$$\varepsilon_i^{(G)}\varepsilon_j^{(G)}\frac{g_G^2}{m_*^2} \sim \varepsilon_i\varepsilon_j\frac{g_*^2}{m_*^2}\frac{g_M}{g_*}, \quad\quad\quad \text{(A.11)}$$

and is always enhanced compared to scenarios where $g_* \sim g_G$ (and comparable in case $g_* \sim g_M$).

The leading $O(\varepsilon^2)$ corrections to the Higgs potential are dominated by diagrams with $n$ Higgses emerging from a single vertex with the top-partners, of order $g_G^2 g_M^{n-2}$. Diagrams with multiple Higgs vertices are suppressed because in that case, one would have to pay the price of multiple $g_G^2$ insertions, which measure the top-partners' couplings. Conservatively assuming again fully composite $t_R$, with $\varepsilon_L^{(G)} \sim y_t g_M/g_G^2$, we estimate

$$\delta V_H = \frac{m_*^4 N_c}{16\pi^2}(\varepsilon_L^{(G)})^\dagger \varepsilon_L^{(G)}\frac{g_G^2}{g_M^2}\widehat{V}\left(\frac{g_M H}{m_*}\right). \quad\quad\quad \text{(A.12)}$$

Therefore the corrections to the Higgs mass and quartic coupling read

$$\delta m_H^2 \sim \frac{y_t^2 N_c}{16\pi^2}m_*^2\frac{g_M^2}{g_G^2}, \quad \delta\lambda_H \sim \frac{y_t^2 N_c}{16\pi^2}g_M^2\frac{g_M^2}{g_G^2}, \quad\quad \text{(A.13)}$$

and are both either parametrically larger or comparable than in the one-coupling scenarios considered in the rest of the paper.

# B   Details on the analysis

This appendix collects a few technical details on our analysis. In Appendix B.1 we introduce the most important $SU(2)_L \times U(1)_Y$-invariant effective operators (see Table 4), the Wilson coefficients of which can be derived following the recipe explained in Section 2.2. The rotation to the mass basis is discussed for our flavor symmetric models, and illustrated explicitly in Appendix B.2, where we discuss the expressions of the basic flavor-violating structures $\lambda_\psi \lambda_{\psi'}^\dagger$ that appear at leading order in quark bilinears. The analogous procedure in scenarios of flavor anarchy is discussed in detail, e.g., in [25, 39].

In Appendix B.3 the operators of Table 4 are matched to the low-energy operators commonly used in the study of flavor observables. Finally, an estimate of the leading order contributions to the EDM of the neutron is presented in App. B.4.

Importantly, the list in Tab. 4 is over-complete and redundant *on purpose*. The reason for including redundancies in our list is that this way the power-counting implied by Eq. (2.6) as well as the selection rules enforced by the symmetries of the theory are more transparent. A concrete example will better illustrate this. Suppose we decide to remove the operators $\mathcal{O}_{Hq}^{(1,3)}$ via a field redefinition in favor of four-fermion operators and $\mathcal{O}_{qD}^{(1,3)}$. Then, a superficial use of the EFT (2.6) would completely miss the dominant contribution to $b \to s\ell\ell$. Conversely, had we worked in a basis with $\mathcal{O}_{Hq}^{(1,3)}$ but without $\mathcal{O}_{qD}^{(1,3)}$, the subleading contributions to that observable in scenarios with $P_{\rm LR}$ protection would have been encoded in corrections to four-fermion operators that apparently violate the power-counting of Eq. (2.6). Working with a complete basis would therefore obscure some of the key physical implications of our setup. This is not the case if we consider the entire list of Tab. 4.

## B.1   High-Energy effective operators

In this appendix, we summarize the conventions for the various effective operators considered in our analysis. The SM gauge-invariant dimension-6 operators that are relevant to our analysis are collected in Tab. 4, where

$$D_\mu = \partial_\mu + ig A_\mu \,, \tag{B.1}$$

$$H^\dagger \overleftrightarrow{D}^\mu H = H^\dagger D_\mu H - (D_\mu H)^\dagger H \,, \tag{B.2}$$

$$\widetilde{H}_j = \epsilon_{jk}(H^k)^* \,. \tag{B.3}$$

The operators $\mathcal{O}_\mathcal{X}$ ($\mathcal{X}$ is a label, see Tab. 4) appear in the effective Lagrangian multiplied by Wilson Coefficients $\mathcal{C}_\mathcal{X}$ according to the convention $\mathcal{L}_{\rm EFT} \supset \mathcal{C}_\mathcal{X} \mathcal{O}_\mathcal{X}$. The recipe to write the $\mathcal{C}_\mathcal{X}$'s is discussed in Section 2.2. Their expression in the $SU(2)_L \times U(1)_Y$ invariant basis are defined, up to numbers of order unity, in terms of the parameters $\lambda_\psi, g_*, m_*$ as determined by Eq. (2.6).

The expression of the Wilson coefficients in the mass basis, in which the Yukawa matrices defined by the various incarnations of Eq. (2.7) (see text and Eqs. (4.4, 4.30, 4.55, 5.5, 5.22)) are real and diagonal, are indicated with a "tilde" as

$$\widetilde{\mathcal{C}}_\mathcal{X} \qquad (\text{mass basis}). \tag{B.4}$$

The transformation to the mass basis is obtained via the following rotations of the fermionic fields $q_L, u_R, d_R$ into the mass-eigenstate fields $\widetilde{q}_L, \widetilde{u}_R, \widetilde{d}_R$:

$$u_L = U_u \widetilde{u}_L \,, \qquad d_L = U_d \widetilde{d}_L \,, \qquad u_R = V_u \widetilde{u}_R \,, \qquad d_R = V_d \widetilde{d}_R \,, \tag{B.5}$$

| Bosonic | |
|---|---|
| $\mathcal{O}_H = \frac{1}{2}\partial_\mu(H^\dagger H)\partial^\nu(H^\dagger H)$ | $\mathcal{O}_T = \frac{1}{2}\left(H^\dagger \overleftrightarrow{D}_\mu H\right)\left(H^\dagger \overleftrightarrow{D}^\mu H\right)$ |
| $\mathcal{O}_g = H^\dagger H G^A_{\mu\nu} G^{A\,\mu\nu}$ | $\mathcal{O}_\gamma = H^\dagger H B_{\mu\nu} B^{\mu\nu}$ |
| $\mathcal{O}_W = i\frac{g}{2}\left(H^\dagger \overleftrightarrow{D}^a_\mu H\right) D_\nu W^{a\,\mu\nu}$ | $\mathcal{O}_B = i\frac{g'}{2}\left(H^\dagger \overleftrightarrow{D}_\mu H\right)\partial_\nu B^{\mu\nu}$ |
| $\mathcal{O}_{2W} = \frac{g^2}{2}(D^\mu W^a_{\mu\nu})(D_\rho W^{a\rho\nu})$ | $\mathcal{O}_{2B} = \frac{g'^2}{2}(\partial^\mu B_{\mu\nu})(\partial_\rho B^{\rho\nu})$ |

| Dipoles | |
|---|---|
| $[\mathcal{O}_{uB}]^{ij} = (\bar{q}^i_L \sigma^{\mu\nu} u^j_R)\widetilde{H} B_{\mu\nu}$ | $[\mathcal{O}_{dB}]^{ij} = (\bar{q}^i_L \sigma^{\mu\nu} d^j_R)H B_{\mu\nu}$ |
| $[\mathcal{O}_{uW}]^{ij} = (\bar{q}^i_L \sigma^{\mu\nu} u^j_R)\sigma^a \widetilde{H} W^a_{\mu\nu}$ | $[\mathcal{O}_{dW}]^{ij} = (\bar{q}^i_L \sigma^{\mu\nu} d^j_R)\sigma^a H W^a_{\mu\nu}$ |
| $[\mathcal{O}_{uG}]^{ij} = (\bar{q}^i_L \sigma^{\mu\nu} T^A u^j_R)\widetilde{H} G^A_{\mu\nu}$ | $[\mathcal{O}_{dG}]^{ij} = (\bar{q}^i_L \sigma^{\mu\nu} T^A d^j_R)H G^A_{\mu\nu}$ |

| EW vertices | |
|---|---|
| $\left[\mathcal{O}^{(1)}_{Hq}\right]^{ij} = \left(H^\dagger i\overleftrightarrow{D}_\mu H\right)\bar{q}^i_L \gamma^\mu q^j_L$ | $\left[\mathcal{O}^{(3)}_{Hq}\right]^{ij} = \left(H^\dagger i\sigma^a \overleftrightarrow{D}_\mu H\right)\bar{q}^i_L \gamma^\mu \sigma^a q^j_L$ |
| $[\mathcal{O}_{Hu}]^{ij} = \left(H^\dagger i\overleftrightarrow{D}_\mu H\right)\bar{u}^i_R \gamma^\mu u^j_R$ | $[\mathcal{O}_{Hd}]^{ij} = \left(H^\dagger i\overleftrightarrow{D}_\mu H\right)\bar{d}^i_R \gamma^\mu d^j_R$ |
| $\left[\mathcal{O}^{(1)}_{qD}\right]^{ij} = \bar{q}^i_L \gamma^\nu q^j_L \partial^\mu B_{\mu\nu}$ | $\left[\mathcal{O}^{(3)}_{qD}\right]^{ij} = \bar{q}^i_L \gamma^\nu \sigma^a q^j_L D^\mu W^a_{\mu\nu}$ |
| $[\mathcal{O}_{uD}]^{ij} = \bar{u}^i_R \gamma^\nu u^j_R \partial^\mu B_{\mu\nu}$ | $[\mathcal{O}_{dD}]^{ij} = \bar{d}^i_R \gamma^\nu d^j_R \partial^\mu B_{\mu\nu}$ |

| 4-fermions | |
|---|---|
| $\left[\mathcal{O}^{(1)}_{qq}\right]^{ijkl} = (\bar{q}^i_L \gamma^\mu q^j_L)(\bar{q}^k_L \gamma_\mu q^l_L)$ | $\left[\mathcal{O}^{(3)}_{qq}\right]^{ijkl} = (\bar{q}^i_L \gamma^\mu \sigma^a q^j_L)(\bar{q}^k_L \gamma_\mu \sigma^a q^l_L)$ |
| $[\mathcal{O}_{uu}]^{ijkl} = (\bar{u}^i_R \gamma^\mu u^j_R)(\bar{u}^k_R \gamma_\mu u^l_R)$ | $[\mathcal{O}_{dd}]^{ijkl} = (\bar{d}^i_R \gamma^\mu d^j_R)(\bar{d}^k_R \gamma_\mu d^l_R)$ |
| $\left[\mathcal{O}^{(1)}_{qu}\right]^{ijkl} = (\bar{q}^i_L \gamma^\mu q^j_L)(\bar{u}^k_R \gamma_\mu u^l_R)$ | $\left[\mathcal{O}^{(8)}_{qu}\right]^{ijkl} = (\bar{q}^i_L \gamma^\mu T^A q^j_L)(\bar{u}^k_R \gamma_\mu T^A u^l_R)$ |
| $\left[\mathcal{O}^{(1)}_{qd}\right]^{ijkl} = (\bar{q}^i_L \gamma^\mu q^j_L)(\bar{d}^k_R \gamma_\mu d^l_R)$ | $\left[\mathcal{O}^{(8)}_{qd}\right]^{ijkl} = (\bar{q}^i_L \gamma^\mu T^A q^j_L)(\bar{d}^k_R \gamma_\mu T^A d^l_R)$ |

| Flavor violating Higgs couplings | |
|---|---|
| $[\mathcal{O}_{uH}]^{ij} = (\bar{q}^i_L \widetilde{H} u^j_R)(H^\dagger H)$ | $[\mathcal{O}_{dH}]^{ij} = (\bar{q}^i_L H d^j_R)(H^\dagger H)$ |

Table 4: List of dimension six operators in the Electroweak-invariant basis that are most relevant to our analysis. $T^A = \frac{1}{2}\lambda^A$ are the $SU(3)$ generators, $\lambda^A$ are the Gell-Mann matrices and $\sigma^a$ are the Pauli matrices.

such that

$$Y_u = U_u \widetilde{Y}_u V_u^\dagger, \qquad\qquad Y_d = U_d \widetilde{Y}_d V_d^\dagger, \qquad\qquad V_{\text{CKM}} = U_u^\dagger U_d, \qquad (B.6)$$

where the tilde denotes the diagonalized Yukawa. As a concrete example, the coefficient $\mathcal{C}_{Hu}$ of the operator $\mathcal{O}_{Hu}$ in the mass basis becomes

$$[\widetilde{\mathcal{C}}_{Hu}]^{ij} = [U_u^\dagger \mathcal{C}_{Hu} U_u]^{ij}. \qquad (B.7)$$

Analogously, the reader can derive the Wilson coefficients in the mass basis for all the other operators in Tab. 4. As a reference, in Appendix B.2 we will write the explicit expression of the leading order structures that appear in fermion bilinears.

In each model the matrices $\lambda_\psi$ that control $\mathcal{C}_\mathcal{X}$ have the explicit forms discussed in the text. Similarly, the matrices $U_{u,d}$, $V_{u,d}$ are model-dependent. In models with universality in the right-handed sector (see Sections 4) it is sufficient to show those matrices for pRU. At leading order in an expansion in $y_u/y_c$, $y_c/y_t$, $y_d/y_s$, $y_s/y_b \ll 1$ we find

$$U_u = \begin{pmatrix} 1 & -a^*b\frac{y_u y_c}{y_t^2} & a^*\frac{y_u y_c}{y_t^2} \\ ab^*\frac{y_u y_c}{y_t^2} & 1 & b^*\frac{y_c^2}{y_t^2} \\ -a\frac{y_u y_c}{y_t^2} & -b\frac{y_c^2}{y_t^2} & 1 \end{pmatrix} \qquad V_u = \begin{pmatrix} 1 & -a^*b\frac{y_c^2}{y_t^2} & a^*\frac{y_c}{y_t} \\ ab^*\frac{y_u^2}{y_t^2} & 1 & b^*\frac{y_c}{y_t} \\ -a\frac{y_c}{y_t} & -b\frac{y_c}{y_t} & 1 \end{pmatrix} \qquad (B.8)$$

and

$$U_d = \widetilde{U}_d \begin{pmatrix} 1 & -a'^*b'\frac{y_d y_s}{y_b^2} & a'^*\frac{y_d y_s}{y_b^2} \\ a'b'^*\frac{y_d y_s}{y_b^2} & 1 & b'^*\frac{y_s^2}{y_b^2} \\ -a'\frac{y_d y_s}{y_b^2} & -b'\frac{y_s^2}{y_b^2} & 1 \end{pmatrix} \qquad V_d = \begin{pmatrix} 1 & -a'^*b'\frac{y_s^2}{y_b^2} & a'^*\frac{y_s}{y_b} \\ a'b'^*\frac{y_d^2}{y_b^2} & 1 & b'^*\frac{y_s}{y_b} \\ -a'\frac{y_s}{y_b} & -b'\frac{y_s}{y_b} & 1 \end{pmatrix} \qquad (B.9)$$

where $\widetilde{U}_d$ was introduced in Eq. (4.55). The approximation is reliable as long as $|a|, |b| < y_t/y_c$ and $|a'|, |b'| < y_b/y_s$. Throughout the paper we assume $|a|, |b|, |a'|, |b'| \sim 1$, and so our approximate expressions are accurate. The expressions in puRU are recovered by sending $a' = b' = 0$, whereas RU corresponds to $a = b = a' = b' = 0$. The form of $\widetilde{U}_d$ can always be fixed recalling that $V_{\text{CKM}} = U_u^\dagger U_d$.

In scenarios with Universality in the left-handed sector (see Section 5) it is sufficient to present $U_{u,d}, V_{u,d}$ of pLU. At leading order in the ratio of the quark masses, they read:

$$U_u = \begin{pmatrix} 1 & -ab^*\frac{y_c^2}{y_t^2} & a\frac{y_c}{y_t} \\ a^*b\frac{y_u^2}{y_t^2} & 1 & b\frac{y_c}{y_t} \\ -a^*\frac{y_c}{y_t} & -b^*\frac{y_c}{y_t} & 1 \end{pmatrix} \qquad V_u = \begin{pmatrix} 1 & -ab^*\frac{y_u y_c}{y_t^2} & a\frac{y_u y_c}{y_t^2} \\ a^*b\frac{y_u y_c}{y_t^2} & 1 & b\frac{y_c^2}{y_t^2} \\ -a^*\frac{y_u y_c}{y_t^2} & -b^*\frac{y_c^2}{y_t^2} & 1 \end{pmatrix} \qquad (B.10)$$

and

$$U_d = \widetilde{U}_d \begin{pmatrix} 1 & -a'b'^*\frac{y_s^2}{y_b^2} & a'\frac{y_s}{y_b} \\ a'^*b'\frac{y_d^2}{y_b^2} & 1 & b'\frac{y_s}{y_b} \\ -a'^*\frac{y_s}{y_b} & -b'^*\frac{y_s}{y_b} & 1 \end{pmatrix} \qquad V_d = \begin{pmatrix} 1 & -a'b'^*\frac{y_d y_s}{y_b^2} & a'\frac{y_d y_s}{y_b^2} \\ a'^*b'\frac{y_d y_s}{y_b^2} & 1 & b'\frac{y_s^2}{y_b^2} \\ -a'^*\frac{y_d y_s}{y_b^2} & -b'^*\frac{y_s^2}{y_b^2} & 1 \end{pmatrix} \qquad (B.11)$$

where $\widetilde{U}_d$ was defined in Eq. (5.24) (note that this matrix differs from the one in the pRU scenario). The approximation is accurate as long as $|a|, |b| < y_t/y_c$ and $|a'|, |b'| < y_b/y_s$. As

seen in Sections 5.2, the upper bound cannot be reached otherwise the CKM is not reproduced naturally. In particular, in Sections 5.2 we decided to take $|a|, |b|, |b'| \sim 1$ and $|a'| \sim \lambda_C$. LU has $U_u = V_u = V_d = \mathbf{1}$ and $U_d = V_{\mathrm{CKM}}$.[33]

## B.2 Quark bilinears at leading order

The low-energy dynamics is determined by two basic ingredients, the Lagrangian of Eq. (2.6) and the symmetries. These in turn determine the transformation properties of the $\lambda_\psi$ mixing parameters. In this appendix, we will show how these elements combine into the structure of the low-energy effective Lagrangian in our flavor symmetric models. In particular, we will focus on operators involving fermions, as their coefficients have a non-trivial flavor dependence.

At dimension 6, or lower, the operators of interest contain either two or four elementary quarks. In the Lagrangian of Eq. (2.6), each bilinear in the elementary fermions will be accompanied by the most general combination of the $\lambda_\psi$'s that is compatible with the symmetries. More in detail, this combination transforms in a tensorial representation of $G_{\mathrm{quarks}}$. The basic bilinear building blocks are structures of the form

$$[\lambda_\psi \lambda_{\psi'}^\dagger]^{ij} \overline{\psi}^i \psi'^j, \tag{B.12}$$

with $\psi, \psi'$ any two types of fermions and $i, j$ their flavor indices. At leading order in $\lambda_\psi$, fermion bilinears are weighted by the corresponding $\lambda_\psi \lambda_{\psi'}^\dagger$. Four-fermion operators have thus, at the lowest order, coefficients proportional to the product of two of these structures. Higher order contributions are controlled by further insertions, properly contracted on some of their flavor indices. Diagrammatically, these insertions correspond to loop exchanges of a virtual elementary fermion, such as in Fig. 3.

In the following, we will present several examples of how to build such structures in the case of the Right Universality models. A similar reasoning also applies to the Left Universality models. Furthermore, we will show that, in the mass basis, it is always possible to parametrize all structures $\lambda_\psi \lambda_{\psi'}^\dagger$ in terms of the SM Yukawa and CKM matrices, the diagonal compositeness parameters $\varepsilon_\psi$ and the mixing matrices $U_{u/d}$ and $V_{u/d}$ that diagonalize the Yukawas. This parametrization allows for a clearer understanding of the flavor-violating structure of each model. The complete list of such structures for scenarios with universality in the right-handed quarks are reported in Table 5, while those with universality in the left-handed quarks are reported in Table 6.

In the following, the $\lambda_\psi \lambda_{\psi'}^\dagger$ tensors are defined in the flavor basis, while $S_{\overline{\psi}\psi'}$ denote the corresponding expression defined in the relevant mass basis (which basis, up or down, will be evident case by case).

**RR structures** Consider the MFV RU scenario first. There the only structures involving two right handed quarks are

$$\lambda_u \lambda_u^\dagger \qquad \text{and} \qquad \lambda_d \lambda_d^\dagger. \tag{B.13}$$

Other structures, such as $\lambda_u^\dagger \lambda_d$ (associated with $\bar{u}_R d_R$), are forbidden by the $U(3)_U \times U(3)_D$ symmetry of the strong sector. In the mass basis, since the right handed mixings are diagonal

---

[33]As emphasized in Sections 5.2, because $\widetilde{U}_d$ is 2-dimensional some of the parameters $a, b, a', b'$ are necessary to reproduce the CKM.

(see (4.4)), we immediately find

$$S^{\text{RU}}_{\bar{u}_R u_R} = V_u^\dagger \lambda_u \lambda_u^\dagger V_u = \varepsilon_u^2 g_*^2 \qquad \text{and} \qquad S^{\text{RU}}_{\bar{d}_R d_R} = V_d^\dagger \lambda_d \lambda_d^\dagger V_d = \varepsilon_d^2 g_*^2 \,. \tag{B.14}$$

In this model, there is clearly no flavor violation in the right handed sector.

Instead, for puRU, the RR structure in the up sector is controlled by the linear combination

$$\begin{aligned}
&\lambda_u^{(2)}[\lambda_u^{(2)}]^\dagger + r\lambda_u^{(1)}[\lambda_u^{(1)}]^\dagger \\
&= \left(\lambda_u^{(2)}[\lambda_u^{(2)}]^\dagger + \varepsilon_u^2/\varepsilon_{u_3}^2 \lambda_u^{(1)}[\lambda_u^{(1)}]^\dagger\right) + \left(r - \varepsilon_u^2/\varepsilon_{u_3}^2\right)\lambda_u^{(1)}[\lambda_u^{(1)}]^\dagger \\
&= \varepsilon_u^2 g_*^2 \mathbf{1} + \left(r - \varepsilon_u^2/\varepsilon_{u_3}^2\right)\lambda_u^{(1)}[\lambda_u^{(1)}]^\dagger \,,
\end{aligned} \tag{B.15}$$

with $r$ a $O(1)$ real number. In the second line we have employed the definitions in (4.30) and added and subtracted the term $\varepsilon_u^2/\varepsilon_{u_3}^2 \lambda_u^{(1)}[\lambda_u^{(1)}]^\dagger$ to reproduce the identity matrix plus a flavor non-universal term. In the mass basis, this becomes

$$S^{\text{puRU}}_{\bar{u}_R u_R} = \varepsilon_u^2 g_*^2 (\mathbf{1} - V_u^\dagger \widetilde{P} V_u) + r\varepsilon_{u_3}^2 g_*^2 V_u^\dagger \widetilde{P} V_u \,, \tag{B.16}$$

with $\widetilde{P}$ the projector onto the third family

$$\widetilde{P} = \frac{1}{\varepsilon_{u_3}^2 g_*^2}\lambda_u^{(1)}[\lambda_u^{(1)}]^\dagger = \begin{pmatrix} 0 & 0 & 0 \\ 0 & 0 & 0 \\ 0 & 0 & 1 \end{pmatrix} \,. \tag{B.17}$$

This shows that in puRU, flavor violation in the up-right sector is controlled by the small mixing angles of $V_u$. The down sector is instead still diagonal and flavor universal as in RU.

In pRU (see Eq. (4.55)) the up-sector stays the same and a similar result also holds in the down sector. By the same procedure, we find

$$S^{\text{pRU}}_{\bar{d}_R d_R} = \varepsilon_d^2 g_*^2 (\mathbf{1} - V_d^\dagger \widetilde{P} V_d) + r'\varepsilon_{d_3}^2 g_*^2 V_d^\dagger \widetilde{P} V_d \tag{B.18}$$

where of course $r'$ is in general independent from $r$.

**LR structures**  We now look at the structures involving one left and one right handed quark. It is clear that the natural objects to compare these structures to are the Standard Model Yukawa. For RU there are only two such structures that we can build, namely

$$\lambda_{q_u}\lambda_u^\dagger \sim g_* Y_u \qquad \text{and} \qquad \lambda_{q_d}\lambda_d^\dagger \sim g_* Y_d \,. \tag{B.19}$$

In the mass basis, we have

$$S^{\text{RU}}_{\bar{u}_L u_R} = g_* \widetilde{Y}_u \quad \text{and} \quad S^{\text{RU}}_{\bar{d}_L d_R} = g_* \widetilde{Y}_d \,. \tag{B.20}$$

The mixed up-down structures instead read

$$S^{\text{RU}}_{\bar{d}_L u_R} = U_d^\dagger \lambda_{q_u}\lambda_u^\dagger V_u = V_{\text{CKM}}^\dagger S^{\text{RU}}_{\bar{u}_L u_R} \qquad \text{and} \qquad S^{\text{RU}}_{\bar{u}_L d_R} = U_u^\dagger \lambda_{q_d}\lambda_d^\dagger V_d = V_{\text{CKM}} S^{\text{RU}}_{\bar{d}_L d_R} \,. \tag{B.21}$$

The situation changes in puRU. Indeed, in this case, the structure is given by a linear combination of two terms

$$\lambda_{q_u}^{(2)}[\lambda_u^{(2)}]^\dagger + r\lambda_{q_u}^{(1)}[\lambda_u^{(1)}]^\dagger \,. \tag{B.22}$$

In Eq. (4.31) we defined the Yukawa to correspond to $r = 1$, but every other operator involving an up-left and an up-right quark contains this same structure with a different value for $r$. By adding and subtracting $\lambda_{q_u}^{(1)}[\lambda_u^{(1)}]^\dagger$, we can rewrite (B.22) in terms of the Standard Model Yukawa, plus a term that contains all the flavor-violation. Such a term is proportional to $r - 1$, so this parameter describes the "misalignment" of that structure with respect to the SM. Explicitly, in the mass basis, we can write

$$S_{\bar{u}_L u_R}^{\text{puRU}} = g_* \widetilde{Y}_u + (r - 1) g_* y_t U_u^\dagger \widetilde{P} V_u \,. \tag{B.23}$$

The $\bar{d}_L u_R$ structure in the mass basis can be obtained by multiplying the previous one by $V_{\text{CKM}}$ on the left, see Table 5. The structures involving $d_R$ are the same as in RU.

In pRU a structure like (B.23) also appears in the down sector where we get

$$S_{\bar{d}_L d_R}^{\text{pRU}} = g_* \widetilde{Y}_d + (r' - 1) g_* y_b U_d^\dagger \widetilde{U}_d \widetilde{P} V_d \,. \tag{B.24}$$

The remaining ones are shown in Table 5.

**LL structures**   Finally, we look at the structures containing two left handed quarks. For these ones, flavor violation appears already in RU models. That is because we can build a linear combination

$$\lambda_{q_u} \lambda_{q_u}^\dagger + r \lambda_{q_d} \lambda_{q_d}^\dagger \,, \tag{B.25}$$

which is not diagonalizable at the same time as the Yukawas. Using the expressions in (4.4), in the bass basis we find

$$S_{\bar{u}_L u_L}^{\text{RU}} = \frac{1}{\varepsilon_u^2} \widetilde{Y}_u^2 + \frac{r}{\varepsilon_d^2} V_{\text{CKM}} \widetilde{Y}_d^2 V_{\text{CKM}}^\dagger \,, \tag{B.26}$$

$$S_{\bar{u}_L d_L}^{\text{RU}} = \frac{1}{\varepsilon_u^2} \widetilde{Y}_u^2 V_{\text{CKM}} + \frac{r}{\varepsilon_d^2} V_{\text{CKM}} \widetilde{Y}_d^2 \,, \tag{B.27}$$

$$S_{\bar{d}_L d_L}^{\text{RU}} = \frac{1}{\varepsilon_u^2} V_{\text{CKM}}^\dagger \widetilde{Y}_u^2 V_{\text{CKM}} + \frac{r}{\varepsilon_d^2} \widetilde{Y}_d^2 \,. \tag{B.28}$$

Notice that each structure can be obtained from the other by multiplying to the left or right with $V_{\text{CKM}}$.

In the same way, the models puRU and pRU add additional terms to this combination, and consequently additional sources of flavor violation. For puRU we have three terms

$$\lambda_{q_u}^{(2)}[\lambda_{q_u}^{(2)}]^\dagger + r \lambda_{q_u}^{(1)}[\lambda_{q_u}^{(1)}]^\dagger + r' \lambda_{q_d} \lambda_{q_d}^\dagger \,. \tag{B.29}$$

Again, adding and subtracting the term $\lambda_{q_u}^{(1)}[\lambda_{q_u}^{(1)}]^\dagger$ with a suitable coefficient we can write in the mass basis

$$S_{\bar{u}_L u_L}^{\text{puRU}} = \frac{1}{\varepsilon_u^2} \widetilde{Y}_u^2 + \left( \frac{r}{\varepsilon_{u_3}^2} - \frac{1}{\varepsilon_u^2} \right) y_t^2 U_u^\dagger \widetilde{P} U_u + \frac{r'}{\varepsilon_d^2} V_{\text{CKM}} \widetilde{Y}_d^2 V_{\text{CKM}}^\dagger \,. \tag{B.30}$$

The additional source of flavor violation is controlled by the mixing matrix $U_u$ and weighted by $1/\varepsilon_{u_3}^2$. The LL structures involving the down quarks can be derived analogously and are reported in Table 5.

Finally, in pRU we have four independent structures

$$\lambda_{q_u}^{(2)}[\lambda_{q_u}^{(2)}]^\dagger + r\lambda_{q_u}^{(1)}[\lambda_{q_u}^{(1)}]^\dagger + r'\lambda_{q_d}^{(2)}[\lambda_{q_d}^{(2)}]^\dagger + r''\lambda_{q_d}^{(1)}[\lambda_{q_d}^{(1)}]^\dagger \,. \tag{B.31}$$

The expressions in the mass basis can be derived along the same lines as done above. For example, the up-quark structures become

$$\begin{aligned}
S_{\bar{u}_L u_L}^{\text{pRU}} =& \frac{1}{\varepsilon_u^2}\widetilde{Y}_u^2 + \left(\frac{r}{\varepsilon_{u_3}^2} - \frac{1}{\varepsilon_u^2}\right) y_t^2 U_u^\dagger \widetilde{P} U_u \\
&+ r' V_{\text{CKM}} \left[\frac{1}{\varepsilon_d^2}\widetilde{Y}_d^2 + \left(\frac{r''}{\varepsilon_{d_3}^2} - \frac{1}{\varepsilon_d^2}\right) y_b^2 U_d^\dagger \widetilde{U}_d \widetilde{P} \widetilde{U}_d^\dagger U_d\right] V_{\text{CKM}}^\dagger \,.
\end{aligned} \tag{B.32}$$

The remaining structures are collected in Table 5.

## B.3  Matching to the low-energy effective operators

Many constraining observables are studied below the weak scale. Their physics is thus conventionally described by low-energy effective operators $\mathcal{O}_\mathcal{Y}$ where the electroweak sector has been integrated out. In this appendix, we perform the matching between the coefficients of these $\mathcal{O}_\mathcal{Y}$ and the SM-invariant operators of Tab. 4. The Wilson coefficients of the low-energy operators $\mathcal{O}_\mathcal{Y}$ below the weak scale are by definition always in the mass basis and are labeled as $C_\mathcal{Y}$. Instead, we remind the reader that $\mathcal{C}_\mathcal{X}$ and $\widetilde{\mathcal{C}}_\mathcal{X}$ denote respectively the Wilson coefficients of the operators of Tab. 4 in the gauge and mass basis.

We first consider the rare leptonic decays of $B_s$ mesons. In all the models we study the dominant transitions come from the operators in Tab. 4 that modify the electroweak couplings. At low energies, after integrating out the electroweak bosons, these operators match to four fermion operators that are conventionally parameterized as [77]

$$\begin{aligned}
\mathcal{H}_{\text{eff}} = -\frac{4G_F}{\sqrt{2}} V_{\text{CKM}}^{tb}(V_{\text{CKM}}^{ts})^* \frac{e^2}{16\pi^2} \big[& C_{10}(\bar{s}_L\gamma^\mu b_L)(\bar{\ell}\gamma_\mu\gamma^5\ell) + C_{10}'(\bar{s}_R\gamma^\mu b_R)(\bar{\ell}\gamma_\mu\gamma^5\ell) + \\
&+ C_9(\bar{s}_L\gamma^\mu b_L)(\bar{\ell}\gamma_\mu\ell) + C_9'(\bar{s}_R\gamma^\mu b_R)(\bar{\ell}\gamma_\mu\ell)\big] + hc \,.
\end{aligned} \tag{B.33}$$

Explicitly, the matching is given by

$$C_{10} = \frac{2\sqrt{2}\pi^2}{G_F e^2} \frac{1}{V_{\text{CKM}}^{tb}(V_{\text{CKM}}^{ts})^*} \left([\widetilde{\mathcal{C}}_{Hq}^{(1)}]^{23} + [\widetilde{\mathcal{C}}_{Hq}^{(3)}]^{23}\right) \,, \tag{B.34}$$

$$C_{10}' = \frac{2\sqrt{2}\pi^2}{G_F e^2} \frac{1}{V_{\text{CKM}}^{tb}(V_{\text{CKM}}^{ts})^*} [\widetilde{\mathcal{C}}_{Hd}]^{23} \,, \tag{B.35}$$

$$C_9 = C_{10}\left(-1 + 4\sin^2\theta_W\right) - \frac{4\sqrt{2}\pi^2}{G_F e} \frac{1}{V_{\text{CKM}}^{tb}(V_{\text{CKM}}^{ts})^*} \left(\cos\theta_W[\widetilde{\mathcal{C}}_{qD}^{(1)}]^{23} - \sin\theta_W[\widetilde{\mathcal{C}}_{qD}^{(3)}]^{23}\right) \,, \tag{B.36}$$

$$C_9' = C_{10}'\left(-1 + 4\sin^2\theta_W\right) - \frac{4\sqrt{2}\pi^2}{G_F e} \frac{1}{V_{\text{CKM}}^{tb}(V_{\text{CKM}}^{ts})^*} \cos\theta_W[\widetilde{\mathcal{C}}_{dD}]^{23} \,, \tag{B.37}$$

where the Wilson Coefficients are all in the mass basis. The operators $\mathcal{O}_{Hq}^{(1,3)}, \mathcal{O}_{Hd}$ of Tab. 4 modify the coupling of the SM fermion to the $Z$, and after the vector boson is integrated out give rise to four-fermion operators coupling the flavor-changing quark current to the $Z$

| | Right Univ. | Partial Up Right Univ. | Partial Right Univ. |
|---|---|---|---|
| $S_{\bar{u}_L u_L}$ | $\frac{1}{\varepsilon_u^2}\widetilde{Y}_u^2 +$ $\frac{r}{\varepsilon_d^2}V_{\mathrm{CKM}}\widetilde{Y}_d^2 V_{\mathrm{CKM}}^\dagger$ | $\frac{1}{\varepsilon_u^2}\widetilde{Y}_u^2 + \left(\frac{r}{\varepsilon_{u_3}^2} - \frac{1}{\varepsilon_u^2}\right)y_t^2 U_u^\dagger \widetilde{P} U_u$ $+ \frac{r'}{\varepsilon_d^2}V_{\mathrm{CKM}}\widetilde{Y}_d^2 V_{\mathrm{CKM}}^\dagger$ | $\frac{1}{\varepsilon_u^2}\widetilde{Y}_u^2 + \left(\frac{r}{\varepsilon_{u_3}^2} - \frac{1}{\varepsilon_u^2}\right)y_t^2 U_u^\dagger \widetilde{P} U_u +$ $r'V_{\mathrm{CKM}}\left[\frac{1}{\varepsilon_d^2}\widetilde{Y}_d^2 + \left(\frac{r''}{\varepsilon_{d_3}^2} - \frac{1}{\varepsilon_d^2}\right)y_b^2 U_d^\dagger \widetilde{U}_d \widetilde{P}\widetilde{U}_d^\dagger U_d\right] V_{\mathrm{CKM}}^\dagger$ |
| $S_{\bar{u}_R u_R}$ | $\varepsilon_u^2 g_*^2$ | $\varepsilon_u^2 g_*^2(\mathbf{1} - V_u^\dagger \widetilde{P} V_u) + r\varepsilon_{u_3}^2 g_*^2 V_u^\dagger \widetilde{P} V_u$ | $\varepsilon_u^2 g_*^2(\mathbf{1} - V_u^\dagger \widetilde{P} V_u) + r\varepsilon_{u_3}^2 g_*^2 V_u^\dagger \widetilde{P} V_u$ |
| $S_{\bar{u}_L u_R}$ | $g_*\widetilde{Y}_u$ | $g_*\widetilde{Y}_u + (r-1)g_* y_t U_u^\dagger \widetilde{P} V_u$ | $g_*\widetilde{Y}_u + (r-1)g_* y_t U_u^\dagger \widetilde{P} V_u$ |
| $S_{\bar{d}_L d_L}$ | $V_{\mathrm{CKM}}^\dagger S_{\bar{u}_L u_L} V_{\mathrm{CKM}}$ | $V_{\mathrm{CKM}}^\dagger S_{\bar{u}_L u_L} V_{\mathrm{CKM}}$ | $V_{\mathrm{CKM}}^\dagger S_{\bar{u}_L u_L} V_{\mathrm{CKM}}$ |
| $S_{\bar{d}_R d_R}$ | $\varepsilon_d^2 g_*^2$ | $\varepsilon_d^2 g_*^2$ | $\varepsilon_d^2 g_*^2(\mathbf{1} - V_d^\dagger \widetilde{P} V_d) + r\varepsilon_{d_3}^2 g_*^2 V_d^\dagger \widetilde{P} V_d$ |
| $S_{\bar{d}_L d_R}$ | $g_*\widetilde{Y}_d$ | $g_*\widetilde{Y}_d$ | $g_*\widetilde{Y}_d + (r-1)g_* y_b U_d^\dagger \widetilde{U}_d \widetilde{P} V_d$ |
| $S_{\bar{u}_L d_L}$ | $S_{\bar{u}_L u_L} V_{\mathrm{CKM}}$ | $S_{\bar{u}_L u_L} V_{\mathrm{CKM}}$ | $S_{\bar{u}_L u_L} V_{\mathrm{CKM}}$ |
| $S_{\bar{u}_R d_R}$ | $0$ | $0$ | $0$ |
| $S_{\bar{u}_L d_R}$ | $V_{\mathrm{CKM}}S_{\bar{d}_L d_R}$ | $V_{\mathrm{CKM}}S_{\bar{d}_L d_R}$ | $V_{\mathrm{CKM}}S_{\bar{d}_L d_R}$ |
| $S_{\bar{d}_L u_R}$ | $V_{\mathrm{CKM}}^\dagger S_{\bar{u}_L u_R}$ | $V_{\mathrm{CKM}}^\dagger S_{\bar{u}_L u_R}$ | $V_{\mathrm{CKM}}^\dagger S_{\bar{u}_L u_R}$ |

Table 5: Leading order structures of the coefficients $S_{\overline{\psi}\psi'}$ of the quark bilinears in the mass basis for models with universality in the right-handed sector (see Section B.2 for details). Flavor indices are implicit. The parameters $r, r', r''$ are unknown, non-universal $O(1)$ numbers. The matrix $\widetilde{P}$ is defined in Eq. (4.38).

| | Left Univ. | Partial Left Univ. |
|---|---|---|
| $S_{\bar{u}_L u_L}$ | $\varepsilon_q^2 g_*^2$ | $\varepsilon_q^2 g_*^2 + (r\varepsilon_{q_3}^2 - \varepsilon_q^2)g_*^2 U_u^\dagger \widetilde{P} U_u$ |
| $S_{\bar{u}_R u_R}$ | $\frac{1}{\varepsilon_q^2}\widetilde{Y}_u^2$ | $\frac{1}{\varepsilon_q^2}\widetilde{Y}_u^2 + \left(\frac{r}{\varepsilon_{q_3}^2} - \frac{1}{\varepsilon_q^2}\right)y_t^2 V_u^\dagger \widetilde{P} V_u$ |
| $S_{\bar{u}_L u_R}$ | $g_*\widetilde{Y}_u$ | $g_*\widetilde{Y}_u + (r-1)g_* y_t U_u^\dagger \widetilde{P} V_u$ |
| $S_{\bar{d}_L d_L}$ | $V_{\mathrm{CKM}}^\dagger S_{\bar{u}_L u_L} V_{\mathrm{CKM}}$ | $V_{\mathrm{CKM}}^\dagger S_{\bar{u}_L u_L} V_{\mathrm{CKM}}$ |
| $S_{\bar{d}_R d_R}$ | $\frac{1}{\varepsilon_q^2}\widetilde{Y}_d^2$ | $\frac{1}{\varepsilon_q^2}\widetilde{Y}_d^2 + \left(\frac{r}{\varepsilon_{q_3}^2} - \frac{1}{\varepsilon_q^2}\right)y_b^2 V_d^\dagger \widetilde{P} V_d$ |
| $S_{\bar{d}_L d_R}$ | $g_*\widetilde{Y}_d$ | $g_*\widetilde{Y}_d + (r-1)g_* y_b U_d^\dagger \widetilde{P} V_d$ |
| $S_{\bar{u}_L d_L}$ | $S_{\bar{u}_L u_L} V_{\mathrm{CKM}}$ | $S_{\bar{u}_L u_L} V_{\mathrm{CKM}}$ |
| $S_{\bar{u}_R d_R}$ | $\frac{1}{\varepsilon_q^2}\widetilde{Y}_u V_{\mathrm{CKM}}\widetilde{Y}_d$ | $\frac{1}{\varepsilon_q^2}\widetilde{Y}_u V_{\mathrm{CKM}}\widetilde{Y}_d + \left(\frac{r}{\varepsilon_{q_3}^2} - \frac{1}{\varepsilon_q^2}\right) y_t y_b V_u^\dagger \widetilde{P} V_d$ |
| $S_{\bar{u}_L d_R}$ | $V_{\mathrm{CKM}}S_{\bar{d}_L d_R}$ | $V_{\mathrm{CKM}}S_{\bar{d}_L d_R}$ |
| $S_{\bar{d}_L u_R}$ | $V_{\mathrm{CKM}}^\dagger S_{\bar{u}_L u_R}$ | $V_{\mathrm{CKM}}^\dagger S_{\bar{u}_L u_R}$ |

Table 6: Same as in Table 5, but for models with universality in the left-handed sector. The parameters $r, r' = O(1)$ are unknown and non-universal.

current (see for instance [78]). Notice that the resulting $C_9^{(\prime)}$ are suppressed by a factor $(1 - 4\sin^2\theta_w) \sim 0.08$ compared to the $C_{10}^{(\prime)}$ ones. In addition, the operators $\mathcal{O}_{qD}^{(1,3)}, \mathcal{O}_{dD}$ give rise to flavor-violating anomalous couplings to both the photon and the $Z$. The exchange of a virtual photon creates flavor-violating interactions between the quark current and the EM current that contribute only to the $C_9^{(\prime)}$ operators. Virtual $Z$ exchange is instead suppressed at low energies by a factor $q^2/m_Z^2$, with $q^2$ the invariant mass of the lepton pair, and is therefore negligible.

The $b \to s\gamma$ transitions are parametrized by the following effective Hamiltonian

$$\mathcal{H}_{\text{eff}} = -\frac{4G_F}{\sqrt{2}} V_{\text{CKM}}^{tb}(V_{\text{CKM}}^{ts})^* \frac{m_b e}{16\pi^2} F^{\mu\nu} \left(C_7 \bar{s}_L \sigma_{\mu\nu} b_R + C_7' \bar{s}_R \sigma_{\mu\nu} b_L\right) + \text{hc} .. \tag{B.38}$$

These operators match to the operator of Tab. 4 as

$$C_7 = \frac{4\sqrt{2}\pi^2}{G_F e} \frac{1}{V_{\text{CKM}}^{tb}(V_{\text{CKM}}^{ts})^*} \frac{v}{m_b\sqrt{2}} [\widetilde{\mathcal{C}}_{d\gamma}]^{23}, \tag{B.39}$$

$$C_7' = \frac{4\sqrt{2}\pi^2}{G_F e} \frac{1}{V_{\text{CKM}}^{tb}(V_{\text{CKM}}^{ts})^*} \frac{v}{m_b\sqrt{2}} [\widetilde{\mathcal{C}}_{d\gamma}^*]^{32}. \tag{B.40}$$

In the above expressions we introduced the combination

$$[\widetilde{\mathcal{C}}_{f\gamma}]^{ij} = \cos\theta_W [\widetilde{\mathcal{C}}_{fB}]^{ij} - \sin\theta_W [\widetilde{\mathcal{C}}_{fW}]^{ij} \tag{B.41}$$

which contributes for example to $b \to s\gamma$ and the quark EDMs. In particular, the anomalous electric and magnetic moments of the fermion $f_i$ are

$$d_f^{ii} = \sqrt{2}\, v \operatorname{Im}[\widetilde{\mathcal{C}}_{f\gamma}]^{ii} \tag{B.42}$$

$$a_f^{ii} = (g_{f_i} - 2)/2 = 2\sqrt{2}\, v\, m_{f_i} \operatorname{Re}[\widetilde{\mathcal{C}}_{f\gamma}]^{ii}/e.$$

Similar considerations hold for the dipole operators that involve gluons.

The last class of low-energy operators we are interested in are the four fermion operators that modify $\Delta F = 2$ transitions. The standard parametrization can be found in [79]. The subset that is relevant to our analysis is

$$\mathcal{O}_1^{f_if_j} = (\bar{f}_{jL}^\alpha \gamma_\mu f_{iL}^\alpha)(\bar{f}_{jL}^\beta \gamma^\mu f_{iL}^\beta), \quad \mathcal{O}_1'^{f_if_j} = (\bar{f}_{jR}^\alpha \gamma_\mu f_{iR}^\alpha)(\bar{f}_{jR}^\beta \gamma^\mu f_{iR}^\beta), \tag{B.43}$$

$$\mathcal{O}_2^{f_if_j} = (\bar{f}_{jR}^\alpha f_{iL}^\alpha)(\bar{f}_{jR}^\beta f_{iL}^\beta), \quad \mathcal{O}_2'^{f_if_j} = (\bar{f}_{jL}^\alpha f_{iR}^\alpha)(\bar{f}_{jL}^\beta f_{iR}^\beta) \tag{B.44}$$

$$\mathcal{O}_4^{f_if_j} = (\bar{f}_{jR}^\alpha f_{iL}^\alpha)(\bar{f}_{jL}^\beta f_{iR}^\beta), \quad \mathcal{O}_5^{f_if_j} = (\bar{f}_{jR}^\alpha f_{iL}^\beta)(\bar{f}_{jL}^\beta f_{iR}^\alpha), \tag{B.45}$$

where the indices $\alpha$ and $\beta$ are the $SU(3)$ color indices. The Wilson coefficients of $\mathcal{O}_1^{f_if_j}$, $\mathcal{O}_1'^{f_if_j}$, $\mathcal{O}_4^{f_if_j}$, $\mathcal{O}_5^{f_if_j}$ match at tree-level with the SM-gauge-invariant dimension-6 operators in Tab. 4 as [78]

$$C_1^{f_if_j} = [\widetilde{\mathcal{C}}_{qq}^{(1)}]^{jiji} + [\widetilde{\mathcal{C}}_{qq}^{(3)}]^{jiji}, \tag{B.46}$$

$$C_1'^{f_if_j} = [\widetilde{\mathcal{C}}_{rr}]^{jiji}, \tag{B.47}$$

$$C_4^{f_if_j} = -[\widetilde{\mathcal{C}}_{qr}^{(8)}]^{jiji}, \tag{B.48}$$

$$C_5^{f_if_j} = -2[\widetilde{\mathcal{C}}_{qr}^{(1)}]^{jiji} + \frac{1}{3}[\widetilde{\mathcal{C}}_{qr}^{(8)}]^{jiji}, \tag{B.49}$$

where $r = u, d$ depending on whether the $f$ on the left-hand side corresponds to up or down type quarks respectively. The operators $\mathcal{O}_2^{f_if_j}$, $\mathcal{O}_2'^{f_if_j}$ are generated at dimension-8, and will be discussed in App. C.4.1.

## B.4   Neutron EDM

The exact expression of the neutron EDM in terms of the Wilson coefficients of CP-odd operators involving quarks, gluons and photons is unknown. The electric dipole moments of the up and down quarks, $d_u^{11}, d_d^{11}$ (see (B.42)), are expected to enter $d_n$ with coefficients of order unity. Analogous NDA considerations instead imply that the chromo-electric dipoles must appear multiplied by an overall factor $\sim e/(4\pi)$. Because in our models the electric and chromo-electric dipoles are expected to have comparable coefficients, the impact of the former is presumably subleading and will be ignored.

Considerations based on the $SU(3)$ chiral symmetry and naive dimensional analysis indicate that also the EDM of the strange quark should contribute sizably. In perturbation theory such effect is seen as a strange loop coupled to the photon via $d_d^{22}$ and attached to the hadronic states via virtual gluons. Because the virtuality of the gluons and the strange quark are of order the QCD scale, there is a priori no reason for such contributions to be small. The only parametric suppression we can identify is a factor of $1/N_c$ due to the presence of an additional fermion loop. These considerations lead us to infer that the electric dipoles of the three light quarks appear in $d_n$ according to the expression

$$d_n = c_u d_u^{11} + c_d d_d^{11} + c_s \frac{1}{N_c} d_d^{22}, \tag{B.50}$$

with $c_{u,d,s}$ real numbers naively of order unity. We should note however that OZI-suppressed processes are experimentally observed to be a bit rarer than suggested by a naive $1/N_c$ argument. Perhaps $|c_s| \sim 1/10$ would not be too unrealistic, then. Given that in the real world $N_c = 3$, the strange quark may contribute very significantly if indeed $|c_s| \sim 0.1 \div 1$, as we naively expect. The potential relevance of the strange quark has already been emphasized by other authors (see e.g. [80]) but, as far as we know, the phenomenological consequences of the estimate in Eq. (B.50) have not been fully appreciated in the literature. Actually, the contribution $\propto c_s$ is often ignored. For example, in Ref. [81] the diagrams with virtual strange quarks were not considered.

Lattice simulations partially confirm our expectation finding that $c_u \sim -0.2$, $c_d \sim +0.8$ [17], but our guess for $c_s$ seems completely off. The paper [61] actually finds an unexpectedly large accidental suppression $|c_s| \sim 1/100$. To better appreciate this result it is instructive to have a look at Table 2 of [82], where additional numerical data of that analysis is presented. From the first two lines of the table we see that the authors find that OZI-suppressed contributions to the nucleon matrix elements of scalar and axial-vector operators involving the $u$ and $d$ quarks are remarkably well estimated by a simple factor $1/N_c$, with virtually no accidental cancellations. Yet, the disconnected contributions to the nucleon matrix element of *tensor* operators with $u, d$ are suppressed by the accidental $\sim 1/10$ found in other OZI-suppressed quantities (see the third line). If the same was found for the strange quark we would have $|c_s| \sim 0.1$. What is striking however is that the matrix element of the tensor operator of the $s$-quark in line 4 is further suppressed compared to the analogous disconnected $u, d$ contribution by an additional $1/10$. And it is this further suppression that leads to the value $|c_s| \sim 1/100$ quoted in [61].[34] Given our difficulty in understanding parametrically the nature of such large accidental suppression, in this paper we conservatively contemplate also the implications of $c_s$ of order unity. We warn the reader that, however, $c_s \sim 1$ may be a

---

[34]Of course the distinction between "connected" and "disconnected" diagrams ceases to make rigorous sense when considering $u, d$-operators. LV would like to thank T. Bhattacharya for discussions on the results of [61].

significant overestimate. Our aim is to emphasize the phenomenological relevance of $c_s$ and hopefully motivate further investigations of this quantity both via analytical methods as well as non-perturbative techniques.

Finally, the contribution to $d_n$ from quarks heavier than the QCD scale $\Lambda_{\text{QCD}}$ is captured, below the heavy quark mass, by CP-odd operators involving the light degrees of freedom. Consider the charm quark first. At 1-loop order its EDM is mapped into dimension-8 operators with gluons and photons [83]

$$\mathcal{L}_{\mu<m_c} \supset c d_u^{22}(m_c)\frac{g_s^3(m_c)}{16\pi^2}\frac{1}{m_c^3}FGG\widetilde{G}, \tag{B.51}$$

where $c = O(1)$. Using naive dimensional analysis, the resulting contribution to the neutron EDM reads

$$d_n \sim d_u^{22}(m_c)\left(\frac{g_s(m_c)}{4\pi}\frac{\Lambda_{\text{QCD}}}{m_c}\right)^3. \tag{B.52}$$

The RG evolution from $\sim m_c$ down to the neutron mass scale has been ignored because it would introduce a correction of the same order as the uncertainty in our NDA estimate. Next consider the charm chromo-electric dipole moment $\widetilde{d}_f = \sqrt{2}\,v\,\text{Im}[\widetilde{\mathcal{C}}_{fG}]$. Matching it at 1-loop to the pure gluon operator in Eq. (2.13), and again ignoring the RG evolution down to the GeV, we have

$$d_n \sim \widetilde{d}_u^{22}(m_c)\frac{g_s^2(m_c)}{16\pi^2}\left(\frac{e}{4\pi}\frac{\Lambda_{\text{QCD}}}{m_c}\right). \tag{B.53}$$

Observing that $\alpha_s(m_c) \sim 0.4$ and $\Lambda_{\text{QCD}}/m_c \sim 0.3$ we approximately find $d_n \sim 2\times 10^{-4}\,d_u^{22}(m_c)$ and $d_n \sim 2\times 10^{-4}\,\widetilde{d}_u^{22}(m_c)$. In our models, we expect $d_u^{22}$ and $\widetilde{d}_u^{22}$ to be parametrically of the same order. As a result, the contributions in (B.52) and (B.53) are numerically comparable. However, in all our models the impact of the c-quark turns out to be negligible compared to those in (B.50). This is even more so for the dipoles of the bottom and top quarks. In our paper, we can thus safely approximate the neutron EDM by the expression in Eq. (B.50).

## C    Constraints and prospects

In the following we collect the phenomenological constraints on the models presented in the main text. In Appendix C.1 we discuss the current bounds associated with purely bosonic interactions, whereas the electroweak $\widehat{T}$ is analyzed in Appendix C.2. Then we move to the flavor-dependent (both diagonal and off-diagonal) constraints in Appendix C.3. In Appendix C.4 we also include a separate discussion of a few interesting bounds that for our flavor models are subdominant in the range of parameter space explored in this paper. Finally, in App. C.5 we show how the sensitivity to such observables is expected to improve in the next decades.

The input parameters we use throughout the paper are reported in Tables 7 and 8.

### C.1    Bosonic (aka universal) operators

The SILH Lagrangian [24] contains the full set of operators that modify the physics at the Z-pole as well as the Higgs couplings irrespective of the flavor structure of the underlying model. They are therefore referred to as "universal" operators [74, 86]. In Tab. 4 we collect the subset of operators that is more relevant phenomenologically.

| $y_u$ | $y_c$ | $y_t$ | $y_d$ | $y_s$ | $y_b$ |
|---|---|---|---|---|---|
| $5.9 \times 10^{-6}$ | $3.0 \times 10^{-3}$ | $0.81$ | $13 \times 10^{-6}$ | $0.25 \times 10^{-3}$ | $13 \times 10^{-3}$ |

| $y_e$ | $y_\mu$ | $y_\tau$ | $g_s$ | $g$ | $g'$ |
|---|---|---|---|---|---|
| $2.9 \times 10^{-6}$ | $0.60 \times 10^{-3}$ | $10 \times 10^{-3}$ | $1.0$ | $0.63$ | $0.36$ |

Table 7: SM couplings defined in the $\overline{\text{MS}}$ scheme and renormalized at the scale $\mu = 3$ TeV. The RG evolution is performed through DSixTools [84]. The initial values of the parameters are taken at the $m_W$ scale from [85].

| $\lambda_C$ | $A$ | $\rho$ | $\eta$ |
|---|---|---|---|
| $0.23$ | $0.79$ | $0.14$ | $0.36$ |

Table 8: CKM inputs parameters in the Wolfenstein parametrization, taken from [65].

According to the rules reviewed in Sections 2.2, the Wilson coefficients of $\mathcal{O}_g$, $\mathcal{O}_\gamma$, $\mathcal{O}_H$, $\mathcal{O}_{W,B}$, $\mathcal{O}_{2W}$ read

$$\mathcal{C}_g = c_g \frac{g_s^2}{m_*^2}, \quad \mathcal{C}_\gamma = c_\gamma \frac{g'^2}{m_*^2}, \quad \mathcal{C}_H = c_H \frac{g_*^2}{m_*^2}, \quad \mathcal{C}_{W,B} = c_{W,B} \frac{1}{m_*^2}, \quad \mathcal{C}_{2W} = c_{2W} \frac{1}{g_*^2 m_*^2}, \quad \text{(C.1)}$$

where $c_{g,\gamma,H,W,B,2W}$ are expected to be order unity. Current 95% C.L. constraints on $\mathcal{C}_g$ and $\mathcal{C}_\gamma$, essentially driven by the Higgs couplings to gluons and photons, are extracted from [87] and read

$$m_* \gtrsim 17\sqrt{c_g} \text{ TeV}. \quad \text{(C.2)}$$

Generic SILH scenarios with $c_g \sim 1$ are therefore robustly outside the direct reach of LHC. In order to obtain more compelling new physics models for the TeV scale the coefficients $c_{g,\gamma}$ must be somewhat suppressed. This may be achieved assuming the Higgs is a pseudo-Nambu Goldstone boson. In that case $c_g, c_\gamma$ are expected to be proportional to the same small couplings that control the Higgs potential [88]. The natural expectation $c_g \sim y_t^2/16\pi^2$, $c_\gamma \sim 3y_t^2/16\pi^2$ is enough to relax Eq. (C.2) to

$$m_* \gtrsim 1 \div 2 \text{ TeV}. \quad \text{(C.3)}$$

The current bound on $\mathcal{C}_H$ is extracted from [87], those on $\widehat{S} = (\mathcal{C}_W + \mathcal{C}_B)m_W^2$ and $\mathcal{C}_{2W}$ respectively from [70] and [89].[35] Taking $c_{H,W,B,2W} = 1$ we obtain

$$\mathcal{C}_H \to m_* \gtrsim 0.85g_* \text{ TeV}, \quad \mathcal{C}_W + \mathcal{C}_B \to m_* \gtrsim 2.4 \text{ TeV}, \quad \mathcal{C}_{2W} \to m_* \gtrsim 4.6/g_* \text{ TeV}. \quad \text{(C.4)}$$

Note that following our conventions $c_{W,B} = 1$ the bound on the $\widehat{S}$ parameter is on the quantity $2/m_*^2$, consistently with [45].

In the main text we assume that a suppression of $c_{g,\gamma}$ is present and work under the hypothesis that (C.4) represent the most stringent constraints from the bosonic operators of Tab. 4.

---

[35]Notice that the measurement of [89] reports a negative $\mathcal{C}_{2W}$, consistent with $\mathcal{C}_{2W} = 0$ only at $2\sigma$ level. The powercounting in Eq. (C.1), however, is only consistent with a positive $\mathcal{C}_{2W}$ [90]. Waiting for additional experimental inputs, we neglect such issue and we consider half of the 95% experimental interval in [89].

## C.2   Corrections to $\widehat{T}$

As anticipated in App. A.1, the coefficient of the operator $\mathcal{O}_T$, or more concretely the so-called electroweak $\widehat{T}$ parameter [74], is associated with the violation of the custodial symmetry. The mixing with the elementary fermions in Eqs. (3.1, 5.1) can explicitly break custodial symmetry and consequently loops of the elementary fermions can give rise to corrections to $\widehat{T}$.

Referring to the embeddings for the composite fermions listed in Tab. 3 we can identify these contributions to $\Delta\widehat{T}$ by means of spurion analysis. Starting with left-handed fermions, if $\mathcal{O}_{q_{(u/d)}} \in (\mathbf{2},\mathbf{1})_{1/6}$ there are no corrections to $\widehat{T}$, whereas in all other cases we find

$$\Delta\widehat{T} \sim \frac{N_c \lambda_L^4 v^2}{16\pi^2 m_*^2} \, . \tag{C.5}$$

In models with right-handed universality the leading contribution comes from $t_L$ partner, in particular we have $\lambda_L = \lambda_{q_u} \sim y_t/\varepsilon_u$ (RU) or $\lambda_L = \lambda_{q_u} \sim y_t/\varepsilon_{u_3}$ (puRU, pRU). Imposing the experimental bound [53] we get

$$m_* \gtrsim \begin{cases} 0.50/\varepsilon_u^2 \text{ TeV} & \text{RU} \\ 0.50/\varepsilon_{u_3}^2 \text{ TeV} & \text{puRU, pRU} \end{cases} \tag{C.6}$$

In the most optimistic scenario for RU, $\varepsilon_u \sim 0.96/\sqrt{g_*}$ and the above bound is subdominant compared to those found in the text. In puRU and pRU the ideal setup has $\varepsilon_{u_3} \sim 1$ and the same conclusion can be drawn. In models with universality in the left-handed sector ($\lambda_L = \lambda_q \sim \varepsilon_q g_*$ or $\lambda_L = \lambda_q \sim \varepsilon_{q_3} g_*$) we get

$$m_* \gtrsim \begin{cases} 0.76\,(\varepsilon_q g_*)^2 \text{ TeV} & \text{LU} \\ 0.76\,(\varepsilon_{q_3} g_*)^2 \text{ TeV} & \text{pLU} \end{cases} \tag{C.7}$$

The best case scenarios are here obtained for $\varepsilon_q \lesssim \varepsilon_{q_3} \sim y_t/g_*$ and the resulting constraint is not competitive with those discussed in the text. Hence, the contributions to the $\widehat{T}$ parameter from the left-handed sector are under control in the most attractive scenarios discussed in the paper.

Contributions to $\widehat{T}$ from the right-handed sector are dominated by $t_R$. These are absent only if $\mathcal{O}_u \in (\mathbf{1},\mathbf{1})$. Otherwise

$$\mathcal{O}_u \in (\mathbf{1},\mathbf{3}): \quad \Delta\widehat{T} \sim \frac{N_c \lambda_u^2 g_*^2 v^2}{16\pi^2 m_*^2} \, , \qquad \mathcal{O}_u \in (\mathbf{1},\mathbf{2}): \quad \Delta\widehat{T} \sim \frac{N_c \lambda_u^4 v^2}{16\pi^2 m_*^2} \, . \tag{C.8}$$

and we get the following bounds:

$$\mathcal{O}_u \in (\mathbf{1},\mathbf{3}): \qquad m_* \gtrsim \begin{cases} 0.76\,\varepsilon_u g_*^2 \text{ TeV} & \text{RU} \\ 0.76\,\varepsilon_{u_3} g_*^2 \text{ TeV} & \text{puRU, pRU} \\ 0.61\,g_*/\varepsilon_q \text{ TeV} & \text{LU} \\ 0.61\,g_*/\varepsilon_{q_3} \text{ TeV} & \text{pLU} \end{cases} \tag{C.9}$$

and

$$\mathcal{O}_u \in (\mathbf{1},\mathbf{2}): \qquad m_* \gtrsim \begin{cases} 0.76\,\varepsilon_u^2 g_*^2 \text{ TeV} & \text{RU} \\ 0.76\,\varepsilon_{u_3}^2 g_*^2 \text{ TeV} & \text{puRU, pRU} \\ 0.49\,/\varepsilon_q^2 \text{ TeV} & \text{LU} \\ 0.49\,/\varepsilon_{q_3}^2 \text{ TeV} & \text{pLU} \end{cases} \tag{C.10}$$

In conclusion, assuming the optimal option of Eq. (A.2) for the composite partners representations (or even (A.3) for models with a single composite doublet), we are guaranteed that corrections to $\widehat{T}$ are sufficiently small in all the most compelling models. In the other cases $\widehat{T}$ can represent a competitive bound, as discussed in the main text.

## C.3   Flavor observables

In this section, we list the dominant flavor-dependent experimental bounds that we use throughout the paper. The associated *tree-level* constraints on our flavor models are then collected in various tables. For certain processes, most notably the neutron EDM and dipole corrections to $B \to X_s\gamma$, radiative contributions should of course be taken into account when the tree-level effects are absent. The corresponding coefficients *are not* reported in the tables, but instead discussed separately for each model in the main text. Additional constraints are discussed in detail in App. C.4 and analyzed in the main text. The most relevant leptonic observables are discussed in the main text.

All the bounds on models with universality in the right-handed quark sector and arising at tree-level are collected in Tab. 9 ($\Delta F = 0$), Tab. 11 ($\Delta F = 1$) and Tab. 13 ($\Delta F = 2$), all those for models with universality in the left-handed quark sector are shown in Tab. 10 ($\Delta F = 0$), Tab. 12 ($\Delta F = 1$) and Tab. 14 ($\Delta F = 2$). In the tables, for each operator we show the leading order expression of the Wilson coefficient (conveniently derived using Table 5 and Table 6) and the 95% C.L. constraint on $m_*$ in TeV. By $\times$ we indicate that the operator is not generated at leading order. By — we emphasize that the bound allows $m_* \sim 1$ TeV for any value of the $\varepsilon_\psi$ parameters identified in the bulk of the paper. The label "w/$P_{LR}$" in Tables 11 and 12 emphasizes that the constraint is derived assuming custodial protection, see App. A.1 and the main text for details. In parenthesis, we show contributions that are naively suppressed but may become large for small values of $\varepsilon_\psi$.

The observables from which we derived our constraints are listed in the following.

- **$\Delta\mathbf{F} = \mathbf{0}$** In this class we find the dijet angular distributions, the neutron EDMs, and flavor-diagonal corrections to the vector couplings.

  The bounds from dijet angular distributions are taken from the CMS [49] and ATLAS [50] results based on early run 2 analyses. In both measurements, the collaborations consider the effect of $(\bar{u}_R\gamma^\mu u_R + \bar{d}_R\gamma^\mu d_R)^2$. To extract the bounds for up-type and down-type quarks separately, we rescaled the bounds of [50] by a corresponding factor, which we extracted from the projections of [51]. The result is $\widetilde{\mathcal{C}}_{uu} \lesssim 0.017, 0.044\,\mathrm{TeV}^{-2}$, $\widetilde{\mathcal{C}}_{dd} \lesssim 0.08, 0.11\,\mathrm{TeV}^{-2}$ and $\widetilde{\mathcal{C}}_{qq} \lesssim 0.013, 0.037\,\mathrm{TeV}^{-2}$. We also use the projections of Ref. [51] for the run 3 of LHC, which read $\widetilde{\mathcal{C}}_{uu} \lesssim 0.0062, 0.015\,\mathrm{TeV}^{-2}$, $\widetilde{\mathcal{C}}_{dd} \lesssim 0.03, 0.038\,\mathrm{TeV}^{-2}$ and $\widetilde{\mathcal{C}}_{qq} \lesssim 0.005, 0.012\,\mathrm{TeV}^{-2}$.

  For what concerns the neutron EDM, we use (B.50) where the quark EDMs are RG evolved down to $\sim 1$ GeV, and impose $|d_n| \leq 1.8 \times 10^{-26} e$ cm [35].

  Anomalous $Z$-couplings to all quarks but the top are very well constrained by LEP measurements. We take the bounds from [53] and present only the dominant ones. The unquoted constraints are under control in the range of interest for the new physics parameters. The constraints on the anomalous couplings of $b_L$ can be avoided by assuming custodial protection, as reviewed in Appendix A.1. In particular, for models with (Partial-)Right universality, where we assume two doublets mixing with each SM

| Coefficient | Right Univ. | | Partial Up Right Univ. | | Partial Right Univ. | |
|---|---|---|---|---|---|---|
| $[\widetilde{\mathcal{C}}_{Hq}^{(1)} + \widetilde{\mathcal{C}}_{Hq}^{(3)}]^{33}$ | $\frac{y_t^2}{m_*^2\varepsilon_u^2}\left(\frac{y_b^2}{m_*^2\varepsilon_d^2}\right)$ | $\frac{2.2\div2.8}{\varepsilon_u}\left(\frac{0.05}{\varepsilon_d}\right)$ | $\frac{y_t^2}{m_*^2\varepsilon_{u_3}^2}\left(\frac{y_b^2}{m_*^2\varepsilon_d^2}\right)$ | $\frac{2.2\div2.8}{\varepsilon_{u_3}}\left(\frac{0.05}{\varepsilon_d}\right)$ | $\frac{y_t^2}{m_*^2\varepsilon_{u_3}^2}\left(\frac{y_b^2}{m_*^2\varepsilon_{d_3}^2}\right)$ | $\frac{2.2\div2.8}{\varepsilon_{u_3}}\left(\frac{0.05}{\varepsilon_{d_3}}\right)$ |
| $[\widetilde{\mathcal{C}}_{Hq}^{(3)}]^{33}$ | $\frac{y_t^2}{m_*^2\varepsilon_u^2}$ | $\frac{0.9}{\varepsilon_u}$ | $\frac{y_t^2}{m_*^2\varepsilon_{u_3}^2}$ | $\frac{0.9}{\varepsilon_{u_3}}$ | $\frac{y_t^2}{m_*^2\varepsilon_{u_3}^2}$ | $\frac{0.9}{\varepsilon_{u_3}}$ |
| $[\widetilde{\mathcal{C}}_{qD}^{(3)}]^{33}$ | $g\frac{y_t^2}{g_*^2 m_*^2\varepsilon_u^2}$ | $\frac{1.1\div1.3}{g_*\varepsilon_u}$ | $g\frac{y_t^2}{g_*^2 m_*^2\varepsilon_u^2}$ | $\frac{1.1\div1.3}{g_*\varepsilon_{u_3}}$ | $g\frac{y_t^2}{g_*^2 m_*^2\varepsilon_u^2}$ | $\frac{1.1\div1.3}{g_*\varepsilon_{u_3}}$ |
| $[\widetilde{\mathcal{C}}_{Hu}]^{11}$ | $\frac{\varepsilon_u^2 g_*^2}{m_*^2}$ | $1.7\div2.1 g_*\varepsilon_u$ | $\frac{\varepsilon_u^2 g_*^2}{m_*^2}$ | $1.7\div2.1 g_*\varepsilon_u$ | $\frac{\varepsilon_u^2 g_*^2}{m_*^2}$ | $1.7\div2.1 g_*\varepsilon_u$ |
| $[\widetilde{\mathcal{C}}_{Hu}]^{33}$ | $\frac{\varepsilon_u^2 g_*^2}{m_*^2}$ | $0.7 g_*\varepsilon_u$ | $\frac{\varepsilon_{u_3}^2 g_*^2}{m_*^2}$ | $0.7 g_*\varepsilon_{u_3}$ | $\frac{\varepsilon_{d_3}^2 g_*^2}{m_*^2}$ | $0.7 g_*\varepsilon_{u_3}$ |
| $[\widetilde{\mathcal{C}}_{Hd}]^{11,22}$ | $\frac{\varepsilon_d^2 g_*^2}{m_*^2}$ | $1.4\div1.6 g_*\varepsilon_d$ | $\frac{\varepsilon_d^2 g_*^2}{m_*^2}$ | $1.4\div1.6 g_*\varepsilon_d$ | $\frac{\varepsilon_d^2 g_*^2}{m_*^2}$ | $1.4\div1.6 g_*\varepsilon_d$ |
| $[\widetilde{\mathcal{C}}_{Hd}]^{33}$ | $\frac{\varepsilon_d^2 g_*^2}{m_*^2}$ | $1.0\div1.5 g_*\varepsilon_d$ | $\frac{\varepsilon_d^2 g_*^2}{m_*^2}$ | $1.0\div1.5 g_*\varepsilon_d$ | $\frac{\varepsilon_{d_3}^2 g_*^2}{m_*^2}$ | $1.0\div1.5 g_*\varepsilon_{d_3}$ |

Table 9: Main bounds on anomalous Z/W couplings in models with (Partial) RU. For each operator, we show the leading order expression of the Wilson coefficient and the 95% C.L. constraint on $m_*$ in TeV (see App. C.3 for more details). By $\times$ we indicate that the operator is not generated at leading order. By — we emphasize that the bound allows $m_* \sim 1$ TeV for any value of the $\varepsilon_\psi$ parameters identified in the bulk of the paper. The $\mathrm{P_{LR}}$ symmetry can be invoked to remove some of the bounds. See App. A.1 for more details.

| Coefficient | Left Univ. | | Partial Left Univ. | |
|---|---|---|---|---|
| $[\widetilde{\mathcal{C}}_{Hq}^{(1)} + \widetilde{\mathcal{C}}_{Hq}^{(3)}]^{33}$ | $\frac{\varepsilon_q^2 g_*^2}{m_*^2}$ | $2.8\div3.5\, g_*\varepsilon_q$ | $\frac{\varepsilon_{q_3}^2 g_*^2}{m_*^2}$ | $2.8\div3.5\, g_*\varepsilon_{q_3}$ |
| $[\widetilde{\mathcal{C}}_{Hq}^{(3)}]^{11}$ | $\frac{\varepsilon_q^2 g_*^2}{m_*^2}$ | $4.5\div5.0\, g_*\varepsilon_q$ | $\frac{\varepsilon_q^2 g_*^2}{m_*^2}$ | $4.5\div5.0\, g_*\varepsilon_q$ |
| $[\widetilde{\mathcal{C}}_{Hq}^{(3)}]^{33}$ | $\frac{g_*^2\varepsilon_q^2}{m_*^2}$ | $1.1\, g_*\varepsilon_q$ | $\frac{g_*^2\varepsilon_{q_3}^2}{m_*^2}$ | $1.1\, g_*\varepsilon_{q_3}$ |
| $[\widetilde{\mathcal{C}}_{qD}^{(3)}]^{33}$ | $\frac{g\varepsilon_q^2}{m_*^2}$ | $1.3\div1.7\,\varepsilon_q$ | $\frac{g\varepsilon_{q_3}^2}{m_*^2}$ | $1.3\div1.7\,\varepsilon_{q_3}$ |
| $[\widetilde{\mathcal{C}}_{Hu}]^{33}$ | $\frac{y_t^2}{\varepsilon_q^2 m_*^2}$ | $\frac{0.5}{\varepsilon_q}$ | $\frac{y_t^2}{\varepsilon_{q_3}^2 m_*^2}$ | $\frac{0.5}{\varepsilon_{q_3}}$ |
| $[\widetilde{\mathcal{C}}_{Hd}]^{33}$ | $\frac{y_b^2}{\varepsilon_q^2 m_*^2}$ | — | $\frac{y_b^2}{\varepsilon_{q_3}^2 m_*^2}$ | — |

Table 10: Same as Tab. 9 in models with (Partial) Left Universality.

| Coefficient | Right Univ. | | Partial Up Right Univ. | | Partial Right Univ. | |
|---|---|---|---|---|---|---|
| $C_{10}$ | $\frac{1}{2}\frac{4\sqrt{2}\pi^2}{G_F m_*^2}\frac{y_t^2}{e^2\varepsilon_u^2}$ | $\frac{6.5\div8.3}{\varepsilon_u}$ | $\frac{1}{2}\frac{4\sqrt{2}\pi^2}{G_F m_*^2}\frac{y_t^2}{e^2\varepsilon_{u_3}^2}$ | $\frac{6.5\div8.3}{\varepsilon_{u_3}}$ | $\frac{1}{2}\frac{4\sqrt{2}\pi^2}{G_F m_*^2}\frac{y_t^2}{e^2\varepsilon_{u_3}^2}$ | $\frac{6.5\div8.3}{\varepsilon_{u_3}}$ |
| $C_{10}'$ | $\times$ | | $\times$ | | $\frac{b'}{2}\frac{y_s/y_b}{V^{tb}_{\mathrm{CKM}}(V^{ts}_{\mathrm{CKM}})^*}\frac{4\sqrt{2}\pi^2 g_*^2}{G_F m_*^2 e^2}\varepsilon_{d_3}^2$ | $7.1 g_*\varepsilon_{d_3}$ |
| $C_9(\mathrm{w/P_{LR}})$ | $\frac{4\sqrt{2}\pi^2}{G_F m_*^2}\frac{y_t^2}{g_*^2\varepsilon_u^2}$ | $\frac{1.2\div3.5}{g_*\varepsilon_u}$ | $\frac{4\sqrt{2}\pi^2}{G_F m_*^2}\frac{y_t^2}{g_*^2\varepsilon_{u_3}^2}$ | $\frac{1.2\div3.5}{g_*\varepsilon_{u_3}}$ | $\frac{4\sqrt{2}\pi^2}{G_F m_*^2}\frac{y_t^2}{g_*^2\varepsilon_{u_3}^2}$ | $\frac{1.2\div3.5}{g_*\varepsilon_{u_3}}$ |
| $C_9'(\mathrm{w/P_{LR}})$ | $\times$ | | $\times$ | | $b'\frac{y_s/y_b}{V^{tb}_{\mathrm{CKM}}(V^{ts}_{\mathrm{CKM}})^*}\frac{4\sqrt{2}\pi^2}{G_F m_*^2}\varepsilon_{d_3}^2$ | $(1.0\div2.4)\varepsilon_{d_3}$ |
| $C_7$ | $\times$ | | $\times$ | | $\frac{m_s^2}{m_b^2}\frac{b'4\sqrt{2}\pi^2}{G_F V^{tb}_{\mathrm{CKM}}(V^{ts}_{\mathrm{CKM}})^* m_*^2}$ | $0.69\div1.0$ |
| $C_7'$ | $\times$ | | $\times$ | | $\frac{m_s}{m_b}\frac{b'4\sqrt{2}\pi^2}{G_F V^{tb}_{\mathrm{CKM}}(V^{ts}_{\mathrm{CKM}})^* m_*^2}$ | $4.5\div5.2$ |

Table 11: Main bounds from tree-level $b \to s$ transitions in models with (Partial) RU. For each operator we show the leading order expression of the Wilson coefficient, including its dependence on the parameters $a, b, a', b'$, and the 95% C.L. constraint on $m_*$ in TeV, assuming $a, b, a', b'$ are of order unity. By $\times$ we indicate that the operator is not generated at leading order. By — we emphasize that the bound allows $m_* \sim 1$ TeV for any value of the $\varepsilon_\psi$ parameters identified in the bulk of the paper. Imposing $\mathrm{P_{LR}}$ (see Appendix A.1) the constraint from $C_{10}, C_{10}'$ can be removed and $B_s \to \mu\mu$ is controlled by the coefficients $C_9, C_9'$ reported in the table and labelled "w/$\mathrm{P_{LR}}$".

| Coefficient | Left Univ. | Partial Left Univ. | |
|---|---|---|---|
| $C_{10}$ | $\times$ | $\frac{1}{2}\frac{4\sqrt{2}\pi^2}{G_F m_*^2}\frac{g_*^2\varepsilon_{q_3}^2}{e^2}$ | $8.0\div10\,g_*\varepsilon_{q_3}$ |
| $C_{10}'$ | $\times$ | $\frac{1}{2}\frac{\sqrt{2}}{G_F}\frac{4\pi^2}{e^2}\frac{b'}{V^{tb}_{\mathrm{CKM}}(V^{ts}_{\mathrm{CKM}})^*}\frac{y_s^2}{m_*^2\varepsilon_q^2}$ | $\frac{0.01}{\varepsilon_q}$ |
| $C_9(\mathrm{w/P_{LR}})$ | $\times$ | $\frac{4\sqrt{2}\pi^2}{G_F m_*^2}\varepsilon_{q_3}^2$ | $1.5\div4.3\,\varepsilon_{q_3}$ |
| $C_9'(\mathrm{w/P_{LR}})$ | $\times$ | $\frac{\sqrt{2}}{G_F}\frac{4\pi^2}{g_*^2}\frac{b'}{V^{tb}_{\mathrm{CKM}}(V^{ts}_{\mathrm{CKM}})^*}\frac{y_s^2}{m_*^2\varepsilon_q^2}$ | — |
| $C_7$ | $\times$ | $\frac{m_s}{m_b}\frac{4\sqrt{2}\pi^2 b'}{G_F V^{tb}_{\mathrm{CKM}}(V^{ts}_{\mathrm{CKM}})^* m_*^2}$ | $4.9\div7.5$ |
| $C_7'$ | $\times$ | $\frac{m_s^2}{m_b^2}\frac{4\sqrt{2}\pi^2 b'}{G_F V^{tb}_{\mathrm{CKM}}(V^{ts}_{\mathrm{CKM}})^* m_*^2}$ | $0.63\div0.72$ |

Table 12: Same as Tab. 11 for models with (Partial) Left Universality.

| Coefficient | Right Univ. | | Partial Up Right Univ. | | Partial Right Univ. | |
|---|---|---|---|---|---|---|
| $\operatorname{Im} C_1^{sd}$ | $2\frac{A^4 y_t^4 \eta \lambda_C^{10}}{m_*^2 g_*^2 \varepsilon_u^4}$ | $\frac{5.4}{g_* \varepsilon_u^2}$ | $2\frac{A^4 y_t^4 \eta \lambda_C^{10}}{m_*^2 g_*^2 \varepsilon_{u_3}^4}(2\frac{A^2 y_c^2 y_t^2 \eta \lambda_C^6}{m_*^2 g_*^2 \varepsilon_{u_3}^2 \varepsilon_u^2})$ | $\frac{5.4}{g_* \varepsilon_{u_3}^2}(\frac{0.5}{g_* \varepsilon_{u_3}\varepsilon_u})$ | $2\frac{A^4 y_t^4 \eta \lambda_C^{10}}{m_*^2 g_*^2 \varepsilon_{u_3}^4}(2\frac{A^2 y_c^2 y_t^2 \eta \lambda_C^6}{m_*^2 g_*^2 \varepsilon_{u_3}^2 \varepsilon_u^2})$ | $\frac{5.4}{g_* \varepsilon_{u_3}^2}(\frac{0.5}{g_* \varepsilon_{u_3}\varepsilon_u})$ |
| $\operatorname{Im} C_1'^{sd}$ | $\times$ | | $\times$ | | $(a'b'^*)\frac{2 y_s^4}{y_b^4}\frac{g_*^2\varepsilon_{d_3}^4}{m_*^2}$ | $10\, g_* \varepsilon_{d_3}^2$ |
| $\operatorname{Im} C_4^{sd}$ | $\times$ | | $\times$ | | $(a'b'^*)^2 \frac{y_s^5 y_d}{y_b^4 m_*^2}$ | — |
| $C_1^{bd}$ | $\frac{A^2 \lambda_C^6 y_t^4}{m_*^2 g_*^2 \varepsilon_u^4}$ | $\frac{6.6}{g_* \varepsilon_u^2}$ | $\frac{A^2 \lambda_C^6 y_t^4}{m_*^2 g_*^2 \varepsilon_{u_3}^4}$ | $\frac{6.6}{g_* \varepsilon_{u_3}^2}$ | $\frac{A^2 \lambda_C^6 y_t^4}{m_*^2 g_*^2 \varepsilon_{u_3}^4}$ | $\frac{6.6}{g_* \varepsilon_{u_3}^2}$ |
| $C_1'^{bd}$ | $\times$ | | $\times$ | | $a'^2 \frac{y_s^2}{y_b^2}\frac{g_*^2\varepsilon_{d_3}^4}{m_*^2}$ | $22\, g_* \varepsilon_{d_3}^2$ |
| $C_4^{bd}$ | $\times$ | | $\times$ | | $a'^2 \frac{y_s^2 y_d}{y_b m_*^2}$ | — |
| $C_1^{bs}$ | $\frac{A^2 \lambda_C^4 y_t^4}{g_*^2 \varepsilon_u^4 m_*^2}$ | $\frac{5.6}{g_* \varepsilon_u^2}$ | $\frac{A^2 \lambda_C^4 y_t^4}{g_*^2 \varepsilon_{u_3}^4 m_*^2}$ | $\frac{5.6}{g_* \varepsilon_{u_3}^2}$ | $\frac{A^2 \lambda_C^4 y_t^4}{g_*^2 \varepsilon_{u_3}^4 m_*^2}$ | $\frac{5.6}{g_* \varepsilon_{u_3}^2}$ |
| $C_1'^{bs}$ | $\times$ | | $\times$ | | $b'^2 \frac{y_s^2}{y_b^2}\frac{g_*^2\varepsilon_{d_3}^4}{m_*^2}$ | $4.1\, g_* \varepsilon_{d_3}^2$ |
| $C_4^{bs}$ | $\times$ | | $\times$ | | $b'^2 \frac{y_s^3}{y_b m_*^2}$ | — |
| $\operatorname{Im} C_1^{cu}$ | $2\frac{A^2 y_b^2 y_s^2 \eta \lambda_C^6}{m_*^2 g_*^2 \varepsilon_d^4}$ | — | $2\frac{A^2 y_b^2 y_s^2 \eta \lambda_C^6}{m_*^2 g_*^2 \varepsilon_d^4}$ | — | $2\frac{A^2 y_b^2 y_s^2 \eta \lambda_C^6}{m_*^2 g_*^2 \varepsilon_{d_3}^2 \varepsilon_d^2}$ | — |
| $\operatorname{Im} C_1'^{cu}$ | $\times$ | | $(ab)^2 \frac{y_c^4}{y_t^4}\frac{g_*^2\varepsilon_{u_3}^2}{m_*^2}$ | $0.1\, g_* \varepsilon_{u_3}^2$ | $(ab)^2 \frac{y_c^4}{y_t^4}\frac{g_*^2\varepsilon_{u_3}}{m_*^2}$ | $0.1\, g_* \varepsilon_{u_3}^2$ |
| $\operatorname{Im} C_4^{cu}$ | $\times$ | | $(ab)^2 \frac{y_c^5 y_u}{y_t^4 m_*^2}$ | — | $(ab)^2 \frac{y_c^5 y_u}{y_t^4 m_*^2}$ | — |

Table 13: Bounds from tree-level $\Delta F = 2$ transitions in models with (Partial) RU. For each operator, we show the leading order expression of the Wilson coefficient, including its dependence on the parameters $a, b, a', b'$, and the 95% C.L. constraint on $m_*$ in TeV, assuming $a, b, a', b'$ are of order unity. By $\times$ we indicate that the operator is not generated at leading order. By — we emphasize that the bound allows $m_* \sim 1$ TeV for any value of the $\varepsilon_\psi$ parameters identified in the bulk of the paper. The experimental bounds are taken from Ref. [52].

| Coefficient | Left Univ. | Partial Left Univ. | |
|---|---|---|---|
| $\operatorname{Im} C_1^{sd}$ | $\times$ | $2\frac{A^4 \eta \lambda_C^{10} \varepsilon_{q_3}^4 g_*^2}{m_*^2}$ | $8.1\, g_* \varepsilon_{q_3}^2$ |
| $\operatorname{Im} C_1'^{sd}$ | $\times$ | $(a'b'^*)^2 \frac{y_d^2 y_s^6}{y_b^4 g_*^2 \varepsilon_q^4 m_*^2}$ | — |
| $\operatorname{Im} C_4^{sd}$ | $\times$ | $(a'b'^*)^2 \frac{y_s^5 y_d}{y_b^4 m_*^2}$ | — |
| $C_1^{bd}$ | $\times$ | $A^2 \lambda_C^6 g_*^2 \varepsilon_{q_3}^4$ | $10\, g_* \varepsilon_{q_3}^2$ |
| $C_1'^{bd}$ | $\times$ | $a'^2 \frac{y_s^2 y_d^2}{g_*^2 m_*^2 \varepsilon_q^4}$ | — |
| $C_4^{bd}$ | $\times$ | $a'^2 \frac{y_s^2 y_d}{y_b m_*^2}$ | — |
| $C_1^{bs}$ | $\times$ | $A^2 \lambda_C^4 g_*^2 \varepsilon_{q_3}^4$ | $8.5\, g_* \varepsilon_{q_3}^2$ |
| $C_1'^{bs}$ | $\times$ | $b'^2 \frac{y_s^4}{g_*^2 \varepsilon_q^4 m_*^2}$ | — |
| $C_4^{bs}$ | $\times$ | $b'^2 \frac{y_s^3}{y_b m_*^2}$ | — |
| $\operatorname{Im} C_1^{cu}$ | $\times$ | $(ab)^2 \frac{y_c^4 \varepsilon_{q_3}^4 g_*^2}{y_t^4}$ | $0.1\, g_* \varepsilon_{q_3}^2$ |
| $\operatorname{Im} C_1'^{cu}$ | $\times$ | $(ab)^2 \frac{y_c^6 y_u^2}{y_t^4 g_*^2 m_*^2 \varepsilon_q^4}$ | — |
| $\operatorname{Im} C_4^{cu}$ | $\times$ | $(ab)^2 \frac{y_c^5 y_u}{y_t^4 m_*^2}$ | — |

Table 14: Same as Tab. 13 for models with (Partial) Left Universality.

one, it is possible to suppress at the same time all the bounds on $[\widetilde{\mathcal{C}}_{Hq}^{(1)} + \widetilde{\mathcal{C}}_{Hq}^{(3)}]^{33}$, $\widetilde{\mathcal{C}}_{Hu}$, $\widetilde{\mathcal{C}}_{Hd}$, except for the bounds in the parenthesis of the first row. The latter is however quite mild. For models with (Partial-)Left universality and only one partner for each SM doublet, we can choose to protect at the same time only $\widetilde{\mathcal{C}}_{Hq}^{(1)} + \widetilde{\mathcal{C}}_{Hq}^{(3)}$ and $\widetilde{\mathcal{C}}_{Hu}$ but not $\widetilde{\mathcal{C}}_{Hd}$. Still, the bound from the latter is not relevant in the interesting range of parameters. Even in the presence of custodial protection, there are residual anomalous $b_L$-couplings from the operators $\mathcal{O}_{qD}^{(1)}$ and $\mathcal{O}_{qD}^{(3)}$ (see the main text for details).

Importantly, current data favor a negative anomalous $Zb_Rb_R$ coupling. The constraint on $\widetilde{\mathcal{C}}_{Hd}^{33}$ shown in [53] is therefore non-symmetric and not centered around zero. As already explained in Sec. 3.2 we give a measure of the current uncertainty on this quantity presenting two bounds. One is obtained imposing the coefficient does not exceed the weaker bound $|\widetilde{\mathcal{C}}_{Hd}^{33}|v^2 < 0.06$; the other by requiring it does not exceed half of the full interval: $|\widetilde{\mathcal{C}}_{Hd}^{33}|v^2 < (0.06 - 0.009)/2$.

Anomalous couplings of the top quark are instead tested by LHC measurements. The bounds are extracted from Fig. 1 of [59], where individual constraints on the third components of $\widetilde{\mathcal{C}}_{Hq}^{(1,3)} = (y_t^2/2)\widetilde{\mathcal{C}}_{\varphi Q}^{1,3}$, and $\widetilde{\mathcal{C}}_{Hu} = (y_t^2/2)\widetilde{\mathcal{C}}_{\varphi t}$ are shown. To estimate the bound on $[\widetilde{\mathcal{C}}_{Hu}]^{33}$ we take the single operator reach shown by the bright green column on $\widetilde{\mathcal{C}}_{\varphi t}$ in that figure; the one on $[\widetilde{\mathcal{C}}_{Hq}^{(3)} - \widetilde{\mathcal{C}}_{Hq}^{(1)}]^{33}$ is taken from the dark green lines associated to $\widetilde{\mathcal{C}}_{\varphi Q}^{1,3}$, where only the LHC constraint is taken into account.

Models with Left Universality modify significantly the couplings to the $W$, see Section 5.1.2. The best bound on a universal correction $\delta g_W$ comes from tests of unitarity of the CKM and reads $(1 + \delta g_W)^2 \sum_i \left|V_{\text{CKM}}^{ui}\right|^2 - 1 = (1.5 \pm 0.7) \times 10^{-3}$ [60] (see Section 5.1.2 for details).

- **$\Delta \mathbf{F} = \mathbf{1}$** In this class the most relevant bounds arise from $b \to s$ transitions.

  We extract the bound on $C_{10}$ from Fig. 3 of [54] (the data-driven contour at zero $C_{9,\mu}$), rescaled by a factor $\sqrt{3.84/6}$ in order to take into account the fewer degrees of freedom of our model. Following our prescription we take the larger boundary of the interval and half of the interval: $|C_{10}| \leq 0.38, 0.23$.

  The bound on $|C_{10}'|$ is taken from half of the interval shown in Fig. 4 of [54]: $|C_{10}'| < 0.24$.

  For $C_9$ we provide a more and a less conservative choice. The weaker bound is obtained from the extreme boundary of Fig. 7 in [54], where in a more "model independent" fashion the unknown hadronic contributions are directly fitted from the data. For the strongest bound we take half of the interval of the "model dependent" bound in Tab. 1 of [54]. Concretely we impose $|C_9| < 2.2, 0.4$.

  The bound on $|C_9'|$ is taken from Tab. 1 of [54], considering the largest boundary and half of the interval: $|C_9'| \leq 2.2, 0.4$. Notice that the stronger value is compatible with Tab. 6 of [33].

  For $C_7, C_7'$ we include the effect of the RG evolution from 3 TeV down to $m_b$, which results in weakening the bound by a factor $\sim 2$ (see for instance [39] and references therein). We include this effect and we take the bounds for $C_7$ and $C_7'$ from [55] and [58], respectively. For $C_7$ we consider the two boundaries of the data-driven interval

in [55] to account for the uncertainties in the bound. We get $\mathrm{Re}[C_7] \leq 0.099, 0.042$ and $\mathrm{Re}[C_7'] \leq 0.12, 0.089$.

- **$\Delta F = 2$** The experimental bounds on these observables are extracted from [52], we report here only the most relevant ones for our analysis

$$
\begin{aligned}
&|C_1^{bs}| \leq (210\,\mathrm{TeV})^{-2}\,, && |C_1^{bd}| \leq (1.1 \times 10^3\,\mathrm{TeV})^{-2}\,, \\
&\mathrm{Im}[C_1^{sd}] \leq (2.6 \times 10^4\,\mathrm{TeV})^{-2}\,, && \mathrm{Im}[C_1^{cu}] \leq (10^4\,\mathrm{TeV})^{-2}\,, \\
&|C_4^{bs}| \leq (840\,\mathrm{TeV})^{-2}\,, && |C_4^{bd}| \leq (2.4 \times 10^3\,\mathrm{TeV})^{-2}\,, \\
&\mathrm{Im}[C_4^{sd}] \leq (3.6 \times 10^5\,\mathrm{TeV})^{-2}\,, && \mathrm{Im}[C_4^{cu}] \leq (3.7 \times 10^4\,\mathrm{TeV})^{-2}\,. \\
&|C_2^{bs}| \leq (600\,\mathrm{TeV})^{-2}\,, && |C_2^{bd}| \leq (2 \times 10^3\,\mathrm{TeV})^{-2}\,, \\
&\mathrm{Im}[C_2^{sd}] \leq (2.5 \times 10^5\,\mathrm{TeV})^{-2}\,, && \mathrm{Im}[C_2^{cu}] \leq (2.5 \times 10^4\,\mathrm{TeV})^{-2}\,.
\end{aligned}
\tag{C.11}
$$

Completely analogous constraints apply to the right-handed counterparts $(C_1')$. The previous bounds are renormalized at $\mu = 3\,\mathrm{TeV}$. The renormalization has only a marginal impact of the order of $few$-percent on most coefficients. Larger running effects impact $C_4$ (see for instance [84]), but that is less relevant in our analysis.

- **Lepton observables**

  Collider bounds on the coefficients of the operators $\mathcal{O}_{ee,\ell\ell}$ are taken from [64] and read $\widetilde{\mathcal{C}}_{ee}^{1111}, \widetilde{\mathcal{C}}_{\ell\ell}^{1111} < 0.066\,\mathrm{TeV}^{-2}$.

  Modified $Z$ couplings to leptons are constrained by LEP/SLC. We take the bounds from [53].

  The constraint on the rare decay $\mu \to e\gamma$ currently reads $\mathrm{BR}(\mu \to e\gamma) < 3.1 \times 10^{-13}$ [15] and roughly translates into $|[\widetilde{\mathcal{C}}_{e\gamma}]^{21,12}| \lesssim 1.8 \times 10^{-10}/\,\mathrm{TeV}^2$. For $\tau \to \mu\gamma$ we impose $|[\widetilde{\mathcal{C}}_{e\gamma}]^{32,23}| \lesssim 2.7 \times 10^{-6}/\,\mathrm{TeV}^2$ [25].

  Among the constraints from $\mu \to e$ conversions in nuclei, currently transitions in gold set the strongest bound, i.e. $\mathrm{BR}(\mu\,\mathrm{Au} \to e\,\mathrm{Au}) < 7 \times 10^{-13}$ [65]. This roughly becomes $|\widetilde{\mathcal{C}}_{He}^{21,12}| \lesssim 4.9 \times 10^{-6}/\mathrm{TeV}^2$ and $|\widetilde{\mathcal{C}}_{H\ell}^{21,21}| \lesssim 4.9 \times 10^{-6}/\mathrm{TeV}^2$ [25].

  Finally, the bound $|d_e| \leq 4.1 \times 10^{-30} e$ cm on the electron EDM [38] reads $|\widetilde{\mathcal{C}}_{e\gamma}^{11}| \lesssim 1.8 \times 10^{-13}/\mathrm{TeV}^2$.

## C.4 Additional flavor-violating processes

In the following, we discuss a few constraints that in generic models for flavor, e.g. anarchic scenarios of partial compositeness, can be important, whereas in models with partial universality turn out to be subdominant compared to those discussed in the main text.

### C.4.1 Modified Higgs couplings to fermions

The definition of Yukawa couplings $Y_u, Y_d$ and quark masses adopted throughout the paper did not take into account the impact of higher-dimensional operators. In this appendix, we show why such an approximation is well justified. We limit ourselves to dimension-6 operators, but we do not expect the consideration of even higher dimensions to affect that conclusion.

At dimension-6, two operators affect the quark masses and the Higgs couplings to quarks: $\mathcal{O}_{uH,dH}$ (see Table 4). Including them, the up-quark and down-quark mass matrices in the gauge basis read

$$M_u = \frac{v}{\sqrt{2}}\left(Y_u + \frac{1}{2}\mathcal{C}_{uH}v^2\right), \qquad M_d = \frac{v}{\sqrt{2}}\left(Y_d + \frac{1}{2}\mathcal{C}_{dH}v^2\right), \tag{C.12}$$

where $Y_{u,d}$ are the Yukawa couplings in the renormalizable part of the theory and $\mathcal{C}_{uH,dH}$ are the Wilson coefficients of $\mathcal{O}_{uH,dH}$. The associated linear Higgs coupling to the quark bilinear $\overline{u_L}u_R$ is $\frac{1}{\sqrt{2}}Y_u^{\rm eff} = \frac{1}{\sqrt{2}}Y_u + \frac{3}{2}\mathcal{C}_{uH}v^2$, and similarly for the down sector. Combining these relations we obtain

$$Y_u^{\rm eff} = \sqrt{2}\frac{M_u}{v} + \mathcal{C}_{uH}v^2, \qquad Y_d^{\rm eff} = \sqrt{2}\frac{M_d}{v} + \mathcal{C}_{dH}v^2. \tag{C.13}$$

The matrices that diagonalize $M_{u,d}$ have exactly the same form as the $U_u, V_u, U_d, V_d$ shown in App. B.1 up to a correction of order $g_*^2 v^2/m_*^2$ in the parameters $y_{u,c,t}, y_{d,s,b}, a, b, a', b'$ that enter the definition of $Y_{u,d}$ in our models. The error committed in replacing those matrices with $U_u, V_u, U_d, V_d$ effectively corresponds to neglecting the effect of dimension-8 operators, and is thus justified at the level of accuracy we are interested in.

In scenarios with MFV, at leading order in the spurions $\lambda_\psi$, $\mathcal{C}_{uH}$ and $\mathcal{C}_{dH}$ are proportional to the corresponding Yukawa matrices. That results in universal and flavor invariant corrections to the $h\bar{q}q$ couplings, which in the up and down sector take the form (in an obvious notation and using $v/f \equiv g_* v/m_*$)

$$Y_u^{\rm eff} = \sqrt{2}\frac{M_u}{v}\left(1 + c_u\frac{v^2}{f^2}\right), \qquad Y_d^{\rm eff} = \sqrt{2}\frac{M_d}{v}\left(1 + c_d\frac{v^2}{f^2}\right) \qquad ({\rm RU,\ LU}), \tag{C.14}$$

with $c_u \neq c_d$ in general. Non-universal and flavor-violating corrections only arise from loops and involve higher powers of the Yukawas and the same GIM suppression as in the SM. These effects are therefore experimentally irrelevant.

In models with partial universality non-universal and flavor breaking corrections already arise at leading order in the $\lambda_\psi$. In puRU that occurs only in the up sector, where two independent structures contribute to $\mathcal{C}_{uH}$, while the couplings to $d$-quarks maintain the same universal MFV structure of Eq. (C.14). Focusing then on the up sector of puRU, and working at leading non-trivial order in $\lambda_\psi$, the coupling matrix of Eq. (C.13) in the mass basis reads

$$[\widetilde{\mathcal{C}}_{uH}v^2]^{ij} = \frac{g_* v^2}{m_*^2}\left\{U_u^\dagger\left[c_u\lambda_q^{(2)}[\lambda_u^{(2)}]^\dagger + (c_u + c_t)\lambda_q^{(1)}[\lambda_u^{(1)}]^\dagger\right]V_u\right\}^{ij}, \tag{C.15}$$

with $c_{u,t}$ numbers of order unity. Recalling the definition of $Y_u$ given in (4.31), we can add and subtract a term of the form $c_u\lambda_q^{(1)}[\lambda_u^{(1)}]^\dagger$ inside the square parenthesis, and re-write (C.15) as

$$[\widetilde{\mathcal{C}}_{uH}v^2]^{ij} = \frac{g_*^2 v^2}{m_*^2}\left\{c_u Y_u + c_t U_u^\dagger\frac{\lambda_q^{(1)}[\lambda_u^{(1)}]^\dagger}{g_*}V_u\right\}^{ij} \tag{C.16}$$

$$= \frac{g_*^2 v^2}{m_*^2}\left\{c_u\begin{pmatrix} y_u & 0 & 0 \\ 0 & y_c & 0 \\ 0 & 0 & y_t \end{pmatrix} + c_t y_t\begin{pmatrix} |a|^2\frac{y_u y_c^2}{y_t^3} & a^*b\frac{y_u y_c^2}{y_t^3} & -a^*\frac{y_u y_c}{y_t^2} \\ ab^*\frac{y_c^3}{y_t^3} & |b|^2\frac{y_c^3}{y_t^3} & -b^*\frac{y_c^2}{y_t^2} \\ -a\frac{y_c}{y_t} & -b\frac{y_c}{y_t} & 1 \end{pmatrix}\right\}^{ij}.$$

In pRU, we have an expression analogous to (C.16) for the up sector, and a similar one for the down sector

$$[\widetilde{\mathcal{C}}_{dH}v^2]^{ij} = \frac{g_*^2 v^2}{m_*^2}\left\{ c_d \begin{pmatrix} y_d & 0 & 0 \\ 0 & y_s & 0 \\ 0 & 0 & y_b \end{pmatrix} + c_b y_b \begin{pmatrix} |a'|^2\frac{y_d y_s^2}{y_b^3} & a'^* b'\frac{y_d y_s^2}{y_b^3} & -a'^*\frac{y_d y_s}{y_b^2} \\ a'b'^*\frac{y_s^3}{y_b^3} & |b'|^2\frac{y_s^3}{y_b^3} & -b'^*\frac{y_s^2}{y_b^2} \\ -a'\frac{y_s}{y_b} & -b'\frac{y_s}{y_b} & 1 \end{pmatrix}^{ij} \right\}. \tag{C.17}$$

Finally, in pLU we have

$$[\widetilde{\mathcal{C}}_{uH}v^2]^{ij} = \frac{g_*^2 v^2}{m_*^2}\left\{ c_u \begin{pmatrix} y_u & 0 & 0 \\ 0 & y_c & 0 \\ 0 & 0 & y_t \end{pmatrix} + c_t y_t \begin{pmatrix} |a|^2\frac{y_u y_c^2}{y_t^3} & ab^*\frac{y_c^3}{y_t^3} & -a\frac{y_c}{y_t} \\ a^*b\frac{y_u y_c^2}{y_t^3} & |b|^2\frac{y_c^3}{y_t^3} & -b\frac{y_c}{y_t} \\ -a^*\frac{y_u y_c}{y_t^2} & -b^*\frac{y_c^2}{y_t^2} & 1 \end{pmatrix}^{ij} \right\} \tag{C.18}$$

$$[\widetilde{\mathcal{C}}_{dH}v^2]^{ij} = \frac{g_*^2 v^2}{m_*^2}\left\{ c_d \begin{pmatrix} y_d & 0 & 0 \\ 0 & y_s & 0 \\ 0 & 0 & y_b \end{pmatrix} + c_b y_b \begin{pmatrix} |a'|^2\frac{y_d y_s^2}{y_b^3} & a'b'^*\frac{y_s^3}{y_b^3} & -a'\frac{y_s}{y_b} \\ a'^*b'\frac{y_d y_s^2}{y_b^3} & |b'|^2\frac{y_s^3}{y_b^3} & -b'\frac{y_s}{y_b} \\ -a'^*\frac{y_d y_s}{y_b^2} & -b'^*\frac{y_s^2}{y_b^2} & 1 \end{pmatrix}^{ij} \right\}, \tag{C.19}$$

where of course $c_{u,t,d,b}$ in general differ from those of puRU and pRU.

All of the corrections are proportional to $(g_* v/m_*)^2$ and are constituted of two independent terms. The first, proportional to $c_{u,d}$, is exactly aligned with the SM mass matrix and the second, proportional to $c_{t,b}$, is flavor non-universal. In puRU and pRU the non-universal correction controlled by $c_t$ mainly involves the couplings $\bar{t}_L u_R^i$ whereas the one controlled by $c_b$ is dominantly on $\bar{b}_L d_R^i$. Conversely, in pLU the larger non-universal corrections are on $\bar{u}_L^i t_R$ and $\bar{d}_L^i b_R$. Those non-universal effects may be sizable compared to the universal ones even in the regime considered throughout the paper in which $|a|, |b|, |a'|, |b'|$ are not much larger than unity. Yet, as we will see now, the phenomenological consequences are usually very modest if not negligible.

The modified Higgs couplings mediate anomalous Higgs decays, top decays as well as novel $\Delta F \neq 0$ transitions, in particular the tree-level Higgs exchange affects $\Delta F = 2$ observables [91]. We begin with the latter. After integrating out the Higgs, the coefficients $\mathcal{C}_{fH}$ generate a contribution to the coefficients of the operators in Eqs. (B.44), (B.45). Explicitly, the relation is given by

$$C_2^{f_i f_j} = \frac{v^4}{2m_h^2}([\widetilde{\mathcal{C}}_{fH}]^{ij*})^2 \tag{C.20}$$

$$C_2'^{f_i f_j} = \frac{v^4}{2m_h^2}([\widetilde{\mathcal{C}}_{fH}]^{ji})^2 \tag{C.21}$$

$$C_4^{f_i f_j} = \frac{v^4}{m_h^2}[\widetilde{\mathcal{C}}_{fH}]^{ij*}[\widetilde{\mathcal{C}}_{fH}]^{ji}. \tag{C.22}$$

We report in Tab. 15 a parametric estimate of those Wilson coefficients and the corresponding experimental bounds. As already emphasized, all those bounds are on the quantity $m_*/g_*$ that is also constrained by $\mathcal{C}_H$, see Eq. (C.4). However, those found in Tab. 15 are numerically negligible in the fiducial range of parameter space in which $a, b, a', b'$ are at most of order unity, due to the strong suppression of flavor-violating transitions characterizing our models, see for

example Eq. (C.16). The strongest is obtained in the case of Partial Right Universality and reads

$$m_*/g_* \gtrsim 0.4 \, \mathrm{TeV} \,, \qquad (C.23)$$

and is weaker than what found in Eq. (C.4), as anticipated. For comparison, note that in flavor anarchic scenarios of partial compositeness, additional assumptions are needed in order for Higgs-mediated flavor-violation to be compatible with data. A natural suppression occurs in models where the Higgs is a pseudo-Nambu-Goldstone boson, see [28]. No such assumption is necessary here.

| Coefficient | Partial Up Right Univ. | | Partial Right Univ. | | Partial Left Univ. | |
|---|---|---|---|---|---|---|
| $\mathrm{Im}\, C_2^{sd}$ | $\times$ | | $(a'b'^*)^2 \frac{y_s^4 y_d^2}{y_b^4} \frac{g_*^4 v^4}{2m_*^4 m_h^2}$ | — | $(a'b'^*)^2 \frac{y_s^6}{y_b^4} \frac{g_*^4 v^4}{2m_*^4 m_h^2}$ | — |
| $\mathrm{Im}\, C_2'^{sd}$ | $\times$ | | $(a'b'^*)^2 \frac{y_s^6}{y_b^4} \frac{g_*^4 v^4}{2m_*^4 m_h^2}$ | — | $(a'b'^*)^2 \frac{y_s^4 y_d^2}{y_b^4} \frac{g_*^4 v^4}{2m_*^4 m_h^2}$ | — |
| $\mathrm{Im}\, C_4^{sd}$ | $\times$ | | $(a'b'^*)^2 \frac{y_s^5 y_d}{y_b^4} \frac{g_*^4 v^4}{m_*^4 m_h^2}$ | — | $(a'b'^*)^2 \frac{y_s^5 y_d}{y_b^4} \frac{g_*^4 v^4}{m_*^4 m_h^2}$ | — |
| $C_2^{bd}$ | $\times$ | | $a'^2 \frac{y_s^2 y_d^2}{y_b^2} \frac{g_*^4 v^4}{2m_*^4 m_h^2}$ | — | $a'^2 y_s^2 \frac{g_*^4 v^4}{2m_*^4 m_h^2}$ | $0.2\, g_*$ |
| $C_2'^{bd}$ | $\times$ | | $a'^2 y_s^2 \frac{g_*^4 v^4}{2m_*^4 m_h^2}$ | $0.4\, g_*$ | $a'^2 \frac{y_s^2 y_d^2}{y_b^2} \frac{g_*^4 v^4}{2m_*^4 m_h^2}$ | — |
| $C_4^{bd}$ | $\times$ | | $a'^2 \frac{y_s^2 y_d}{y_b} \frac{g_*^4 v^4}{m_*^4 m_h^2}$ | $0.1\, g_*$ | $a'^2 \frac{y_s^2 y_d}{y_b} \frac{g_*^4 v^4}{m_*^4 m_h^2}$ | — |
| $C_2^{bs}$ | $\times$ | | $b'^2 \frac{y_s^4}{y_b^2} \frac{g_*^4 v^4}{2m_*^4 m_h^2}$ | — | $b'^2 y_s^2 \frac{g_*^4 v^4}{2m_*^4 m_h^2}$ | $0.2\, g_*$ |
| $C_2'^{bs}$ | $\times$ | | $b'^2 y_s^2 \frac{g_*^4 v^4}{2m_*^4 m_h^2}$ | $0.2\, g_*$ | $b'^2 \frac{y_s^4}{y_b^2} \frac{g_*^4 v^4}{2m_*^4 m_h^2}$ | — |
| $C_4^{bs}$ | $\times$ | | $b'^2 \frac{y_s^3}{y_b} \frac{g_*^4 v^4}{m_*^4 m_h^2}$ | $0.1\, g_*$ | $b'^2 \frac{y_s^3}{y_b} \frac{g_*^4 v^4}{m_*^4 m_h^2}$ | $0.1\, g_*$ |
| $\mathrm{Im}\, C_2^{cu}$ | $(ab)^2 \frac{y_c^4 y_u^2}{y_t^4} \frac{g_*^4 v^4}{2m_*^4 m_h^2}$ | — | $(ab)^2 \frac{y_c^4 y_u^2}{y_t^4} \frac{g_*^4 v^4}{2m_*^4 m_h^2}$ | — | $(ab)^2 \frac{y_c^6}{y_t^4} \frac{g_*^4 v^4}{2m_*^4 m_h^2}$ | — |
| $\mathrm{Im}\, C_2'^{cu}$ | $(ab)^2 \frac{y_c^6}{y_t^4} \frac{g_*^4 v^4}{2m_*^4 m_h^2}$ | — | $(ab)^2 \frac{y_c^6}{y_t^4} \frac{g_*^4 v^4}{2m_*^4 m_h^2}$ | — | $(ab)^2 \frac{y_c^4 y_u^2}{y_t^4} \frac{g_*^4 v^4}{2m_*^4 m_h^2}$ | — |
| $\mathrm{Im}\, C_4^{cu}$ | $(ab)^2 \frac{y_c^5 y_u}{y_t^4} \frac{g_*^4 v^4}{m_*^4 m_h^2}$ | — | $(ab)^2 \frac{y_c^5 y_u}{y_t^4} \frac{g_*^4 v^4}{m_*^4 m_h^2}$ | — | $(ab)^2 \frac{y_c^5 y_u}{y_t^4} \frac{g_*^4 v^4}{m_*^4 m_h^2}$ | — |

Table 15: Constraints from $\Delta F = 2$ transitions mediated by tree-level Higgs exchange for models with Partial Right and Left Universality. For each operator, we report the leading order expression of the Wilson coefficient, including its dependence on the parameters $a, b, a', b'$, and the 95% constraint on $m_*$ in TeV, assuming $a, b, a', b'$ are of order unity, except for pLU where $a' \sim \lambda_C$ (see Eq. (5.27)). The bounds for pRU and pLU are parametrically the same, but those on pLU are numerically slightly weaker due to the fact that we took $a' \sim \lambda_C$. With — and $\times$ we indicate respectively that the bound is negligible and that the operator is not generated at leading order.

Anomalous Higgs decays into quarks are also induced by $\mathcal{C}_{uH,dH}$ in general. In pRU and pLU the decay $h \to b\bar{s}$ has a branching ratio of order

$$\mathrm{BR}(h \to b\bar{s}) \sim \left(\frac{g_*^2 v^2}{m_*^2}\right)^2 b'^2 \frac{y_s^2}{y_b^2} \mathrm{BR}(h \to b\bar{b}) = |b'|^2 \, 8 \times 10^{-7} \left(\frac{\mathrm{TeV}}{m_*/g_*}\right)^4 , \qquad (C.24)$$

where $\mathrm{BR}(h \to b\bar{b}) \simeq 0.6$. As far as we can tell, this result is well beyond the sensitivity of any foreseen experiment in our benchmark scenarios with $|b'| \sim 1$. In the extreme regime $|b'| \sim y_b/y_s$[36] the branching ratio grows up to $\mathrm{BR}(h \to b\bar{s}) \sim 4 \times 10^{-3} (g_* \mathrm{TeV}/m_*)^4$, but

---

[36]This requires some fine tuning in pLU, as noticed below (B.11).

other flavor constraints also become increasingly relevant. More precisely, the very same flavor-violating Yukawa coupling inducing $h \to b\bar{s}$ also controls a contribution

$$\Delta C_2^{(\prime)bs} = c_b^2 \frac{y_s^2 b'^2}{2m_h^2} \left( \frac{g_*^2 v^2}{m_*^2} \right)^2 \tag{C.25}$$

to the coefficient $C_2^{(\prime)bs}$ of the operators $\mathcal{O}_2^{(\prime)bs}$ affecting $B_s - \overline{B}_s$ mixing. Imposing the bound from the $\Delta F = 2$ observable, Eq. (C.24) is found to satisfy an absolute upper bound

$$\mathrm{BR}(h \to b\bar{s}) = \frac{2|\Delta C_2^{(\prime)bs}|m_h^2}{y_b^2} \mathrm{BR}(h \to b\bar{b}) \lesssim 3 \times 10^{-4} \tag{C.26}$$

which applies not only to pRU and pLU, but also to any model with modified Higgs couplings [92]. This branching ratio seems below any foreseeable sensitivity (see for instance [93]).

Other anomalous Higgs decays into quarks are similarly suppressed. Yet, models with partial universality in the leptonic sector (see Section 6) would allow $h \to \tau\bar{\mu}$. The resulting branching ratio has the same structure as (C.24):

$$\mathrm{BR}(h \to \tau\bar{\mu}) \sim \left( \frac{g_*^2 v^2}{m_*^2} \right)^2 b_\ell^2 \frac{y_\mu^2}{y_b^2} \mathrm{BR}(h \to b\bar{b}) = |b_\ell|^2 \, 5 \times 10^{-6} \left( \frac{\mathrm{TeV}}{m_*/g_*} \right)^4 . \tag{C.27}$$

In our benchmark scenarios with $|b_\ell| \sim 1$ this is far too small to be detected [70]. In the large $|b_\ell|$ limit, i.e. $|b_\ell| \sim y_\tau/y_\mu$, the otherwise subdominant constraint from $\tau \to \mu\gamma$ would set a strong limit on $m_*$, see (6.10). Taking that into account we find

$$\mathrm{BR}(h \to \tau\bar{\mu}) \lesssim 3 \times 10^{-5} \left( \frac{g_*^2}{16\pi^2 c} \right)^2 , \tag{C.28}$$

where $c$ is the coefficient of the dipole in (6.10). The result is thus below the sensitivity of HL-LHC [70] unless the coefficient of the dipole operator is accidentally suppressed, i.e. unless $|c| < g_*^2/16\pi^2$.

Let us next consider anomalous top decays. Those involving the Higgs boson, i.e. $t \to qh$ with $q = u, c$, are dominantly mediated by $\mathcal{O}_{uH,dH}$. The operators in the classes "dipoles" and "EW vertices" of Table 4 contribute respectively to $t \to qg, q\gamma$ and to $t \to qZ$. Explicit expressions for the decay rates are collected in [94]. For all our models with partial universality we find that such rates scale as

$$\Gamma(t \to qh) = \frac{y_t^2}{32\pi} m_t \left( 1 - \frac{m_h^2}{m_t^2} \right)^2 \left( \frac{g_*^2 v^2}{m_*^2} \theta_q \right)^2 \tag{C.29}$$

$$\Gamma_{\mathrm{EW}}(t \to qZ) = \frac{g^2}{32\pi \cos^2\theta_W} \frac{m_t^3}{m_Z^2} \left( 1 - \frac{m_Z^2}{m_t^2} \right)^2 \left( 1 + 2\frac{m_Z^2}{m_t^2} \right) \left( \frac{g_*^2 v^2 \varepsilon_3^2}{m_*^2} \theta_q \right)^2 \tag{C.30}$$

$$\Gamma_{\mathrm{dip.}}(t \to qg) = \frac{g_s^2}{4\pi} m_t \left( c_{\mathrm{dip.}} \frac{y_t^2 v^2}{m_*^2} \theta_q \right)^2 , \tag{C.31}$$

with $\theta_u \sim ay_c/y_t$ for $q = u$ and $\theta_c \sim by_c/y_t$ for $q = c$. The parameter $\varepsilon_3$ is more model-dependent: we have $\varepsilon_3 = \varepsilon_{u_3} \gtrsim \varepsilon_u$ in puRU and pRU whereas $\varepsilon_3 = \varepsilon_{q_3} \gtrsim \varepsilon_q$ in pLU. The rates (C.29) and (C.30) are comparable for $\varepsilon_3 \sim 1$, but the former dominates for smaller $\varepsilon_3$. Furthermore, (C.31) is always smaller because by assumption $g_* \gtrsim y_t, g_s$. That might be even

more suppressed if dipoles come with a 1-loop suppression factor $c_{\text{dip.}} \sim g_*^2/16\pi^2$. For these reasons, we will measure the typical size of the anomalous top decays by evaluating (C.29). Recalling that the total decay width of the top is approximately $\Gamma_t = 1.4\,\text{GeV}$, and assuming the fiducial values $|a|, |b| \sim 1$, we find

$$\text{BR}(t \to ch/uh) \sim \begin{cases} 0 & \text{RU, LU} \\ 9.0 \times 10^{-9} \left(\frac{\text{TeV}}{m_*/g_*}\right)^4 & \text{puRU, pRU, pLU}, \end{cases} \tag{C.32}$$

which are virtually undetectable. The branching ratio into $u$ or the one into $c$ may nevertheless be significantly enhanced by taking a maximal $|a| \sim y_t/y_c$ or a maximal $|b| \sim y_t/y_c$ respectively. Both coefficients cannot be simultaneously large compatibly with $D - \overline{D}$ meson oscillations, though. So, taking for instance $|a| \ll 1$ and $|b| \sim y_t/y_c$ the mixing $\theta_c \sim 1$ gets larger than in anarchic models whereas $D - \overline{D}$ meson oscillation remains SM-like. In that limit we find $\text{BR}(t \to ch) \sim 7 \times 10^{-4}(g_*\text{TeV}/m_*)^4$, a value observable at HL-LHC [70]. The actual reliability of this result would however require a more dedicated study of all the other experimental constraints, that so far have been estimated numerically only under the hypothesis $|a|, |b| \sim 1$. This interesting study will be left for future work.

In models with partial universality in the right-handed sector, an additional contribution to $\Gamma_{\text{EW}}(t \to qZ)$ comes from left-left transitions. It is proportional to $V_{\text{CKM}} \widetilde{Y}_b^2 V_{\text{CKM}}^\dagger/\varepsilon_d^2$ in puRU and to $V_{\text{CKM}} \widetilde{Y}_b^2 V_{\text{CKM}}^\dagger/\varepsilon_{d_3}^2$ in pRU (see Table 5). This contribution is smaller than (C.29) for any $\varepsilon_d \gtrsim 0.05/\sqrt{g_*}$ in puRU or $\varepsilon_d, \varepsilon_{d_3} \gtrsim 0.05/\sqrt{g_*}$ in pRU, which is also the region we focused on in the paper, as emphasized in Sections 4.2.4 and 4.3.3. Yet, even with smaller $\varepsilon_d, \varepsilon_{d_3}$ the rate remains tiny. Indeed, one has

$$\text{BR}_{\text{EW}}(t \to qZ) \sim \frac{g^2}{16\pi} \frac{m_t}{\Gamma_t} \left(\frac{\lambda_C^2 y_b^2 v^2}{\varepsilon_{d,d_3}^2 m_*^2}\right)^2 \lesssim 2 \times 10^{-8} \qquad (\text{puRU, pRU}), \tag{C.33}$$

which is always negligible, on account of the neutron EDM constraints in Eqs. (4.40) and (4.71).

In flavor anarchy, the structure of the flavor non-universal Higgs couplings is different than in our flavor models, but the resulting effects are always phenomenologically irrelevant because of the much larger scale $m_*$.

### C.4.2   Direct CP-violation in the $D$ system

A potentially important $\Delta F = 1$ constraint comes from direct CP violation in the hadronic decays of $D$ mesons. In particular, we consider the observable

$$\Delta a_{\text{CP}} \equiv a_{K^+K^-} - a_{\pi^+\pi^-} \tag{C.34}$$

whose latest measured value [95]

$$\Delta a_{\text{CP}} = (-15.4 \pm 2.9) \times 10^{-4} \tag{C.35}$$

is roughly compatible with the expected SM prediction.

The associated $c \to u$ transitions are affected by four-fermion operators as well as

$$\mathcal{H}_{\text{eff}} = \frac{G_F}{\sqrt{2}} \frac{m_c}{4\pi^2} \left(C_8 \bar{u}_L \sigma^{\mu\nu} g_s G_{\mu\nu} c_R + C_8' \bar{c}_L \sigma^{\mu\nu} g_s G_{\mu\nu} u_R\right) + hc\,, \tag{C.36}$$

see for example [96]. We will first focus on the latter Hamiltonian, and comment on the effect of four-fermion operators at the end.

The new physics contribution to $\Delta a_{\text{CP}}$ may be estimated as

$$\Delta a_{\text{CP}}^{\text{NP}} \sim \frac{2}{|V_{\text{CKM}}^{us} V_{\text{CKM}}^{cs}|} \text{Im}[\Delta R^{\text{NP}}] \text{Im}[C_8^{(')}] , \tag{C.37}$$

where $\text{Im}[\Delta R^{\text{NP}}]$ is the ratio between the hadronic matrix elements of the subdominant new physics operator and the dominant SM contribution. The value of $\text{Im}[\Delta R^{\text{NP}}]$ is not known, but is not expected to exceed $O(1)$ (see, e.g. [94, 97] and references therein). The coefficient $C_8^{(')}$ is generically non-vanishing in the three models with Partial Universality discussed in the main text, while it is absent at that order in scenarios with MFV. For models with Partial (Up) Right universality we estimate

$$\text{Im}[C_8'] \sim c \, \text{Im}[ab^*] \frac{4\pi^2}{m_*^2} \frac{\sqrt{2}}{G_F} \frac{y_c^2}{y_t^2} , \tag{C.38}$$

where $c \sim 1$ in generic models or $c \sim g_*^2/16\pi^2$ if dipoles first arise at 1-loop order, while $C_8'$ is further suppressed by $y_u/y_c$. For models with Partial Left Universality, we have instead the same estimate but with $C_8'$ and $C_8$ interchanged. It is worth noting that with flavor anarchy one gets an expression analogous to (C.38) but with the strong suppression $y_c^2/y_t^2$ replaced by the much larger $\lambda_C$ [39]. In conclusion, including an RG suppression $\sim 0.8 \div 0.9$ due to the running down to the $D$-meson mass, we obtain the following new physics contributions

$$\Delta a_{\text{CP}}^{\text{NP}} \sim 6 \times 10^{-4} \, c \, \text{Im}[ab^*] \text{Im}[\Delta R^{\text{NP}}] \times \begin{cases} 0 & \text{RU, LU} \\ \left(\frac{0.9 \, \text{TeV}}{m_*}\right)^2 & \text{puRU, pRU, pLU} \\ \left(\frac{120 \, \text{TeV}}{m_*}\right)^2 & \text{Anarchy} \end{cases} . \tag{C.39}$$

Taking $c \, \text{Im}[ab^*] \text{Im}[\Delta R^{\text{NP}}] \sim 1$ and $m_* \sim 1$ TeV the result is comparable to the experimental uncertainty shown in (C.35) in all our flavor models. Since the other bounds discussed in the main text suggest that $m_* \gtrsim 2 \div 3$ TeV, however, we conclude that this observable may become relevant only if $c \, \text{Im}[ab^*] \text{Im}[\Delta R^{\text{NP}}] \gtrsim 5 \div 10$, which seems rather unlikely especially when dipoles are of 1-loop size ($c \sim g_*^2/16\pi^2$). The constraint (C.39) is instead rather significant in models with flavor anarchy.

In analogy to (C.37), a four-fermion operator $\mathcal{O}_i$ with the appropriate $c \to u$ flavor structure may contribute to the asymmetry as $\Delta a_{\text{CP}}^{\text{NP}} \sim 2 \, \text{Im}[\Delta R_i^{\text{NP}}] \text{Im}[C_i^{\text{NP}}/C^{\text{SM}}]$, where $\Delta R_i^{\text{NP}}$ is the ratio of matrix elements of $\mathcal{O}_i$ and of $\mathcal{O}^{\text{SM}}$, the latter being the four-fermion operator controlling the dominant SM contribution, whereas $C_i^{\text{NP}}$ is the new physics contribution to the Wilson coefficient of $\mathcal{O}_i$ and $C^{\text{SM}} \sim (G_F/\sqrt{2}) V_{\text{CKM}}^{us}[V_{\text{CKM}}^{cs}]^*$ the coefficient of $\mathcal{O}^{\text{SM}}$ predicted in the SM. Again, models with complete universality have vanishing coefficients at leading order. In puRU and pRU the most important four-fermion operators are of the form $\bar{c}_R \gamma^\mu u_R \bar{f} \gamma_\mu f$ and have coefficients conservatively of order $ab^* \varepsilon_{u3}^2 g_*^2 (y_c^2/y_t^2)/m_*^2$, dominantly induced by $\mathcal{O}_{Hu}$ via $Z^0$-exchange. Conversely, in pLU the transition is dominantly induced by $\bar{u}_L \gamma^\mu u_L \bar{f} \gamma_\mu f$ with coefficients conservatively of order $a^* b \, \varepsilon_{q3}^2 g_*^2 (y_c^2/y_t^2)/m_*^2$. In either case we get

$$\Delta a_{\text{CP}}^{\text{NP}} \lesssim 2 \, \text{Im}[\Delta R_i^{\text{NP}}] \frac{g_*^2 \sqrt{2}}{m_*^2 G_F} \frac{\text{Im}[ab^*] y_c^2/y_t^2}{|V_{\text{CKM}}^{us} V_{\text{CKM}}^{cs}|} . \tag{C.40}$$

The expression is thus parametrically identical to (C.39) but with $c \sim g_*^2/16\pi^2$. Hence, CP-violation in $D$ decays does not appear to be an efficient probe of our flavor-symmetry scenarios unless the relevant hadronic form factors are larger than naively expected.

### C.4.3  Rare Kaon decays

Other potentially interesting observables are the rare decays $K \to \pi\nu\bar{\nu}$, $K \to \pi\mu\bar{\mu}$ [98, 99] and $K_{L,S} \to \mu\bar{\mu}$. We discuss in detail the current constraints from $K^+ \to \pi^+\nu\bar{\nu}$. The other observables lead to bounds that are at most comparable at present.

The relevant operators are the $s \to d$ analog of Eq. (B.33), which here we re-write as

$$\mathcal{L}_{\text{eff}} = \frac{4G_F}{\sqrt{2}} \frac{\alpha_{\text{em}}}{4\pi} \left[ C_{L,\nu}(\bar{d}_L\gamma^\mu s_L)(\bar{\nu}\gamma_\mu P_L\nu) + C_{R,\nu}(\bar{d}_R\gamma^\mu s_R)(\bar{\nu}\gamma_\mu P_L\nu) \right]. \tag{C.41}$$

By P-invariance the axial part of the quark currents does not contribute to the QCD matrix elements $\langle\pi|(\bar{d}\gamma^\mu P_L s)|K\rangle = \langle\pi|(\bar{d}\gamma^\mu s)|K\rangle/2 = \langle\pi|(\bar{d}\gamma^\mu P_R s)|K\rangle$. It follows that

$$\Gamma(K^+ \to \pi^+\nu\bar{\nu}) = \Gamma(K^+ \to \pi^+\nu\bar{\nu})_{\text{SM}} \left| 1 + \frac{C_{L,\nu}^{\text{NP}} + C_{R,\nu}^{\text{NP}}}{C_{L,\nu}^{\text{SM}}} \right|^2, \tag{C.42}$$

with the superscripts $^{\text{SM}}$ and $^{\text{NP}}$ indicating the SM and new physics contributions respectively. Because in our scenarios the coefficients $C_{L/R,\nu}^{\text{NP}}$ are a priori uncorrelated, the bounds we obtain by turning on an operator at a time are exactly the same. Note also that in our scenarios the neutrino current is flavor universal up to negligible corrections.

We extract the bounds from [100]. The authors define

$$C_{L,\nu}^{\text{NP}} = \frac{\sqrt{2}}{4G_F} \frac{4\pi}{\alpha_{\text{em}}} \frac{1}{2}[\widetilde{\mathcal{C}}_{Hq}^{(1)} + \widetilde{\mathcal{C}}_{Hq}^{(3)}]^{12} \equiv \frac{\sqrt{2}}{4G_F} \frac{4\pi}{\alpha_{\text{em}}} \frac{\widetilde{c}_{L,\nu}}{\Lambda^2}, \tag{C.43}$$

$$C_{R,\nu}^{\text{NP}} = \frac{\sqrt{2}}{4G_F} \frac{4\pi}{\alpha_{\text{em}}} \frac{1}{2}[\widetilde{\mathcal{C}}_{Hd}]^{12} \equiv \frac{\sqrt{2}}{4G_F} \frac{4\pi}{\alpha_{\text{em}}} \frac{\widetilde{c}_{R,\nu}}{\Lambda^2},$$

and obtain $\text{Re}[\widetilde{c}_{L/R,\nu}] \in [-1.8, 4.4] \times 10^{-4}$, $\text{Im}[\widetilde{c}_{L/R,\nu}] \in [-2.8, 3.5] \times 10^{-4}$ for $\Lambda = 1$ TeV. The interval is asymmetric due to the possibility of a partial cancellation with the SM contribution. To avoid this unlikely coincidence we conservatively impose

$$\left| \frac{\widetilde{c}_{L/R,\nu}}{\Lambda^2} \right| < 1.8 \times 10^{-4} \, \text{TeV}^{-2}. \tag{C.44}$$

The resulting constraints on the models' parameters are presented in Table 16. Consistently, the bounds are parametrically the same as those from $C_{10}, C'_{10}$, but on the new physics scale $m_*$ are numerically a factor $\sim 10$ weaker than those arising from $B$ decays. As we emphasized several times already, also the constraints in Table 16 may be relaxed by invoking the $P_{\text{LR}}$ symmetry.

## C.5  Prospects

In this appendix, we list the expected improvement over the next 10-20 years in the experimental sensitivity on the most constraining observables discussed in Sections C.1, C.2 and C.3. Our main goal is to obtain a qualitative picture of how the allowed parameter space of our models will evolve in the near future. The result of our analysis is summarized in the right panels of Figs. 5, 7, 8, 6, 9 and discussed in the conclusions.

| Coefficient | Right Univ. | | Partial Up Right Univ. | | Partial Right Univ. | |
|---|---|---|---|---|---|---|
| $[\widetilde{\mathcal{C}}_{Hq}^{(1/3)}]^{12}$ | $\frac{A^2\lambda_C^5\, y_t^2}{m_*^2\varepsilon_u^2}$ | $\frac{0.8}{\varepsilon_u}$ | $\frac{A^2\lambda_C^5\, y_t^2}{m_*^2\varepsilon_{u_3}^2}$ | $\frac{0.8}{\varepsilon_{u_3}}$ | $\frac{A^2\lambda_C^5\, y_t^2}{m_*^2\varepsilon_{u_3}^2}$ | $\frac{0.8}{\varepsilon_{u_3}}$ |
| $[\widetilde{\mathcal{C}}_{Hd}]^{12}$ | $\times$ | | $\times$ | | $\frac{a'b'y_s^2 g_*^2 \varepsilon_{d_3}^2}{y_b^2 m_*^2}$ | $1\, g_*\varepsilon_{d_3}$ |

| Coefficient | Left Univ. | Partial Left Univ. | |
|---|---|---|---|
| $[\widetilde{\mathcal{C}}_{Hq}^{(1/3)}]^{12}$ | $\times$ | $\frac{A^2\lambda_C^5 g_*^2 \varepsilon_{q_3}^2}{m_*^2}$ | $1\, g_*\varepsilon_{q_3}$ |
| $[\widetilde{\mathcal{C}}_{Hd}]^{12}$ | $\times$ | $\frac{a'b'y_d y_s^3}{y_b^2 \varepsilon_q^2 m_*^2}$ | — |

Table 16: Constraints from $K \to \pi\nu\nu$ decays. For each operator, we report the leading order expression of the Wilson coefficient, including its dependence on $a, b, a', b'$, and the constraint on $m_*$ in TeV, assuming $a, b, a', b'$ are of order unity. By $\times$ we indicate that the operator is not generated at leading order. By — we emphasize that the bound allows $m_* \sim 1$ TeV for any value of the $\varepsilon_\psi$ parameters identified in the bulk of the paper.

- **Bosonic operators.** HL-LHC projections for the bounds on $\mathcal{O}_H$, $\mathcal{O}_W$ and $\mathcal{O}_{2W}$ are obtained respectively from [70], [70], [101] and read

$$\mathcal{C}_H \to m_* \gtrsim 1.1\, g_*\, \text{TeV}\,, \qquad \mathcal{C}_W \to m_* \gtrsim 4.8\, \text{TeV}\,, \qquad \mathcal{C}_{2W} \to m_* \gtrsim 8.0/g_*\, \text{TeV} \tag{C.45}$$

  The sensitivity on the electroweak $\widehat{T}$ parameter will improve by a factor of 2 [70].

- **Semileptonic $B$ decays.** The sensitivity on $C_{10}$ and $C_9$ will improve roughly by a factor 3, see Fig. 7.6 of [10]. If no discovery is made, semileptonic $B$ decays will thus remain the most constraining flavor-violating observables in all our flavor models without $P_{\text{LR}}$ protection.

- $b \to s$ **transitions.** Belle II will increase the statistics of $B \to X_s\gamma$ but, unfortunately, the uncertainty on $C_7$ will soon be dominated by the theoretical error. In Table 61 of [11] the authors estimate approximately an $\sim 2$ improvement in the experimental sensitivity on $C_7$, which we assume in our projections. Regarding $C_7'$, the dominant constraint is expected from a measurement of $B^0 \to K^*e^+e^-$ at LHCb. The experimental uncertainty should go from 11% to roughly 2% [10], suggesting an improvement of roughly a factor $\sim 5$.

- **$\Delta F = 2$ observables.** The sensitivity on the coefficients of $\Delta F = 2$ four-fermion operators is expected to improve by roughly a factor of 4 at the HL-LHC, see Fig. 25 of [69]. The sensitivity on both $B$ and $K$ meson oscillation will increase at a comparable rate, so the relative relevance of these observables will not be qualitatively affected in the near future.

- **Neutron EDM.** The sensitivity on the neutron EDM is expected to improve at least by a factor 10 and potentially even 100 in the most optimist projections [17]. In the projections in the right panels of Figs. 5, 7, 8, 6, 9 we conservatively assumed a factor 10 improvement and a coefficient of the strange-dipole contribution $|c_s| \sim 0.01$ (see Eq. (B.50)), as found in [61].

- **Unitarity of CKM.** The sensitivity on the anomalous couplings of fermions to the $W$ boson are particularly relevant in scenarios with left universality. In particular, a

flavor universal shift (essentially a modification of the Fermi constant, experimentally detected as a violation of unitarity of the CKM) currently provides a significant bound in LU. The uncertainty on this quantity is mostly theoretical and it is therefore difficult to estimate how it will evolve in the future. For definiteness we assume an improvement of $1/2$ in the sensitivity.

— **Dijets.** These currently provide significant constraints especially in the RU and LU scenarios, where there is no separation between the compositeness of the light and the heavy families. According to [51], the new physics reach in dijet observables is not expected to change significantly with the full HL-LHC luminosity because theory uncertainties, especially due to PDFs, already dominate. Nevertheless, an improvement by a factor of 2 or 3 appears to be reasonable [102]. Conservatively we thus assume a factor of 2 improvement on the sensitivity on the Wilson coefficients constrained by dijet searches.

— **Anomalous vector couplings to the top.** The sensitivity on $[\widetilde{\mathcal{C}}_{Hu}]^{33}$ and $[\widetilde{\mathcal{C}}_{Hq}]^{33}$ are also expected to improve [59]. However, those coefficients are not very relevant to our analysis because constraints from flavor violation and bounds from bosonic operators are always more constraining.

— **Electron EDM.** This is expected to improve by a factor $\sim 10$ (see for instance [72]) compared to the current best bound we adopted here.

— $\mu \to e$ **conversions.** Regarding the leptonic sector, the bound on $\mathrm{BR}(\mu \to e\gamma)$ is expected to improve by a factor $\sim 5$ [15]. The branching ration for $\mu \to e$ conversions in nuclei, currently dominated by the constraint $\mathrm{BR}(\mu\,\mathrm{Au} \to e\,\mathrm{Au}) < 7 \times 10^{-13}$, will improve by a remarkable factor of $10^4$ by the Mu2e experiment [13]. An equally remarkable improvement is expected for $\mu \to eee$ by Mu3e [14].

## C.6   Direct resonance production

Resonance searches in scenarios of partial compositeness may be correlated to our study of the indirect signatures via the Lagrangian in Eq. (2.3). Here we limit our discussion to a list of some of the relevant literature on the subject. We focus on colored fermionic resonances, aka top partners, and vector resonances, which are the two most generic implications of SILH scenarios.

- **Top partners** Pair-production via QCD interactions of colored fermionic resonances is constrained by current data only in the low-mass range $\sim 1 \div 2$ TeV [103]. Preliminary FCC-hh projections suggest a direct reach of up to 9 TeV [104] for the highest target luminosity of $10$ ab$^{-1}$.

  Single-production can probe top-partners up to slightly higher scales, $m_* \sim 2 \div 3$ TeV, but is more model-dependent, see [29, 103, 105] for the analysis of a few benchmark scenarios. FCC-hh projections [104] indicate a direct reach of $10 \div 20$ TeV in the best case scenarios.

  The direct reach of a 10+ TeV muon collider on top partners is, instead, roughly half of the center of mass energy of the collider, ranging from 10 to 30 TeV in the current proposals [71].

- **Spin-1 resonances** Vectors can kinetically mix with the SM gauge bosons, and the resulting phenomenology is rather model-independent. Current constraints cover roughly up to $4 \div 5$ [106] (see also [89] for a recent experimental results). FCC-hh projections can be found in Ref. [70] and are expected to explore the mass range $m_* \sim 10 \div 15\,\mathrm{TeV}$, or even higher.

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
