# Peer review of "Exploring the Flavor Symmetry Landscape"

_SciPost Physics_

## Round 2 · Referee Report · Anonymous (Referee 1) · 2024-4-29

Strengths
2 The paper is very clearly written and well structured.
3 The list of observables is well chosen and all the material necessary to cross-check numerical values is given.
Report
The paper is well written and clearly structured. A reader's guide in section 3. 2 makes it easy to navigate through the following sections.
The observables for the phenomenological study are well chosen and the numerical boundaries on the parameters of different models can be checked easily with the materials provided, including the extensive appendices.
The plots provided and the corresponding explanations of results nicely summarize the findings of the study.
I recommend the paper for publications without any further changes.
Recommendation
Publish (meets expectations and criteria for this Journal)

---

## Round 2 · Referee Report · Anonymous (Referee 2) · 2024-5-28

Strengths
1- The paper is a comprehensive update to the partial compositeness paradigm taking into account the latest collider, EW, and flavor data. 2- By restricting to the partial compositeness framework, the authors are able to work with an EFT that keeps the associated dynamical correlations. 3- The analysis is exhaustive, complete, and accurate. Organization of the paper is excellent. 4- The important point that new physics can still lie around a few TeV (if protected via appropriate flavor symmetries) is re-emphasized, as well as the fact that current indirect experiments can still make progress in the near future.
Weaknesses
1- While clearly useful, the paper is largely an update and collection of established results in the flavor and composite Higgs literature. 2- Keeping dynamical correlations between Wilson coefficients is emphasized as an advantage over SMEFT analyses, but no clear strategy to exploit these correlations to maximize discovery potential is given. 3- While FCC-hh is often mentioned, the future collider discussion does not include any implications for FCC-ee (currently proposed to precede FCC-hh by ~25 years).
Report
Their main result is that the lowest allowed new physics scale is realized for options featuring an intermediate flavor symmetry, such as (variations of) $U(2)$, rather than the maximal symmetry of MFV. While less optimal models may be explored indirectly in the near future via flavor and CP violating observables, they find that the optimal ones are best explored at a future high-energy collider.
While the analysis is comprehensive and sound, a potential drawback of the paper is that the conclusions of the previous paragraph are known in the literature. I say potential because, while known, this result is important enough to be worth re-emphasizing in the PC context. In addition, SMEFT analyses are criticized due to the lack of dynamical correlations, but no clear plan is provided to exploit the correlations in their approach to increase discovery potential. Rather, the focus is on the interplay of flavor data together with flavor-conserving constraints from colliders and EW precision observables to set lower bounds on the new physics scale. While this is interesting, this interplay has certainly been pointed out before. Also it seems that many of the bounds are dominated by a single observable (the usual suspects well-known in the composite Higgs literature), which opens the question of how much one really benefits from keeping correlations.
Overall, the paper clearly meets the journal’s acceptance criteria and adds to the literature as a comprehensive and modern reference on flavor in the context of composite Higgs modes. I am therefore happy to recommend it for publication after some minor points are addressed.
Requested changes
1- I would like the authors to clarify the advantage to the correlations between EFT operators kept in their approach, and in particular if one can exploit these correlations to devise experimental search strategies to improve discovery potential at current/and future machines.
2- Second, given the focus on FCC-hh in the paper, I think comments on the indirect potential of FCC-ee are in order, given that the machine is part of the same program and is proposed to arrive much earlier.
3- Finally, since the optimal flavor symmetries identified in this work appear very close to $U(2)^n$ symmetries appearing in the literature that have long been recognized to provide a good compromise for simultaneously passing flavor, collider, and EW bounds in a wide variety of contexts (i.e. in EFT analyses and in concrete models), I think a more exhaustive inclusion of references would be appropriate.
Recommendation
Ask for minor revision

---

## Editorial Decision

unknown